# FANTASTIC PRETRAINING OPTIMIZERS AND WHERE TO FIND THEM

**Kaiyue Wen**
Stanford University
kaiyuew@stanford.edu

**David Hall**
Stanford University
dlwh@cs.stanford.edu

**Tengyu Ma**
Stanford University
tengyuma@stanford.edu

**Percy Liang**
Stanford University
pliang@cs.stanford.edu

## ABSTRACT

AdamW has long been the dominant optimizer in language model pretraining, despite numerous claims that alternative optimizers offer 1.4 to 2× speedup. We posit that two methodological shortcomings have obscured fair comparisons and hindered practical adoption: (i) unequal hyperparameter tuning and (ii) limited or misleading evaluation setups. To address these two issues, we conduct a systematic study of ten deep learning optimizers across four model scales (0.1B–1.2B parameters) and data-to-model ratios (1–8× the Chinchilla optimum). We find that fair and informative comparisons require rigorous hyperparameter tuning and evaluations across a range of model scales and data-to-model ratios, performed at the end of training. First, optimal hyperparameters for one optimizer may be suboptimal for another, making blind hyperparameter transfer unfair. Second, the actual speedup of many proposed optimizers over well-tuned baselines is lower than claimed and decreases with model size to only 1.1× for 1.2B parameter models. Thirdly, comparing intermediate checkpoints before reaching the target training budgets can be misleading, as rankings between two optimizers can flip during training due to learning rate decay. Through our thorough investigation, we find that all the fastest optimizers such as Muon and Soap, use matrices as preconditioners — multiplying gradients with matrices rather than entry-wise scalars. However, the speedup of matrix-based optimizers is inversely proportional to model scale, decreasing from 1.4× over AdamW for 0.1B parameter models to merely 1.1× for 1.2B parameter models.

## 1    INTRODUCTION

Pretraining has been the most computationally expensive component in the training pipeline for large language models, accounting for over 95% of the cost in DeepSeek V3 DeepSeek-AI et al. (2025b), and the additional RL training cost in DeepSeek R1 DeepSeek-AI et al. (2025a) is also comparatively much smaller. Until recently, AdamW has been the standard optimizer. Recent studies have introduced novel optimizers that claim to accelerate pretraining by 1.4× to 2× compared to AdamW Liu et al. (2024a); Vyas et al. (2025); Liu et al. (2025a); Yuan et al. (2025); Liang et al. (2025); Wang et al. (2025); Liu et al. (2025c); Pethick et al. (2025); Ma et al. (2025), yet these optimizers have not yet been widely adopted in real-world pretraining DeepSeek-AI et al. (2024); Yang et al. (2025); Grattafiori et al. (2024), with notable exceptions of Kimi K2 Team et al. (2025b), which uses the Muon-clip optimizer (a variant of Muon Jordan et al. (2024)), and GLM 4.5 Team et al. (2025a), which uses Muon.

We pinpoint two problems in optimizer evaluation in the evaluation process of these optimizers that both undermine confidence in new methods and limit their practical adoption. First, some baselines suffer from improperly tuned hyperparameters. Second, many experiments are confined to smaller-scale settings, leaving unanswered questions about how these optimizers perform in broader,

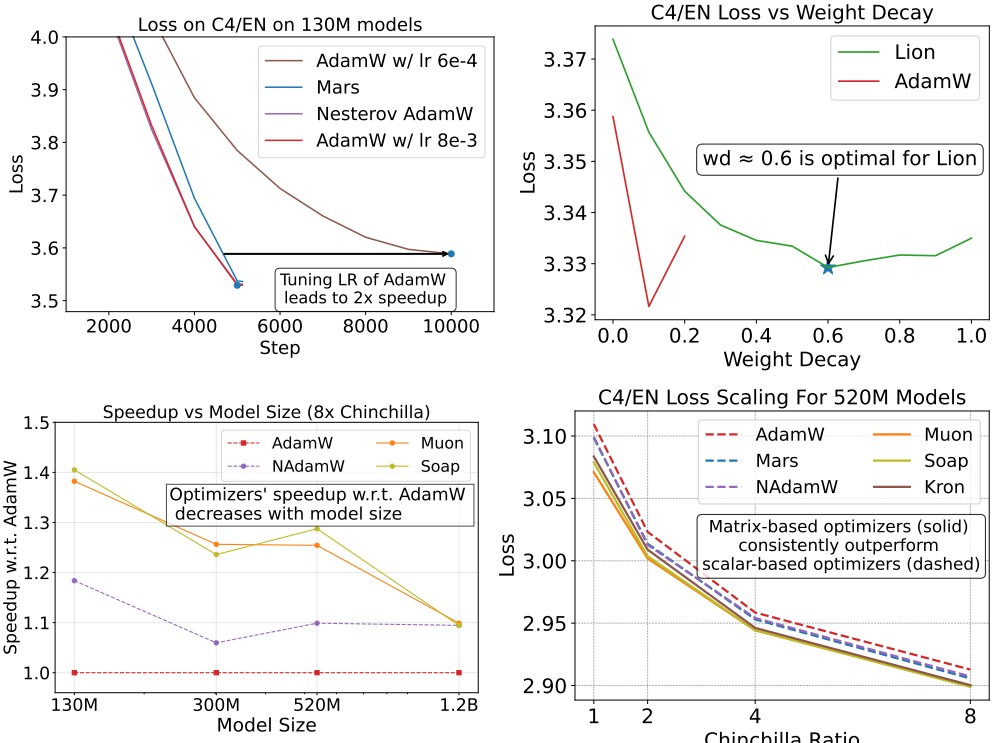

Figure 1: **Top Left**: The commonly used AdamW baseline for optimizer design is under-tuned. Up to a 2× speedup is achievable by tuning a single hyperparameter (learning rate) in the GPT-3 recipe Brown et al. (2020) for a 100M model (adopted in Liu et al. (2024a); Wen et al. (2024); Yuan et al. (2025); Liang et al. (2025); Wang et al. (2025)), highlighting the importance of proper hyperparameter optimization. **Top Right**: Fixing hyperparameters across optimizers does not guarantee fair comparison. Shared hyperparameters such as learning rate and weight decay are commonly set to a constant in previous studies. However, even conceptually similar optimizers may correspond to very different optimal hyperparameters. **Bottom Left**: Speedup decays with model size. While some optimizers show high (1.3-1.4×) speedup over AdamW on models under 1B parameters, the speedup decays with model size to only 1.1× for 1.2B parameters. **Bottom Right**: Matrix-based optimizers consistently outperform scalar-based optimizers. The loss curves for three scalar-based optimizers (AdamW, Nesterov AdamW, Mars) and three matrix-based optimizers (Kron, Soap, Muon) trained with different Chinchilla ratios of data are shown. Matrix-based optimizers achieve a consistent speedup over scalar-based optimizers. Furthermore, the three matrix-based optimizers converge to a similar loss in an overtrained setting.

more realistic scenarios. To address these issues, we benchmark eleven optimizers, including AdamW, in a rigorously controlled setup, focusing on two key questions:

1. **How to ensure hyperparameter optimality?** Previous works typically rely on manual hyperparameter selection and keep common hyperparameters such as learning rate and weight decay fixed across optimizers (e.g. Liu et al. (2025a)). This tuning process may result in a weak baseline, as the hyperparameters chosen may favor the proposed optimizers rather than for AdamW. We validate this concern by showing that tuning just one hyperparameter in the widely adopted GPT-3 recipe (introduced in Brown et al. (2020) and used in Liu et al. (2024a); Wen et al. (2024); Yuan et al. (2025); Liu et al. (2025d;c); Liang et al. (2025); Wang et al. (2025)) can yield a 2× speedup for pretraining (Figure 1, top left). To address, this we perform coordinate descent in the hyperparameter space on N hyperparameters for all eleven optimizers, iterating until convergence across for each of six different settings for models with up to 0.5B parameters to ensure that the hyperparameters are near-optimal for each setting.

2. **How do speedups differ across different scaling regimes?** Large language models are trained in different regimes: depending on the model size, the data-to-model ratio (the number of tokens

over the number of parameters) can range from 1 to more than 50 times the Chinchilla optimal (about 20 based on Hoffmann et al. (2022)). Typical optimizer experiments in $1\times$ Chinchilla optimal regimes raise concerns about a new optimizer's effectiveness in high data-to-model ratio. To address this, we benchmark the optimizers in four distinct data-to-model ratios ($1\times, 2\times, 4\times$ and $8\times$ the Chinchilla optimal regime) and scale up to 1.2B parameter models (following Everett et al. (2024); Zhang et al. (2025a)).

Concretely, we employ the Llama 2 architecture Touvron et al. (2023b); Grattafiori et al. (2024) (ranging from 0.1B to 1.2B parameters) and a data mixture similar to OLMo 2's OLMo et al. (2025). We focus on the final validation loss on the C4-EN mixture as a known proxy for downstream performance Bhagia et al. (2024), while also tracking exact downstream performance on various benchmarks. As previous works Vyas et al. (2025); Liu et al. (2025a) show that the step-wise computation overhead of matrix-based optimizers can be reduced to under $10\%$ through proper implementation, we primarily compare algorithms by the number of tokens needed to reach a given loss.

Our empirical results show the necessity of careful hyperparameter tuning and end-of-training evaluations across a range of model scales and data-to-model ratios:

1. **Hyperparameter transfer between optimizers is non-trivial.** Even similar optimizers may need very different hyperparameters (e.g., Lion's optimal weight decay $\approx 0.6$ vs. AdamW's $\approx 0.1$, Figure 1, top right), so fixing hyperparameters across optimizers can lead to unfair comparisons.

2. **The speedup of new optimizers is lower than claimed and diminishes with model size.** Many reported speedups of $2\times$ simply reflect a weak baseline. Against our well-tuned AdamW baseline, the speedup of alternative optimizers does not exceed $1.4\times$ (Figure 4). Furthermore, while new optimizers such as Muon and Soap show $1.3\times$ speedups for small models (0.1B), the speedups diminish to around $1.1\times$ for 1.2B parameter models at $8\times$ Chinchilla Ratio (Figure 1, bottom left), a regime that is not tested in previous works studying the scaling law of these optimizers [1].

3. **Early-stage loss curves can mislead significantly.** During learning rate decay, loss curves of different optimizers may cross multiple times (Figure 6), so judging optimizers using intermediate checkpoints may result in a different ranking than comparing models at the target training budget.

Our benchmarking also reveals new insights about optimizer design:

1. **Matrix-based optimizers consistently outperform scalar-based optimizers for small models.** *Scalar-based optimizers* (e.g., AdamW, Lion, Mars, etc.) update each parameter individually using scalar operations. After proper tuning, all scalar-based optimizers achieve similar optimization speeds to AdamW, with an average speedup ratio of less than $1.2\times$. *Matrix-based optimizers* (e.g., Kron, Muon, Soap, etc.) leverage the inherent matrix structure of neural network parameters and precondition gradients using matrix multiplication. Despite their diverse update rules, matrix-based optimizers all deliver approximately a $1.3\times$ speedup over AdamW (Figure 1, bottom right) for models under 520M parameters.

2. **Optimal choice of optimizer shifts depends on data-to-model ratios.** A winner in the $1\times$ Chinchilla regime may be suboptimal when data-to-model ratio increases. For example, while Muon is consistently the best optimizer in smaller Chinchilla ratio regimes, it is outperformed by Kron and Soap when the data-to-model ratio increases to 8x or larger (Figures 4 and 5).

## 2 RELATED WORKS

Due to page limit, a more detailed discussion of the related work is deferred to Section B.

**Optimizers for Pretraining.** Research on optimization for deep learning builds on classical stochastic methods Robbins and Monro (1951); Nesterov (1983); Duchi et al. (2011) and neural network–specific insights Sutskever et al. (2013), leading to adaptive methods such as Adam Kingma and Ba (2017) and its variants (e.g., AdamW Loshchilov and Hutter (2019), Nesterov Adam Dozat (2016)). Subsequent

---

[1] The original Soap paper Vyas et al. (2025) investigate model sizes up to 0.6B parameters. and Kimi's paper on Muon Liu et al. (2025a) only considers $1\times$ Chinchilla regime.

improvements have addressed convergence guarantees, variance reduction Yuan et al. (2025); Pagliardini et al. (2024), memory efficiency Shazeer and Stern (2018); Zhang et al. (2025b), and large-batch training You et al. (2020). In parallel, matrix-preconditioned methods such as Shampoo Gupta et al. (2018), Muon Jordan et al. (2024), and Scion Pethick et al. (2025) have emerged, alongside approaches leveraging Hessian information Yao et al. (2021) or symbolic discovery Chen et al. (2023). Another popular line of research is designing better optimizers specifically for pretraining. These optimizers are our main objects of study. An (incomplete) list of optimizers and their claimed speedups over AdamW includes Sophia Liu et al. (2024a) (2×), Soap Vyas et al. (2025) (1.4×), Muon Jordan et al. (2024); Liu et al. (2025a) (2×), MARS Yuan et al. (2025) (2×), Cautious AdamW Liang et al. (2025) (2×), Block-wise Learning Rate Adam Wang et al. (2025) (2×), FOCUS Liu et al. (2025c) (2×), SWAN Ma et al. (2025) (2×), DION Ahn et al. (2025) (3×), and SPlus Frans et al. (2025) (2×). We present a comparison of setups with these prior works in Section I.

**Re-evaluation Methodology.** Our work is a rigorous evaluation of optimizers for pretraining. Rigorous evaluation has been an important part of deep learning research to clarify the current status of research and move the community forward. Jiang et al. (2019) critically examines metrics for predicting LLMs' generalization capability and has facilitated research on understanding loss landscape sharpness and generalization and new optimizers such as SAM Foret et al. (2021). Reevaluations for optimizers in different settings than us have also shown insight on optimizer design Schmidt et al. (2021); Zhao et al. (2025); Kasimbeg et al. (2025).

**Comparison with concurrent work Semenov et al. (2025)** Semenov et al. (2025) study the same problem of benchmarking optimizers for pretraining and reach similar high-level conclusions on the importance of weight decay, learning rate decay, and the benefits of variance-reduced AdamW variants. Our results diverge in the relative ranking of matrix-level optimizers: while we find Muon competitive with or surpassing Mars, they report the opposite. We attribute this to differences in experimental regimes—our larger-batch TPU-v5lite setting favors matrix-level methods, whereas their smaller-batch GPU setting amplifies the advantages of variance reduction. Additional differences in learning rate sweeps and Muon tuning strategies further contribute to the discrepancy, underscoring the sensitivity of optimizer benchmarking to batch size, hardware, and hyperparameter tuning.

# 3 METHODOLOGY

This section outlines our experimental design and evaluation protocol. Section 3.1 introduces the general setup, followed by a three-phase hyperparameter-tuning framework: Phase I (Section 3.2) uses coordinate-descent sweeps across model sizes and data ratios to identify scaling-sensitive parameters; Phase II (Section 3.3) refines these on mid-scale settings and selects optimizers; Phase III (Section 3.4) extrapolates hyperparameter scaling laws to 1.2B parameter and 16x Chinchilla regime. Optimal configurations and loss trends appear in Section D.

| Optimizer | References | Alg |
|---|---|---|
| **Baseline** | | |
| AdamW | Loshchilov and Hutter (2019) | Alg. 1 |
| **Variance-reduced AdamW Variants** | | |
| NadamW | Dozat (2016) | Alg. 2 |
| Mars | Yuan et al. (2025) | Alg. 5 |
| Cautious | Liang et al. (2025) | Alg. 7 |
| **Memory-efficient Optimizers** | | |
| Lion | Chen et al. (2023) | Alg. 3 |
| Adam-mini | Zhang et al. (2025b) | Alg. 6 |
| **Matrix-based Optimizers** | | |
| Muon | Jordan et al. (2024) | Alg. 8 |
| Scion | Pethick et al. (2025) | Alg. 9 |
| Kron (PSGD) | Li (2018a; 2022) | Alg. 10 |
| Soap | Vyas et al. (2025) | Alg. 11 |
| **Hessian-Approximation Optimizers** | | |
| Sophia | Liu et al. (2024a) | Alg. 4 |

Table 1: Optimizers under study

## 3.1 GENERAL EXPERIMENTAL SETUP

Following OLMo et al. (2025), we conduct all our experiments on a large-scale pretraining corpus composed of three publicly available datasets, tokenized with the Llama3 tokenizer: DCLM-baseline (3.8 trillion tokens, Li et al. (2025b)), StarCoder Data (0.25 trillion tokens, Lozhkov et al. (2024)), and ProofPile 2 (55 billion tokens, Azerbayev et al. (2024)).

| Stage | LR | WD | min lr ratio | Warmup | Max Grad Norm | Batch | Val. Loss |
|---|---|---|---|---|---|---|---|
| Init | 0.008 | 0.1 | 0 | 1000 | 1 | 256 | 3.298 |
| Round 1 | 0.008 | 0.1 | 0 | 2000 | 1 | 256 | 3.282 |
| Round 2 | 0.008 | 0.1 | 0 | 2000 | 1 | 128 | 3.263 |
| Best | 0.008 | 0.1 | 0 | 2000 | 2 | 128 | 3.263 |

Table 2: Illustrative coordinate-descent steps for AdamW on the 130M 1× Chinchilla regime. Unchanged hyperparameters: $\beta_1 = 0.9, \beta_2 = 0.98, \epsilon = 10^{-10}$.

Our benchmarks cover four model sizes derived from the architecture Touvron et al. (2023a;b); Grattafiori et al. (2024), with approximately 130M, 300M, 520M, and 1.2B parameters. Each variant uses a fixed sequence length of 4,096 and 32 transformer layers (following Liu et al. (2024b)), differing only in hidden dimension, intermediate dimension, and number of attention heads. Detailed hyperparameters are summarized in Table 4. Training is implemented in JAX and executed on TPU v5 hardware. We employ a mixed-precision scheme (parameters in fp32 and activations in bf16). For each model, we use 20 times its non-embedding parameter count to compute the Chinchilla optimal data-to-model ratio based on Hoffmann et al. (2022). We will use $n\times$ Chinchilla to represent training the models for $n$ times the Chinchilla optimal number of tokens.

Our primary evaluation metric for the model is the language modeling loss on the English split of the C4 dataset Raffel et al. (2023), which has been shown to be a strong proxy for downstream performance Bhagia et al. (2024). We also track downstream accuracy and bits-per-byte on the following suite of benchmarks: ARC (Easy and Challenge) Clark et al. (2018), BoolQ Clark et al. (2019), COPA Gordon et al. (2012), CommonsenseQA Talmor et al. (2018), HellaSwag Zellers et al. (2019), LAMBADA Paperno et al. (2016), OpenBookQA Mihaylov et al. (2018), PIQA Bisk et al. (2020), WSC273 Levesque et al. (2012), and Winogrande Sakaguchi et al. (2020).

Our study includes a wide range of eleven optimizers listed in Table 1. Due to the page limit, we defer the exact algorithm descriptions of these optimizers to Section C. We selected these eleven optimizers according to three guiding principles: (i) include widely adopted baselines such as AdamW and Lion; (ii) cover recently proposed optimizers; and (iii) when multiple methods share a similar update rule, choose a few representative algorithms. These choices ensure both breadth and depth in our comparison. We group the optimizers under study into five general classes:

1. Our baseline algorithm is AdamW, with $m_t$ and $v_t$ being first- and second-order momentum of gradient. AdamW's update rule is $w_{t+1} = w_t - \eta \frac{m_t}{\sqrt{v_t}+\epsilon} - \eta \lambda w_t$.

2. Reducing the variance of updates is a shared motivation behind many optimizers. For example, Nesterov AdamW incorporates the Nesterov lookahead technique to estimate gradients more accurately, with the following update rule: $w_{t+1} = w_t - \eta \frac{\beta_1 \mathbf{m_t} + (1-\beta_1)\mathbf{g_t}}{\sqrt{v_t}+\epsilon} - \eta \lambda w_t$.

3. Optimizers such as Lion aim to reduce the memory required by AdamW by only keeping the first-order momentum. Lion is observed to perform better than AdamW at a small scale Liang et al. (2025): $w_{t+1} = w_t - \eta \operatorname{sign}(\beta_2 m_t + (1 - \beta_2)g_t)$, where $m_t$ is the first-order momentum.

4. Other optimizers like Muon leverage the matrix structure of neural networks and perform preconditioning of gradients through matrix multiplication instead of scalar multiplication. For Muon, the key operation is called Newton-Schulz: $\operatorname{NS}(M) = M(aM + bM^\top M + c(M^\top M)^2)$. With appropriate $a, b, c$, one can prove that $\operatorname{NS}^{(5)}(M) \approx \arg\max_{\|O\|_{\text{op}}=1} \operatorname{Tr}(O^\top M)$ when $\|M\|_{\text{op}} < 1$. Muon has the following update rule: $w_{t+1} = w_t - \eta \operatorname{NS}^{(5)}(\beta_2 m_t + (1 - \beta_2)g_t)$. This is done for all the matrices except the token classification head and the embedding in the network, which are optimized by AdamW.

5. Optimizers including Sophia are motivated by the famous Newton's method and use Hessian-vector product to approximate the diagonalized version of the Hessian matrix empirically.[2]

## 3.2 PHASE I: FINE-GRAINED HYPERPARAMETER COORDINATE DESCENT

Fixed or lightly tuned baselines can severely understate an optimizer's capability and lead to overstated speedup claims. For instance, a single tweak to the learning-rate schedule with peak learning rate 6e-4

---

[2] We defer the result of Sophia to Section D.4.

| Optimizer | Scaling-Sensitive Hyperparameters |
|---|---|
| AdamW | learning rate, warmup, weight decay, batch size |
| Nesterov AdamW | learning rate, warmup |
| Muon | learning rate |
| Soap | learning rate, warmup, block size |

Table 3: Scaling-sensitive hyperparameters identified in Phase I (Section 3.2) for a selected subset of optimizers, see full list at Table 5. To reduce Soap's memory, we apply parameter blocking as in Anil et al. (2021).

in the GPT-3 recipe Brown et al. (2020) can produce nearly a 2× speedup (Figure 1, left). To avoid such artifacts, we perform an exhaustive, one-at-a-time sweep over each hyperparameter, identify the set of near-best configurations in each regime, and then determine which knobs truly require re-tuning as scale changes.

For each optimizer, we define a discrete grid for every hyperparameter (e.g., for AdamW, our swept hyperparameters include learning rate, weight decay, warmup steps, $\beta_1$, $\beta_2$, $\epsilon$, gradient-norm clipping, and batch size). Starting from a initial hyperparameter configuration similar to the one in the original paper or is commonly used in pretraining in each iteration, we hold all but one hyperparameter fixed at the best values, then search the whole grid for that parameter, and accept the new value if the validation loss improves by more than $\Delta_1 = 3 \times 10^{-3}$. We repeat passes until no parameter update yields further significant gain. We perform this sweeping for 6 different settings, namely 130M, 300M, 500M at 1× Chinchilla and 130M at 2×, 4×, 8× Chinchilla. One exemplary hyperparameter optimization procedure for AdamW on a model with 300M parameters and 1× Chinchilla is shown in Table 2.

**Result of Phase I.** By the end of Phase I, we have, for each optimizer and each of the six regimes (130M, 300M, 500M at 1×; 130M at 2×, 4×, 8× Chinchilla), identified a coordinate-wise local optimum of hyperparameters— when all hyperparameters except one are held fixed, minimizes validation loss.

### 3.3 Phase II: Coordinate Descent on Scaling-Sensitive Hyperparameters

While the extensive coordinate descent guarantees coordinate-wise optimality, it is too costly to perform on larger-scale experiments. We empirically observe two crucial properties of the sweeping that allow us to simplify the descent procedure: (1) Losses are sensitive to only a subset of hyperparameters; many have little effect on performance when perturbed from their optimal values. (2) Among the sensitive hyperparameters, the optimal settings for most remain stable across scales, so tuning is needed only at smaller scales.

Building on these observations, we can simplify our hyperparameter search by identifying scaling-sensitive parameters, which (i) crucially influence the final performance, and (ii) change with respect to the model scale. We define the **approximate-optimal configuration** for a given regime as the set of all hyperparameter tuples whose final loss lies within $\Delta_2$ =6.4e-3 of the regime's best-observed loss $L_r^*$, where $r$ denotes a regime/setting, that is, $r$ is a pair of choice of model size and data budget. Concretely, let $c_h$ denotes a hyperparameter (where $h$ is the index) and $c$ be the tuple of all hyperparameters.

1. For each regime $r$, let $\mathcal{C}_r = \{ c : L(c) \leq L_r^* + \Delta_2 \}$ be all the hyperparameter configurations in our coordinate descent procedure that yield approximately optimal loss.

2. A hyperparameter $c_h$ is called **scaling-insensitive** if there exists a single value $v_h$ such that there exists $c \in \mathcal{C}_r$ such that $c_h = v_h$ for every regime $r$ in Phase I. Otherwise, $c_h$ is **scaling-sensitive**, meaning its optimal value shifts depending on model size or data budget.

We carry forward only the scaling-sensitive hyperparameters (shown in Table 5) into Phase II, thereby focusing our next round of coordinate-descent sweeps on the hyperparameters that truly require re-tuning across scaling regimes. We then perform sweeping for another 6 different settings, namely 300M, 500M at 2×, 4×, 8× Chinchilla.

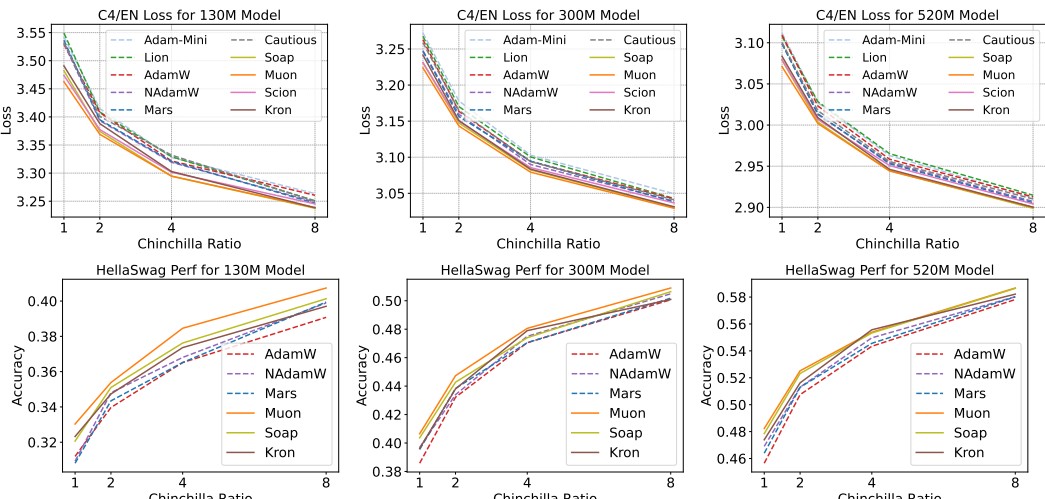

Figure 2: **Main Results For Phase I & II.** Top: We plot the validation loss on C4/EN for the experiments in Phase I and Phase II. Every point corresponds to the optimal loss achieved at the corresponding Chinchilla ratio for each optimizer. Bottom: we plot the HellaSwag performance corresponding to the selected run for a subset of optimizers: the AdamW baseline, the top 2 most performant scalar-based optimizers, and the top 3 most performant matrix-based optimizers. Analysis is deferred to Section 4.1.

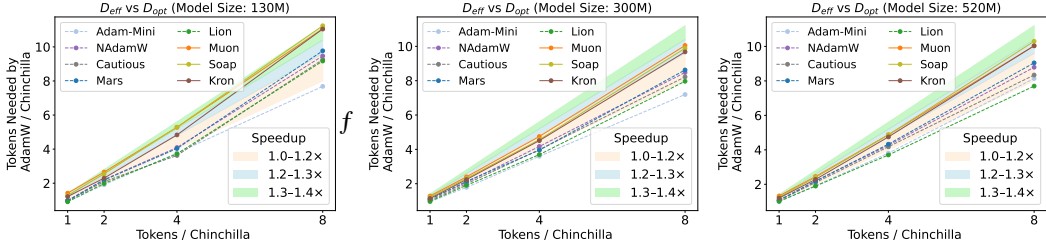

Figure 3: **Speedup of different optimizers across scale.** We estimate the speedup of different optimizers by fitting a scaling law for AdamW and then map the loss of different optimizers to the corresponding equivalent data budget. We observe that (i) The highest speedup is capped at 1.4×; (ii) matrix-based optimizers consistently outperform scalar-based optimizers and show an increasing speedup with data budget.

**Result of Phase II.** Combined with the results in Phase I, we obtain a set of near-optimal hyperparameters and their corresponding losses for 12 different settings (130M, 300M, 500M and 1×, 2×, 4×, 8× Chinchilla).

To quantify the speedup of different optimizers over the baseline AdamW, we fit how AdamW's loss scales with data budget $D$ for each model size $N$ with the following functional form: $\hat{L}_N(D) = \alpha_N D^{-B_N} + \beta_N$. Suppose an optimizer achieves loss $L_{\text{optimizer}}$ at data budget $D_{\text{Optimizer}}$. We calculate the corresponding data budget needed for AdamW to achieve this loss, denoted by $D_{\text{AdamW}}$ by finding the solution to the equation $\hat{L}_N(D_{\text{AdamW}}) = L_{\text{optimizer}}$. We then use $D_{\text{AdamW}}/D_{\text{Optimizer}}$ as the estimated speedup ratio.

Through this set of experiments, we observed that matrix-based optimizers consistently outperform scalar-based optimizers, but all optimizers' speedup ratios over AdamW do not exceed 1.4×.

### 3.4 PHASE III: HYPERPARAMETER SCALING LAW FOR FURTHER EXTRAPOLATION

Having obtained optimized hyperparameter settings from Phase II (Section 3.3), we now fit a smooth scaling law that predicts the optimal value of each scaling-sensitive hyperparameter as a function of

Figure 4: **Speedup of different optimizers across scale.** We estimate the speedup of different optimizers by fitting a scaling law for AdamW and then map the loss of different optimizers to the corresponding equivalent data budget. We observe that (i) The highest speedup is capped at $1.4\times$; (ii) matrix-based optimizers consistently outperform scalar-based optimizers and show an increasing speedup with data budget.

model size $N$ and data budget $D$. Concretely, we model the optimal value for each scaling-sensitive hyperparameter $h$ as: $h(N, D) = \alpha N^{-A} D^{-B} + \beta$, where $A$, $B$, $\alpha$, and $\beta$ are learned coefficients.

We estimate these parameters via non-linear least-squares on the 12 observed $(N, D, h)$ triples for each optimizer, minimizing the squared error between predicted and actual optimal hyperparameter values. To test the quality of our prediction, we ran a full Phase I sweep at $N = 1.2B$ and Chinchilla = 1 for AdamW. Comparing the identified optimum against our fitted hyperparameters, we observe that our predicted hyperparameters yield a final loss within 3e-3 of the optimal configuration, showing that our hyperparameter scaling law can effectively predict the optimal hyperparameters. We then performed two case studies to further extrapolate our benchmarking:

1. To test the effect of scaling up model sizes, we train 1.2B models using AdamW, Nesterov AdamW, and Muon on 1 to 8× Chinchilla ratio.

2. To further test the effect of different optimizers when data-to-model ratios are high, we train 130M and 300M models using AdamW, Nesterov AdamW, Muon, and Soap on 16x Chinchilla ratio.

**Results of Phase III.** We demonstrate two potential shortcomings of Muon optimizers, which is the best optimizers in Phase I and Phase II, through experiments in this phase: (i) while Muon's speedup persists for models up to 1.2B parameters, the speedup decreases to under 1.2×; (ii) With a 16× Chinchilla ratio, Soap outperform Muon on the 130M and 300M models.

## 4 EMPIRICAL FINDINGS

### 4.1 MAIN RESULTS

**Results on 0.1B–0.5B-parameter models.** Figure 2 shows the validation loss curves for the 130M, 300M, and 520M models with varying Chinchilla ratios (1 to 8) in our benchmark. We further show that HellaSwag accuracy improvements closely mirror validation-loss gains. This is consistent with prior works that show lower losses translates to better downstream accuracy Bhagia et al. (2024); Liu et al. (2025b). Across all model scales and compute budgets, both the variance-reduced Adam variants (NAdamW, Mars, Cautious) and the matrix-based optimizers deliver speedups over the AdamW baseline. However, no method achieves 2× step-wise acceleration claimed in previous literature. We note that Soap Vyas et al. (2025) is one of the few works that conduct independent hyperparameter sweeping for the baseline, and it indeed reports a speedup closest to the actual observed improvement. Following the methodology defined in Section 3.3, we calculate the estimated speedup ratio of different optimizers in Figure 4 and the highest speedup ratio is 1.4×. From the measured speedups, three patterns stand out in this computation regime:

1. *Matrix-based methods outperform scalar-based methods. The speedup ratio increases with data budget yet decreases with model size.* For every model size, matrix-based optimizers (Soap, Muon, Kron, Scion—solid curves) consistently drive validation loss below that of their scalar-based counterparts (dashed curves). In the base (1× Chinchilla) compute regime, Muon performs best, but at 8× Chinchilla compute, the advantage shifts to Soap and Kron. As shown by the super-linear trend in Figure 4, the speedup ratios of these three optimizers grow with increasing data budget. To the best of authors' knowledge, this dependency on data budget is not noted in prior works, which typically only experiment on one data-to-model ratio.

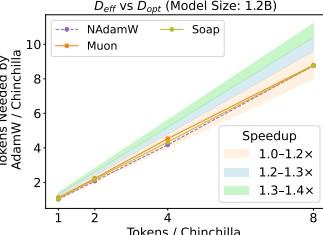 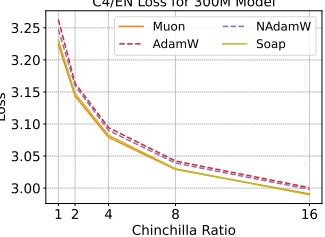

Figure 5: **Case Studies.** Left: Validation loss scaling on 1.2B model for AdamW, NAdamW, Muon and Soap. Muon and Soap still offer significant speedup over AdamW but no longer significantly speed up over NAdamW. Mid: Estimated speedup ratio with the same methodology Figure 4, we observe that Muon and Soap's speedup decays with model size to only $1.1\times$. Last: Experiment with 300M $16\times$ Chinchilla setting, Soap outperforms Muon when data-to-model ratio further increases.

However, the greatest gains occur on the 130M model, after which the speedup decreases to roughly 1.3× for larger model sizes. We will affirm this observation further in the experiments of 1.2B models, showing that the speedup ratios of matrix-based optimizers decrease to $1.1\times$ for the 1.2B model.

2. *Variance-reduction techniques provide a small but clear lift.* Within the scalar-based family, all variance-reduced Adam variants (NAdamW, Mars, Cautious) surpass vanilla AdamW—except for a small lag at the smallest experiment. Muon combines matrix-based updates with Nesterov momentum, illustrating how variance reduction compounds with matrix adaptation can yield greater efficiency. This result is very different from the $2\times$ speedup reported in prior works and we attribute this disparity to the better-tuned baseline.

3. *Memory-efficient variants of AdamW closely track the performance of AdamW.* The two memory-efficient AdamW variants (Lion, Adam-mini)—despite their reduced auxiliary state—closely track the performance of AdamW, with a slowdown of at most $5\%$ and sometimes even perform better than AdamW. Interestingly, Lion and Adam-mini show different scaling trends regarding model size: the disadvantage of Lion relative to AdamW widens while the disadvantage of Adam-mini over AdamW narrows.

**Results on 1.2B-parameters Models.** Using the hyperparameter scaling law we fit (Section 3.3), we scale up the model size to 1.2B to examine how the speedup of optimizers scales with model size. We observe that NAdamW, Muon, and Soap still deliver speedup over AdamW, but the speedup deminishes to $\approx 1.1\times$ for all these optimizers (Figure 5, Left and Mid) and no longer leads to downstream improvements (Table 6). While the exact reason for the speedup decay is not clear, we conjecture that the landscape-adaptation advantage of matrix-based optimizers plays a smaller role at the largest model scale we tested, where the noise factor may be more important for training dynamics, leading to similar performance between Muon and Nesterov AdamW.

**High data-to-model Ratio.** In our previous experiments, Muon is outperformed by Soap in the $8\times$ Chinchilla regime for the 130M and 520M models. We further train three 300M models to $16\times$ Chinchilla and verify that Muon is no longer the optimal optimizer when the data-to-model ratio increases (Figure 5, right). We conjecture that the second-order momentum maintained by Soap and Kron becomes more effective, and adaptivity to heterogeneity in parameter directions may lead to a larger speedup when the data-to-model ratio increases. We also perform similar experiments on 130M models and reach the same results (deferred to Section D.5).

## 4.2 NECESSITY OF RIGOROUS BENCHMARKING

Our systematic sweeps uncover both universal optimization principles and surprising optimization-specific nuances, which call for rigorous study when designing future optimizers.

First, we find that even a superior optimizer can underperform a less advanced method when its hyperparameters are not precisely tuned. In our exhaustive grid searches, slight deviations from each optimizer's ideal learning rate or other critical hyperparameter often lead to degradation in validation loss that is large enough to flip the ordering (Figure 6, Left). This hyperparameter sensitivity means manual selection without systematic sweeps will likely produce arbitrary ranking of the optimizers. To further demonstrate this, we plot how validation losses vary when only **one** of the hyperparameters deviates from optimal value for Muon, Soap, Mars on 520M model in $1\times$ Chinchilla regime (Figure 6, Mid). The ordering of the optimizers can easily flip if one chooses a sub-optimal hyperparameter.

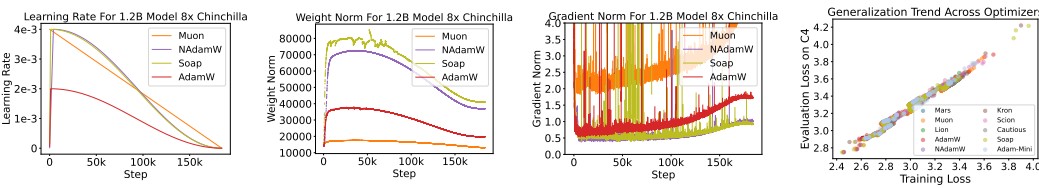

Figure 6: **Necessity of Careful Tuning.** Left: $2\times$ the optimal learning rate diminishes Soap's loss improvement over Mars on 520M 8x experiment; Mid: Variation of loss when only one hyperparameter differs from optimal learning rate and runs converge to within $0.02$ of optimal. The order of optimizers may flip arbitrarily if rigorous tuning is missing. Right: Changing a single hyperparameter like weight decay may lead to misleading faster loss improvement but plateaus later.

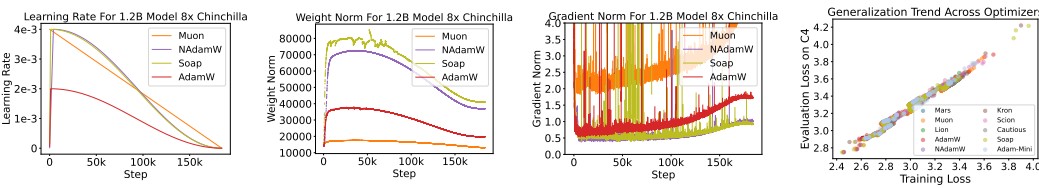

Figure 7: **Common Phenomena Across Optimizers. Left**: Learning rate used for different optimizers. **Middle Left**: Parameter norm of all optimizers shows a similar trend of increment and decrease, closely aligning the increasing and decaying of learning rate schedule. **Middle Right**: Gradient norm increases during learning rate decay. However, this increase does not lead to a loss increase. **Right**: The training loss and evaluation loss follows the same trend for all optimizers.

Second, using the same hyperparameters for different optimizers do not guarantee fair comparison between optimizers. For example, while weight decay is essential across all optimizers for optimal performance, the optimal decay strength varies markedly between optimizers. As shown in (Figure 1, top right), optimal weight decay coefficients differ between the optimizers studied. We also note the optimal weight decay for Kron is larger than the conventional $0.1$ and is approximately $0.5$, which is crucial for Kron to outperform AdamW.

Third, early training behavior can be highly misleading. Validation-loss curves during this initial phase tend to exaggerate performance gaps (Figure 1, bottom right) and, in some cases, even reverse the eventual ranking, (Figure 6, right). Many optimizers exhibit rapid early descent followed by plateauing, meaning that assessments based solely on losses before the end of the training trajectories fail to predict final outcomes.

## 4.3   COMMON PHENOMENA ACROSS OPTIMIZERS

Through logging the evolution of weight and gradient norms, we discovered some shared optimization phenomena across optimizers.

**Parameter norms typically track learning rate decay when there is weight decay** We observe that the parameter norms of all optimizers show a similar pattern of increase and decrease, closely aligning with the increase and decrease of the learning rate if there is a decay phase. However, the absolute values of parameter norms are very different across optimizers (Figure 7, Middle Left).

**Gradient norm increases during learning rate decay.** Across all the optimizers, the gradient norms increase during the training run. However, this increase does not lead to a loss increase. Similar to the parameter norms, the absolute values of gradient norms are not consistent across optimizers. These two phenomena are also reported in the previous work Defazio (2025), where the author also provides a theoretical explanation for both phenomena. We provide additional evidence that these two phenomena are not artifacts of the chosen optimizer AdamW but rather common phenomena across optimizers (Figure 7, Middle Right).

**Different optimizers have similar generalization behavior.** For architecture design, it has been observed that different architectures can have vastly different generalization behaviors Lu et al. (2025). However, this is not the case for optimizers, and the evaluation losses and training losses follow roughly the same trend for all optimizers at the regimes we experimented on (Figure 7, Right).

## ACKNOWLEDGEMENT

This work was supported by the Google TPU Research Cloud (TRC), the Stanford HAI–Google Cloud Credits Program, the DARPA AIQ program, Schmidt Sciences and NSF RI 2045685, and is a part of the Marin Project. The authors thank Evan Walters, Omead Pooladzandi, Jiacheng You, and Zhiyuan Li for discussions and help during our research.

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

## A    Discussion

We benchmarked 11 optimizers in pretraining and found that their true gains over AdamW are much smaller than previously reported. Our results highlight three key lessons: (i) many claimed speedups stem from insufficient hyperparameter tuning, as fair sweeps eliminate most of the apparent advantages; (ii) comparisons based on early or inconsistent evaluation can be misleading, since optimizer rankings often change over the full training trajectory; and (iii) even the best-performing alternatives provide only modest speedups, which further diminish with model scale, dropping to 1.1× at 1.2B parameters. This benchmarking study has the limitation that it does we haven't scaled to models larger than 1.2B parameters. However, we believe that evaluating optimizers on models of comparable size to prior studies is still valuable, as it reveals that insufficient tuning is a major cause of subpar speedups. LLM is used to fix grammar mistakes for this paper.

## B    Extended Related Work

**Optimizers for Deep Learning.** A long line of work has studied optimization for deep learning, incorporating insights from classical optimization literature Robbins and Monro (1951); Nesterov (1983); Duchi et al. (2011) and domain knowledge about deep neural networks Sutskever et al. (2013). (i) Early insights motivated the rise of optimizers that use adaptive learning rates based on second-order momentum Tieleman (2012); Zeiler (2012); Kingma and Ba (2017). Adam Kingma and Ba (2017) later became the default baseline for optimizers with adaptive learning rates. Since then, improvements over Adam and SGD have been proposed, with notable examples including Nesterov Adam Dozat (2016) and AdamW with decoupled weight decay Loshchilov and Hutter (2019). Other improvements include addressing the convergence of Adam on convex loss Reddi et al. (2018); Zaheer et al. (2018); Taniguchi et al. (2024), considering interpolation between Adam and SGD to improve generalization Luo et al. (2019); Xie et al. (2022), performing further variance reductions on optimizer updates Liu et al. (2021); Zhang et al. (2019); Yuan et al. (2025); Xie et al. (2024); Pagliardini et al. (2024); Zhuang et al. (2020), incorporating momentum on weights Ivgi et al. (2023); Defazio et al. (2024b), allowing easier hyperparameter tuning Defazio and Mishchenko (2023); Mishchenko and Defazio (2024); Defazio et al. (2024a), reducing memory usage by incorporating the structure of neural networks Shazeer and Stern (2018); Zhang et al. (2025b); Zhu et al. (2025); Luo et al. (2023); Modoranu et al. (2024); Zhao et al. (2024), and modifying the algorithm to allow larger batch sizes You et al. (2017; 2020). (ii) Starting from Preconditioned SGD Li (2018a) and Shampoo Gupta et al. (2018), another line of optimizer design Morwani et al. (2024); Eschenhagen et al. (2023); Martens and Grosse (2020); Li (2018b); Eschenhagen et al. (2025) began to incorporate matrix preconditioners instead of simple scalar preconditioners. Techniques including learning rate grafting Agarwal et al. (2020), blocking, and distributed methodology Anil et al. (2021) have since been proposed. These matrix-based approaches later led to the theory of modular duality in deep learning optimization Bernstein and Newhouse (2024a;b); Large et al. (2024) and new optimizers including Muon Jordan et al. (2024) and Scion Pethick et al. (2025). (iii) Motivated by Newton's algorithm, there has been a line of work that tries to incorporate Hessian information Becker and Cun (1989); Yao et al. (2021); Schaul et al. (2013); Yao et al. (2021). (iv) Symbolic discovery of optimizers Chen et al. (2023) has discovered a memory-efficient SignGD Bernstein et al. (2018) variant called Lion, which claims to outperform Adam on a wide range of tasks.

**Optimization for Pretraining.** Since Brown et al. (2020), the cost of pretraining has increased dramatically. One of the challenges is how to choose hyperparameters with minimal cost. A pivotal line of work in this direction is the tensor program series that allows for extrapolating some hyperparameters across scales Yang (2020b;c;a; 2021); Yang and Littwin (2023); Yang et al. (2022; 2024). Empirical results suggest that fitting a power law to scale hyperparameters is now common practice for large models Li et al. (2025a); Everett et al. (2024); Liu et al. (2025a); DeepSeek-AI et al. (2024); Zhang et al. (2025a). We have also incorporated this approach in our paper. Another popular line of research is designing better optimizers specifically for pretraining. These optimizers are our main objects of study. An (incomplete) list of optimizers and their claimed speedups over AdamW includes Sophia Liu et al. (2024a) (2×), Soap Vyas et al. (2025) (1.4×), Muon Jordan et al. (2024); Liu et al. (2025a) (2×), MARS Yuan et al. (2025) (2×), Cautious AdamW Liang et al. (2025) (2×), Block-wise Learning Rate Adam Wang et al. (2025) (2×), FOCUS Liu et al. (2025c) (2×),

SWAN Ma et al. (2025) (2×), DION Ahn et al. (2025) (3×), and SPlus Frans et al. (2025) (2×). We present a comparison of setups with these prior works in Section I.

**Benchmarking Optimizers** Schmidt et al. (2021) re-evaluated optimizers at that time and showed that (i) which optimizer is optimal is problem-specific, and (ii) rigorous hyperparameter tuning is required and unequal tuning can account for most of the claimed speedup. Unlike Schmidt et al. (2021), we evaluate the LLM pretraining task and include modern optimizers they did not test. However, we arrive at a similar conclusion: rigorous and fair hyperparameter tuning is still not the norm, but rather the exception in optimizer design research. Zhao et al. (2025) also examines how different optimizers perform on the pretraining tasks. However, the focus of Zhao et al. (2025) is on understanding loss structure, and the optimizers tested in the paper are shown to have slower convergence compared to AdamW, whereas the optimizers tested in this paper show small but significant improvements in convergence speed. The Algoperf competition Kasimbeg et al. (2025) evaluates different optimizers across different settings and arrives at a similar conclusion that (i) matrix-based optimizers and (ii) denoising methods such as Nesterov momentum lead to speedup over AdamW. Our works focus on the pretraining setting and investigates the effect of scaling dataset and model sizes.

**Comparison with concurrent work Semenov et al. (2025)** Semenov et al. (2025) investigates the same topic of benchmarking optimizers for pre-training. These two works agree on many high-level points, e.g., (i) non-zero weight decay and decaying to a small learning rate are essential for pretraining; (ii) variance-reduced AdamW variants such as Mars show non-trivial speedups over vanilla AdamW.

However, our results differ in the relative performance of matrix-level optimizers. Our paper shows that matrix-level optimizers such as Muon can achieve significant speedups over variance-reduced versions of AdamW, such as Mars, their study finds that AdEMaMix and Mars outperform Muon. Our initial investigation suggests this discrepancy largely stems from the differences in the batch sizes used in experiments. Their most extensively tuned experiments on 130M models use a batch size of only 0.1M and 0.02M tokens, whereas our experiments operate with tuned batch sizes that are not smaller than 0.4M tokens. This difference is likely a result of different hardware regimes. We leverage 128 TPU-v5lite chips (which are approximately equivalent to 12 H100 GPUs), where only larger batches (larger than 0.4M) can fully utilize the parallelism in the compute. On the other side, their experiments appear to be primarily conducted on 1-8 H100 GPUs, where smaller batches might be preferable. Since Mars and AdEMAMix both perform gradient averaging and variance reduction, these methods are advantageous in their noise-dominated small-batch regime, whereas in our larger-batch setting these benefits diminish and matrix-level optimizers become more competitive.

In their larger scale experiments, Semenov et al. (2025) increase both model size to 720M parameters and batch size to 1M tokens, respectively, which is closer to our setting. However, we differ in hyperparameter tuning methodology: (i) we conduct more extensive sweeps over learning rates on our 520M scales experiments and generally find higher values (4e-3 to 8e-3) compared to their choices (1e-3 to 2e-3), which are transferred from smaller scale and for some optimizers are not tuned; and (ii) for Muon, we separately tune the learning rates for the embedding layers and the matrix weights, which also improves its performance. These differences highlight the sensitivity of optimizer benchmarking to hardware and tuning strategies, underscoring the importance of carefully controlled experimental design when comparing optimizer performance.

## C  OPTIMIZER DEFINITIONS

In this section, we present the algorithm for each optimizer we evaluated.

Throughout this section, we use the following notation: $w_t$ for model parameters, $g_t$ for gradients at step $t$, $\eta$ for learning rate, $\lambda$ for weight decay, $\beta_1, \beta_2$ for moment decay rates, $\epsilon$ for numerical stability, $g_{\text{norm}}$ for gradient norm, and $m, v$ for first and second moments. All operations are element-wise unless specified.

**Algorithm 1** AdamW

**Hyperparameters:** $\beta_1$, $\beta_2$, $\epsilon$, $\eta$, $\lambda$, $g_{\mathrm{norm}}$
**State:** $m$, $v$
**Update Rule:**

$$\hat{g}_t = g_t \min\{1, \frac{g_{\mathrm{norm}}}{\|g_t\|_2}\}$$
$$m_t = \beta_1\, m_{t-1} + (1 - \beta_1)\, \hat{g}_t,$$
$$v_t = \beta_2\, v_{t-1} + (1 - \beta_2)\, \hat{g}_t^2,$$
$$\hat{m}_t = \frac{m_t}{1 - \beta_1^t}, \quad \hat{v}_t = \frac{v_t}{1 - \beta_2^t},$$
$$w_{t+1} = w_t - \eta\, \frac{\hat{m}_t}{\sqrt{\hat{v}_t} + \epsilon}\; -\; \eta\, \lambda\, w_t.$$

**Algorithm 2** Nesterov AdamW

**Hyperparameters:** $\beta_1$, $\beta_2$, $\epsilon$, $\eta$, $\lambda$, $g_{\mathrm{norm}}$
**State:** $m$, $v$
**Update Rule:**

$$\hat{g}_t = g_t \min\{1, \frac{g_{\mathrm{norm}}}{\|g_t\|_2}\}$$
$$m_t = \beta_1\, m_{t-1} + (1 - \beta_1)\, \hat{g}_t,$$
$$v_t = \beta_2\, v_{t-1} + (1 - \beta_2)\, \hat{g}_t^2,$$
$$\tilde{m}_t = \beta_1\, m_t + (1 - \beta_1)\, \hat{g}_t,$$
$$\hat{m}_t = \frac{\tilde{m}_t}{1 - \beta_1^t}, \quad \hat{v}_t = \frac{v_t}{1 - \beta_2^t},$$
$$w_{t+1} = w_t - \eta\, \frac{\hat{m}_t}{\sqrt{\hat{v}_t} + \epsilon}\; -\; \eta\, \lambda\, w_t.$$

**Algorithm 3** Lion

**Hyperparameters:** $\beta_1, \beta_2$, $\eta$, $\epsilon$, $\lambda$, $g_{\mathrm{norm}}$
**State:** $m$
**Update Rule:**

$$\hat{g}_t = g_t \min\{1, \frac{g_{\mathrm{norm}}}{\|g_t\|_2}\}$$
$$\hat{m}_t = \beta_1\, m_{t-1} + (1 - \beta_1)\, \hat{g}_t,$$
$$m_{t+1} = \beta_2\, m_{t-1} + (1 - \beta_2)\, \hat{g}_t,$$
$$w_{t+1} = w_t - \eta\, \mathrm{sign}(\hat{m}_t) - \eta\, \lambda\, w_t.$$

**Algorithm 4** Sophia-H

**Hyperparameters:** $\{\eta_t\}_{t=1}^{T}, \ \lambda, \ k, \ \beta_1, \ \beta_2, \ \gamma, \ \epsilon$

**State:** $m_0 = 0, \ h_{1-k} = 0, \ \theta_1$

**For** $t = 1, \ldots, T$:

$$g_t = \nabla_\theta L_t(\theta_t)$$

$$m_t = \beta_1 \, m_{t-1} + (1 - \beta_1) \, g_t$$

If $t \bmod k = 1$:

$$r \sim \{\pm 1\}^d, \quad v = g_t \cdot r, \quad u = \nabla_\theta v,$$

$$\hat{h} \ = \ r \odot u$$

$$h_t = \beta_2 \, h_{t-k} + (1 - \beta_2) \, \hat{h}$$

Else: $\quad h_t = h_{t-1}$

$$\theta_t \ = \ \theta_t \ - \ \eta_t \, \lambda \, \theta_t$$

$$\theta_{t+1} = \theta_t \ - \ \eta_t \, \mathrm{clip}\Big(\tfrac{m_t}{\max(\gamma \, h_t, \epsilon)}, \, 1\Big)$$

**Algorithm 5** MARS

**Hyperparameters:** $\beta_1, \ \beta_2, \ \gamma, \ \epsilon, \ \eta, \ \lambda, \ g_{\mathrm{norm}}$

**State:** $m, \ v, \ g_{t-1}$

**Update Rule:**

$$c_t = g_t + \gamma \, \frac{\beta_1}{1 - \beta_1} \, (g_t - g_{t-1}),$$

$$\hat{c}_t = c_t \min\{1, \frac{g_{\mathrm{norm}}}{\|c_t\|_2}\},$$

$$m_t = \beta_1 \, m_{t-1} + (1 - \beta_1) \, \hat{c}_t,$$

$$v_t = \beta_2 \, v_{t-1} + (1 - \beta_2) \, \hat{c}_t^2,$$

$$\hat{m}_t = \frac{m_t}{1 - \beta_1^t}, \quad \hat{v}_t = \frac{v_t}{1 - \beta_2^t},$$

$$w_{t+1} = w_t - \eta \, \frac{\hat{m}_t}{\sqrt{\hat{v}_t} + \epsilon} - \eta \, \lambda \, w_t.$$

**Algorithm 6** Adam-mini

**Hyperparameters:** $\beta_1$, $\beta_2$, $\epsilon$, $\eta$, $\lambda$, $g_{\text{norm}}$
**State:** $m$ (with the same shape as $w$), $v$ (one scalar for each block)
**Setup:**

1. Partition all parameters into `param_blocks`: please refer to Zhang et al. (2025b) for the exact partition scheme.

2. We will use $w_{t,b}$ and $m_{t,b}$ to denote the parameters in block $b$ at step $t$

**Update**

$$\hat{g}_t = g_t \min\{1, \frac{g_{\text{norm}}}{\|g_t\|_2}\}$$
$$m_t = \beta_1 \, m_{t-1} + (1 - \beta_1) \, \hat{g}_t,$$
$$\hat{m}_t = \frac{m_t}{1 - \beta_1^t},$$

**for each block** $b \in$ `param_blocks`

$$v_{t,b} = \beta_2 \, v_{t-1,b} + (1 - \beta_2)\text{mean}(g_{t,b}^2)$$
$$\hat{v}_{t,b} = \frac{v_{t,b}}{1 - \beta_2^t}.$$

$$w_{t,b} = w_{t-1,b} - \eta \, \hat{m}_{t,b}/(\sqrt{v_{t,b}} + \epsilon) - \eta \, \lambda \, w_{t-1,b}.$$

**Algorithm 7** Cautious

**Hyperparameters:** $\beta_1$, $\beta_2$, $\epsilon$, $\eta$, $\lambda$, $g_{\text{norm}}$
**State:** $m$, $v$
**Update Rule:**

$$\hat{g}_t = g_t \min\{1, \frac{g_{\text{norm}}}{\|g_t\|_2}\}$$
$$m_t = \beta_1 \, m_{t-1} + (1 - \beta_1) \, \hat{g}_t,$$
$$v_t = \beta_2 \, v_{t-1} + (1 - \beta_2) \, \hat{g}_t^2,$$
$$\hat{m}_t = \frac{m_t}{1 - \beta_1^t}, \quad \hat{v}_t = \frac{v_t}{1 - \beta_2^t},$$
$$u_t = \frac{\hat{m}_t}{\sqrt{\hat{v}_t} + \epsilon}, \quad s_t = \mathbb{I}(u_t \cdot \hat{g}_t > 0),$$
$$\hat{u}_t = \frac{u_t \cdot s_t}{\text{mean}(s_t)}, \quad w_{t+1} = w_t - \eta \, \hat{u}_t - \eta \, \lambda \, w_t.$$

**Algorithm 8** Muon

**Hyperparameters:** $\beta,\ \eta,\ \epsilon, \beta_1, \beta_2, \epsilon_{Adam}, \eta_{Adam}, \lambda, g_{\mathrm{norm}}$
**State:** $m$
**Update Rule For Weights in LM Head, Embedding, and LayerNorm:** Same as AdamW
**Update Rule For Matrices in Transformer Layer :**

$$\hat{g}_t = g_t \min\{1, \frac{g_{\mathrm{norm}}}{\|g_t\|_2}\}$$
$$m_t = \beta\, m_{t-1} + \hat{g}_t,$$
$$u = \beta\, m_t + \hat{g}_t,$$
$$u = \mathrm{NewtonSchulz}(u, \text{steps} = 5),$$
$$s = \sqrt{\max\left(1, \tfrac{\mathrm{rows}(w)}{\mathrm{cols}(w)}\right)},$$
$$u = s\, u,$$
$$w_{t+1} = w_t - \eta\, u - \eta\, \lambda\, w_t.$$

**Newton–Schulz Orthogonalization (Operating on Matrices):**

$$X \leftarrow \frac{X}{\|X\| + \epsilon}, \quad \text{transpose} \leftarrow (\mathrm{rows}(X) > \mathrm{cols}(X)),$$
$$\text{if transpose: } X \leftarrow X^\top,$$
$$\text{for } i = 1\ldots 5 :$$
$$\quad A = X\,X^\top,$$
$$\quad B = -4.7750\,A + 2.0315\,A^2,$$
$$\quad X \leftarrow 3.4445\,X + B\,X,$$
$$\text{if transpose: } X \leftarrow X^\top, \quad \text{return } X.$$

**Algorithm 9** Scion

**Hyperparameters:** $\beta,\ \eta,\ \epsilon, \beta_1, \beta_2, \eta_{\mathrm{SignGD}}, g_{\mathrm{norm}}$
**State:** $m$
**Update Rule For Matrices in LM Head and Embedding:**

$$\hat{g}_t = g_t \min\{1, \frac{g_{\mathrm{norm}}}{\|g_t\|_2}\}$$
$$m_t = \beta_1\, m_{t-1} + (1 - \beta_1)\, \hat{g}_t,$$
$$w_{t+1} = w_t - \eta_{\mathrm{SignGD}} \,\mathrm{sign}\big(\hat{m}_t\big).$$

**Update Rule For Matrices in Transformer Layer :** Same as Muon

---

**Algorithm 10** PSGD Kron

---

**Hyperparameters:** $\beta_1$, $\eta$, $\lambda$, $\epsilon$, $g_{\text{norm}}$, normalize_grads, partition_grads_into_blocks, merge_small_dims, block_size, target_merged_dim_size, $p_{\text{upd}}(t)$,
**Setup:**
If merge_small_dims is True, then try merging small dimensions into a single dimension with size target_merged_dim_size greedily.
If partition_grads_into_blocks is True, then partition all parameters into block_size $\times$ block_size blocks.
Define unfold$_i$ as the function that unfolds all dimensions except the $i$-th dimension into a single dimension and fold$_i$ as the inverse function.
**State (per block $\ell$):** Denote $w^{(\ell)}$ as the parameters in block $\ell$ and assume it has shape $d_1 \times d_2 \times \cdots \times d_n$.

$$\mu^{(\ell)}, \text{ with the same shape as } w^{(\ell)},$$

$$Q_i^{(\ell)} \ (i = 1, \ldots, n), \text{ a lower-triangular matrix with shape } d_i \times d_i$$

1.

$$\text{if } g_{\text{norm}} > 0 : \hat{g}_t = g_t \min\{1, \frac{g_{\text{norm}}}{\|g_t\|_2}\}, \text{else } \hat{g}_t = g_t$$

$$\text{if normalize\_grads} : \hat{g}_t = \frac{\hat{g}_t}{\|\hat{g}_t\|_2 + \epsilon}, \text{else } \hat{g}_t = \hat{g}_t$$

$$\mu_t \leftarrow \beta_1 \mu_{t-1} + (1 - \beta_1) \hat{g}_t, \quad \hat{\mu}_t \leftarrow \frac{\mu_t}{1 - \beta_1^t}.$$

2. **Balance check:** $\text{bal\_ctr} \leftarrow \text{bal\_ctr} + 1$. If $\text{bal\_ctr} \geq 100$, then $Q_i^{(\ell)} \leftarrow \text{balance}(Q_i^{(\ell)})$, $\text{bal\_ctr} \leftarrow 0$. **Balance function:** $\text{balance}(Q_i^{(\ell)})$ performs the following steps:

   (a) Compute the maximum absolute value for each row/column of $Q_i^{(\ell)}$: $\text{norms}_i = \max_{j,k} |Q_i^{(\ell)}[j, k]|$

   (b) Calculate the geometric mean: $\text{gmean}_i = (\Pi_{j=1}^n \text{norms}_i[j])^{\frac{1}{d_i}}$

   (c) Scale each element: $Q_i^{(\ell)} \leftarrow Q_i^{(\ell)} \cdot \frac{\text{gmean}_i}{\text{norms}_i}$

   (d) Return the balanced $Q_i^{(\ell)}$

3. **Preconditioner update:** with probability $p_{\text{upd}}(t)$,

   (a) **Random probe:** $V^{(\ell)} \leftarrow \text{tree\_random\_like}(g_t^{(\ell)})$.

   (b) **Dampen:** $\hat{\mu}_t^{(\ell)} \leftarrow \hat{\mu}_t^{(\ell)} + \epsilon \cdot \text{mean}(|\hat{\mu}_t^{(\ell)}|) \cdot V^{(\ell)}$

   (c) **Conjugate sketch ($B$):**

   $$X^{(0)} = V^{(\ell)}, \quad X^{(i)} = \text{fold}_i \left( \left( Q_i^{(\ell)} \right)^{-T} \text{unfold}_i(X^{(i-1)}) \right), \quad B = X^{(n)}.$$

   (d) **Pre-sketch ($A$):**

   $$Y^{(0)} = \hat{\mu}^{(\ell)}, \quad Y^{(i)} = \text{fold}_i \left( Q_i^{(\ell)} \text{unfold}_i(Y^{(i-1)}) \right), \quad A = Y^{(n)}.$$

   (e) For each $i$ in $1, \ldots, n$,

   $$M_i = \text{unfold}_i(A), \quad C_i = \text{unfold}_i(B),$$

   $$T_1 = M_i M_i^T, \quad T_2 = C_i C_i^T, \quad s = \max_{u,v} |(T_1 + T_2)_{u,v}|,$$

   $$Q_i^{(\ell)} \leftarrow Q_i^{(\ell)} - \alpha \frac{T_1 - T_2}{s} Q_i^{(\ell)}.$$

4. **(Precondition gradient)**

   $$G^{(0)} = \hat{\mu}^{(\ell)}, \quad G^{(i)} = \text{fold}_i \left( Q_i^{(\ell)} \text{unfold}_i(G^{(i-1)}) \right), \quad \tilde{g}_t^{(\ell)} = G^{(d)}.$$

27

5. **(Weigh-decay & update)**

   $$\tilde{g}_t^{(\ell)} \leftarrow \tilde{g}_t^{(\ell)} + \lambda w_{t-1}^{(\ell)}, \quad w_t^{(\ell)} \leftarrow w_{t-1}^{(\ell)} - \eta \tilde{g}_t^{(\ell)}.$$

---

**Algorithm 11** SOAP

---

**Hyperparameters:** $\beta_1$, $\beta_2$, $\mu$, $k$, $\epsilon$, block_size, $g_{\mathrm{norm}}$
**State:** $m$, $v$, $G_A$, $G_B$, $Q_A$, $Q_B$
Partition all parameters into block_size $\times$ block_size block
**Update Rule For Each Block:**

$$\hat{g}_t = g_t \min\{1, \frac{g_{\mathrm{norm}}}{\|g_t\|_2}\}$$

$$\hat{g}_t = Q_A\,\hat{g}_t\,Q_B,$$

$$m_t = \beta_1\,m_{t-1} + (1 - \beta_1)\,\hat{g}_t, \quad v_t = \beta_2\,v_{t-1} + (1 - \beta_2)\,\hat{g}_t^2,$$

$$\hat{m}_t = \frac{m_t}{1 - \beta_1^t}, \quad \hat{v}_t = \frac{v_t}{1 - \beta_2^t},$$

$$w_{t+1} = w_t - \eta_t\,Q_A^\top\Big(\frac{\hat{m}_t}{\sqrt{\hat{v}_t} + \epsilon}\Big)Q_B^\top,$$

$$G_A = \mu\,G_A + (1 - \mu)\,\hat{g}_t\,\hat{g}_t^\top + \epsilon I, \quad G_B = \mu\,G_B + (1 - \mu)\,\hat{g}_t^\top\,\hat{g}_t + \epsilon I,$$

$$\text{if } t \bmod k = 0: \quad Q_A = \mathrm{QR}(G_A\,Q_A), \quad Q_B = \mathrm{QR}(G_B\,Q_B).$$

---

| Batch Size | 524288 | 1048576 | 2097152 |
|---|---|---|---|
| adamw | 3.046 | 3.055 | |
| muon | 3.029 | | 3.056 |

## C.1 Equivalence between Our Parameterization and Constrained Scion

We note that our parameterization for Scion is equivalent to a constrained-Scion formulation, with the only difference being that we did not separately rescale the weight decay for the embedding and language-model head parameters. Concretely, one can establish a mapping between the two schemes as follows.

Let us consider a linear layer with weight $W$, input dimension $d_{\mathrm{in}}$, and output dimension $d_{\mathrm{out}}$. Suppose the constrained-Scion update is written in the form

$$\Delta W = -\eta'\,r\,O - \eta'\,W,$$

where $O$ is the orthogonalized momentum (after appropriate scaling), $\eta'$ is the "base" learning rate, and $r$ is a spectral radius parameter for constrained Scion.

In contrast, our version applies an update

$$\Delta W = -\eta\,O - \eta\,\lambda\,W,$$

where $\eta$ is the learning rate and $\lambda$ is the weight decay coefficient.

We can match the two updates by selecting

$$\eta = \eta'\,r, \qquad \lambda = \frac{1}{r}$$

in the constrained Scion setting (or $\lambda = 0$ in the unconstrained case). This ensures that

$$-\eta\,O - \eta\,\lambda\,W = -\,\eta'\,r\,O - \eta'\,W.$$

For layers involving the "sign" component (e.g. the signed momentum $S$), the constrained-Scion variant further scales by a factor $\frac{1}{d_{\mathrm{in}}}$. Thus the update takes the form

$$\Delta W = -\eta'\,r\,\frac{S}{d_{\mathrm{in}}} - \eta'\,W,$$

which corresponds to

$$\eta = \frac{\eta'\,r}{d_{\mathrm{in}}}, \qquad \lambda = \frac{d_{\mathrm{in}}}{r}$$

in the original formulation with weight decay.

| Optimizer | Scaling-Sensitive Hyperparameters |
|---|---|
| AdamW | `learning rate, warmup, weight decay, batch size` |
| Nesterov AdamW | `learning rate, warmup` |
| Lion | `learning rate, beta2` |
| Adam-mini | `learning rate, weight decay, warmup` |
| Cautious | `learning rate, batch size` |
| Mars | `learning rate, warmup, beta1` |
| Scion | `learning rate, beta1, # decay steps in WSD` Hu et al. (2024) |
| Muon | `learning rate` |
| Soap | `learning rate, warmup, block size` |
| Kron | `learning rate` |

Table 5: Scaling-sensitive hyperparameters identified in Phase I (Section 3.2). To reduce the memory required by Soap, we apply the parameter blocking as in Anil et al. (2021)

## D    OMITTED EXPERIMENTS

### D.1    ARCHITECTURE HYPERPARAMETER DETAILS

| Model | Params | Seq Len | Hidden Dim | Inter Dim | # Layers | # Heads |
|---|---|---|---|---|---|---|
| Llama-130M | 130M | 4096 | 512 | 2048 | 32 | 8 |
| Llama-300M | 300M | 4096 | 768 | 3072 | 32 | 12 |
| Llama-520M | 520M | 4096 | 1024 | 4096 | 32 | 16 |
| Llama-1.2B | 1.2B | 4096 | 1536 | 6144 | 32 | 24 |

Table 4: Detailed architecture hyperparameters for each model size we studied.

### D.2    SCALING SENSITIVE HYPERPARAMETERS

### D.3    SCALING LAW

Based on Sections E and F, we fitted our scaling law for the 1.2B run of Muon, NAdamW and AdamW, we round the fitted value to our grid of hyperparameter and used hyperparameters are shown in Section G. After training the models, we fitted a scaling law for Muon and AdamW of the following form:

$$L(N, D) = \alpha N^{-A} + \beta D^{-B} + \gamma \tag{1}$$

The fitted values are

- For AdamW, $\alpha = 21.4289, A = 0.1555, \beta = 276.4235, B = 0.2804, \gamma = 1.7324$, with a RMS error of $3 \times 10^{-3}$.

- For Muon $\alpha = 32.7458, A = 0.1864, \beta = 59.0221, B = 0.2074, \gamma = 1.8063$ , with a RMS error of $5 \times 10^{-3}$.

This scaling law predicts that when the parameter scale reaches 7B, Muon will actually result in a higher loss compared to AdamW in $1\times$ Chinchilla regime.

### D.4    SOPHIA EXPERIMENTS

We performed a Phase I experiment with Sophia, with detailed hyperparameter settings shown in Section E. We found that Sophia tends to underperform AdamW in smaller compute regimes and eventually slightly outperforms AdamW when either the model size or the data size increases.

We also note that the evaluaton in Sophia Liu et al. (2024a) largely follows the correct procedure of the comparisons in our paper — the peak learning rate was tuned to be optimal for the baseline.

However, as the data loader in the codebase used did not fully randomize the order of the data, the optimal peak learning rate in that code base for AdamW is significantly smaller than the optimal peak learning rate for a fully randomized data loader setting. It turns out that Sophia doesn't offer significant speedup over AdamW.

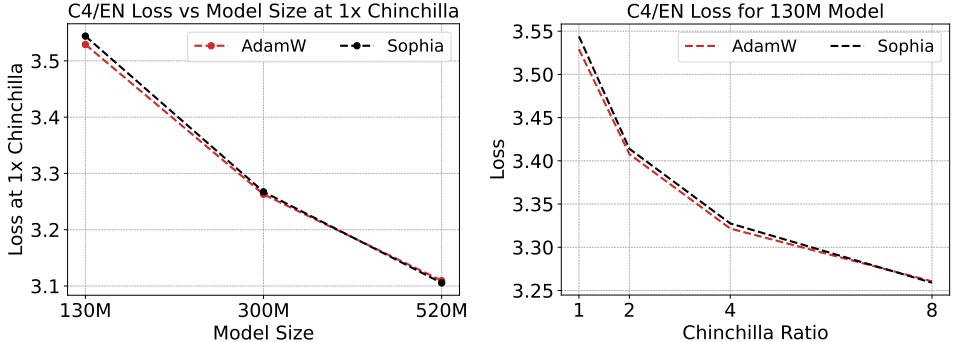

Figure 8: **Sophia Experiments.** Left: Loss curve of Sophia and AdamW in $1\times$ Chinchilla setting. Right: Loss curve of Sophia and AdamW for 130M model size.

### D.5 HIGH DATA-TO-MODEL RATIO

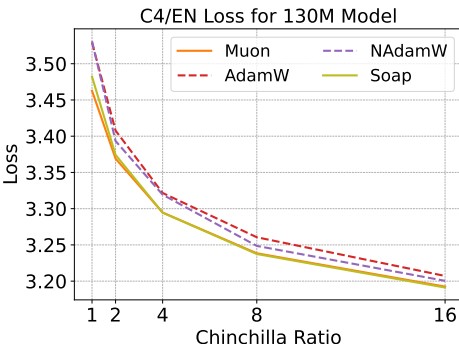

Figure 9: **More Case Studies.** Experiment with 130M $16\times$ Chinchilla setting, SOAP outperforms Muon in the overtraining setting.

### D.6 EVALUATION PERFORMANCE

Table 6: Benchmark performance of 1.2B models with different optimizers and Chinchilla scaling.

| Optimizer | LAMBADA | OpenBook | Wino | PIQA | BoolQ | WSC273 | Hella | ARC-C. | ARC-E | COPA | Avg |
|-----------|---------|----------|------|------|-------|--------|-------|--------|-------|------|-----|
| **8x Chinchilla (193B)** | | | | | | | | | | | |
| AdamW | 67.16 | 41.40 | 64.96 | 76.12 | 68.59 | 82.78 | 67.56 | 43.43 | 74.49 | 85.00 | 67.15 |
| NAdamW | 67.84 | 40.20 | 64.80 | 77.15 | 68.10 | 83.52 | 67.40 | 43.34 | 73.61 | 81.00 | 66.70 |
| Muon | 67.53 | 39.80 | 67.09 | 77.09 | 68.81 | 80.95 | 67.67 | 43.34 | 73.53 | 84.00 | 66.98 |

Table 7: Evaluation Performance for Mars, Model Size = 130m

| DATA SIZE / PERFORMANCE METRIC | 2B | 5B | 10B | 21B |
|---|---|---|---|---|
| LAMBADA OPENAI | 0.327 | 0.377 | 0.402 | 0.451 |
| OPENBOOKQA | 0.274 | 0.304 | 0.298 | 0.302 |
| WINOGRANDE | 0.521 | 0.512 | 0.526 | 0.522 |
| PIQA | 0.625 | 0.652 | 0.662 | 0.678 |
| BOOLQ | 0.616 | 0.504 | 0.569 | 0.557 |
| WSC273 | 0.557 | 0.560 | 0.579 | 0.590 |
| HELLASWAG 0SHOT | 0.308 | 0.343 | 0.365 | 0.399 |
| ARC CHALLENGE | 0.225 | 0.258 | 0.272 | 0.247 |
| ARC EASY | 0.462 | 0.517 | 0.532 | 0.544 |
| COPA | 0.650 | 0.690 | 0.660 | 0.710 |
| FINAL C4 LOSS | 3.537 | 3.396 | 3.323 | 3.247 |

Table 8: Evaluation Performance for Mars, Model Size = 300m

| DATA SIZE / PERFORMANCE METRIC | 6B | 12B | 24B | 48B |
|---|---|---|---|---|
| LAMBADA OPENAI | 0.450 | 0.493 | 0.514 | 0.538 |
| OPENBOOKQA | 0.300 | 0.312 | 0.328 | 0.328 |
| WINOGRANDE | 0.534 | 0.557 | 0.549 | 0.561 |
| PIQA | 0.680 | 0.686 | 0.695 | 0.720 |
| BOOLQ | 0.480 | 0.533 | 0.612 | 0.608 |
| WSC273 | 0.586 | 0.608 | 0.667 | 0.667 |
| HELLASWAG 0SHOT | 0.395 | 0.437 | 0.471 | 0.502 |
| ARC CHALLENGE | 0.253 | 0.269 | 0.298 | 0.307 |
| ARC EASY | 0.547 | 0.580 | 0.608 | 0.616 |
| COPA | 0.700 | 0.740 | 0.720 | 0.730 |
| FINAL C4 LOSS | 3.249 | 3.158 | 3.097 | 3.040 |

Table 9: Evaluation Performance for Mars, Model Size = 520m

| DATA SIZE / PERFORMANCE METRIC | 10B | 21B | 42B | 85B |
|---|---|---|---|---|
| LAMBADA OPENAI | 0.513 | 0.548 | 0.576 | 0.601 |
| OPENBOOKQA | 0.330 | 0.336 | 0.366 | 0.378 |
| WINOGRANDE | 0.548 | 0.559 | 0.586 | 0.601 |
| PIQA | 0.704 | 0.717 | 0.725 | 0.735 |
| BOOLQ | 0.559 | 0.598 | 0.625 | 0.631 |
| WSC273 | 0.601 | 0.696 | 0.736 | 0.780 |
| HELLASWAG 0SHOT | 0.462 | 0.513 | 0.546 | 0.579 |
| ARC CHALLENGE | 0.294 | 0.317 | 0.333 | 0.351 |
| ARC EASY | 0.611 | 0.648 | 0.665 | 0.676 |
| COPA | 0.730 | 0.760 | 0.750 | 0.750 |
| FINAL C4 LOSS | 3.101 | 3.015 | 2.955 | 2.906 |

Table 10: Evaluation Performance for Muon, Model Size = 130m

| DATA SIZE
PERFORMANCE METRIC | 2B | 5B | 10B | 21B |
|---|---|---|---|---|
| LAMBADA OPENAI | 0.358 | 0.396 | 0.431 | 0.456 |
| OPENBOOKQA | 0.286 | 0.288 | 0.318 | 0.300 |
| WINOGRANDE | 0.515 | 0.518 | 0.530 | 0.541 |
| PIQA | 0.640 | 0.660 | 0.670 | 0.678 |
| BOOLQ | 0.571 | 0.504 | 0.583 | 0.452 |
| WSC273 | 0.553 | 0.564 | 0.579 | 0.582 |
| HELLASWAG 0SHOT | 0.330 | 0.354 | 0.381 | 0.407 |
| ARC CHALLENGE | 0.231 | 0.244 | 0.269 | 0.276 |
| ARC EASY | 0.493 | 0.511 | 0.543 | 0.559 |
| COPA | 0.650 | 0.720 | 0.730 | 0.660 |
| FINAL C4 LOSS | 3.464 | 3.369 | 3.296 | 3.240 |

Table 11: Evaluation Performance for Muon, Model Size = 300m

| DATA SIZE
PERFORMANCE METRIC | 6B | 12B | 24B | 48B |
|---|---|---|---|---|
| LAMBADA OPENAI | 0.459 | 0.492 | 0.529 | 0.550 |
| OPENBOOKQA | 0.302 | 0.310 | 0.330 | 0.346 |
| WINOGRANDE | 0.504 | 0.534 | 0.548 | 0.570 |
| PIQA | 0.679 | 0.689 | 0.695 | 0.714 |
| BOOLQ | 0.482 | 0.555 | 0.599 | 0.570 |
| WSC273 | 0.615 | 0.641 | 0.659 | 0.656 |
| HELLASWAG 0SHOT | 0.406 | 0.447 | 0.481 | 0.509 |
| ARC CHALLENGE | 0.270 | 0.289 | 0.290 | 0.316 |
| ARC EASY | 0.564 | 0.591 | 0.620 | 0.638 |
| COPA | 0.670 | 0.730 | 0.700 | 0.700 |
| FINAL C4 LOSS | 3.224 | 3.143 | 3.079 | 3.032 |

Table 12: Evaluation Performance for Muon, Model Size = 520m

| DATA SIZE
PERFORMANCE METRIC | 10B | 21B | 42B | 85B |
|---|---|---|---|---|
| LAMBADA OPENAI | 0.520 | 0.559 | 0.597 | 0.611 |
| OPENBOOKQA | 0.328 | 0.340 | 0.342 | 0.368 |
| WINOGRANDE | 0.560 | 0.567 | 0.594 | 0.590 |
| PIQA | 0.713 | 0.718 | 0.730 | 0.736 |
| BOOLQ | 0.590 | 0.555 | 0.633 | 0.613 |
| WSC273 | 0.659 | 0.667 | 0.762 | 0.736 |
| HELLASWAG 0SHOT | 0.482 | 0.525 | 0.554 | 0.587 |
| ARC CHALLENGE | 0.300 | 0.318 | 0.351 | 0.348 |
| ARC EASY | 0.623 | 0.640 | 0.671 | 0.674 |
| COPA | 0.730 | 0.760 | 0.730 | 0.780 |
| FINAL C4 LOSS | 3.073 | 3.002 | 2.945 | 2.900 |

Table 13: Evaluation Performance for Lion, Model Size = 130m

| DATA SIZE
PERFORMANCE METRIC | 2B | 5B | 10B | 21B |
|---|---|---|---|---|
| LAMBADA OPENAI | 0.316 | 0.380 | 0.417 | 0.450 |
| OPENBOOKQA | 0.280 | 0.296 | 0.312 | 0.292 |
| WINOGRANDE | 0.503 | 0.518 | 0.519 | 0.521 |
| PIQA | 0.629 | 0.649 | 0.663 | 0.680 |
| BOOLQ | 0.603 | 0.507 | 0.419 | 0.489 |
| WSC273 | 0.564 | 0.538 | 0.590 | 0.623 |
| HELLASWAG 0SHOT | 0.304 | 0.343 | 0.366 | 0.402 |
| ARC CHALLENGE | 0.229 | 0.272 | 0.262 | 0.272 |
| ARC EASY | 0.464 | 0.500 | 0.536 | 0.539 |
| COPA | 0.700 | 0.670 | 0.730 | 0.730 |
| FINAL C4 LOSS | 3.552 | 3.409 | 3.331 | 3.252 |

Table 14: Evaluation Performance for Lion, Model Size = 300m

| DATA SIZE
PERFORMANCE METRIC | 6B | 12B | 24B | 48B |
|---|---|---|---|---|
| LAMBADA OPENAI | 0.428 | 0.471 | 0.509 | 0.535 |
| OPENBOOKQA | 0.306 | 0.320 | 0.326 | 0.346 |
| WINOGRANDE | 0.506 | 0.530 | 0.532 | 0.564 |
| PIQA | 0.676 | 0.686 | 0.699 | 0.713 |
| BOOLQ | 0.486 | 0.490 | 0.609 | 0.593 |
| WSC273 | 0.579 | 0.615 | 0.608 | 0.667 |
| HELLASWAG 0SHOT | 0.386 | 0.428 | 0.466 | 0.502 |
| ARC CHALLENGE | 0.267 | 0.296 | 0.294 | 0.322 |
| ARC EASY | 0.535 | 0.586 | 0.600 | 0.618 |
| COPA | 0.690 | 0.730 | 0.710 | 0.760 |
| FINAL C4 LOSS | 3.268 | 3.170 | 3.100 | 3.046 |

Table 15: Evaluation Performance for Lion, Model Size = 520m

| DATA SIZE
PERFORMANCE METRIC | 10B | 21B | 42B | 85B |
|---|---|---|---|---|
| LAMBADA OPENAI | 0.507 | 0.543 | 0.569 | 0.597 |
| OPENBOOKQA | 0.336 | 0.334 | 0.356 | 0.358 |
| WINOGRANDE | 0.560 | 0.548 | 0.581 | 0.590 |
| PIQA | 0.699 | 0.713 | 0.726 | 0.740 |
| BOOLQ | 0.600 | 0.546 | 0.612 | 0.631 |
| WSC273 | 0.608 | 0.667 | 0.685 | 0.700 |
| HELLASWAG 0SHOT | 0.463 | 0.509 | 0.545 | 0.571 |
| ARC CHALLENGE | 0.303 | 0.320 | 0.340 | 0.371 |
| ARC EASY | 0.603 | 0.638 | 0.664 | 0.673 |
| COPA | 0.710 | 0.750 | 0.740 | 0.770 |
| FINAL C4 LOSS | 3.108 | 3.029 | 2.965 | 2.915 |

Table 16: Evaluation Performance for NAdamW, Model Size = 130m

| DATA SIZE
PERFORMANCE METRIC | 2B | 5B | 10B | 21B |
|---|---|---|---|---|
| LAMBADA OPENAI | 0.325 | 0.383 | 0.413 | 0.446 |
| OPENBOOKQA | 0.284 | 0.286 | 0.308 | 0.322 |
| WINOGRANDE | 0.515 | 0.507 | 0.546 | 0.507 |
| PIQA | 0.633 | 0.652 | 0.667 | 0.682 |
| BOOLQ | 0.591 | 0.569 | 0.459 | 0.557 |
| WSC273 | 0.535 | 0.560 | 0.612 | 0.597 |
| HELLASWAG 0SHOT | 0.312 | 0.348 | 0.368 | 0.399 |
| ARC CHALLENGE | 0.239 | 0.245 | 0.242 | 0.270 |
| ARC EASY | 0.465 | 0.503 | 0.518 | 0.547 |
| COPA | 0.680 | 0.700 | 0.650 | 0.720 |
| FINAL C4 LOSS | 3.531 | 3.394 | 3.319 | 3.251 |

Table 17: Evaluation Performance for NAdamW, Model Size = 300m

| DATA SIZE
PERFORMANCE METRIC | 6B | 12B | 24B | 48B |
|---|---|---|---|---|
| LAMBADA OPENAI | 0.437 | 0.477 | 0.514 | 0.543 |
| OPENBOOKQA | 0.310 | 0.332 | 0.316 | 0.348 |
| WINOGRANDE | 0.520 | 0.536 | 0.549 | 0.564 |
| PIQA | 0.681 | 0.693 | 0.708 | 0.712 |
| BOOLQ | 0.539 | 0.564 | 0.520 | 0.612 |
| WSC273 | 0.582 | 0.637 | 0.645 | 0.700 |
| HELLASWAG 0SHOT | 0.397 | 0.434 | 0.475 | 0.505 |
| ARC CHALLENGE | 0.264 | 0.275 | 0.292 | 0.307 |
| ARC EASY | 0.553 | 0.589 | 0.605 | 0.632 |
| COPA | 0.710 | 0.730 | 0.700 | 0.750 |
| FINAL C4 LOSS | 3.248 | 3.160 | 3.090 | 3.039 |

Table 18: Evaluation Performance for NAdamW, Model Size = 520m

| DATA SIZE
PERFORMANCE METRIC | 10B | 21B | 42B | 85B |
|---|---|---|---|---|
| LAMBADA OPENAI | 0.500 | 0.551 | 0.577 | 0.599 |
| OPENBOOKQA | 0.332 | 0.350 | 0.358 | 0.342 |
| WINOGRANDE | 0.553 | 0.571 | 0.594 | 0.597 |
| PIQA | 0.695 | 0.719 | 0.731 | 0.738 |
| BOOLQ | 0.613 | 0.606 | 0.639 | 0.618 |
| WSC273 | 0.637 | 0.689 | 0.718 | 0.714 |
| HELLASWAG 0SHOT | 0.469 | 0.513 | 0.550 | 0.580 |
| ARC CHALLENGE | 0.294 | 0.322 | 0.342 | 0.354 |
| ARC EASY | 0.605 | 0.646 | 0.657 | 0.674 |
| COPA | 0.700 | 0.720 | 0.760 | 0.780 |
| FINAL C4 LOSS | 3.100 | 3.013 | 2.955 | 2.907 |

Table 19: Evaluation Performance for Kron, Model Size = 130m

| DATA SIZE
PERFORMANCE METRIC | 2B | 5B | 10B | 21B |
|---|---|---|---|---|
| LAMBADA OPENAI | 0.341 | 0.387 | 0.426 | 0.451 |
| OPENBOOKQA | 0.268 | 0.278 | 0.296 | 0.314 |
| WINOGRANDE | 0.520 | 0.522 | 0.515 | 0.543 |
| PIQA | 0.636 | 0.662 | 0.662 | 0.683 |
| BOOLQ | 0.533 | 0.530 | 0.575 | 0.557 |
| WSC273 | 0.564 | 0.564 | 0.553 | 0.604 |
| HELLASWAG 0SHOT | 0.323 | 0.347 | 0.374 | 0.397 |
| ARC CHALLENGE | 0.235 | 0.259 | 0.256 | 0.281 |
| ARC EASY | 0.487 | 0.519 | 0.543 | 0.572 |
| COPA | 0.700 | 0.690 | 0.710 | 0.710 |
| FINAL C4 LOSS | 3.492 | 3.389 | 3.307 | 3.239 |

Table 20: Evaluation Performance for Kron, Model Size = 300m

| DATA SIZE
PERFORMANCE METRIC | 6B | 12B | 24B | 48B |
|---|---|---|---|---|
| LAMBADA OPENAI | 0.429 | 0.485 | 0.516 | 0.548 |
| OPENBOOKQA | 0.310 | 0.314 | 0.314 | 0.326 |
| WINOGRANDE | 0.527 | 0.534 | 0.542 | 0.571 |
| PIQA | 0.675 | 0.690 | 0.706 | 0.709 |
| BOOLQ | 0.487 | 0.475 | 0.610 | 0.611 |
| WSC273 | 0.546 | 0.619 | 0.659 | 0.718 |
| HELLASWAG 0SHOT | 0.396 | 0.438 | 0.479 | 0.501 |
| ARC CHALLENGE | 0.262 | 0.285 | 0.298 | 0.323 |
| ARC EASY | 0.551 | 0.586 | 0.618 | 0.632 |
| COPA | 0.670 | 0.720 | 0.720 | 0.750 |
| FINAL C4 LOSS | 3.244 | 3.151 | 3.083 | 3.031 |

Table 21: Evaluation Performance for Kron, Model Size = 520m

| DATA SIZE
PERFORMANCE METRIC | 10B | 21B | 42B | 85B |
|---|---|---|---|---|
| LAMBADA OPENAI | 0.512 | 0.549 | 0.569 | 0.603 |
| OPENBOOKQA | 0.338 | 0.332 | 0.350 | 0.352 |
| WINOGRANDE | 0.536 | 0.567 | 0.575 | 0.601 |
| PIQA | 0.721 | 0.713 | 0.724 | 0.739 |
| BOOLQ | 0.529 | 0.560 | 0.537 | 0.559 |
| WSC273 | 0.648 | 0.663 | 0.696 | 0.762 |
| HELLASWAG 0SHOT | 0.474 | 0.516 | 0.556 | 0.582 |
| ARC CHALLENGE | 0.287 | 0.312 | 0.344 | 0.353 |
| ARC EASY | 0.616 | 0.648 | 0.662 | 0.681 |
| COPA | 0.710 | 0.720 | 0.790 | 0.750 |
| FINAL C4 LOSS | 3.084 | 3.009 | 2.946 | 2.900 |

Table 22: Evaluation Performance for Scion, Model Size = 130m

| DATA SIZE | 2B | 5B | 10B | 21B |
|---|---|---|---|---|
| PERFORMANCE METRIC | | | | |
| LAMBADA OPENAI | 0.355 | 0.382 | 0.427 | 0.446 |
| OPENBOOKQA | 0.282 | 0.300 | 0.288 | 0.314 |
| WINOGRANDE | 0.515 | 0.502 | 0.502 | 0.535 |
| PIQA | 0.641 | 0.655 | 0.675 | 0.684 |
| BOOLQ | 0.574 | 0.518 | 0.532 | 0.474 |
| WSC273 | 0.505 | 0.531 | 0.553 | 0.608 |
| HELLASWAG 0SHOT | 0.323 | 0.354 | 0.374 | 0.401 |
| ARC CHALLENGE | 0.247 | 0.241 | 0.276 | 0.264 |
| ARC EASY | 0.501 | 0.513 | 0.541 | 0.557 |
| COPA | 0.650 | 0.660 | 0.660 | 0.680 |
| FINAL C4 LOSS | 3.477 | 3.379 | 3.302 | 3.246 |

Table 23: Evaluation Performance for Scion, Model Size = 300m

| DATA SIZE | 6B | 12B | 24B | 48B |
|---|---|---|---|---|
| PERFORMANCE METRIC | | | | |
| LAMBADA OPENAI | 0.445 | 0.489 | 0.511 | 0.540 |
| OPENBOOKQA | 0.306 | 0.308 | 0.326 | 0.358 |
| WINOGRANDE | 0.531 | 0.550 | 0.555 | 0.566 |
| PIQA | 0.684 | 0.689 | 0.701 | 0.712 |
| BOOLQ | 0.576 | 0.610 | 0.574 | 0.585 |
| WSC273 | 0.597 | 0.612 | 0.685 | 0.696 |
| HELLASWAG 0SHOT | 0.406 | 0.442 | 0.478 | 0.506 |
| ARC CHALLENGE | 0.272 | 0.285 | 0.298 | 0.316 |
| ARC EASY | 0.564 | 0.592 | 0.625 | 0.633 |
| COPA | 0.680 | 0.730 | 0.700 | 0.740 |
| FINAL C4 LOSS | 3.232 | 3.152 | 3.086 | 3.039 |

Table 24: Evaluation Performance for Scion, Model Size = 520m

| DATA SIZE | 10B | 21B | 42B | 85B |
|---|---|---|---|---|
| PERFORMANCE METRIC | | | | |
| LAMBADA OPENAI | 0.521 | 0.552 | 0.575 | 0.601 |
| OPENBOOKQA | 0.338 | 0.362 | 0.366 | 0.374 |
| WINOGRANDE | 0.569 | 0.551 | 0.605 | 0.586 |
| PIQA | 0.704 | 0.717 | 0.736 | 0.743 |
| BOOLQ | 0.575 | 0.615 | 0.620 | 0.641 |
| WSC273 | 0.674 | 0.692 | 0.736 | 0.736 |
| HELLASWAG 0SHOT | 0.481 | 0.520 | 0.549 | 0.581 |
| ARC CHALLENGE | 0.289 | 0.314 | 0.346 | 0.354 |
| ARC EASY | 0.625 | 0.647 | 0.667 | 0.686 |
| COPA | 0.730 | 0.730 | 0.750 | 0.770 |
| FINAL C4 LOSS | 3.080 | 3.007 | 2.952 | 2.904 |

Table 25: Evaluation Performance for Cautious, Model Size = 130m

| DATA SIZE | 2B | 5B | 10B | 21B |
|---|---|---|---|---|
| PERFORMANCE METRIC | | | | |
| LAMBADA OPENAI | 0.314 | 0.385 | 0.420 | 0.458 |
| OPENBOOKQA | 0.298 | 0.280 | 0.308 | 0.318 |
| WINOGRANDE | 0.515 | 0.520 | 0.519 | 0.508 |
| PIQA | 0.638 | 0.652 | 0.664 | 0.669 |
| BOOLQ | 0.580 | 0.512 | 0.512 | 0.550 |
| WSC273 | 0.560 | 0.527 | 0.593 | 0.604 |
| HELLASWAG 0SHOT | 0.314 | 0.347 | 0.371 | 0.401 |
| ARC CHALLENGE | 0.239 | 0.244 | 0.261 | 0.270 |
| ARC EASY | 0.476 | 0.508 | 0.532 | 0.544 |
| COPA | 0.690 | 0.720 | 0.710 | 0.690 |
| FINAL C4 LOSS | 3.535 | 3.403 | 3.334 | 3.253 |

Table 26: Evaluation Performance for Cautious, Model Size = 300m

| DATA SIZE | 6B | 12B | 24B | 48B |
|---|---|---|---|---|
| PERFORMANCE METRIC | | | | |
| LAMBADA OPENAI | 0.437 | 0.479 | 0.510 | 0.544 |
| OPENBOOKQA | 0.316 | 0.314 | 0.326 | 0.328 |
| WINOGRANDE | 0.515 | 0.533 | 0.530 | 0.576 |
| PIQA | 0.677 | 0.699 | 0.707 | 0.711 |
| BOOLQ | 0.547 | 0.492 | 0.584 | 0.574 |
| WSC273 | 0.568 | 0.608 | 0.634 | 0.667 |
| HELLASWAG 0SHOT | 0.397 | 0.432 | 0.477 | 0.505 |
| ARC CHALLENGE | 0.267 | 0.280 | 0.292 | 0.308 |
| ARC EASY | 0.541 | 0.569 | 0.606 | 0.618 |
| COPA | 0.670 | 0.730 | 0.760 | 0.720 |
| FINAL C4 LOSS | 3.260 | 3.165 | 3.094 | 3.043 |

Table 27: Evaluation Performance for Cautious, Model Size = 520m

| DATA SIZE | 10B | 21B | 42B | 85B |
|---|---|---|---|---|
| PERFORMANCE METRIC | | | | |
| LAMBADA OPENAI | 0.509 | 0.549 | 0.578 | 0.604 |
| OPENBOOKQA | 0.318 | 0.338 | 0.342 | 0.364 |
| WINOGRANDE | 0.548 | 0.575 | 0.571 | 0.605 |
| PIQA | 0.705 | 0.715 | 0.733 | 0.739 |
| BOOLQ | 0.601 | 0.611 | 0.582 | 0.606 |
| WSC273 | 0.619 | 0.718 | 0.659 | 0.736 |
| HELLASWAG 0SHOT | 0.467 | 0.514 | 0.550 | 0.583 |
| ARC CHALLENGE | 0.305 | 0.315 | 0.344 | 0.358 |
| ARC EASY | 0.609 | 0.649 | 0.676 | 0.689 |
| COPA | 0.730 | 0.760 | 0.770 | 0.770 |
| FINAL C4 LOSS | 3.100 | 3.017 | 2.956 | 2.910 |

Table 28: Evaluation Performance for SOAP, Model Size = 130m

| DATA SIZE | 2B | 5B | 10B | 21B |
| --- | --- | --- | --- | --- |
| PERFORMANCE METRIC | | | | |
| LAMBADA OPENAI | 0.344 | 0.404 | 0.434 | 0.458 |
| OPENBOOKQA | 0.280 | 0.270 | 0.300 | 0.318 |
| WINOGRANDE | 0.511 | 0.510 | 0.523 | 0.517 |
| PIQA | 0.643 | 0.652 | 0.666 | 0.676 |
| BOOLQ | 0.569 | 0.491 | 0.519 | 0.559 |
| WSC273 | 0.513 | 0.568 | 0.597 | 0.612 |
| HELLASWAG 0SHOT | 0.321 | 0.351 | 0.376 | 0.401 |
| ARC CHALLENGE | 0.247 | 0.255 | 0.270 | 0.270 |
| ARC EASY | 0.475 | 0.519 | 0.537 | 0.571 |
| COPA | 0.690 | 0.710 | 0.720 | 0.710 |
| FINAL C4 LOSS | 3.487 | 3.376 | 3.295 | 3.240 |

Table 29: Evaluation Performance for SOAP, Model Size = 300m

| DATA SIZE | 6B | 12B | 24B | 48B |
| --- | --- | --- | --- | --- |
| PERFORMANCE METRIC | | | | |
| LAMBADA OPENAI | 0.001 | 0.492 | 0.518 | 0.541 |
| OPENBOOKQA | 0.256 | 0.324 | 0.324 | 0.334 |
| WINOGRANDE | 0.506 | 0.534 | 0.546 | 0.555 |
| PIQA | 0.546 | 0.689 | 0.706 | 0.711 |
| BOOLQ | 0.402 | 0.593 | 0.565 | 0.593 |
| WSC273 | 0.498 | 0.619 | 0.703 | 0.670 |
| HELLASWAG 0SHOT | 0.257 | 0.443 | 0.478 | 0.506 |
| ARC CHALLENGE | 0.219 | 0.281 | 0.305 | 0.324 |
| ARC EASY | 0.309 | 0.590 | 0.612 | 0.632 |
| COPA | 0.590 | 0.740 | 0.720 | 0.700 |
| FINAL C4 LOSS | 5.437 | 3.147 | 3.082 | 3.030 |

Table 30: Evaluation Performance for SOAP, Model Size = 520m

| DATA SIZE | 10B | 21B | 42B | 85B |
| --- | --- | --- | --- | --- |
| PERFORMANCE METRIC | | | | |
| LAMBADA OPENAI | 0.513 | 0.555 | 0.582 | 0.606 |
| OPENBOOKQA | 0.308 | 0.312 | 0.368 | 0.350 |
| WINOGRANDE | 0.548 | 0.562 | 0.594 | 0.583 |
| PIQA | 0.695 | 0.714 | 0.729 | 0.739 |
| BOOLQ | 0.563 | 0.623 | 0.596 | 0.643 |
| WSC273 | 0.648 | 0.703 | 0.722 | 0.755 |
| HELLASWAG 0SHOT | 0.478 | 0.523 | 0.553 | 0.586 |
| ARC CHALLENGE | 0.292 | 0.345 | 0.334 | 0.356 |
| ARC EASY | 0.610 | 0.656 | 0.655 | 0.678 |
| COPA | 0.740 | 0.760 | 0.740 | 0.770 |
| FINAL C4 LOSS | 3.079 | 3.004 | 2.957 | 2.899 |

Table 31: Evaluation Performance for AdamW, Model Size = 130m

| DATA SIZE
PERFORMANCE METRIC | 2B | 5B | 10B | 21B |
|---|---|---|---|---|
| LAMBADA OPENAI | 0.322 | 0.380 | 0.410 | 0.442 |
| OPENBOOKQA | 0.306 | 0.290 | 0.298 | 0.292 |
| WINOGRANDE | 0.506 | 0.519 | 0.515 | 0.530 |
| PIQA | 0.632 | 0.663 | 0.665 | 0.674 |
| BOOLQ | 0.568 | 0.415 | 0.553 | 0.576 |
| WSC273 | 0.582 | 0.553 | 0.564 | 0.623 |
| HELLASWAG 0SHOT | 0.312 | 0.340 | 0.365 | 0.391 |
| ARC CHALLENGE | 0.229 | 0.255 | 0.262 | 0.270 |
| ARC EASY | 0.463 | 0.497 | 0.523 | 0.544 |
| COPA | 0.660 | 0.650 | 0.720 | 0.730 |
| FINAL C4 LOSS | 3.529 | 3.409 | 3.322 | 3.262 |

Table 32: Evaluation Performance for AdamW, Model Size = 300m

| DATA SIZE
PERFORMANCE METRIC | 6B | 12B | 24B | 48B |
|---|---|---|---|---|
| LAMBADA OPENAI | 0.435 | 0.484 | 0.500 | 0.537 |
| OPENBOOKQA | 0.300 | 0.312 | 0.328 | 0.318 |
| WINOGRANDE | 0.519 | 0.530 | 0.542 | 0.552 |
| PIQA | 0.673 | 0.691 | 0.697 | 0.713 |
| BOOLQ | 0.454 | 0.496 | 0.508 | 0.602 |
| WSC273 | 0.586 | 0.626 | 0.667 | 0.692 |
| HELLASWAG 0SHOT | 0.386 | 0.432 | 0.471 | 0.501 |
| ARC CHALLENGE | 0.269 | 0.295 | 0.304 | 0.310 |
| ARC EASY | 0.535 | 0.581 | 0.612 | 0.616 |
| COPA | 0.700 | 0.700 | 0.690 | 0.710 |
| FINAL C4 LOSS | 3.264 | 3.166 | 3.094 | 3.043 |

Table 33: Evaluation Performance for AdamW, Model Size = 520m

| DATA SIZE
PERFORMANCE METRIC | 10B | 21B | 42B | 85B |
|---|---|---|---|---|
| LAMBADA OPENAI | 0.500 | 0.540 | 0.570 | 0.591 |
| OPENBOOKQA | 0.320 | 0.336 | 0.342 | 0.356 |
| WINOGRANDE | 0.539 | 0.586 | 0.583 | 0.601 |
| PIQA | 0.705 | 0.719 | 0.732 | 0.739 |
| BOOLQ | 0.555 | 0.615 | 0.596 | 0.609 |
| WSC273 | 0.604 | 0.656 | 0.703 | 0.729 |
| HELLASWAG 0SHOT | 0.456 | 0.507 | 0.543 | 0.578 |
| ARC CHALLENGE | 0.299 | 0.311 | 0.323 | 0.349 |
| ARC EASY | 0.613 | 0.639 | 0.669 | 0.688 |
| COPA | 0.740 | 0.680 | 0.710 | 0.780 |
| FINAL C4 LOSS | 3.110 | 3.023 | 2.958 | 2.913 |

Table 34: Evaluation Performance for Adam-Mini, Model Size = 130m

| DATA SIZE
PERFORMANCE METRIC | 2B | 5B | 10B | 21B |
|---|---|---|---|---|
| LAMBADA OPENAI | 0.317 | 0.369 | 0.412 | 0.444 |
| OPENBOOKQA | 0.286 | 0.282 | 0.282 | 0.302 |
| WINOGRANDE | 0.525 | 0.500 | 0.515 | 0.528 |
| PIQA | 0.640 | 0.655 | 0.662 | 0.670 |
| BOOLQ | 0.506 | 0.496 | 0.445 | 0.583 |
| WSC273 | 0.571 | 0.557 | 0.546 | 0.634 |
| HELLASWAG 0SHOT | 0.310 | 0.337 | 0.364 | 0.390 |
| ARC CHALLENGE | 0.241 | 0.247 | 0.255 | 0.264 |
| ARC EASY | 0.465 | 0.495 | 0.525 | 0.555 |
| COPA | 0.690 | 0.710 | 0.730 | 0.700 |
| FINAL C4 LOSS | 3.542 | 3.416 | 3.328 | 3.266 |

Table 35: Evaluation Performance for Adam-Mini, Model Size = 300m

| DATA SIZE
PERFORMANCE METRIC | 6B | 12B | 24B | 48B |
|---|---|---|---|---|
| LAMBADA OPENAI | 0.429 | 0.468 | 0.509 | 0.535 |
| OPENBOOKQA | 0.300 | 0.318 | 0.318 | 0.350 |
| WINOGRANDE | 0.502 | 0.528 | 0.545 | 0.586 |
| PIQA | 0.688 | 0.693 | 0.683 | 0.701 |
| BOOLQ | 0.480 | 0.546 | 0.609 | 0.573 |
| WSC273 | 0.586 | 0.546 | 0.593 | 0.689 |
| HELLASWAG 0SHOT | 0.381 | 0.424 | 0.464 | 0.492 |
| ARC CHALLENGE | 0.270 | 0.292 | 0.294 | 0.300 |
| ARC EASY | 0.545 | 0.579 | 0.607 | 0.625 |
| COPA | 0.700 | 0.720 | 0.740 | 0.730 |
| FINAL C4 LOSS | 3.272 | 3.178 | 3.103 | 3.049 |

Table 36: Evaluation Performance for Adam-Mini, Model Size = 520m

| DATA SIZE
PERFORMANCE METRIC | 10B | 21B | 42B | 85B |
|---|---|---|---|---|
| LAMBADA OPENAI | 0.500 | 0.538 | 0.576 | 0.604 |
| OPENBOOKQA | 0.306 | 0.326 | 0.330 | 0.368 |
| WINOGRANDE | 0.534 | 0.561 | 0.571 | 0.594 |
| PIQA | 0.707 | 0.721 | 0.723 | 0.733 |
| BOOLQ | 0.543 | 0.626 | 0.619 | 0.536 |
| WSC273 | 0.619 | 0.700 | 0.733 | 0.744 |
| HELLASWAG 0SHOT | 0.459 | 0.505 | 0.541 | 0.576 |
| ARC CHALLENGE | 0.288 | 0.327 | 0.332 | 0.352 |
| ARC EASY | 0.613 | 0.638 | 0.654 | 0.671 |
| COPA | 0.710 | 0.720 | 0.760 | 0.760 |
| FINAL C4 LOSS | 3.112 | 3.027 | 2.966 | 2.912 |

# E    HYPERPARAMETER ABLATION IN PHASE I

We reported the results for the optimizers we swept in Phase I. The result is formulated as follows: the first row shows the approximately best configuration found and the following rows show the results for the 1-dimensional ablations centered around the found configuration. The loss presented here is the final loss on the C4/EN validation set.

## E.1 SWEEPING RESULTS FOR ADAMW

Table 37: Hyperparameter ablation for AdamW on 130m on 1x Chinchilla Data

| $\beta_1$ | $\beta_2$ | $\epsilon$ | $\eta$ | $g_{\mathrm{norm}}$ | $\eta_{min}$ | BSZ | warmup | $\lambda$ | Loss | Link |
|---|---|---|---|---|---|---|---|---|---|---|
| 0.9 | 0.98 | 1e-20 | 0.008 | 1 | 0 | 128 | 2000 | 0.1 | 3.529 | 0 |
| 0.95 | – | – | – | – | – | – | – | – | 3.539 | 1 |
| 0.98 | – | – | – | – | – | – | – | – | 3.882 | 2 |
| – | 0.9 | – | – | – | – | – | – | – | 3.545 | 3 |
| – | 0.95 | – | – | – | – | – | – | – | 3.535 | 4 |
| – | – | 1e-25 | – | – | – | – | – | – | 3.529 | 5 |
| – | – | 1e-15 | – | – | – | – | – | – | 3.531 | 6 |
| – | – | 1e-10 | – | – | – | – | – | – | 3.531 | 7 |
| – | – | – | 0.004 | – | – | – | – | – | 3.550 | 8 |
| – | – | – | 0.016 | – | – | – | – | – | 3.538 | 9 |
| – | – | – | 0.032 | – | – | – | – | – | 7.781 | 10 |
| – | – | – | – | 0 | – | – | – | – | 3.534 | 11 |
| – | – | – | – | 2.0 | – | – | – | – | 3.534 | 12 |
| – | – | – | – | – | – | 256 | – | – | 3.611 | 13 |
| – | – | – | – | – | – | – | 500 | – | 7.452 | 14 |
| – | – | – | – | – | – | – | 1000 | – | 3.532 | 15 |
| – | – | – | – | – | – | – | 4000 | – | 3.575 | 16 |
| – | – | – | – | – | – | – | – | 0 | 3.545 | 17 |
| – | – | – | – | – | – | – | – | 0.2 | 3.536 | 18 |

Table 38: Hyperparameter ablation for AdamW on 130m on 2x Chinchilla Data

| $\beta_1$ | $\beta_2$ | $\epsilon$ | $\eta$ | $g_{\mathrm{norm}}$ | $\eta_{min}$ | BSZ | warmup | $\lambda$ | Loss | Link |
|---|---|---|---|---|---|---|---|---|---|---|
| 0.9 | 0.98 | 1e-20 | 0.008 | 1 | 0 | 128 | 2000 | 0.1 | 3.409 | 0 |
| 0.95 | – | – | – | – | – | – | – | – | 3.417 | 1 |
| 0.98 | – | – | – | – | – | – | – | – | 7.557 | 2 |
| – | 0.9 | – | – | – | – | – | – | – | 3.423 | 3 |
| – | 0.95 | – | – | – | – | – | – | – | 3.413 | 4 |
| – | – | 1e-25 | – | – | – | – | – | – | 3.409 | 5 |
| – | – | 1e-15 | – | – | – | – | – | – | 3.409 | 6 |
| – | – | 1e-10 | – | – | – | – | – | – | 3.410 | 7 |
| – | – | – | 0.004 | – | – | – | – | – | 3.420 | 8 |
| – | – | – | 0.016 | – | – | – | – | – | 3.419 | 9 |
| – | – | – | 0.032 | – | – | – | – | – | 7.840 | 10 |
| – | – | – | – | 0 | – | – | – | – | 3.410 | 11 |
| – | – | – | – | 2.0 | – | – | – | – | 3.408 | 12 |
| – | – | – | – | – | – | 256 | – | – | 3.437 | 13 |
| – | – | – | – | – | – | 512 | – | – | 3.527 | 14 |
| – | – | – | – | – | – | – | 500 | – | 7.277 | 15 |
| – | – | – | – | – | – | – | 1000 | – | 3.413 | 16 |
| – | – | – | – | – | – | – | 4000 | – | 3.415 | 17 |
| – | – | – | – | – | – | – | – | 0 | 3.436 | 18 |
| – | – | – | – | – | – | – | – | 0.2 | 3.415 | 19 |

Table 39: Hyperparameter ablation for AdamW on 130m on 4x Chinchilla Data

| $\beta_1$ | $\beta_2$ | $\epsilon$ | $\eta$ | $g_{\mathrm{norm}}$ | $\eta_{min}$ | BSZ | warmup | $\lambda$ | Loss | Link |
|---|---|---|---|---|---|---|---|---|---|---|
| 0.9 | 0.98 | 1e-20 | 0.008 | 1 | 0 | 128 | 2000 | 0.1 | 3.322 | 0 |
| 0.95 | – | – | – | – | – | – | – | – | 3.330 | 1 |
| 0.98 | – | – | – | – | – | – | – | – | 3.416 | 2 |
| – | 0.9 | – | – | – | – | – | – | – | 3.338 | 3 |
| – | 0.95 | – | – | – | – | – | – | – | 3.329 | 4 |
| – | – | 1e-25 | – | – | – | – | – | – | 3.322 | 5 |
| – | – | 1e-15 | – | – | – | – | – | – | 3.323 | 6 |
| – | – | 1e-10 | – | – | – | – | – | – | 3.324 | 7 |
| – | – | – | 0.004 | – | – | – | – | – | 3.329 | 8 |
| – | – | – | 0.016 | – | – | – | – | – | 3.337 | 9 |
| – | – | – | 0.032 | – | – | – | – | – | 7.562 | 10 |
| – | – | – | – | 0 | – | – | – | – | 3.327 | 11 |
| – | – | – | – | 2.0 | – | – | – | – | 3.324 | 12 |
| – | – | – | – | – | – | 256 | – | – | 3.331 | 13 |
| – | – | – | – | – | – | 512 | – | – | 3.373 | 14 |
| – | – | – | – | – | – | 1024 | – | – | 3.480 | 15 |
| – | – | – | – | – | – | – | 500 | – | 7.262 | 16 |
| – | – | – | – | – | – | – | 1000 | – | 3.327 | 17 |
| – | – | – | – | – | – | – | 4000 | – | 3.325 | 18 |
| – | – | – | – | – | – | – | – | 0 | 3.359 | 19 |
| – | – | – | – | – | – | – | – | 0.2 | 3.335 | 20 |

Table 40: Hyperparameter ablation for AdamW on 130m on 8x Chinchilla Data

| $\beta_1$ | $\beta_2$ | $\epsilon$ | $\eta$ | $g_{\mathrm{norm}}$ | $\eta_{min}$ | BSZ | warmup | $\lambda$ | Loss | Link |
|---|---|---|---|---|---|---|---|---|---|---|
| 0.9 | 0.98 | 1e-20 | 0.008 | 1 | 0 | 256 | 1000 | 0.1 | 3.262 | 0 |
| 0.95 | – | – | – | – | – | – | – | – | 3.273 | 1 |
| 0.98 | – | – | – | – | – | – | – | – | 3.430 | 2 |
| – | 0.9 | – | – | – | – | – | – | – | 3.272 | 3 |
| – | 0.95 | – | – | – | – | – | – | – | 3.266 | 4 |
| – | – | 1e-25 | – | – | – | – | – | – | 3.262 | 5 |
| – | – | 1e-15 | – | – | – | – | – | – | 3.263 | 6 |
| – | – | 1e-10 | – | – | – | – | – | – | 3.261 | 7 |
| – | – | – | 0.004 | – | – | – | – | – | 3.270 | 8 |
| – | – | – | 0.016 | – | – | – | – | – | 7.435 | 9 |
| – | – | – | 0.032 | – | – | – | – | – | 7.658 | 10 |
| – | – | – | – | 0 | – | – | – | – | 3.263 | 11 |
| – | – | – | – | 2.0 | – | – | – | – | 3.264 | 12 |
| – | – | – | – | – | – | 128 | – | – | 3.264 | 13 |
| – | – | – | – | – | – | 512 | – | – | 3.286 | 14 |
| – | – | – | – | – | – | 1024 | – | – | 3.328 | 15 |
| – | – | – | – | – | – | – | 500 | – | 3.278 | 16 |
| – | – | – | – | – | – | – | 2000 | – | 3.263 | 17 |
| – | – | – | – | – | – | – | 4000 | – | 3.262 | 18 |
| – | – | – | – | – | – | – | – | 0 | 3.310 | 19 |
| – | – | – | – | – | – | – | – | 0.2 | 3.269 | 20 |

Table 41: Hyperparameter ablation for AdamW on 300m on 1x Chinchilla Data

| $\beta_1$ | $\beta_2$ | $\epsilon$ | $\eta$ | $g_{\mathrm{norm}}$ | $\eta_{min}$ | BSZ | warmup | $\lambda$ | Loss | Link |
|---|---|---|---|---|---|---|---|---|---|---|
| 0.9 | 0.98 | 1e-10 | 0.008 | 1 | 0 | 128 | 2000 | 0.1 | 3.264 | 0 |
| 0.95 | – | – | – | – | – | – | – | – | 3.271 | 1 |
| 0.98 | – | – | – | – | – | – | – | – | 7.351 | 2 |
| – | 0.9 | – | – | – | – | – | – | – | 3.280 | 3 |
| – | 0.95 | – | – | – | – | – | – | – | 3.269 | 4 |
| – | – | 1e-25 | – | – | – | – | – | – | 3.265 | 5 |
| – | – | 1e-20 | – | – | – | – | – | – | 3.265 | 6 |
| – | – | 1e-15 | – | – | – | – | – | – | 3.263 | 7 |
| – | – | – | 0.004 | – | – | – | – | – | 3.272 | 8 |
| – | – | – | 0.016 | – | – | – | – | – | 7.760 | 9 |
| – | – | – | 0.032 | – | – | – | – | – | 7.784 | 10 |
| – | – | – | – | 0 | – | – | – | – | 3.263 | 11 |
| – | – | – | – | 2.0 | – | – | – | – | 3.263 | 12 |
| – | – | – | – | – | – | 256 | – | – | 3.282 | 13 |
| – | – | – | – | – | – | 512 | – | – | 3.367 | 14 |
| – | – | – | – | – | – | – | 500 | – | 7.704 | 15 |
| – | – | – | – | – | – | – | 1000 | – | 7.759 | 16 |
| – | – | – | – | – | – | – | 4000 | – | 3.270 | 17 |
| – | – | – | – | – | – | – | – | 0 | 3.303 | 18 |
| – | – | – | – | – | – | – | – | 0.2 | 3.275 | 19 |

Table 42: Hyperparameter ablation for AdamW on 520m on 1x Chinchilla Data

| $\beta_1$ | $\beta_2$ | $\epsilon$ | $\eta$ | $g_{\mathrm{norm}}$ | $\eta_{min}$ | BSZ | warmup | $\lambda$ | Loss | Link |
|---|---|---|---|---|---|---|---|---|---|---|
| 0.9 | 0.98 | 1e-10 | 0.004 | 1 | 0 | 256 | 1000 | 0.2 | 3.110 | 0 |
| 0.95 | – | – | – | – | – | – | – | – | 3.112 | 1 |
| 0.98 | – | – | – | – | – | – | – | – | 3.229 | 2 |
| – | 0.9 | – | – | – | – | – | – | – | 3.116 | 3 |
| – | 0.95 | – | – | – | – | – | – | – | 3.111 | 4 |
| – | – | 1e-25 | – | – | – | – | – | – | 3.116 | 5 |
| – | – | 1e-20 | – | – | – | – | – | – | 3.116 | 6 |
| – | – | 1e-15 | – | – | – | – | – | – | 3.115 | 7 |
| – | – | – | 0.008 | – | – | – | – | – | 7.837 | 8 |
| – | – | – | 0.016 | – | – | – | – | – | 7.756 | 9 |
| – | – | – | 0.032 | – | – | – | – | – | 7.680 | 10 |
| – | – | – | – | 0 | – | – | – | – | 3.114 | 11 |
| – | – | – | – | 2.0 | – | – | – | – | 3.118 | 12 |
| – | – | – | – | – | – | 128 | – | – | 7.630 | 13 |
| – | – | – | – | – | – | 512 | – | – | 3.169 | 14 |
| – | – | – | – | – | – | 1024 | – | – | 3.302 | 15 |
| – | – | – | – | – | – | – | 500 | – | 3.165 | 16 |
| – | – | – | – | – | – | – | 2000 | – | 3.113 | 17 |
| – | – | – | – | – | – | – | 4000 | – | 3.126 | 18 |
| – | – | – | – | – | – | – | – | 0 | 7.270 | 19 |
| – | – | – | – | – | – | – | – | 0.1 | 3.135 | 20 |

## E.2 SWEEPING RESULTS FOR CAUTIOUS

Table 43: Hyperparameter ablation for Cautious on 130m on 1x Chinchilla Data

| $\beta_1$ | $\beta_2$ | $\epsilon$ | $\eta$ | $g_{\mathrm{norm}}$ | $\eta_{min}$ | BSZ | warmup | $\lambda$ | Loss | Link |
|---|---|---|---|---|---|---|---|---|---|---|
| 0.95 | 0.98 | 1e-15 | 0.008 | 1 | 0 | 128 | 2000 | 0.1 | 3.535 | 0 |
| 0.8 | – | – | – | – | – | – | – | – | 6.698 | 1 |
| 0.9 | – | – | – | – | – | – | – | – | 3.549 | 2 |
| 0.98 | – | – | – | – | – | – | – | – | 3.534 | 3 |
| – | 0.9 | – | – | – | – | – | – | – | 3.551 | 4 |
| – | 0.95 | – | – | – | – | – | – | – | 3.543 | 5 |
| – | – | 1e-25 | – | – | – | – | – | – | 3.537 | 6 |
| – | – | 1e-20 | – | – | – | – | – | – | 3.537 | 7 |
| – | – | 1e-10 | – | – | – | – | – | – | 3.536 | 8 |
| – | – | – | 0.016 | – | – | – | – | – | 3.539 | 9 |
| – | – | – | 0.032 | – | – | – | – | – | 7.802 | 10 |
| – | – | – | – | 0 | – | – | – | – | 3.534 | 11 |
| – | – | – | – | 2.0 | – | – | – | – | 3.532 | 12 |
| – | – | – | – | – | – | 256 | – | – | 3.610 | 13 |
| – | – | – | – | – | – | – | 500 | – | 3.572 | 14 |
| – | – | – | – | – | – | – | 1000 | – | 3.535 | 15 |
| – | – | – | – | – | – | – | 4000 | – | 3.582 | 16 |
| – | – | – | – | – | – | – | – | 0 | 3.552 | 17 |
| – | – | – | – | – | – | – | – | 0.2 | 3.537 | 18 |

Table 44: Hyperparameter ablation for Cautious on 130m on 2x Chinchilla Data

| $\beta_1$ | $\beta_2$ | $\epsilon$ | $\eta$ | $g_{\mathrm{norm}}$ | $\eta_{min}$ | BSZ | warmup | $\lambda$ | Loss | Link |
|---|---|---|---|---|---|---|---|---|---|---|
| 0.98 | 0.98 | 1e-15 | 0.008 | 2 | 0 | 128 | 2000 | 0.1 | 3.403 | 0 |
| 0.8 | – | – | – | – | – | – | – | – | 7.417 | 1 |
| 0.9 | – | – | – | – | – | – | – | – | 3.427 | 2 |
| 0.95 | – | – | – | – | – | – | – | – | 3.413 | 3 |
| – | 0.9 | – | – | – | – | – | – | – | >10 | 4 |
| – | 0.95 | – | – | – | – | – | – | – | 3.408 | 5 |
| – | – | 1e-25 | – | – | – | – | – | – | 3.404 | 6 |
| – | – | 1e-20 | – | – | – | – | – | – | 3.404 | 7 |
| – | – | 1e-10 | – | – | – | – | – | – | 3.404 | 8 |
| – | – | – | 0.016 | – | – | – | – | – | 3.416 | 9 |
| – | – | – | 0.032 | – | – | – | – | – | 3.513 | 10 |
| – | – | – | – | 0 | – | – | – | – | 3.415 | 11 |
| – | – | – | – | 1.0 | – | – | – | – | 3.401 | 12 |
| – | – | – | – | – | – | 256 | – | – | 3.422 | 13 |
| – | – | – | – | – | – | 512 | – | – | 3.499 | 14 |
| – | – | – | – | – | – | – | 500 | – | 3.453 | 15 |
| – | – | – | – | – | – | – | 1000 | – | 3.412 | 16 |
| – | – | – | – | – | – | – | 4000 | – | 3.410 | 17 |
| – | – | – | – | – | – | – | – | 0 | 3.436 | 18 |
| – | – | – | – | – | – | – | – | 0.2 | 3.405 | 19 |

Table 45: Hyperparameter ablation for Cautious on 130m on 8x Chinchilla Data

| $\beta_1$ | $\beta_2$ | $\epsilon$ | $\eta$ | $g_{\text{norm}}$ | $\eta_{min}$ | BSZ | warmup | $\lambda$ | Loss | Link |
|---|---|---|---|---|---|---|---|---|---|---|
| 0.98 | 0.98 | 1e-15 | 0.008 | 2 | 0 | 256 | 2000 | 0.1 | 3.253 | 0 |
| 0.8 | – | – | – | – | – | – | – | – | 7.395 | 1 |
| 0.9 | – | – | – | – | – | – | – | – | 3.271 | 2 |
| 0.95 | – | – | – | – | – | – | – | – | 3.259 | 3 |
| – | 0.9 | – | – | – | – | – | – | – | >10 | 4 |
| – | 0.95 | – | – | – | – | – | – | – | 3.253 | 5 |
| – | – | 1e-25 | – | – | – | – | – | – | 3.251 | 6 |
| – | – | 1e-20 | – | – | – | – | – | – | 3.251 | 7 |
| – | – | 1e-10 | – | – | – | – | – | – | 3.250 | 8 |
| – | – | – | 0.016 | – | – | – | – | – | 3.257 | 9 |
| – | – | – | 0.032 | – | – | – | – | – | 3.383 | 10 |
| – | – | – | – | 0 | – | – | – | – | 3.254 | 11 |
| – | – | – | – | 1.0 | – | – | – | – | 3.251 | 12 |
| – | – | – | – | – | – | 128 | – | – | 3.265 | 13 |
| – | – | – | – | – | – | 512 | – | – | 3.261 | 14 |
| – | – | – | – | – | – | 1024 | – | – | 3.294 | 15 |
| – | – | – | – | – | – | – | 500 | – | 3.276 | 16 |
| – | – | – | – | – | – | – | 1000 | – | 3.253 | 17 |
| – | – | – | – | – | – | – | 4000 | – | 3.255 | 18 |
| – | – | – | – | – | – | – | – | 0 | 3.291 | 19 |
| – | – | – | – | – | – | – | – | 0.2 | 3.253 | 20 |

Table 46: Hyperparameter ablation for Cautious on 300m on 1x Chinchilla Data

| $\beta_1$ | $\beta_2$ | $\epsilon$ | $\eta$ | $g_{\text{norm}}$ | $\eta_{min}$ | BSZ | warmup | $\lambda$ | Loss | Link |
|---|---|---|---|---|---|---|---|---|---|---|
| 0.98 | 0.98 | 1e-25 | 0.008 | 2 | 0 | 128 | 2000 | 0.1 | 3.260 | 0 |
| 0.9 | – | – | – | – | – | – | – | – | 3.286 | 1 |
| 0.95 | – | – | – | – | – | – | – | – | 3.271 | 2 |
| – | 0.9 | – | – | – | – | – | – | – | >10 | 3 |
| – | 0.95 | – | – | – | – | – | – | – | >10 | 4 |
| – | – | 1e-25 | – | – | – | – | – | – | 3.260 | 5 |
| – | – | 1e-15 | – | – | – | – | – | – | 3.260 | 6 |
| – | – | – | – | 0 | – | – | – | – | 3.274 | 7 |
| – | – | – | – | 1 | – | – | – | – | 3.259 | 8 |
| – | – | – | – | – | – | 256 | – | – | 3.270 | 9 |
| – | – | – | – | – | – | – | 1000 | – | 7.352 | 10 |
| – | – | – | – | – | – | – | 4000 | – | 3.264 | 11 |
| – | – | – | – | – | – | – | – | 0.0 | 3.310 | 12 |
| – | – | – | – | – | – | – | – | 0.2 | 3.266 | 13 |

Table 47: Hyperparameter ablation for Cautious on 520m on 1x Chinchilla Data

| $\beta_1$ | $\beta_2$ | $\epsilon$ | $\eta$ | $g_{\mathrm{norm}}$ | $\eta_{min}$ | BSZ | warmup | $\lambda$ | Loss | Link |
|------|------|-------|-------|---|---|-----|------|-----|-------|----|
| 0.98 | 0.98 | 1e-25 | 0.008 | 1 | 0 | 256 | 2000 | 0.1 | 3.100 | 0 |
| 0.8 | – | – | – | – | – | – | – | – | >10 | 1 |
| 0.9 | – | – | – | – | – | – | – | – | 7.264 | 2 |
| 0.95 | – | – | – | – | – | – | – | – | 3.108 | 3 |
| – | 0.9 | – | – | – | – | – | – | – | >10 | 4 |
| – | 0.95 | – | – | – | – | – | – | – | 3.105 | 5 |
| – | – | 1e-25 | – | – | – | – | – | – | 3.100 | 6 |
| – | – | 1e-20 | – | – | – | – | – | – | 3.100 | 7 |
| – | – | 1e-15 | – | – | – | – | – | – | 3.101 | 8 |
| – | – | 1e-10 | – | – | – | – | – | – | 3.101 | 9 |
| – | – | – | 0.016 | – | – | – | – | – | 3.125 | 10 |
| – | – | – | 0.032 | – | – | – | – | – | 7.662 | 11 |
| – | – | – | – | 0 | – | – | – | – | 3.123 | 12 |
| – | – | – | – | 2.0 | – | – | – | – | 3.102 | 13 |
| – | – | – | – | – | – | 128 | – | – | >10 | 14 |
| – | – | – | – | – | – | 512 | – | – | 3.123 | 15 |
| – | – | – | – | – | – | – | 500 | – | >10 | 16 |
| – | – | – | – | – | – | – | 1000 | – | 3.119 | 17 |
| – | – | – | – | – | – | – | 4000 | – | 3.107 | 18 |
| – | – | – | – | – | – | – | – | 0 | 3.131 | 19 |
| – | – | – | – | – | – | – | – | 0.2 | 3.105 | 20 |

### E.3 SWEEPING RESULTS FOR LION

Table 48: Hyperparameter ablation for Lion on 130m on 1x Chinchilla Data

| $\beta_1$ | $\beta_2$ | $\eta$ | $g_{\mathrm{norm}}$ | $\eta_{min}$ | BSZ | warmup | $\lambda$ | Loss | Link |
|------|------|--------|------|------|------|--------|------|--------|------|
| 0.9 | 0.95 | 0.002 | 1 | 0 | 128 | 2000 | 0.7 | 3.552 | 0 |
| 0.8 | – | – | – | – | – | – | – | 3.575 | 1 |
| 0.95 | – | – | – | – | – | – | – | 3.557 | 2 |
| 0.98 | – | – | – | – | – | – | – | 7.644 | 3 |
| – | 0.9 | – | – | – | – | – | – | 3.550 | 4 |
| – | 0.98 | – | – | – | – | – | – | 3.630 | 5 |
| – | – | 0.0005 | – | – | – | – | – | 3.579 | 6 |
| – | – | 0.001 | – | – | – | – | – | 3.549 | 7 |
| – | – | 0.004 | – | – | – | – | – | 7.806 | 8 |
| – | – | 0.008 | – | – | – | – | – | 7.828 | 9 |
| – | – | – | 0 | – | – | – | – | 3.559 | 10 |
| – | – | – | 2.0 | – | – | – | – | 3.557 | 11 |
| – | – | – | – | – | 256 | – | – | 3.643 | 12 |
| – | – | – | – | – | – | 500 | – | 7.840 | 13 |
| – | – | – | – | – | – | 1000 | – | 7.739 | 14 |
| – | – | – | – | – | – | 4000 | – | 3.601 | 15 |
| – | – | – | – | – | – | – | 0 | 3.570 | 16 |
| – | – | – | – | – | – | – | 0.1 | 3.562 | 17 |
| – | – | – | – | – | – | – | 0.2 | 3.557 | 18 |
| – | – | – | – | – | – | – | 0.3 | 3.555 | 19 |
| – | – | – | – | – | – | – | 0.4 | 3.553 | 20 |
| – | – | – | – | – | – | – | 0.5 | 3.551 | 21 |
| – | – | – | – | – | – | – | 0.6 | 3.549 | 22 |
| – | – | – | – | – | – | – | 0.8 | 3.554 | 23 |
| – | – | – | – | – | – | – | 0.9 | 3.556 | 24 |
| – | – | – | – | – | – | – | 1 | 3.557 | 25 |

Table 49: Hyperparameter ablation for Lion on 130m on 2x Chinchilla Data

| $\beta_1$ | $\beta_2$ | $\eta$ | $g_{\text{norm}}$ | $\eta_{min}$ | BSZ | warmup | $\lambda$ | Loss | Link |
|---|---|---|---|---|---|---|---|---|---|
| 0.9 | 0.95 | 0.001 | 1 | 0 | 128 | 2000 | 0.7 | 3.414 | 0 |
| 0.8 | – | – | – | – | – | – | – | 3.436 | 1 |
| 0.95 | – | – | – | – | – | – | – | 3.418 | 2 |
| 0.98 | – | – | – | – | – | – | – | 7.313 | 3 |
| – | 0.9 | – | – | – | – | – | – | 3.433 | 4 |
| – | 0.98 | – | – | – | – | – | – | 3.409 | 5 |
| – | – | 0.0005 | – | – | – | – | – | 3.435 | 6 |
| – | – | 0.002 | – | – | – | – | – | 3.414 | 7 |
| – | – | 0.004 | – | – | – | – | – | 3.489 | 8 |
| – | – | 0.008 | – | – | – | – | – | 7.826 | 9 |
| – | – | – | 0 | – | – | – | – | 3.414 | 10 |
| – | – | – | 2.0 | – | – | – | – | 3.413 | 11 |
| – | – | – | – | – | 256 | – | – | 3.447 | 12 |
| – | – | – | – | – | 512 | – | – | 3.540 | 13 |
| – | – | – | – | – | – | 500 | – | 3.435 | 14 |
| – | – | – | – | – | – | 1000 | – | 3.415 | 15 |
| – | – | – | – | – | – | 4000 | – | 3.423 | 16 |
| – | – | – | – | – | – | – | 0.4 | 3.419 | 17 |
| – | – | – | – | – | – | – | 0.5 | 3.418 | 18 |
| – | – | – | – | – | – | – | 0.6 | 3.414 | 19 |
| – | – | – | – | – | – | – | 0.8 | 3.414 | 20 |
| – | – | – | – | – | – | – | 0.9 | 3.413 | 21 |
| – | – | – | – | – | – | – | 1 | 3.413 | 22 |

Table 50: Hyperparameter ablation for Lion on 130m on 4x Chinchilla Data

| $\beta_1$ | $\beta_2$ | $\eta$ | $g_{\text{norm}}$ | $\eta_{min}$ | BSZ | warmup | $\lambda$ | Loss | Link |
|---|---|---|---|---|---|---|---|---|---|
| 0.9 | 0.95 | 0.001 | 1 | 0 | 128 | 2000 | 0.7 | 3.331 | 0 |
| 0.8 | – | – | – | – | – | – | – | 3.345 | 1 |
| 0.95 | – | – | – | – | – | – | – | 3.335 | 2 |
| 0.98 | – | – | – | – | – | – | – | 7.330 | 3 |
| – | 0.9 | – | – | – | – | – | – | 3.345 | 4 |
| – | 0.98 | – | – | – | – | – | – | 3.340 | 5 |
| – | – | 0.0005 | – | – | – | – | – | 3.338 | 6 |
| – | – | 0.002 | – | – | – | – | – | 3.342 | 7 |
| – | – | 0.004 | – | – | – | – | – | 7.406 | 8 |
| – | – | 0.008 | – | – | – | – | – | >10 | 9 |
| – | – | – | 0 | – | – | – | – | 3.333 | 10 |
| – | – | – | 2.0 | – | – | – | – | 3.334 | 11 |
| – | – | – | – | – | 256 | – | – | 3.346 | 12 |
| – | – | – | – | – | 512 | – | – | 3.386 | 13 |
| – | – | – | – | – | 1024 | – | – | 3.492 | 14 |
| – | – | – | – | – | – | 500 | – | 3.364 | 15 |
| – | – | – | – | – | – | 1000 | – | 3.336 | 16 |
| – | – | – | – | – | – | 4000 | – | 3.333 | 17 |
| – | – | – | – | – | – | – | 0.4 | 3.335 | 18 |
| – | – | – | – | – | – | – | 0.5 | 3.333 | 19 |
| – | – | – | – | – | – | – | 0.6 | 3.329 | 20 |
| – | – | – | – | – | – | – | 0.8 | 3.332 | 21 |
| – | – | – | – | – | – | – | 0.9 | 3.332 | 22 |
| – | – | – | – | – | – | – | 1 | 3.335 | 23 |

Table 51: Hyperparameter ablation for Lion on 130m on 8x Chinchilla Data

| $\beta_1$ | $\beta_2$ | $\eta$ | $g_{\text{norm}}$ | $\eta_{min}$ | BSZ | warmup | $\lambda$ | Loss | Link |
|---|---|---|---|---|---|---|---|---|---|
| 0.9 | 0.98 | 0.001 | 1 | 0 | 128 | 2000 | 0.7 | 3.252 | 0 |
| 0.8 | – | – | – | – | – | – | – | 3.287 | 1 |
| 0.95 | – | – | – | – | – | – | – | 3.264 | 2 |
| 0.98 | – | – | – | – | – | – | – | 3.286 | 3 |
| – | 0.9 | – | – | – | – | – | – | 3.277 | 4 |
| – | 0.95 | – | – | – | – | – | – | 3.263 | 5 |
| – | – | 0.0005 | – | – | – | – | – | 3.254 | 6 |
| – | – | 0.002 | – | – | – | – | – | 3.310 | 7 |
| – | – | 0.004 | – | – | – | – | – | 7.829 | 8 |
| – | – | 0.008 | – | – | – | – | – | NaN | 9 |
| – | – | – | 0 | – | – | – | – | 3.329 | 10 |
| – | – | – | 2.0 | – | – | – | – | 3.265 | 11 |
| – | – | – | – | – | 256 | – | – | 3.260 | 12 |
| – | – | – | – | – | 512 | – | – | 3.287 | 13 |
| – | – | – | – | – | 1024 | – | – | 3.342 | 14 |
| – | – | – | – | – | – | 500 | – | 3.336 | 15 |
| – | – | – | – | – | – | 1000 | – | 3.273 | 16 |
| – | – | – | – | – | – | 4000 | – | 3.258 | 17 |
| – | – | – | – | – | – | – | 0.4 | 3.256 | 18 |
| – | – | – | – | – | – | – | 0.5 | 3.251 | 19 |
| – | – | – | – | – | – | – | 0.6 | 3.252 | 20 |
| – | – | – | – | – | – | – | 0.8 | 3.258 | 21 |
| – | – | – | – | – | – | – | 0.9 | 3.259 | 22 |
| – | – | – | – | – | – | – | 1 | 3.261 | 23 |

Table 52: Hyperparameter ablation for Lion on 300m on 1x Chinchilla Data

| $\beta_1$ | $\beta_2$ | $\eta$ | $g_{\text{norm}}$ | $\eta_{min}$ | BSZ | warmup | $\lambda$ | Loss | Link |
|---|---|---|---|---|---|---|---|---|---|
| 0.9 | 0.95 | 0.001 | 1 | 0 | 128 | 2000 | 0.6 | 3.268 | 0 |
| 0.8 | – | – | – | – | – | – | – | 3.283 | 1 |
| 0.95 | – | – | – | – | – | – | – | 3.271 | 2 |
| 0.98 | – | – | – | – | – | – | – | 7.690 | 3 |
| – | 0.9 | – | – | – | – | – | – | 3.283 | 4 |
| – | 0.98 | – | – | – | – | – | – | 3.271 | 5 |
| – | – | 0.0005 | – | – | – | – | – | 3.286 | 6 |
| – | – | 0.002 | – | – | – | – | – | 3.283 | 7 |
| – | – | 0.004 | – | – | – | – | – | 7.817 | 8 |
| – | – | 0.008 | – | – | – | – | – | 7.863 | 9 |
| – | – | – | 0 | – | – | – | – | 3.274 | 10 |
| – | – | – | 2.0 | – | – | – | – | 3.269 | 11 |
| – | – | – | – | – | 256 | – | – | 3.295 | 12 |
| – | – | – | – | – | 512 | – | – | 3.367 | 13 |
| – | – | – | – | – | – | 500 | – | 7.607 | 14 |
| – | – | – | – | – | – | 1000 | – | 3.278 | 15 |
| – | – | – | – | – | – | 4000 | – | 3.277 | 16 |
| – | – | – | – | – | – | – | 0.4 | 3.272 | 17 |
| – | – | – | – | – | – | – | 0.5 | 3.271 | 18 |
| – | – | – | – | – | – | – | 0.7 | 3.268 | 19 |
| – | – | – | – | – | – | – | 0.8 | 3.268 | 20 |
| – | – | – | – | – | – | – | 0.9 | 3.269 | 21 |
| – | – | – | – | – | – | – | 1 | 3.269 | 22 |

Table 53: Hyperparameter ablation for Lion on 520m on 1x Chinchilla Data

| $\beta_1$ | $\beta_2$ | $\eta$ | $g_{\mathrm{norm}}$ | $\eta_{min}$ | BSZ | warmup | $\lambda$ | Loss | Link |
|---|---|---|---|---|---|---|---|---|---|
| 0.9 | 0.95 | 0.001 | 1 | 0 | 128 | 2000 | 0.7 | 3.108 | 0 |
| 0.8 | – | – | – | – | – | – | – | 3.117 | 1 |
| 0.95 | – | – | – | – | – | – | – | 3.121 | 2 |
| 0.98 | – | – | – | – | – | – | – | 7.577 | 3 |
| – | 0.9 | – | – | – | – | – | – | 3.119 | 4 |
| – | 0.98 | – | – | – | – | – | – | 3.208 | 5 |
| – | – | 0.0005 | – | – | – | – | – | 3.113 | 6 |
| – | – | 0.002 | – | – | – | – | – | 7.734 | 7 |
| – | – | 0.004 | – | – | – | – | – | 8.046 | 8 |
| – | – | 0.008 | – | – | – | – | – | NaN | 9 |
| – | – | – | 0 | – | – | – | – | 3.128 | 10 |
| – | – | – | 2.0 | – | – | – | – | 3.109 | 11 |
| – | – | – | – | – | 256 | – | – | 3.117 | 12 |
| – | – | – | – | – | 512 | – | – | 3.151 | 13 |
| – | – | – | – | – | 1024 | – | – | 3.252 | 14 |
| – | – | – | – | – | – | 500 | – | 7.403 | 15 |
| – | – | – | – | – | – | 1000 | – | 7.119 | 16 |
| – | – | – | – | – | – | 4000 | – | 3.115 | 17 |
| – | – | – | – | – | – | – | 0.4 | 3.109 | 18 |
| – | – | – | – | – | – | – | 0.5 | 3.108 | 19 |
| – | – | – | – | – | – | – | 0.6 | 3.108 | 20 |
| – | – | – | – | – | – | – | 0.8 | 3.109 | 21 |
| – | – | – | – | – | – | – | 0.9 | 3.112 | 22 |
| – | – | – | – | – | – | – | 1 | 3.115 | 23 |

### E.4 SWEEPING RESULTS FOR MARS

Table 54: Hyperparameter ablation for Mars on 130m on 1x Chinchilla Data

| $\beta_1$ | $\beta_2$ | $\epsilon$ | $\gamma$ | $\eta$ | $g_{\mathrm{norm}}$ | $\eta_{min}$ | BSZ | warmup | $\lambda$ | Loss | Link |
|---|---|---|---|---|---|---|---|---|---|---|---|
| 0.9 | 0.95 | 1e-25 | 0.025 | 0.016 | 1 | 0 | 128 | 2000 | 0.1 | 3.537 | 0 |
| 0.8 | – | – | – | – | – | – | – | – | – | 3.568 | 1 |
| 0.95 | – | – | – | – | – | – | – | – | – | 3.548 | 2 |
| 0.98 | – | – | – | – | – | – | – | – | – | 3.586 | 3 |
| – | 0.9 | – | – | – | – | – | – | – | – | 3.548 | 4 |
| – | 0.98 | – | – | – | – | – | – | – | – | 3.536 | 5 |
| – | 0.99 | – | – | – | – | – | – | – | – | 3.562 | 6 |
| – | – | 1e-30 | – | – | – | – | – | – | – | 3.537 | 7 |
| – | – | 1e-25 | – | – | – | – | – | – | – | 3.537 | 8 |
| – | – | 1e-20 | – | – | – | – | – | – | – | 3.537 | 9 |
| – | – | 1e-15 | – | – | – | – | – | – | – | 3.537 | 10 |
| – | – | 1e-10 | – | – | – | – | – | – | – | 3.537 | 11 |
| – | – | – | 0.0125 | – | – | – | – | – | – | 3.540 | 12 |
| – | – | – | 0.05 | – | – | – | – | – | – | 3.542 | 13 |
| – | – | – | 0.1 | – | – | – | – | – | – | 3.553 | 14 |
| – | – | – | – | 0.008 | – | – | – | – | – | 3.538 | 15 |
| – | – | – | – | 0.032 | – | – | – | – | – | 7.574 | 16 |
| – | – | – | – | – | – | – | 256 | – | – | 3.616 | 17 |
| – | – | – | – | – | – | – | – | 500 | – | >10 | 18 |
| – | – | – | – | – | – | – | – | 1000 | – | >10 | 19 |
| – | – | – | – | – | – | – | – | 4000 | – | 3.596 | 20 |
| – | – | – | – | – | – | – | – | – | 0 | 3.552 | 21 |
| – | – | – | – | – | – | – | – | – | 0.2 | 3.561 | 22 |

Table 55: Hyperparameter ablation for Mars on 130m on 2x Chinchilla Data

| $\beta_1$ | $\beta_2$ | $\epsilon$ | $\gamma$ | $\eta$ | $g_{\mathrm{norm}}$ | $\eta_{min}$ | BSZ | warmup | $\lambda$ | Loss | Link |
|---|---|---|---|---|---|---|---|---|---|---|---|
| 0.95 | 0.98 | 1e-25 | 0.025 | 0.008 | 1 | 0 | 128 | 2000 | 0.1 | 3.396 | 0 |
| 0.8 | – | – | – | – | – | – | – | – | – | 3.421 | 1 |
| 0.9 | – | – | – | – | – | – | – | – | – | 3.402 | 2 |
| 0.98 | – | – | – | – | – | – | – | – | – | 3.401 | 3 |
| – | 0.9 | – | – | – | – | – | – | – | – | 3.409 | 4 |
| – | 0.95 | – | – | – | – | – | – | – | – | 3.400 | 5 |
| – | 0.99 | – | – | – | – | – | – | – | – | 3.398 | 6 |
| – | – | 1e-30 | – | – | – | – | – | – | – | 3.396 | 7 |
| – | – | 1e-20 | – | – | – | – | – | – | – | 3.396 | 8 |
| – | – | 1e-15 | – | – | – | – | – | – | – | 3.398 | 9 |
| – | – | 1e-10 | – | – | – | – | – | – | – | 3.397 | 10 |
| – | – | – | 0.0125 | – | – | – | – | – | – | 3.398 | 11 |
| – | – | – | 0.05 | – | – | – | – | – | – | 3.404 | 12 |
| – | – | – | 0.1 | – | – | – | – | – | – | 3.415 | 13 |
| – | – | – | – | 0.016 | – | – | – | – | – | 3.402 | 14 |
| – | – | – | – | 0.032 | – | – | – | – | – | 3.441 | 15 |
| – | – | – | – | – | – | – | 256 | – | – | 3.427 | 16 |
| – | – | – | – | – | – | – | 512 | – | – | 3.524 | 17 |
| – | – | – | – | – | – | – | – | 500 | – | 3.400 | 18 |
| – | – | – | – | – | – | – | – | 1000 | – | 3.395 | 19 |
| – | – | – | – | – | – | – | – | 4000 | – | 3.409 | 20 |
| – | – | – | – | – | – | – | – | – | 0 | 3.430 | 21 |
| – | – | – | – | – | – | – | – | – | 0.2 | 3.404 | 22 |

Table 56: Hyperparameter ablation for Mars on 130m on 4x Chinchilla Data

| $\beta_1$ | $\beta_2$ | $\epsilon$ | $\gamma$ | $\eta$ | $g_{\mathrm{norm}}$ | $\eta_{min}$ | BSZ | warmup | $\lambda$ | Loss | Link |
|---|---|---|---|---|---|---|---|---|---|---|---|
| 0.9 | 0.98 | 1e-25 | 0.025 | 0.008 | 1 | 0 | 128 | 2000 | 0.1 | 3.323 | 0 |
| 0.8 | – | – | – | – | – | – | – | – | – | 3.336 | 1 |
| 0.95 | – | – | – | – | – | – | – | – | – | 3.326 | 2 |
| 0.98 | – | – | – | – | – | – | – | – | – | 3.350 | 3 |
| – | 0.9 | – | – | – | – | – | – | – | – | 3.337 | 4 |
| – | 0.95 | – | – | – | – | – | – | – | – | 3.330 | 5 |
| – | 0.99 | – | – | – | – | – | – | – | – | 3.324 | 6 |
| – | – | 1e-30 | – | – | – | – | – | – | – | 3.323 | 7 |
| – | – | 1e-25 | – | – | – | – | – | – | – | 3.323 | 8 |
| – | – | 1e-20 | – | – | – | – | – | – | – | 3.323 | 9 |
| – | – | 1e-15 | – | – | – | – | – | – | – | 3.321 | 10 |
| – | – | 1e-10 | – | – | – | – | – | – | – | 3.322 | 11 |
| – | – | – | 0.0125 | – | – | – | – | – | – | 3.323 | 12 |
| – | – | – | 0.05 | – | – | – | – | – | – | 3.324 | 13 |
| – | – | – | 0.1 | – | – | – | – | – | – | 3.330 | 14 |
| – | – | – | – | 0.016 | – | – | – | – | – | 3.337 | 15 |
| – | – | – | – | 0.032 | – | – | – | – | – | 8.642 | 16 |
| – | – | – | – | – | – | – | 256 | – | – | 3.336 | 17 |
| – | – | – | – | – | – | – | 512 | – | – | 3.384 | 18 |
| – | – | – | – | – | – | – | 1024 | – | – | 3.525 | 19 |
| – | – | – | – | – | – | – | – | 500 | – | 3.328 | 20 |
| – | – | – | – | – | – | – | – | 1000 | – | 3.321 | 21 |
| – | – | – | – | – | – | – | – | 4000 | – | 3.327 | 22 |
| – | – | – | – | – | – | – | – | – | 0 | 3.359 | 23 |
| – | – | – | – | – | – | – | – | – | 0.2 | 3.333 | 24 |

Table 57: Hyperparameter ablation for Mars on 130m on 8x Chinchilla Data

| $\beta_1$ | $\beta_2$ | $\epsilon$ | $\gamma$ | $\eta$ | $g_{\text{norm}}$ | $\eta_{min}$ | BSZ | warmup | $\lambda$ | Loss | Link |
|---|---|---|---|---|---|---|---|---|---|---|---|
| 0.98 | 0.99 | 1e-25 | 0.025 | 0.008 | 1 | 0 | 128 | 2000 | 0.1 | 3.247 | 0 |
| 0.8 | – | – | – | – | – | – | – | – | – | 3.272 | 1 |
| 0.9 | – | – | – | – | – | – | – | – | – | 3.252 | 2 |
| 0.95 | – | – | – | – | – | – | – | – | – | 3.256 | 3 |
| – | 0.9 | – | – | – | – | – | – | – | – | NaN | 4 |
| – | 0.95 | – | – | – | – | – | – | – | – | 3.257 | 5 |
| – | 0.98 | – | – | – | – | – | – | – | – | 3.249 | 6 |
| – | – | 1e-30 | – | – | – | – | – | – | – | 3.247 | 7 |
| – | – | 1e-25 | – | – | – | – | – | – | – | 3.247 | 8 |
| – | – | 1e-20 | – | – | – | – | – | – | – | 3.247 | 9 |
| – | – | 1e-15 | – | – | – | – | – | – | – | 3.247 | 10 |
| – | – | 1e-10 | – | – | – | – | – | – | – | 3.248 | 11 |
| – | – | – | 0.0125 | – | – | – | – | – | – | 3.255 | 12 |
| – | – | – | 0.05 | – | – | – | – | – | – | 3.252 | 13 |
| – | – | – | 0.1 | – | – | – | – | – | – | 3.410 | 14 |
| – | – | – | – | 0.016 | – | – | – | – | – | 3.262 | 15 |
| – | – | – | – | 0.032 | – | – | – | – | – | 3.322 | 16 |
| – | – | – | – | – | – | – | 256 | – | – | 3.250 | 17 |
| – | – | – | – | – | – | – | 512 | – | – | 3.277 | 18 |
| – | – | – | – | – | – | – | 1024 | – | – | 3.339 | 19 |
| – | – | – | – | – | – | – | – | 500 | – | 3.250 | 20 |
| – | – | – | – | – | – | – | – | 1000 | – | 3.248 | 21 |
| – | – | – | – | – | – | – | – | 4000 | – | 3.251 | 22 |
| – | – | – | – | – | – | – | – | – | 0 | 3.292 | 23 |
| – | – | – | – | – | – | – | – | – | 0.2 | 3.265 | 24 |

Table 58: Hyperparameter ablation for Mars on 300m on 1x Chinchilla Data

| $\beta_1$ | $\beta_2$ | $\epsilon$ | $\gamma$ | $\eta$ | $g_{\mathrm{norm}}$ | $\eta_{min}$ | BSZ | warmup | $\lambda$ | Loss | Link |
|---|---|---|---|---|---|---|---|---|---|---|---|
| 0.98 | 0.98 | 1e-25 | 0.05 | 0.008 | 1 | 0 | 128 | 1000 | 0.1 | 3.249 | 0 |
| 0.8 | – | – | – | – | – | – | – | – | – | 7.899 | 1 |
| 0.9 | – | – | – | – | – | – | – | – | – | 3.259 | 2 |
| 0.95 | – | – | – | – | – | – | – | – | – | 3.247 | 3 |
| – | 0.9 | – | – | – | – | – | – | – | – | NaN | 4 |
| – | 0.95 | – | – | – | – | – | – | – | – | 3.256 | 5 |
| – | 0.99 | – | – | – | – | – | – | – | – | 3.247 | 6 |
| – | – | 1e-30 | – | – | – | – | – | – | – | 3.249 | 7 |
| – | – | 1e-20 | – | – | – | – | – | – | – | 3.250 | 8 |
| – | – | 1e-15 | – | – | – | – | – | – | – | 3.250 | 9 |
| – | – | 1e-10 | – | – | – | – | – | – | – | 3.252 | 10 |
| – | – | – | 0.0125 | – | – | – | – | – | – | 3.259 | 11 |
| – | – | – | 0.025 | – | – | – | – | – | – | 3.249 | 12 |
| – | – | – | 0.1 | – | – | – | – | – | – | 3.280 | 13 |
| – | – | – | – | 0.016 | – | – | – | – | – | 3.265 | 14 |
| – | – | – | – | 0.032 | – | – | – | – | – | 3.410 | 15 |
| – | – | – | – | – | – | – | 256 | – | – | 3.288 | 16 |
| – | – | – | – | – | – | – | 512 | – | – | 3.389 | 17 |
| – | – | – | – | – | – | – | 1024 | – | – | 3.624 | 18 |
| – | – | – | – | – | – | – | – | 500 | – | 3.254 | 19 |
| – | – | – | – | – | – | – | – | 2000 | – | 3.254 | 20 |
| – | – | – | – | – | – | – | – | 4000 | – | 3.269 | 21 |
| – | – | – | – | – | – | – | – | – | 0 | 3.312 | 22 |
| – | – | – | – | – | – | – | – | – | 0.2 | 3.262 | 23 |

Table 59: Hyperparameter ablation for Mars on 520m on 1x Chinchilla Data

| $\beta_1$ | $\beta_2$ | $\epsilon$ | $\gamma$ | $\eta$ | $g_{\mathrm{norm}}$ | $\eta_{min}$ | BSZ | warmup | $\lambda$ | Loss | Link |
|---|---|---|---|---|---|---|---|---|---|---|---|
| 0.95 | 0.95 | 1e-25 | 0.025 | 0.008 | 1 | 0 | 256 | 2000 | 0.1 | 3.101 | 0 |
| 0.8 | – | – | – | – | – | – | – | – | – | >10 | 1 |
| 0.9 | – | – | – | – | – | – | – | – | – | 3.108 | 2 |
| 0.98 | – | – | – | – | – | – | – | – | – | 3.107 | 3 |
| – | 0.9 | – | – | – | – | – | – | – | – | 3.108 | 4 |
| – | 0.98 | – | – | – | – | – | – | – | – | 3.099 | 5 |
| – | 0.99 | – | – | – | – | – | – | – | – | 3.102 | 6 |
| – | – | 1e-30 | – | – | – | – | – | – | – | 3.101 | 7 |
| – | – | 1e-20 | – | – | – | – | – | – | – | 3.101 | 8 |
| – | – | 1e-15 | – | – | – | – | – | – | – | 3.101 | 9 |
| – | – | 1e-10 | – | – | – | – | – | – | – | 3.101 | 10 |
| – | – | – | 0.0125 | – | – | – | – | – | – | 3.100 | 11 |
| – | – | – | 0.05 | – | – | – | – | – | – | 3.103 | 12 |
| – | – | – | 0.1 | – | – | – | – | – | – | 3.113 | 13 |
| – | – | – | – | 0.016 | – | – | – | – | – | 3.111 | 14 |
| – | – | – | – | 0.032 | – | – | – | – | – | NaN | 15 |
| – | – | – | – | – | – | – | 128 | – | – | 3.102 | 16 |
| – | – | – | – | – | – | – | 512 | – | – | 3.130 | 17 |
| – | – | – | – | – | – | – | 1024 | – | – | 3.224 | 18 |
| – | – | – | – | – | – | – | – | 500 | – | 3.128 | 19 |
| – | – | – | – | – | – | – | – | 1000 | – | 3.105 | 20 |
| – | – | – | – | – | – | – | – | 4000 | – | 3.112 | 21 |
| – | – | – | – | – | – | – | – | – | 0 | 3.131 | 22 |
| – | – | – | – | – | – | – | – | – | 0.2 | 3.103 | 23 |

## E.5 SWEEPING RESULTS FOR NADAMW

Table 60: Hyperparameter ablation for NAdamW on 130m on 1x Chinchilla Data

| $\beta_1$ | $\beta_2$ | $\epsilon$ | $\eta$ | $g_{\mathrm{norm}}$ | $\eta_{min}$ | BSZ | warmup | $\lambda$ | Loss | Link |
|---|---|---|---|---|---|---|---|---|---|---|
| 0.95 | 0.98 | 1e-25 | 0.008 | 1 | 0 | 128 | 2000 | 0.1 | 3.531 | 0 |
| 0.8 | – | – | – | – | – | – | – | – | 4.764 | 1 |
| 0.9 | – | – | – | – | – | – | – | – | 3.552 | 2 |
| 0.98 | – | – | – | – | – | – | – | – | 3.585 | 3 |
| – | 0.9 | – | – | – | – | – | – | – | 3.552 | 4 |
| – | 0.95 | – | – | – | – | – | – | – | 3.535 | 5 |
| – | – | 1e-20 | – | – | – | – | – | – | 3.531 | 6 |
| – | – | 1e-15 | – | – | – | – | – | – | 3.533 | 7 |
| – | – | 1e-10 | – | – | – | – | – | – | 3.531 | 8 |
| – | – | – | 0.016 | – | – | – | – | – | 3.545 | 9 |
| – | – | – | 0.032 | – | – | – | – | – | >10 | 10 |
| – | – | – | – | 0 | – | – | – | – | 3.537 | 11 |
| – | – | – | – | 2.0 | – | – | – | – | 3.539 | 12 |
| – | – | – | – | – | – | 256 | – | – | 3.624 | 13 |
| – | – | – | – | – | – | – | 500 | – | 3.646 | 14 |
| – | – | – | – | – | – | – | 1000 | – | 3.545 | 15 |
| – | – | – | – | – | – | – | 4000 | – | 3.577 | 16 |
| – | – | – | – | – | – | – | – | 0 | 3.547 | 17 |
| – | – | – | – | – | – | – | – | 0.2 | 3.535 | 18 |

Table 61: Hyperparameter ablation for NAdamW on 130m on 2x Chinchilla Data

| $\beta_1$ | $\beta_2$ | $\epsilon$ | $\eta$ | $g_{\mathrm{norm}}$ | $\eta_{min}$ | BSZ | warmup | $\lambda$ | Loss | Link |
|---|---|---|---|---|---|---|---|---|---|---|
| 0.95 | 0.98 | 1e-10 | 0.008 | 1 | 0 | 128 | 2000 | 0.1 | 3.394 | 0 |
| 0.8 | – | – | – | – | – | – | – | – | 3.452 | 1 |
| 0.9 | – | – | – | – | – | – | – | – | 3.408 | 2 |
| 0.98 | – | – | – | – | – | – | – | – | 3.402 | 3 |
| – | 0.9 | – | – | – | – | – | – | – | 3.403 | 4 |
| – | 0.95 | – | – | – | – | – | – | – | 3.399 | 5 |
| – | – | 1e-25 | – | – | – | – | – | – | 3.394 | 6 |
| – | – | 1e-20 | – | – | – | – | – | – | 3.394 | 7 |
| – | – | 1e-15 | – | – | – | – | – | – | 3.393 | 8 |
| – | – | – | 0.016 | – | – | – | – | – | 3.406 | 9 |
| – | – | – | 0.032 | – | – | – | – | – | 7.675 | 10 |
| – | – | – | – | 0 | – | – | – | – | 3.400 | 11 |
| – | – | – | – | 2.0 | – | – | – | – | 3.396 | 12 |
| – | – | – | – | – | – | 256 | – | – | 3.423 | 13 |
| – | – | – | – | – | – | 512 | – | – | 3.520 | 14 |
| – | – | – | – | – | – | – | 500 | – | 3.424 | 15 |
| – | – | – | – | – | – | – | 1000 | – | 3.400 | 16 |
| – | – | – | – | – | – | – | 4000 | – | 3.405 | 17 |
| – | – | – | – | – | – | – | – | 0 | 3.421 | 18 |
| – | – | – | – | – | – | – | – | 0.2 | 3.398 | 19 |

Table 62: Hyperparameter ablation for NAdamW on 130m on 4x Chinchilla Data

| $\beta_1$ | $\beta_2$ | $\epsilon$ | $\eta$ | $g_{\text{norm}}$ | $\eta_{min}$ | BSZ | warmup | $\lambda$ | Loss | Link |
|---|---|---|---|---|---|---|---|---|---|---|
| 0.95 | 0.98 | 1e-10 | 0.008 | 1 | 0 | 128 | 2000 | 0.1 | 3.319 | 0 |
| 0.8 | – | – | – | – | – | – | – | – | 7.085 | 1 |
| 0.9 | – | – | – | – | – | – | – | – | 3.331 | 2 |
| 0.98 | – | – | – | – | – | – | – | – | 3.349 | 3 |
| – | 0.9 | – | – | – | – | – | – | – | 3.332 | 4 |
| – | 0.95 | – | – | – | – | – | – | – | 3.327 | 5 |
| – | – | 1e-25 | – | – | – | – | – | – | 3.321 | 6 |
| – | – | 1e-20 | – | – | – | – | – | – | 3.321 | 7 |
| – | – | 1e-15 | – | – | – | – | – | – | 3.323 | 8 |
| – | – | – | 0.016 | – | – | – | – | – | 3.343 | 9 |
| – | – | – | 0.032 | – | – | – | – | – | 7.733 | 10 |
| – | – | – | – | 0 | – | – | – | – | 3.324 | 11 |
| – | – | – | – | 2.0 | – | – | – | – | 3.323 | 12 |
| – | – | – | – | – | – | 256 | – | – | 3.332 | 13 |
| – | – | – | – | – | – | 512 | – | – | 3.372 | 14 |
| – | – | – | – | – | – | 1024 | – | – | 3.496 | 15 |
| – | – | – | – | – | – | – | 500 | – | 6.917 | 16 |
| – | – | – | – | – | – | – | 1000 | – | 3.328 | 17 |
| – | – | – | – | – | – | – | 4000 | – | 3.321 | 18 |
| – | – | – | – | – | – | – | – | 0 | 3.359 | 19 |
| – | – | – | – | – | – | – | – | 0.2 | 3.330 | 20 |

Table 63: Hyperparameter ablation for NAdamW on 130m on 8x Chinchilla Data

| $\beta_1$ | $\beta_2$ | $\epsilon$ | $\eta$ | $g_{\text{norm}}$ | $\eta_{min}$ | BSZ | warmup | $\lambda$ | Loss | Link |
|---|---|---|---|---|---|---|---|---|---|---|
| 0.95 | 0.98 | 1e-10 | 0.008 | 1 | 0 | 128 | 2000 | 0.1 | 3.251 | 0 |
| 0.8 | – | – | – | – | – | – | – | – | 3.282 | 1 |
| 0.9 | – | – | – | – | – | – | – | – | 3.257 | 2 |
| 0.98 | – | – | – | – | – | – | – | – | 3.253 | 3 |
| – | 0.9 | – | – | – | – | – | – | – | 3.260 | 4 |
| – | 0.95 | – | – | – | – | – | – | – | 3.255 | 5 |
| – | – | 1e-25 | – | – | – | – | – | – | 3.251 | 6 |
| – | – | 1e-20 | – | – | – | – | – | – | 3.251 | 7 |
| – | – | 1e-15 | – | – | – | – | – | – | 3.250 | 8 |
| – | – | – | 0.016 | – | – | – | – | – | 3.274 | 9 |
| – | – | – | 0.032 | – | – | – | – | – | 7.670 | 10 |
| – | – | – | – | 0 | – | – | – | – | 3.253 | 11 |
| – | – | – | – | 2.0 | – | – | – | – | 3.250 | 12 |
| – | – | – | – | – | – | 256 | – | – | 3.249 | 13 |
| – | – | – | – | – | – | 512 | – | – | 3.270 | 14 |
| – | – | – | – | – | – | 1024 | – | – | 3.321 | 15 |
| – | – | – | – | – | – | – | 500 | – | 3.274 | 16 |
| – | – | – | – | – | – | – | 1000 | – | 3.255 | 17 |
| – | – | – | – | – | – | – | 4000 | – | 3.252 | 18 |
| – | – | – | – | – | – | – | – | 0 | 3.286 | 19 |
| – | – | – | – | – | – | – | – | 0.2 | 3.265 | 20 |

Table 64: Hyperparameter ablation for NAdamW on 300m on 1x Chinchilla Data

| $\beta_1$ | $\beta_2$ | $\epsilon$ | $\eta$ | $g_{\mathrm{norm}}$ | $\eta_{min}$ | BSZ | warmup | $\lambda$ | Loss | Link |
|---|---|---|---|---|---|---|---|---|---|---|
| 0.95 | 0.98 | 1e-10 | 0.008 | 1 | 0 | 128 | 2000 | 0.1 | 3.248 | 0 |
| 0.8 | – | – | – | – | – | – | – | – | 7.542 | 1 |
| 0.9 | – | – | – | – | – | – | – | – | 7.054 | 2 |
| 0.98 | – | – | – | – | – | – | – | – | 3.254 | 3 |
| – | 0.9 | – | – | – | – | – | – | – | 3.263 | 4 |
| – | 0.95 | – | – | – | – | – | – | – | 3.256 | 5 |
| – | – | 1e-25 | – | – | – | – | – | – | 3.250 | 6 |
| – | – | 1e-20 | – | – | – | – | – | – | 3.250 | 7 |
| – | – | 1e-15 | – | – | – | – | – | – | 3.250 | 8 |
| – | – | – | 0.016 | – | – | – | – | – | 7.538 | 9 |
| – | – | – | 0.032 | – | – | – | – | – | 7.695 | 10 |
| – | – | – | – | 0 | – | – | – | – | 3.260 | 11 |
| – | – | – | – | 2.0 | – | – | – | – | 3.250 | 12 |
| – | – | – | – | – | – | 256 | – | – | 3.272 | 13 |
| – | – | – | – | – | – | 512 | – | – | 3.344 | 14 |
| – | – | – | – | – | – | – | 500 | – | 7.744 | 15 |
| – | – | – | – | – | – | – | 1000 | – | 7.421 | 16 |
| – | – | – | – | – | – | – | 4000 | – | 3.256 | 17 |
| – | – | – | – | – | – | – | – | 0 | 3.283 | 18 |
| – | – | – | – | – | – | – | – | 0.2 | 3.257 | 19 |

## E.6 SWEEPING RESULTS FOR ADAM-MINI

Table 65: Hyperparameter ablation for Adam-Mini on 130m on 1x Chinchilla Data

| $\beta_1$ | $\beta_2$ | $\epsilon$ | $\eta$ | $g_{\mathrm{norm}}$ | $\eta_{min}$ | BSZ | warmup | $\lambda$ | Loss | Link |
|---|---|---|---|---|---|---|---|---|---|---|
| 0.9 | 0.98 | 1e-15 | 0.008 | 1 | 0 | 128 | 2000 | 0.1 | 3.542 | 0 |
| 0.95 | – | – | – | – | – | – | – | – | 3.560 | 1 |
| 0.98 | – | – | – | – | – | – | – | – | 7.733 | 2 |
| – | 0.9 | – | – | – | – | – | – | – | 3.554 | 3 |
| – | 0.95 | – | – | – | – | – | – | – | 3.545 | 4 |
| – | – | 1e-25 | – | – | – | – | – | – | 3.546 | 5 |
| – | – | 1e-20 | – | – | – | – | – | – | 3.546 | 6 |
| – | – | 1e-10 | – | – | – | – | – | – | 3.548 | 7 |
| – | – | – | 0.004 | – | – | – | – | – | 3.558 | 8 |
| – | – | – | 0.016 | – | – | – | – | – | 7.800 | 9 |
| – | – | – | 0.032 | – | – | – | – | – | 7.825 | 10 |
| – | – | – | – | 0 | – | – | – | – | 3.542 | 11 |
| – | – | – | – | 2.0 | – | – | – | – | 3.542 | 12 |
| – | – | – | – | – | – | 256 | – | – | 3.785 | 13 |
| – | – | – | – | – | – | – | 500 | – | 7.725 | 14 |
| – | – | – | – | – | – | – | 1000 | – | 7.729 | 15 |
| – | – | – | – | – | – | – | 4000 | – | 3.587 | 16 |
| – | – | – | – | – | – | – | – | 0 | 3.566 | 17 |
| – | – | – | – | – | – | – | – | 0.2 | 3.589 | 18 |

Table 66: Hyperparameter ablation for Adam-Mini on 130m on 2x Chinchilla Data

| $\beta_1$ | $\beta_2$ | $\epsilon$ | $\eta$ | $g_{\mathrm{norm}}$ | $\eta_{min}$ | BSZ | warmup | $\lambda$ | Loss | Link |
|---|---|---|---|---|---|---|---|---|---|---|
| 0.9 | 0.98 | 1e-20 | 0.008 | 2 | 0 | 128 | 2000 | 0.1 | 3.416 | 0 |
| 0.95 | – | – | – | – | – | – | – | – | 3.429 | 1 |
| 0.98 | – | – | – | – | – | – | – | – | 7.520 | 2 |
| – | 0.9 | – | – | – | – | – | – | – | 3.422 | 3 |
| – | 0.95 | – | – | – | – | – | – | – | 3.419 | 4 |
| – | – | 1e-25 | – | – | – | – | – | – | 3.416 | 5 |
| – | – | 1e-15 | – | – | – | – | – | – | 3.416 | 6 |
| – | – | 1e-10 | – | – | – | – | – | – | 3.415 | 7 |
| – | – | – | 0.004 | – | – | – | – | – | 3.425 | 8 |
| – | – | – | 0.016 | – | – | – | – | – | 7.796 | 9 |
| – | – | – | 0.032 | – | – | – | – | – | 7.721 | 10 |
| – | – | – | – | 0 | – | – | – | – | 3.416 | 11 |
| – | – | – | – | 1.0 | – | – | – | – | 3.416 | 12 |
| – | – | – | – | – | – | 256 | – | – | 3.487 | 13 |
| – | – | – | – | – | – | 512 | – | – | 3.758 | 14 |
| – | – | – | – | – | – | – | 500 | – | 7.617 | 15 |
| – | – | – | – | – | – | – | 1000 | – | 7.445 | 16 |
| – | – | – | – | – | – | – | 4000 | – | 3.424 | 17 |
| – | – | – | – | – | – | – | – | 0 | 3.447 | 18 |
| – | – | – | – | – | – | – | – | 0.2 | 3.426 | 19 |

Table 67: Hyperparameter ablation for Adam-Mini on 130m on 4x Chinchilla Data

| $\beta_1$ | $\beta_2$ | $\epsilon$ | $\eta$ | $g_{\mathrm{norm}}$ | $\eta_{min}$ | BSZ | warmup | $\lambda$ | Loss | Link |
|---|---|---|---|---|---|---|---|---|---|---|
| 0.9 | 0.98 | 1e-10 | 0.008 | 1 | 0 | 128 | 2000 | 0.1 | 3.328 | 0 |
| 0.95 | – | – | – | – | – | – | – | – | 3.360 | 1 |
| 0.98 | – | – | – | – | – | – | – | – | 7.771 | 2 |
| – | 0.9 | – | – | – | – | – | – | – | 3.337 | 3 |
| – | 0.95 | – | – | – | – | – | – | – | 3.331 | 4 |
| – | – | 1e-25 | – | – | – | – | – | – | 3.331 | 5 |
| – | – | 1e-20 | – | – | – | – | – | – | 3.331 | 6 |
| – | – | 1e-15 | – | – | – | – | – | – | 3.334 | 7 |
| – | – | – | 0.004 | – | – | – | – | – | 3.334 | 8 |
| – | – | – | 0.016 | – | – | – | – | – | 7.717 | 9 |
| – | – | – | 0.032 | – | – | – | – | – | 7.652 | 10 |
| – | – | – | – | 0 | – | – | – | – | 3.329 | 11 |
| – | – | – | – | 2.0 | – | – | – | – | 3.329 | 12 |
| – | – | – | – | – | – | 256 | – | – | 3.363 | 13 |
| – | – | – | – | – | – | 512 | – | – | 3.447 | 14 |
| – | – | – | – | – | – | 1024 | – | – | 3.784 | 15 |
| – | – | – | – | – | – | – | 500 | – | 7.855 | 16 |
| – | – | – | – | – | – | – | 1000 | – | 7.411 | 17 |
| – | – | – | – | – | – | – | 4000 | – | 3.331 | 18 |
| – | – | – | – | – | – | – | – | 0 | 3.364 | 19 |
| – | – | – | – | – | – | – | – | 0.2 | 3.365 | 20 |

Table 68: Hyperparameter ablation for Adam-Mini on 130m on 8x Chinchilla Data

| $\beta_1$ | $\beta_2$ | $\epsilon$ | $\eta$ | $g_{\mathrm{norm}}$ | $\eta_{min}$ | BSZ | warmup | $\lambda$ | Loss | Link |
|---|---|---|---|---|---|---|---|---|---|---|
| 0.9 | 0.98 | 1e-10 | 0.008 | 1 | 0 | 128 | 2000 | 0.1 | 3.266 | 0 |
| 0.95 | – | – | – | – | – | – | – | – | 3.290 | 1 |
| 0.98 | – | – | – | – | – | – | – | – | 7.552 | 2 |
| – | 0.9 | – | – | – | – | – | – | – | 3.281 | 3 |
| – | 0.95 | – | – | – | – | – | – | – | 3.267 | 4 |
| – | – | 1e-25 | – | – | – | – | – | – | 3.267 | 5 |
| – | – | 1e-20 | – | – | – | – | – | – | 3.267 | 6 |
| – | – | 1e-15 | – | – | – | – | – | – | 3.266 | 7 |
| – | – | – | 0.004 | – | – | – | – | – | 3.264 | 8 |
| – | – | – | 0.016 | – | – | – | – | – | 7.614 | 9 |
| – | – | – | 0.032 | – | – | – | – | – | 7.773 | 10 |
| – | – | – | – | 0 | – | – | – | – | 3.268 | 11 |
| – | – | – | – | 2.0 | – | – | – | – | 3.270 | 12 |
| – | – | – | – | – | – | 256 | – | – | 3.281 | 13 |
| – | – | – | – | – | – | 512 | – | – | 3.324 | 14 |
| – | – | – | – | – | – | 1024 | – | – | 3.426 | 15 |
| – | – | – | – | – | – | – | 500 | – | 7.868 | 16 |
| – | – | – | – | – | – | – | 1000 | – | 7.022 | 17 |
| – | – | – | – | – | – | – | 4000 | – | 3.266 | 18 |
| – | – | – | – | – | – | – | – | 0 | 3.304 | 19 |
| – | – | – | – | – | – | – | – | 0.2 | 3.291 | 20 |

Table 69: Hyperparameter ablation for Adam-Mini on 300m on 1x Chinchilla Data

| $\beta_1$ | $\beta_2$ | $\epsilon$ | $\eta$ | $g_{\mathrm{norm}}$ | $\eta_{min}$ | BSZ | warmup | $\lambda$ | Loss | Link |
|---|---|---|---|---|---|---|---|---|---|---|
| 0.9 | 0.98 | 1e-25 | 0.004 | 2 | 0 | 128 | 2000 | 0.2 | 3.272 | 0 |
| 0.95 | – | – | – | – | – | – | – | – | 3.276 | 1 |
| 0.98 | – | – | – | – | – | – | – | – | 8.024 | 2 |
| – | 0.9 | – | – | – | – | – | – | – | 3.279 | 3 |
| – | 0.95 | – | – | – | – | – | – | – | 3.277 | 4 |
| – | – | 1e-25 | – | – | – | – | – | – | 3.272 | 5 |
| – | – | 1e-20 | – | – | – | – | – | – | 3.272 | 6 |
| – | – | 1e-15 | – | – | – | – | – | – | 3.273 | 7 |
| – | – | 1e-10 | – | – | – | – | – | – | 3.273 | 8 |
| – | – | – | 0.008 | – | – | – | – | – | 7.859 | 9 |
| – | – | – | 0.016 | – | – | – | – | – | 7.990 | 10 |
| – | – | – | 0.032 | – | – | – | – | – | 8.581 | 11 |
| – | – | – | – | 0 | – | – | – | – | 3.272 | 12 |
| – | – | – | – | 1.0 | – | – | – | – | 3.272 | 13 |
| – | – | – | – | – | – | 256 | – | – | 5.691 | 14 |
| – | – | – | – | – | – | 512 | – | – | 3.629 | 15 |
| – | – | – | – | – | – | – | 500 | – | 7.020 | 16 |
| – | – | – | – | – | – | – | 1000 | – | 7.042 | 17 |
| – | – | – | – | – | – | – | 4000 | – | 3.280 | 18 |
| – | – | – | – | – | – | – | – | 0 | 3.336 | 19 |
| – | – | – | – | – | – | – | – | 0.1 | 3.280 | 20 |

Table 70: Hyperparameter ablation for Adam-Mini on 520m on 1x Chinchilla Data

| $\beta_1$ | $\beta_2$ | $\epsilon$ | $\eta$ | $g_{\mathrm{norm}}$ | $\eta_{min}$ | BSZ | warmup | $\lambda$ | Loss | Link |
|---|---|---|---|---|---|---|---|---|---|---|
| 0.9 | 0.98 | 1e-10 | 0.004 | 0 | 0 | 128 | 4000 | 0.1 | 3.112 | 0 |
| 0.95 | – | – | – | – | – | – | – | – | 3.113 | 1 |
| 0.98 | – | – | – | – | – | – | – | – | 7.459 | 2 |
| – | 0.9 | – | – | – | – | – | – | – | 3.120 | 3 |
| – | 0.95 | – | – | – | – | – | – | – | 3.115 | 4 |
| – | – | 1e-25 | – | – | – | – | – | – | 3.115 | 5 |
| – | – | 1e-20 | – | – | – | – | – | – | 3.115 | 6 |
| – | – | 1e-15 | – | – | – | – | – | – | 3.112 | 7 |
| – | – | – | 0.008 | – | – | – | – | – | 7.771 | 8 |
| – | – | – | 0.016 | – | – | – | – | – | 7.746 | 9 |
| – | – | – | 0.032 | – | – | – | – | – | 7.778 | 10 |
| – | – | – | – | 1.0 | – | – | – | – | 3.112 | 11 |
| – | – | – | – | 2.0 | – | – | – | – | 3.112 | 12 |
| – | – | – | – | – | – | 256 | – | – | 3.133 | 13 |
| – | – | – | – | – | – | 512 | – | – | 3.200 | 14 |
| – | – | – | – | – | – | – | 500 | – | 7.457 | 15 |
| – | – | – | – | – | – | – | 1000 | – | 7.274 | 16 |
| – | – | – | – | – | – | – | 2000 | – | 7.289 | 17 |
| – | – | – | – | – | – | – | – | 0 | 7.792 | 18 |
| – | – | – | – | – | – | – | – | 0.2 | 3.115 | 19 |

## E.7 SWEEPING RESULTS FOR KRON

Table 71: Hyperparameter ablation for Kron on 130m on 1x Chinchilla Data

| $\beta_1$ | blocksize | $\eta$ | $g_{\mathrm{norm}}$ | $\eta_{min}$ | NormGrad | Blocking | $Init_{pc}$ | $\eta_{pc}$ | $p_{pc}$ | BSZ | $Step_{pc}$ | warmup | $\lambda$ | Loss | Link |
|---|---|---|---|---|---|---|---|---|---|---|---|---|---|---|---|
| 0.95 | 256 | 0.002 | 1 | 0 | True | True | 1 | 0.2 | 0.05 | 128 | 2000 | 1000 | 0.7 | 3.492 | 0 |
| 0.9 | – | – | – | – | – | – | – | – | – | – | – | – | – | 3.500 | 1 |
| 0.98 | – | – | – | – | – | – | – | – | – | – | – | – | – | 3.497 | 2 |
| – | 128 | – | – | – | – | – | – | – | – | – | – | – | – | 3.492 | 3 |
| – | 512 | – | – | – | – | – | – | – | – | – | – | – | – | 3.494 | 4 |
| – | – | 0.0005 | – | – | – | – | – | – | – | – | – | – | – | 3.528 | 5 |
| – | – | 0.001 | – | – | – | – | – | – | – | – | – | – | – | 3.501 | 6 |
| – | – | 0.004 | – | – | – | – | – | – | – | – | – | – | – | 3.514 | 7 |
| – | – | 0.008 | – | – | – | – | – | – | – | – | – | – | – | 7.838 | 8 |
| – | – | – | 0.0 | – | – | – | – | – | – | – | – | – | – | 3.492 | 9 |
| – | – | – | 2.0 | – | – | – | – | – | – | – | – | – | – | 3.492 | 10 |
| – | – | – | – | – | False | – | – | – | – | – | – | – | – | 3.493 | 11 |
| – | – | – | – | – | – | – | – | 0.1 | – | – | – | – | – | 3.498 | 12 |
| – | – | – | – | – | – | – | – | – | 0.1 | – | – | – | – | 3.493 | 13 |
| – | – | – | – | – | – | – | – | – | – | 256 | – | – | – | 3.525 | 14 |
| – | – | – | – | – | – | – | – | – | – | 512 | – | – | – | 3.632 | 15 |
| – | – | – | – | – | – | – | – | – | – | – | 500 | – | – | 3.503 | 16 |
| – | – | – | – | – | – | – | – | – | – | – | 1000 | – | – | 3.498 | 17 |
| – | – | – | – | – | – | – | – | – | – | – | – | 2000 | – | 3.507 | 18 |
| – | – | – | – | – | – | – | – | – | – | – | – | 4000 | – | 3.583 | 19 |
| – | – | – | – | – | – | – | – | – | – | – | – | – | 0.0 | 3.519 | 20 |
| – | – | – | – | – | – | – | – | – | – | – | – | – | 0.5 | 3.491 | 21 |
| – | – | – | – | – | – | – | – | – | – | – | – | – | 0.9 | 3.495 | 22 |

Table 72: Hyperparameter ablation for Kron on 130m on 2x Chinchilla Data

| $\beta_1$ | blocksize | $\eta$ | $g_{\mathrm{norm}}$ | $\eta_{min}$ | NormGrad | Blocking | $Init_{pc}$ | $\eta_{pc}$ | $p_{pc}$ | BSZ | $Step_{pc}$ | warmup | $\lambda$ | Loss | Link |
|---|---|---|---|---|---|---|---|---|---|---|---|---|---|---|---|
| 0.95 | 256 | 0.002 | 1 | 0 | True | True | 1 | 0.2 | 0.05 | 128 | 2000 | 1000 | 0.5 | 3.389 | 0 |
| 0.9 | – | – | – | – | – | – | – | – | – | – | – | – | – | 3.393 | 1 |
| 0.98 | – | – | – | – | – | – | – | – | – | – | – | – | – | 3.391 | 2 |
| – | 512 | – | – | – | – | – | – | – | – | – | – | – | – | 3.390 | 3 |
| – | – | 0.0005 | – | – | – | – | – | – | – | – | – | – | – | 3.409 | 4 |
| – | – | 0.001 | – | – | – | – | – | – | – | – | – | – | – | 3.391 | 5 |
| – | – | 0.004 | – | – | – | – | – | – | – | – | – | – | – | 3.410 | 6 |
| – | – | 0.008 | – | – | – | – | – | – | – | – | – | – | – | 7.458 | 7 |
| – | – | – | 0.0 | – | – | – | – | – | – | – | – | – | – | 3.389 | 8 |
| – | – | – | 2.0 | – | – | – | – | – | – | – | – | – | – | 3.389 | 9 |
| – | – | – | – | – | False | – | – | – | – | – | – | – | – | 3.419 | 10 |
| – | – | – | – | – | – | – | – | 0.1 | – | – | – | – | – | 3.390 | 11 |
| – | – | – | – | – | – | – | – | – | 0.1 | – | – | – | – | 3.388 | 12 |
| – | – | – | – | – | – | – | – | – | – | 256 | – | – | – | 3.402 | 13 |
| – | – | – | – | – | – | – | – | – | – | 512 | – | – | – | 3.438 | 14 |
| – | – | – | – | – | – | – | – | – | – | 1024 | – | – | – | 3.532 | 15 |
| – | – | – | – | – | – | – | – | – | – | – | 500 | – | – | 3.393 | 16 |
| – | – | – | – | – | – | – | – | – | – | – | 1000 | – | – | 3.391 | 17 |
| – | – | – | – | – | – | – | – | – | – | – | – | 2000 | – | 3.392 | 18 |
| – | – | – | – | – | – | – | – | – | – | – | – | 4000 | – | 3.407 | 19 |
| – | – | – | – | – | – | – | – | – | – | – | – | – | 0.0 | 3.420 | 20 |
| – | – | – | – | – | – | – | – | – | – | – | – | – | 0.7 | 3.392 | 21 |
| – | – | – | – | – | – | – | – | – | – | – | – | – | 0.9 | 3.401 | 22 |

Table 73: Hyperparameter ablation for Kron on 130m on 4x Chinchilla Data

| $\beta_1$ | blocksize | $\eta$ | $g_{\mathrm{norm}}$ | $\eta_{min}$ | NormGrad | Blocking | $Init_{pc}$ | $\eta_{pc}$ | $p_{pc}$ | BSZ | $Step_{pc}$ | warmup | $\lambda$ | Loss | Link |
|---|---|---|---|---|---|---|---|---|---|---|---|---|---|---|---|
| 0.95 | 256 | 0.002 | 1 | 0 | True | True | 1 | 0.2 | 0.05 | 128 | 2000 | 1000 | 0.5 | 3.307 | 0 |
| 0.9 | – | – | – | – | – | – | – | – | – | – | – | – | – | 3.316 | 1 |
| 0.98 | – | – | – | – | – | – | – | – | – | – | – | – | – | 3.310 | 2 |
| – | 128 | – | – | – | – | – | – | – | – | – | – | – | – | 3.308 | 3 |
| – | 512 | – | – | – | – | – | – | – | – | – | – | – | – | 3.313 | 4 |
| – | – | 0.0005 | – | – | – | – | – | – | – | – | – | – | – | 3.316 | 5 |
| – | – | 0.001 | – | – | – | – | – | – | – | – | – | – | – | 3.303 | 6 |
| – | – | 0.004 | – | – | – | – | – | – | – | – | – | – | – | 6.703 | 7 |
| – | – | 0.008 | – | – | – | – | – | – | – | – | – | – | – | 7.492 | 8 |
| – | – | – | 0.0 | – | – | – | – | – | – | – | – | – | – | 3.307 | 9 |
| – | – | – | 2.0 | – | – | – | – | – | – | – | – | – | – | 3.307 | 10 |
| – | – | – | – | – | False | – | – | – | – | – | – | – | – | 5.676 | 11 |
| – | – | – | – | – | – | – | – | 0.1 | – | – | – | – | – | 3.311 | 12 |
| – | – | – | – | – | – | – | – | – | 0.1 | – | – | – | – | 3.304 | 13 |
| – | – | – | – | – | – | – | – | – | – | 256 | – | – | – | 3.307 | 14 |
| – | – | – | – | – | – | – | – | – | – | 512 | – | – | – | 3.327 | 15 |
| – | – | – | – | – | – | – | – | – | – | 1024 | – | – | – | 3.370 | 16 |
| – | – | – | – | – | – | – | – | – | – | – | 500 | – | – | 3.311 | 17 |
| – | – | – | – | – | – | – | – | – | – | – | 1000 | – | – | 3.310 | 18 |
| – | – | – | – | – | – | – | – | – | – | – | – | 2000 | – | 3.310 | 19 |
| – | – | – | – | – | – | – | – | – | – | – | – | 4000 | – | 3.317 | 20 |
| – | – | – | – | – | – | – | – | – | – | – | – | – | 0.0 | 3.341 | 21 |
| – | – | – | – | – | – | – | – | – | – | – | – | – | 0.7 | 3.326 | 22 |
| – | – | – | – | – | – | – | – | – | – | – | – | – | 0.9 | 3.340 | 23 |

Table 74: Hyperparameter ablation for Kron on 130m on 8x Chinchilla Data

| $\beta_1$ | blocksize | $\eta$ | $g_{norm}$ | $\eta_{min}$ | NormGrad | Blocking | $Init_{pc}$ | $\eta_{pc}$ | $p_{pc}$ | BSZ | $Step_{pc}$ | warmup | $\lambda$ | Loss | Link |
|---|---|---|---|---|---|---|---|---|---|---|---|---|---|---|---|
| 0.95 | 256 | 0.001 | 1 | 0 | True | True | 1 | 0.2 | 0.1 | 128 | 2000 | 1000 | 0.5 | 3.239 | 0 |
| 0.9 | – | – | – | – | – | – | – | – | – | – | – | – | – | 3.243 | 1 |
| 0.98 | – | – | – | – | – | – | – | – | – | – | – | – | – | 3.239 | 2 |
| – | 512 | – | – | – | – | – | – | – | – | – | – | – | – | 3.239 | 3 |
| – | – | 0.0005 | – | – | – | – | – | – | – | – | – | – | – | 3.245 | 4 |
| – | – | 0.002 | – | – | – | – | – | – | – | – | – | – | – | 3.252 | 5 |
| – | – | 0.004 | – | – | – | – | – | – | – | – | – | – | – | 6.279 | 6 |
| – | – | 0.008 | – | – | – | – | – | – | – | – | – | – | – | 7.504 | 7 |
| – | – | – | 0.0 | – | – | – | – | – | – | – | – | – | – | 3.239 | 8 |
| – | – | – | 2.0 | – | – | – | – | – | – | – | – | – | – | 3.239 | 9 |
| – | – | – | – | – | False | – | – | – | – | – | – | – | – | 3.247 | 10 |
| – | – | – | – | – | – | False | – | – | – | – | – | – | – | 3.240 | 11 |
| – | – | – | – | – | – | – | – | 0.1 | – | – | – | – | – | 3.242 | 12 |
| – | – | – | – | – | – | – | – | – | 0.05 | – | – | – | – | 3.240 | 13 |
| – | – | – | – | – | – | – | – | – | – | 256 | – | – | – | 3.247 | 14 |
| – | – | – | – | – | – | – | – | – | – | 512 | – | – | – | 3.264 | 15 |
| – | – | – | – | – | – | – | – | – | – | 1024 | – | – | – | 3.294 | 16 |
| – | – | – | – | – | – | – | – | – | – | – | 500 | – | – | 3.240 | 17 |
| – | – | – | – | – | – | – | – | – | – | – | 1000 | – | – | 3.239 | 18 |
| – | – | – | – | – | – | – | – | – | – | – | – | 2000 | – | 3.239 | 19 |
| – | – | – | – | – | – | – | – | – | – | – | – | 4000 | – | 3.240 | 20 |
| – | – | – | – | – | – | – | – | – | – | – | – | – | 0.0 | 3.282 | 21 |
| – | – | – | – | – | – | – | – | – | – | – | – | – | 0.7 | 3.241 | 22 |
| – | – | – | – | – | – | – | – | – | – | – | – | – | 0.9 | 3.246 | 23 |

Table 75: Hyperparameter ablation for Kron on 300m on 1x Chinchilla Data

| $\beta_1$ | blocksize | $\eta$ | $g_{norm}$ | $\eta_{min}$ | NormGrad | Blocking | $Init_{pc}$ | $\eta_{pc}$ | $p_{pc}$ | BSZ | $Step_{pc}$ | warmup | $\lambda$ | Loss | Link |
|---|---|---|---|---|---|---|---|---|---|---|---|---|---|---|---|
| 0.95 | 256 | 0.001 | 1 | 0 | True | True | 1 | 0.2 | 0.1 | 128 | 2000 | 1000 | 0.5 | 3.244 | 0 |
| 0.9 | – | – | – | – | – | – | – | – | – | – | – | – | – | 3.248 | 1 |
| 0.98 | – | – | – | – | – | – | – | – | – | – | – | – | – | 3.244 | 2 |
| – | 512 | – | – | – | – | – | – | – | – | – | – | – | – | 3.243 | 3 |
| – | – | 0.0005 | – | – | – | – | – | – | – | – | – | – | – | 3.263 | 4 |
| – | – | 0.002 | – | – | – | – | – | – | – | – | – | – | – | 3.245 | 5 |
| – | – | 0.004 | – | – | – | – | – | – | – | – | – | – | – | 7.636 | 6 |
| – | – | 0.008 | – | – | – | – | – | – | – | – | – | – | – | 6.923 | 7 |
| – | – | – | 0.0 | – | – | – | – | – | – | – | – | – | – | 3.244 | 8 |
| – | – | – | 2.0 | – | – | – | – | – | – | – | – | – | – | 3.244 | 9 |
| – | – | – | – | – | False | – | – | – | – | – | – | – | – | 3.253 | 10 |
| – | – | – | – | – | – | – | – | 0.1 | – | – | – | – | – | 3.247 | 11 |
| – | – | – | – | – | – | – | – | – | 0.05 | – | – | – | – | 3.245 | 12 |
| – | – | – | – | – | – | – | – | – | – | 256 | – | – | – | 3.268 | 13 |
| – | – | – | – | – | – | – | – | – | – | 512 | – | – | – | 3.309 | 14 |
| – | – | – | – | – | – | – | – | – | – | 1024 | – | – | – | 3.393 | 15 |
| – | – | – | – | – | – | – | – | – | – | – | 500 | – | – | 3.248 | 16 |
| – | – | – | – | – | – | – | – | – | – | – | 1000 | – | – | 3.245 | 17 |
| – | – | – | – | – | – | – | – | – | – | – | – | 2000 | – | 3.249 | 18 |
| – | – | – | – | – | – | – | – | – | – | – | – | 4000 | – | 3.255 | 19 |
| – | – | – | – | – | – | – | – | – | – | – | – | – | 0.0 | 3.280 | 20 |
| – | – | – | – | – | – | – | – | – | – | – | – | – | 0.7 | 3.241 | 21 |
| – | – | – | – | – | – | – | – | – | – | – | – | – | 0.9 | 3.243 | 22 |

Table 76: Hyperparameter ablation for Kron on 520m on 1x Chinchilla Data

| $\beta_1$ | blocksize | $\eta$ | $g_{\mathrm{norm}}$ | $\eta_{min}$ | NormGrad | Blocking | $Init_{pc}$ | $\eta_{pc}$ | $p_{pc}$ | BSZ | $Step_{pc}$ | warmup | $\lambda$ | Loss | Link |
|---|---|---|---|---|---|---|---|---|---|---|---|---|---|---|---|
| 0.95 | 256 | 0.001 | 1 | 0 | True | True | 1 | 0.2 | 0.1 | 128 | 2000 | 1000 | 0.5 | 3.084 | 0 |
| 0.9 | – | – | – | – | – | – | – | – | – | – | – | – | – | 3.088 | 1 |
| 0.98 | – | – | – | – | – | – | – | – | – | – | – | – | – | 6.663 | 2 |
| – | 512 | – | – | – | – | – | – | – | – | – | – | – | – | 3.087 | 3 |
| – | – | 0.0005 | – | – | – | – | – | – | – | – | – | – | – | 3.095 | 4 |
| – | – | 0.002 | – | – | – | – | – | – | – | – | – | – | – | 6.412 | 5 |
| – | – | 0.004 | – | – | – | – | – | – | – | – | – | – | – | 6.927 | 6 |
| – | – | 0.008 | – | – | – | – | – | – | – | – | – | – | – | 7.018 | 7 |
| – | – | – | 0.0 | – | – | – | – | – | – | – | – | – | – | 3.084 | 8 |
| – | – | – | 2.0 | – | – | – | – | – | – | – | – | – | – | 3.084 | 9 |
| – | – | – | – | – | False | – | – | – | – | – | – | – | – | 5.487 | 10 |
| – | – | – | – | – | – | – | – | 0.1 | – | – | – | – | – | 3.088 | 11 |
| – | – | – | – | – | – | – | – | – | 0.05 | – | – | – | – | 3.093 | 12 |
| – | – | – | – | – | – | – | – | – | – | 256 | – | – | – | 3.099 | 13 |
| – | – | – | – | – | – | – | – | – | – | 512 | – | – | – | 3.126 | 14 |
| – | – | – | – | – | – | – | – | – | – | – | 500 | – | – | 3.087 | 15 |
| – | – | – | – | – | – | – | – | – | – | – | 1000 | – | – | 3.085 | 16 |
| – | – | – | – | – | – | – | – | – | – | – | – | 2000 | – | 3.090 | 17 |
| – | – | – | – | – | – | – | – | – | – | – | – | 4000 | – | 3.094 | 18 |
| – | – | – | – | – | – | – | – | – | – | – | – | – | 0.0 | 3.127 | 19 |
| – | – | – | – | – | – | – | – | – | – | – | – | – | 0.7 | 3.088 | 20 |
| – | – | – | – | – | – | – | – | – | – | – | – | – | 0.9 | 3.089 | 21 |

### E.8 SWEEPING RESULTS FOR SOAP

Table 77: Hyperparameter ablation for Soap on 130m on 1x Chinchilla Data

| $\beta_1$ | $\beta_2$ | blocksize | $\epsilon$ | $\eta$ | $g_{\mathrm{norm}}$ | $\eta_{min}$ | Blocking | $f_{pc}$ | $\beta_{shampoo}$ | BSZ | warmup | $\lambda$ | Loss | Link |
|---|---|---|---|---|---|---|---|---|---|---|---|---|---|---|
| 0.95 | 0.98 | 256 | 1e-15 | 0.016 | 1 | 0 | True | 1 | 0.95 | 128 | 1000 | 0.1 | 3.483 | 0 |
| 0.8 | – | – | – | – | – | – | – | – | – | – | – | – | 4.868 | 1 |
| 0.9 | – | – | – | – | – | – | – | – | – | – | – | – | 4.547 | 2 |
| – | 0.9 | – | – | – | – | – | – | – | – | – | – | – | 3.496 | 3 |
| – | 0.95 | – | – | – | – | – | – | – | – | – | – | – | 3.491 | 4 |
| – | 0.99 | – | – | – | – | – | – | – | – | – | – | – | 3.482 | 5 |
| – | – | 128 | – | – | – | – | – | – | – | – | – | – | 3.489 | 6 |
| – | – | 512 | – | – | – | – | – | – | – | – | – | – | 3.487 | 7 |
| – | – | – | 1e-20 | – | – | – | – | – | – | – | – | – | 3.513 | 8 |
| – | – | – | 1e-10 | – | – | – | – | – | – | – | – | – | 3.483 | 9 |
| – | – | – | – | 0.004 | – | – | – | – | – | – | – | – | 3.509 | 10 |
| – | – | – | – | 0.008 | – | – | – | – | – | – | – | – | 3.491 | 11 |
| – | – | – | – | – | – | – | – | 5 | – | – | – | – | 3.488 | 12 |
| – | – | – | – | – | – | – | – | – | 0.9 | – | – | – | 3.488 | 13 |
| – | – | – | – | – | – | – | – | – | 0.98 | – | – | – | 3.489 | 14 |
| – | – | – | – | – | – | – | – | – | 0.99 | – | – | – | 3.491 | 15 |
| – | – | – | – | – | – | – | – | – | – | 256 | – | – | 3.523 | 16 |
| – | – | – | – | – | – | – | – | – | – | 512 | – | – | 3.611 | 17 |
| – | – | – | – | – | – | – | – | – | – | – | 500 | – | 3.483 | 18 |
| – | – | – | – | – | – | – | – | – | – | – | 2000 | – | 3.507 | 19 |
| – | – | – | – | – | – | – | – | – | – | – | – | 0 | 3.508 | 20 |
| – | – | – | – | – | – | – | – | – | – | – | – | 0.2 | 3.501 | 21 |

Table 78: Hyperparameter ablation for Soap on 130m on 2x Chinchilla Data

| $\beta_1$ | $\beta_2$ | blocksize | $\epsilon$ | $\eta$ | $g_{\mathrm{norm}}$ | $\eta_{min}$ | Blocking | $f_{pc}$ | $\beta_{shampoo}$ | BSZ | warmup | $\lambda$ | Loss | Link |
|------|------|------|------|------|------|------|------|------|------|------|------|------|------|------|
| 0.95 | 0.99 | 256 | 1e-15 | 0.016 | 1 | 0 | False | 1 | 0.98 | 128 | 500 | 0.1 | 3.376 | 0 |
| 0.8 | – | – | – | – | – | – | – | – | – | – | – | – | 5.163 | 1 |
| 0.9 | – | – | – | – | – | – | – | – | – | – | – | – | 3.397 | 2 |
| – | 0.9 | – | – | – | – | – | – | – | – | – | – | – | 3.393 | 3 |
| – | 0.95 | – | – | – | – | – | – | – | – | – | – | – | 3.384 | 4 |
| – | 0.98 | – | – | – | – | – | – | – | – | – | – | – | 3.380 | 5 |
| – | – | 128 | – | – | – | – | – | – | – | – | – | – | 3.376 | 6 |
| – | – | 512 | – | – | – | – | – | – | – | – | – | – | 3.376 | 7 |
| – | – | – | 1e-20 | – | – | – | – | – | – | – | – | – | 3.384 | 8 |
| – | – | – | 1e-10 | – | – | – | – | – | – | – | – | – | 3.375 | 9 |
| – | – | – | – | 0.004 | – | – | – | – | – | – | – | – | 3.388 | 10 |
| – | – | – | – | 0.008 | – | – | – | – | – | – | – | – | 3.374 | 11 |
| – | – | – | – | – | – | – | – | 10 | – | – | – | – | 3.381 | 12 |
| – | – | – | – | – | – | – | – | – | 0.9 | – | – | – | 3.378 | 13 |
| – | – | – | – | – | – | – | – | – | 0.95 | – | – | – | 3.376 | 14 |
| – | – | – | – | – | – | – | – | – | 0.99 | – | – | – | 3.379 | 15 |
| – | – | – | – | – | – | – | – | – | – | 256 | – | – | 3.385 | 16 |
| – | – | – | – | – | – | – | – | – | – | 512 | – | – | 3.414 | 17 |
| – | – | – | – | – | – | – | – | – | – | 1024 | – | – | 3.479 | 18 |
| – | – | – | – | – | – | – | – | – | – | – | 1000 | – | 3.379 | 19 |
| – | – | – | – | – | – | – | – | – | – | – | 2000 | – | 3.385 | 20 |
| – | – | – | – | – | – | – | – | – | – | – | – | 0 | 3.437 | 21 |
| – | – | – | – | – | – | – | – | – | – | – | – | 0.2 | 3.395 | 22 |

Table 79: Hyperparameter ablation for Soap on 130m on 4x Chinchilla Data

| $\beta_1$ | $\beta_2$ | blocksize | $\epsilon$ | $\eta$ | $g_{\mathrm{norm}}$ | $\eta_{min}$ | Blocking | $f_{pc}$ | $\beta_{shampoo}$ | BSZ | warmup | $\lambda$ | Loss | Link |
|------|------|------|------|------|------|------|------|------|------|------|------|------|------|------|
| 0.95 | 0.99 | 256 | 1e-15 | 0.008 | 1 | 0 | False | 1 | 0.98 | 128 | 500 | 0.1 | 3.295 | 0 |
| 0.8 | – | – | – | – | – | – | – | – | – | – | – | – | 3.328 | 1 |
| 0.9 | – | – | – | – | – | – | – | – | – | – | – | – | 3.302 | 2 |
| – | 0.9 | – | – | – | – | – | – | – | – | – | – | – | 3.312 | 3 |
| – | 0.95 | – | – | – | – | – | – | – | – | – | – | – | 3.303 | 4 |
| – | 0.98 | – | – | – | – | – | – | – | – | – | – | – | 3.298 | 5 |
| – | – | 128 | – | – | – | – | – | – | – | – | – | – | 3.295 | 6 |
| – | – | 512 | – | – | – | – | – | – | – | – | – | – | 3.295 | 7 |
| – | – | – | 1e-20 | – | – | – | – | – | – | – | – | – | 3.297 | 8 |
| – | – | – | 1e-10 | – | – | – | – | – | – | – | – | – | 3.295 | 9 |
| – | – | – | – | 0.004 | – | – | – | – | – | – | – | – | 3.303 | 10 |
| – | – | – | – | 0.016 | – | – | – | – | – | – | – | – | 3.303 | 11 |
| – | – | – | – | – | – | – | – | 10 | – | – | – | – | 3.299 | 12 |
| – | – | – | – | – | – | – | – | – | 0.9 | – | – | – | 3.296 | 13 |
| – | – | – | – | – | – | – | – | – | 0.95 | – | – | – | 3.294 | 14 |
| – | – | – | – | – | – | – | – | – | 0.99 | – | – | – | 3.297 | 15 |
| – | – | – | – | – | – | – | – | – | – | 256 | – | – | 3.305 | 16 |
| – | – | – | – | – | – | – | – | – | – | 512 | – | – | 3.325 | 17 |
| – | – | – | – | – | – | – | – | – | – | 1024 | – | – | 3.358 | 18 |
| – | – | – | – | – | – | – | – | – | – | – | 1000 | – | 3.298 | 19 |
| – | – | – | – | – | – | – | – | – | – | – | 2000 | – | 3.300 | 20 |
| – | – | – | – | – | – | – | – | – | – | – | – | 0 | 3.370 | 21 |
| – | – | – | – | – | – | – | – | – | – | – | – | 0.2 | 3.304 | 22 |

Table 80: Hyperparameter ablation for Soap on 130m on 8x Chinchilla Data

| $\beta_1$ | $\beta_2$ | blocksize | $\epsilon$ | $\eta$ | $g_{\mathrm{norm}}$ | $\eta_{min}$ | Blocking | $f_{pc}$ | $\beta_{shampoo}$ | BSZ | warmup | $\lambda$ | Loss | Link |
|---|---|---|---|---|---|---|---|---|---|---|---|---|---|---|
| 0.95 | 0.99 | 512 | 1e-15 | 0.008 | 1 | 0 | True | 10 | 0.98 | 256 | 1000 | 0.1 | 3.239 | 0 |
| 0.8 | – | – | – | – | – | – | – | – | – | – | – | – | 3.298 | 1 |
| 0.9 | – | – | – | – | – | – | – | – | – | – | – | – | 3.250 | 2 |
| – | 0.9 | – | – | – | – | – | – | – | – | – | – | – | 3.251 | 3 |
| – | 0.95 | – | – | – | – | – | – | – | – | – | – | – | 3.244 | 4 |
| – | 0.98 | – | – | – | – | – | – | – | – | – | – | – | 3.241 | 5 |
| – | – | 128 | – | – | – | – | – | – | – | – | – | – | 3.240 | 6 |
| – | – | 256 | – | – | – | – | – | – | – | – | – | – | 3.242 | 7 |
| – | – | – | 1e-20 | – | – | – | – | – | – | – | – | – | 3.239 | 8 |
| – | – | – | 1e-10 | – | – | – | – | – | – | – | – | – | 3.238 | 9 |
| – | – | – | – | 0.004 | – | – | – | – | – | – | – | – | 3.248 | 10 |
| – | – | – | – | 0.016 | – | – | – | – | – | – | – | – | 3.249 | 11 |
| – | – | – | – | – | – | – | – | – | 0.9 | – | – | – | 3.239 | 12 |
| – | – | – | – | – | – | – | – | – | 0.95 | – | – | – | 3.239 | 13 |
| – | – | – | – | – | – | – | – | – | 0.99 | – | – | – | 3.241 | 14 |
| – | – | – | – | – | – | – | – | – | – | 128 | – | – | 3.242 | 15 |
| – | – | – | – | – | – | – | – | – | – | 512 | – | – | 3.250 | 16 |
| – | – | – | – | – | – | – | – | – | – | 1024 | – | – | 3.276 | 17 |
| – | – | – | – | – | – | – | – | – | – | – | 500 | – | 3.240 | 18 |
| – | – | – | – | – | – | – | – | – | – | – | 2000 | – | 3.240 | 19 |
| – | – | – | – | – | – | – | – | – | – | – | – | 0 | 3.308 | 20 |
| – | – | – | – | – | – | – | – | – | – | – | – | 0.2 | 3.243 | 21 |

Table 81: Hyperparameter ablation for Soap on 300m on 1x Chinchilla Data

| $\beta_1$ | $\beta_2$ | blocksize | $\epsilon$ | $\eta$ | $g_{\mathrm{norm}}$ | $\eta_{min}$ | Blocking | $f_{pc}$ | $\beta_{shampoo}$ | BSZ | warmup | $\lambda$ | Loss | Link |
|---|---|---|---|---|---|---|---|---|---|---|---|---|---|---|
| 0.95 | 0.99 | 512 | 1e-10 | 0.008 | 1 | 0 | True | 10 | 0.9 | 128 | 1000 | 0.1 | 3.231 | 0 |
| 0.8 | – | – | – | – | – | – | – | – | – | – | – | – | 4.544 | 1 |
| 0.9 | – | – | – | – | – | – | – | – | – | – | – | – | 3.251 | 2 |
| – | 0.9 | – | – | – | – | – | – | – | – | – | – | – | 3.247 | 3 |
| – | 0.95 | – | – | – | – | – | – | – | – | – | – | – | 3.240 | 4 |
| – | 0.98 | – | – | – | – | – | – | – | – | – | – | – | 3.234 | 5 |
| – | – | 128 | – | – | – | – | – | – | – | – | – | – | 3.239 | 6 |
| – | – | 256 | – | – | – | – | – | – | – | – | – | – | 3.235 | 7 |
| – | – | – | 1e-20 | – | – | – | – | – | – | – | – | – | 3.233 | 8 |
| – | – | – | 1e-15 | – | – | – | – | – | – | – | – | – | 3.233 | 9 |
| – | – | – | – | 0.004 | – | – | – | – | – | – | – | – | 3.239 | 10 |
| – | – | – | – | 0.016 | – | – | – | – | – | – | – | – | 5.559 | 11 |
| – | – | – | – | – | – | – | – | – | 0.95 | – | – | – | 3.233 | 12 |
| – | – | – | – | – | – | – | – | – | 0.98 | – | – | – | 3.235 | 13 |
| – | – | – | – | – | – | – | – | – | 0.99 | – | – | – | 3.237 | 14 |
| – | – | – | – | – | – | – | – | – | – | 256 | – | – | 3.248 | 15 |
| – | – | – | – | – | – | – | – | – | – | 512 | – | – | 3.283 | 16 |
| – | – | – | – | – | – | – | – | – | – | – | 500 | – | 3.231 | 17 |
| – | – | – | – | – | – | – | – | – | – | – | 2000 | – | 3.235 | 18 |
| – | – | – | – | – | – | – | – | – | – | – | – | 0 | 3.268 | 19 |
| – | – | – | – | – | – | – | – | – | – | – | – | 0.2 | 3.238 | 20 |
| – | – | – | – | – | – | – | – | – | – | – | – | 0.3 | 3.249 | 21 |

Table 82: Hyperparameter ablation for Soap on 520m on 1x Chinchilla Data

| $\beta_1$ | $\beta_2$ | blocksize | $\epsilon$ | $\eta$ | $g_{\mathrm{norm}}$ | $\eta_{min}$ | Blocking | $f_{pc}$ | $\beta_{shampoo}$ | BSZ | warmup | $\lambda$ | Loss | Link |
|---|---|---|---|---|---|---|---|---|---|---|---|---|---|---|
| 0.95 | 0.99 | 512 | 1e-10 | 0.008 | 1 | 0 | True | 10 | 0.95 | 128 | 1000 | 0.1 | 3.079 | 0 |
| 0.8 | – | – | – | – | – | – | – | – | – | – | – | – | 4.630 | 1 |
| 0.9 | – | – | – | – | – | – | – | – | – | – | – | – | 4.316 | 2 |
| – | 0.9 | – | – | – | – | – | – | – | – | – | – | – | 3.097 | 3 |
| – | 0.95 | – | – | – | – | – | – | – | – | – | – | – | 3.090 | 4 |
| – | 0.98 | – | – | – | – | – | – | – | – | – | – | – | 3.085 | 5 |
| – | – | 128 | – | – | – | – | – | – | – | – | – | – | 5.395 | 6 |
| – | – | 256 | – | – | – | – | – | – | – | – | – | – | 4.392 | 7 |
| – | – | – | 1e-20 | – | – | – | – | – | – | – | – | – | 3.082 | 8 |
| – | – | – | 1e-15 | – | – | – | – | – | – | – | – | – | 3.081 | 9 |
| – | – | – | – | 0.004 | – | – | – | – | – | – | – | – | 3.079 | 10 |
| – | – | – | – | 0.016 | – | – | – | – | – | – | – | – | 5.762 | 11 |
| – | – | – | – | – | – | – | – | – | 0.9 | – | – | – | 3.080 | 12 |
| – | – | – | – | – | – | – | – | – | 0.98 | – | – | – | 3.082 | 13 |
| – | – | – | – | – | – | – | – | – | 0.99 | – | – | – | 3.083 | 14 |
| – | – | – | – | – | – | – | – | – | – | 256 | – | – | 3.085 | 15 |
| – | – | – | – | – | – | – | – | – | – | 512 | – | – | 4.215 | 16 |
| – | – | – | – | – | – | – | – | – | – | – | 500 | – | 4.163 | 17 |
| – | – | – | – | – | – | – | – | – | – | – | 2000 | – | 3.081 | 18 |
| – | – | – | – | – | – | – | – | – | – | – | – | 0 | 6.106 | 19 |
| – | – | – | – | – | – | – | – | – | – | – | – | 0.2 | 3.527 | 20 |
| – | – | – | – | – | – | – | – | – | – | – | – | 0.3 | 3.118 | 21 |

### E.9 Sweeping Results for Muon

Table 83: Hyperparameter ablation for Muon on 130m on 1x Chinchilla Data

| $\eta_{adam}$ | $\beta_1$ | $\beta_2$ | Decay(WSD) | $\epsilon$ | $\eta$ | Schedule | $g_{\mathrm{norm}}$ | $\eta_{min}$ | $\beta_{\mathrm{muon}}$ | $\epsilon_{muon}$ | BSZ | warmup | $\lambda$ | Loss | Link |
|---|---|---|---|---|---|---|---|---|---|---|---|---|---|---|---|
| 0.0032 | 0.8 | 0.98 | 0.8 | 1e-15 | 0.016 | linear | 1 | 0 | 0.95 | 1e-05 | 128 | 0 | 0.1 | 3.464 | 0 |
| 0.0016 | – | – | – | – | – | – | – | – | – | – | – | – | – | 3.474 | 1 |
| 0.0048 | – | – | – | – | – | – | – | – | – | – | – | – | – | 3.463 | 2 |
| – | 0.9 | – | – | – | – | – | – | – | – | – | – | – | – | 3.468 | 3 |
| – | 0.95 | – | – | – | – | – | – | – | – | – | – | – | – | 3.467 | 4 |
| – | 0.98 | – | – | – | – | – | – | – | – | – | – | – | – | 3.470 | 5 |
| – | – | 0.9 | – | – | – | – | – | – | – | – | – | – | – | 3.481 | 6 |
| – | – | 0.95 | – | – | – | – | – | – | – | – | – | – | – | 3.469 | 7 |
| – | – | – | 0.2 | – | – | – | – | – | – | – | – | – | – | 3.516 | 8 |
| – | – | – | 0.4 | – | – | – | – | – | – | – | – | – | – | 3.482 | 9 |
| – | – | – | 0.6 | – | – | – | – | – | – | – | – | – | – | 3.468 | 10 |
| – | – | – | 1.0 | – | – | – | – | – | – | – | – | – | – | 3.467 | 11 |
| – | – | – | – | 1e-25 | – | – | – | – | – | – | – | – | – | 3.465 | 12 |
| – | – | – | – | 1e-20 | – | – | – | – | – | – | – | – | – | 3.465 | 13 |
| – | – | – | – | 1e-10 | – | – | – | – | – | – | – | – | – | 3.465 | 14 |
| – | – | – | – | – | 0.008 | – | – | – | – | – | – | – | – | 3.491 | 15 |
| – | – | – | – | – | 0.032 | – | – | – | – | – | – | – | – | 3.484 | 16 |
| – | – | – | – | – | 0.064 | – | – | – | – | – | – | – | – | 3.535 | 17 |
| – | – | – | – | – | 0.128 | – | – | – | – | – | – | – | – | 3.688 | 18 |
| – | – | – | – | – | – | – | 0 | – | – | – | – | – | – | 3.464 | 19 |
| – | – | – | – | – | – | – | 2.0 | – | – | – | – | – | – | 3.464 | 20 |
| – | – | – | – | – | – | – | – | – | 0.8 | – | – | – | – | 3.503 | 21 |
| – | – | – | – | – | – | – | – | – | 0.9 | – | – | – | – | 3.473 | 22 |
| – | – | – | – | – | – | – | – | – | 0.98 | – | – | – | – | 3.474 | 23 |
| – | – | – | – | – | – | – | – | – | – | 1e-25 | – | – | – | 3.465 | 24 |
| – | – | – | – | – | – | – | – | – | – | 1e-20 | – | – | – | 3.465 | 25 |
| – | – | – | – | – | – | – | – | – | – | 1e-15 | – | – | – | 3.465 | 26 |
| – | – | – | – | – | – | – | – | – | – | 1e-10 | – | – | – | 3.465 | 27 |
| – | – | – | – | – | – | – | – | – | – | – | 256 | – | – | 3.504 | 28 |
| – | – | – | – | – | – | – | – | – | – | – | 512 | – | – | 3.601 | 29 |
| – | – | – | – | – | – | – | – | – | – | – | 1024 | – | – | 3.811 | 30 |
| – | – | – | – | – | – | – | – | – | – | – | – | – | 0 | 3.532 | 31 |
| – | – | – | – | – | – | – | – | – | – | – | – | – | 0.2 | 3.483 | 32 |

Table 84: Hyperparameter ablation for Muon on 130m on 2x Chinchilla Data

| $\eta_{adam}$ | $\beta_1$ | $\beta_2$ | Decay(WSD) | $\epsilon$ | $\eta$ | Schedule | $g_{\text{norm}}$ | $\eta_{min}$ | $\beta_{\text{muon}}$ | $\epsilon_{muon}$ | BSZ | warmup | $\lambda$ | Loss | Link |
|---|---|---|---|---|---|---|---|---|---|---|---|---|---|---|---|
| 0.0024 | 0.8 | 0.98 | 0.8 | 1e-15 | 0.008 | linear | 1 | 0 | 0.98 | 1e-05 | 128 | 0 | 0.1 | 3.369 | 0 |
| 0.0008 | – | – | – | – | – | – | – | – | – | – | – | – | – | 3.385 | 1 |
| 0.0016 | – | – | – | – | – | – | – | – | – | – | – | – | – | 3.373 | 2 |
| – | 0.9 | – | – | – | – | – | – | – | – | – | – | – | – | 3.372 | 3 |
| – | 0.95 | – | – | – | – | – | – | – | – | – | – | – | – | 3.373 | 4 |
| – | 0.98 | – | – | – | – | – | – | – | – | – | – | – | – | 3.375 | 5 |
| – | – | 0.9 | – | – | – | – | – | – | – | – | – | – | – | 3.390 | 6 |
| – | – | 0.95 | – | – | – | – | – | – | – | – | – | – | – | 3.377 | 7 |
| – | – | – | 0.2 | – | – | – | – | – | – | – | – | – | – | 3.404 | 8 |
| – | – | – | 0.4 | – | – | – | – | – | – | – | – | – | – | 3.380 | 9 |
| – | – | – | 0.6 | – | – | – | – | – | – | – | – | – | – | 3.371 | 10 |
| – | – | – | 1.0 | – | – | – | – | – | – | – | – | – | – | 3.374 | 11 |
| – | – | – | – | 1e-25 | – | – | – | – | – | – | – | – | – | 3.370 | 12 |
| – | – | – | – | 1e-20 | – | – | – | – | – | – | – | – | – | 3.370 | 13 |
| – | – | – | – | 1e-10 | – | – | – | – | – | – | – | – | – | 3.369 | 14 |
| – | – | – | – | – | 0.016 | – | – | – | – | – | – | – | – | 3.373 | 15 |
| – | – | – | – | – | 0.032 | – | – | – | – | – | – | – | – | 3.406 | 16 |
| – | – | – | – | – | 0.064 | – | – | – | – | – | – | – | – | 3.549 | 17 |
| – | – | – | – | – | 0.128 | – | – | – | – | – | – | – | – | 5.932 | 18 |
| – | – | – | – | – | – | – | 0 | – | – | – | – | – | – | 3.371 | 19 |
| – | – | – | – | – | – | – | 2.0 | – | – | – | – | – | – | 3.370 | 20 |
| – | – | – | – | – | – | – | – | – | 0.8 | – | – | – | – | 3.417 | 21 |
| – | – | – | – | – | – | – | – | – | 0.9 | – | – | – | – | 3.385 | 22 |
| – | – | – | – | – | – | – | – | – | 0.95 | – | – | – | – | 3.371 | 23 |
| – | – | – | – | – | – | – | – | – | – | 1e-25 | – | – | – | 3.370 | 24 |
| – | – | – | – | – | – | – | – | – | – | 1e-20 | – | – | – | 3.370 | 25 |
| – | – | – | – | – | – | – | – | – | – | 1e-15 | – | – | – | 3.370 | 26 |
| – | – | – | – | – | – | – | – | – | – | 1e-10 | – | – | – | 3.370 | 27 |
| – | – | – | – | – | – | – | – | – | – | – | 256 | – | – | 3.400 | 28 |
| – | – | – | – | – | – | – | – | – | – | – | 512 | – | – | 3.465 | 29 |
| – | – | – | – | – | – | – | – | – | – | – | 1024 | – | – | 3.608 | 30 |
| – | – | – | – | – | – | – | – | – | – | – | – | – | 0 | 3.431 | 31 |
| – | – | – | – | – | – | – | – | – | – | – | – | – | 0.2 | 3.378 | 32 |

Table 85: Hyperparameter ablation for Muon on 130m on 4x Chinchilla Data

| $\eta_{adam}$ | $\beta_1$ | $\beta_2$ | Decay(WSD) | $\epsilon$ | $\eta$ | Schedule | $g_{\mathrm{norm}}$ | $\eta_{min}$ | $\beta_{\mathrm{muon}}$ | $\epsilon_{muon}$ | BSZ | warmup | $\lambda$ | Loss | Link |
|---|---|---|---|---|---|---|---|---|---|---|---|---|---|---|---|
| 0.0024 | 0.8 | 0.98 | 0.8 | 1e-15 | 0.008 | linear | 1 | 0 | 0.95 | 1e-05 | 128 | 0 | 0.1 | 3.296 | 0 |
| 0.0008 | – | – | – | – | – | – | – | – | – | – | – | – | – | 3.309 | 1 |
| 0.0016 | – | – | – | – | – | – | – | – | – | – | – | – | – | 3.300 | 2 |
| – | 0.9 | – | – | – | – | – | – | – | – | – | – | – | – | 3.297 | 3 |
| – | 0.95 | – | – | – | – | – | – | – | – | – | – | – | – | 3.299 | 4 |
| – | 0.98 | – | – | – | – | – | – | – | – | – | – | – | – | 3.299 | 5 |
| – | – | 0.9 | – | – | – | – | – | – | – | – | – | – | – | 3.317 | 6 |
| – | – | 0.95 | – | – | – | – | – | – | – | – | – | – | – | 3.305 | 7 |
| – | – | – | 0.2 | – | – | – | – | – | – | – | – | – | – | 3.334 | 8 |
| – | – | – | 0.4 | – | – | – | – | – | – | – | – | – | – | 3.310 | 9 |
| – | – | – | 0.6 | – | – | – | – | – | – | – | – | – | – | 3.301 | 10 |
| – | – | – | 1.0 | – | – | – | – | – | – | – | – | – | – | 3.297 | 11 |
| – | – | – | – | 1e-25 | – | – | – | – | – | – | – | – | – | 3.297 | 12 |
| – | – | – | – | 1e-20 | – | – | – | – | – | – | – | – | – | 3.296 | 13 |
| – | – | – | – | 1e-10 | – | – | – | – | – | – | – | – | – | 3.297 | 14 |
| – | – | – | – | – | 0.016 | – | – | – | – | – | – | – | – | 3.306 | 15 |
| – | – | – | – | – | 0.032 | – | – | – | – | – | – | – | – | 3.347 | 16 |
| – | – | – | – | – | 0.064 | – | – | – | – | – | – | – | – | 3.431 | 17 |
| – | – | – | – | – | 0.128 | – | – | – | – | – | – | – | – | 4.807 | 18 |
| – | – | – | – | – | – | – | 0 | – | – | – | – | – | – | 3.296 | 19 |
| – | – | – | – | – | – | – | 2.0 | – | – | – | – | – | – | 3.296 | 20 |
| – | – | – | – | – | – | – | – | – | 0.8 | – | – | – | – | 3.327 | 21 |
| – | – | – | – | – | – | – | – | – | 0.9 | – | – | – | – | 3.307 | 22 |
| – | – | – | – | – | – | – | – | – | 0.98 | – | – | – | – | 3.295 | 23 |
| – | – | – | – | – | – | – | – | – | – | 1e-25 | – | – | – | 3.296 | 24 |
| – | – | – | – | – | – | – | – | – | – | 1e-20 | – | – | – | 3.296 | 25 |
| – | – | – | – | – | – | – | – | – | – | 1e-15 | – | – | – | 3.296 | 26 |
| – | – | – | – | – | – | – | – | – | – | 1e-10 | – | – | – | 3.297 | 27 |
| – | – | – | – | – | – | – | – | – | – | – | 256 | – | – | 3.308 | 28 |
| – | – | – | – | – | – | – | – | – | – | – | 512 | – | – | 3.346 | 29 |
| – | – | – | – | – | – | – | – | – | – | – | 1024 | – | – | 3.420 | 30 |
| – | – | – | – | – | – | – | – | – | – | – | – | – | 0 | 3.352 | 31 |
| – | – | – | – | – | – | – | – | – | – | – | – | – | 0.2 | 3.316 | 32 |

Table 86: Hyperparameter ablation for Muon on 130m on 8x Chinchilla Data

| $\eta_{adam}$ | $\beta_1$ | $\beta_2$ | Decay(WSD) | $\epsilon$ | $\eta$ | Schedule | $g_{norm}$ | $\eta_{min}$ | $\beta_{muon}$ | $\epsilon_{muon}$ | BSZ | warmup | $\lambda$ | Loss | Link |
|---|---|---|---|---|---|---|---|---|---|---|---|---|---|---|---|
| 0.0024 | 0.8 | 0.98 | 1 | 1e-15 | 0.008 | linear | 1 | 0 | 0.98 | 1e-05 | 128 | 0 | 0.1 | 3.240 | 0 |
| 0.0008 | – | – | – | – | – | – | – | – | – | – | – | – | – | 3.249 | 1 |
| 0.0016 | – | – | – | – | – | – | – | – | – | – | – | – | – | 3.243 | 2 |
| – | 0.9 | – | – | – | – | – | – | – | – | – | – | – | – | 3.240 | 3 |
| – | 0.95 | – | – | – | – | – | – | – | – | – | – | – | – | 3.240 | 4 |
| – | 0.98 | – | – | – | – | – | – | – | – | – | – | – | – | 3.241 | 5 |
| – | – | 0.9 | – | – | – | – | – | – | – | – | – | – | – | 3.260 | 6 |
| – | – | 0.95 | – | – | – | – | – | – | – | – | – | – | – | 3.247 | 7 |
| – | – | – | 0.2 | – | – | – | – | – | – | – | – | – | – | 3.280 | 8 |
| – | – | – | 0.4 | – | – | – | – | – | – | – | – | – | – | 3.257 | 9 |
| – | – | – | 0.6 | – | – | – | – | – | – | – | – | – | – | 3.248 | 10 |
| – | – | – | 0.8 | – | – | – | – | – | – | – | – | – | – | 3.242 | 11 |
| – | – | – | – | 1e-25 | – | – | – | – | – | – | – | – | – | 3.238 | 12 |
| – | – | – | – | 1e-20 | – | – | – | – | – | – | – | – | – | 3.238 | 13 |
| – | – | – | – | 1e-10 | – | – | – | – | – | – | – | – | – | 3.241 | 14 |
| – | – | – | – | – | 0.016 | – | – | – | – | – | – | – | – | 3.253 | 15 |
| – | – | – | – | – | 0.032 | – | – | – | – | – | – | – | – | 3.298 | 16 |
| – | – | – | – | – | 0.064 | – | – | – | – | – | – | – | – | 3.407 | 17 |
| – | – | – | – | – | 0.128 | – | – | – | – | – | – | – | – | 5.568 | 18 |
| – | – | – | – | – | – | – | 0 | – | – | – | – | – | – | 3.240 | 19 |
| – | – | – | – | – | – | – | 2.0 | – | – | – | – | – | – | 3.239 | 20 |
| – | – | – | – | – | – | – | – | – | 0.8 | – | – | – | – | 3.265 | 21 |
| – | – | – | – | – | – | – | – | – | 0.9 | – | – | – | – | 3.250 | 22 |
| – | – | – | – | – | – | – | – | – | 0.95 | – | – | – | – | 3.243 | 23 |
| – | – | – | – | – | – | – | – | – | – | 1e-25 | – | – | – | 3.239 | 24 |
| – | – | – | – | – | – | – | – | – | – | 1e-20 | – | – | – | 3.239 | 25 |
| – | – | – | – | – | – | – | – | – | – | 1e-15 | – | – | – | 3.239 | 26 |
| – | – | – | – | – | – | – | – | – | – | 1e-10 | – | – | – | 3.239 | 27 |
| – | – | – | – | – | – | – | – | – | – | – | 256 | – | – | 3.240 | 28 |
| – | – | – | – | – | – | – | – | – | – | – | 512 | – | – | 3.257 | 29 |
| – | – | – | – | – | – | – | – | – | – | – | 1024 | – | – | 3.302 | 30 |
| – | – | – | – | – | – | – | – | – | – | – | – | – | 0 | 3.310 | 31 |
| – | – | – | – | – | – | – | – | – | – | – | – | – | 0.2 | 3.256 | 32 |

Table 87: Hyperparameter ablation for Muon on 300m on 1x Chinchilla Data

| $\eta_{adam}$ | $\beta_1$ | $\beta_2$ | Decay(WSD) | $\epsilon$ | $\eta$ | Schedule | $g_{\mathrm{norm}}$ | $\eta_{min}$ | $\beta_{\mathrm{muon}}$ | $\epsilon_{muon}$ | BSZ | warmup | $\lambda$ | Loss | Link |
|---|---|---|---|---|---|---|---|---|---|---|---|---|---|---|---|
| 0.0024 | 0.8 | 0.98 | 0.8 | 1e-15 | 0.008 | linear | 1 | 0 | 0.98 | 1e-05 | 128 | 0 | 0.1 | 3.224 | 0 |
| 0.0008 | – | – | – | – | – | – | – | – | – | – | – | – | – | 3.232 | 1 |
| 0.0016 | – | – | – | – | – | – | – | – | – | – | – | – | – | 3.225 | 2 |
| – | 0.9 | – | – | – | – | – | – | – | – | – | – | – | – | 3.226 | 3 |
| – | 0.95 | – | – | – | – | – | – | – | – | – | – | – | – | 3.226 | 4 |
| – | 0.98 | – | – | – | – | – | – | – | – | – | – | – | – | 3.227 | 5 |
| – | – | 0.9 | – | – | – | – | – | – | – | – | – | – | – | 3.241 | 6 |
| – | – | 0.95 | – | – | – | – | – | – | – | – | – | – | – | 3.231 | 7 |
| – | – | – | 0.2 | – | – | – | – | – | – | – | – | – | – | 3.271 | 8 |
| – | – | – | 0.4 | – | – | – | – | – | – | – | – | – | – | 3.240 | 9 |
| – | – | – | 0.6 | – | – | – | – | – | – | – | – | – | – | 3.229 | 10 |
| – | – | – | 1.0 | – | – | – | – | – | – | – | – | – | – | 3.225 | 11 |
| – | – | – | – | 1e-25 | – | – | – | – | – | – | – | – | – | 3.224 | 12 |
| – | – | – | – | 1e-20 | – | – | – | – | – | – | – | – | – | 3.224 | 13 |
| – | – | – | – | 1e-10 | – | – | – | – | – | – | – | – | – | 3.225 | 14 |
| – | – | – | – | – | 0.016 | – | – | – | – | – | – | – | – | 3.236 | 15 |
| – | – | – | – | – | 0.032 | – | – | – | – | – | – | – | – | 3.281 | 16 |
| – | – | – | – | – | 0.064 | – | – | – | – | – | – | – | – | 4.695 | 17 |
| – | – | – | – | – | 0.128 | – | – | – | – | – | – | – | – | 5.976 | 18 |
| – | – | – | – | – | – | – | 0 | – | – | – | – | – | – | 3.224 | 19 |
| – | – | – | – | – | – | – | 2.0 | – | – | – | – | – | – | 3.225 | 20 |
| – | – | – | – | – | – | – | – | – | 0.8 | – | – | – | – | 3.257 | 21 |
| – | – | – | – | – | – | – | – | – | 0.9 | – | – | – | – | 3.235 | 22 |
| – | – | – | – | – | – | – | – | – | 0.95 | – | – | – | – | 3.224 | 23 |
| – | – | – | – | – | – | – | – | – | – | 1e-25 | – | – | – | 3.226 | 24 |
| – | – | – | – | – | – | – | – | – | – | 1e-20 | – | – | – | 3.226 | 25 |
| – | – | – | – | – | – | – | – | – | – | 1e-15 | – | – | – | 3.226 | 26 |
| – | – | – | – | – | – | – | – | – | – | 1e-10 | – | – | – | 3.226 | 27 |
| – | – | – | – | – | – | – | – | – | – | – | 256 | – | – | 3.244 | 28 |
| – | – | – | – | – | – | – | – | – | – | – | 512 | – | – | 3.301 | 29 |
| – | – | – | – | – | – | – | – | – | – | – | 1024 | – | – | 3.419 | 30 |
| – | – | – | – | – | – | – | – | – | – | – | – | – | 0 | 3.285 | 31 |
| – | – | – | – | – | – | – | – | – | – | – | – | – | 0.2 | 3.234 | 32 |

Table 88: Hyperparameter ablation for Muon on 520m on 1x Chinchilla Data

| $\eta_{adam}$ | $\beta_1$ | $\beta_2$ | Decay(WSD) | $\epsilon$ | $\eta$ | Schedule | $g_{\mathrm{norm}}$ | $\eta_{min}$ | $\beta_{\mathrm{muon}}$ | $\epsilon_{muon}$ | BSZ | warmup | $\lambda$ | Loss | Link |
|---|---|---|---|---|---|---|---|---|---|---|---|---|---|---|---|
| 0.0024 | 0.8 | 0.98 | 1 | 1e-25 | 0.008 | linear | 1 | 0 | 0.98 | 1e-05 | 128 | 0 | 0.1 | 3.073 | 0 |
| 0.0008 | – | – | – | – | – | – | – | – | – | – | – | – | – | 3.076 | 1 |
| 0.0016 | – | – | – | – | – | – | – | – | – | – | – | – | – | 3.072 | 2 |
| – | 0.9 | – | – | – | – | – | – | – | – | – | – | – | – | 3.073 | 3 |
| – | 0.95 | – | – | – | – | – | – | – | – | – | – | – | – | 3.074 | 4 |
| – | 0.98 | – | – | – | – | – | – | – | – | – | – | – | – | 3.073 | 5 |
| – | – | 0.9 | – | – | – | – | – | – | – | – | – | – | – | 3.090 | 6 |
| – | – | 0.95 | – | – | – | – | – | – | – | – | – | – | – | 3.080 | 7 |
| – | – | – | 0.2 | – | – | – | – | – | – | – | – | – | – | 3.134 | 8 |
| – | – | – | 0.4 | – | – | – | – | – | – | – | – | – | – | 3.101 | 9 |
| – | – | – | 0.6 | – | – | – | – | – | – | – | – | – | – | 3.086 | 10 |
| – | – | – | 0.8 | – | – | – | – | – | – | – | – | – | – | 3.076 | 11 |
| – | – | – | – | 1e-20 | – | – | – | – | – | – | – | – | – | 3.074 | 12 |
| – | – | – | – | 1e-15 | – | – | – | – | – | – | – | – | – | 3.073 | 13 |
| – | – | – | – | 1e-10 | – | – | – | – | – | – | – | – | – | 3.073 | 14 |
| – | – | – | – | – | 0.016 | – | – | – | – | – | – | – | – | 3.095 | 15 |
| – | – | – | – | – | 0.032 | – | – | – | – | – | – | – | – | 3.147 | 16 |
| – | – | – | – | – | 0.064 | – | – | – | – | – | – | – | – | 7.886 | 17 |
| – | – | – | – | – | 0.128 | – | – | – | – | – | – | – | – | 7.900 | 18 |
| – | – | – | – | – | – | – | 0 | – | – | – | – | – | – | 3.072 | 19 |
| – | – | – | – | – | – | – | 2.0 | – | – | – | – | – | – | 3.071 | 20 |
| – | – | – | – | – | – | – | – | – | 0.8 | – | – | – | – | 3.098 | 21 |
| – | – | – | – | – | – | – | – | – | 0.9 | – | – | – | – | 3.080 | 22 |
| – | – | – | – | – | – | – | – | – | 0.95 | – | – | – | – | 3.073 | 23 |
| – | – | – | – | – | – | – | – | – | – | 1e-25 | – | – | – | 3.072 | 24 |
| – | – | – | – | – | – | – | – | – | – | 1e-20 | – | – | – | 3.072 | 25 |
| – | – | – | – | – | – | – | – | – | – | 1e-15 | – | – | – | 3.072 | 26 |
| – | – | – | – | – | – | – | – | – | – | 1e-10 | – | – | – | 3.072 | 27 |
| – | – | – | – | – | – | – | – | – | – | – | 256 | – | – | 3.080 | 28 |
| – | – | – | – | – | – | – | – | – | – | – | 512 | – | – | 3.115 | 29 |
| – | – | – | – | – | – | – | – | – | – | – | 1024 | – | – | 3.188 | 30 |
| – | – | – | – | – | – | – | – | – | – | – | – | – | 0 | 3.134 | 31 |
| – | – | – | – | – | – | – | – | – | – | – | – | – | 0.2 | 3.091 | 32 |

### E.10 Sweeping Results for Scion

Table 89: Hyperparameter ablation for Scion on 130m on 1x Chinchilla Data

| $\eta_{adam}$ | $\beta_1$ | Decay(WSD) | $\eta$ | Schedule | $g_{\text{norm}}$ | $\eta_{min}$ | $\beta_{\text{muon}}$ | $\epsilon_{scion}$ | BSZ | warmup | $\lambda$ | Loss | Link |
|---|---|---|---|---|---|---|---|---|---|---|---|---|---|
| 0.0032 | 0.95 | 0.8 | 0.016 | linear | 1 | 0 | 0.95 | 1e-15 | 128 | 0 | 0.1 | 3.477 | 0 |
| 0.0016 | – | – | – | – | – | – | – | – | – | – | – | 3.477 | 1 |
| 0.0048 | – | – | – | – | – | – | – | – | – | – | – | 3.481 | 2 |
| 0.0064 | – | – | – | – | – | – | – | – | – | – | – | 3.485 | 3 |
| 0.008 | – | – | – | – | – | – | – | – | – | – | – | 3.488 | 4 |
| 0.0096 | – | – | – | – | – | – | – | – | – | – | – | 3.490 | 5 |
| – | 0.8 | – | – | – | – | – | – | – | – | – | – | 3.493 | 6 |
| – | 0.9 | – | – | – | – | – | – | – | – | – | – | 3.479 | 7 |
| – | 0.98 | – | – | – | – | – | – | – | – | – | – | 3.479 | 8 |
| – | – | 0.0 | – | – | – | – | – | – | – | – | – | 3.779 | 9 |
| – | – | 0.2 | – | – | – | – | – | – | – | – | – | 3.525 | 10 |
| – | – | 0.4 | – | – | – | – | – | – | – | – | – | 3.493 | 11 |
| – | – | 0.6 | – | – | – | – | – | – | – | – | – | 3.480 | 12 |
| – | – | 1.0 | – | – | – | – | – | – | – | – | – | 3.477 | 13 |
| – | – | – | 0.008 | – | – | – | – | – | – | – | – | 3.490 | 14 |
| – | – | – | 0.032 | – | – | – | – | – | – | – | – | 3.509 | 15 |
| – | – | – | 0.064 | – | – | – | – | – | – | – | – | 5.195 | 16 |
| – | – | – | 0.128 | – | – | – | – | – | – | – | – | 6.308 | 17 |
| – | – | – | – | – | 0 | – | – | – | – | – | – | 3.478 | 18 |
| – | – | – | – | – | 2.0 | – | – | – | – | – | – | 3.478 | 19 |
| – | – | – | – | – | – | – | 0.8 | – | – | – | – | 3.480 | 20 |
| – | – | – | – | – | – | – | 0.9 | – | – | – | – | 3.474 | 21 |
| – | – | – | – | – | – | – | 0.98 | – | – | – | – | 3.496 | 22 |
| – | – | – | – | – | – | – | – | 1e-20 | – | – | – | 3.477 | 23 |
| – | – | – | – | – | – | – | – | 1e-10 | – | – | – | 3.477 | 24 |
| – | – | – | – | – | – | – | – | 1e-05 | – | – | – | 3.477 | 25 |
| – | – | – | – | – | – | – | – | – | 256 | – | – | 3.505 | 26 |
| – | – | – | – | – | – | – | – | – | 512 | – | – | 3.583 | 27 |
| – | – | – | – | – | – | – | – | – | 1024 | – | – | 3.761 | 28 |
| – | – | – | – | – | – | – | – | – | – | – | 0 | 3.537 | 29 |
| – | – | – | – | – | – | – | – | – | – | – | 0.2 | 3.495 | 30 |

Table 90: Hyperparameter ablation for Scion on 130m on 2x Chinchilla Data

| $\eta_{adam}$ | $\beta_1$ | Decay(WSD) | $\eta$ | Schedule | $g_{\text{norm}}$ | $\eta_{min}$ | $\beta_{\text{muon}}$ | $\epsilon_{scion}$ | BSZ | warmup | $\lambda$ | Loss | Link |
|---|---|---|---|---|---|---|---|---|---|---|---|---|---|
| 0.0032 | 0.95 | 1 | 0.016 | linear | 1 | 0 | 0.9 | 1e-15 | 128 | 0 | 0.1 | 3.379 | 0 |
| 0.0016 | – | – | – | – | – | – | – | – | – | – | – | 3.378 | 1 |
| 0.0048 | – | – | – | – | – | – | – | – | – | – | – | 3.382 | 2 |
| 0.0064 | – | – | – | – | – | – | – | – | – | – | – | 3.384 | 3 |
| 0.008 | – | – | – | – | – | – | – | – | – | – | – | 3.386 | 4 |
| 0.0096 | – | – | – | – | – | – | – | – | – | – | – | 3.388 | 5 |
| – | 0.8 | – | – | – | – | – | – | – | – | – | – | 3.400 | 6 |
| – | 0.9 | – | – | – | – | – | – | – | – | – | – | 3.383 | 7 |
| – | 0.98 | – | – | – | – | – | – | – | – | – | – | 3.379 | 8 |
| – | – | 0.0 | – | – | – | – | – | – | – | – | – | 3.738 | 9 |
| – | – | 0.2 | – | – | – | – | – | – | – | – | – | 3.437 | 10 |
| – | – | 0.4 | – | – | – | – | – | – | – | – | – | 3.405 | 11 |
| – | – | 0.6 | – | – | – | – | – | – | – | – | – | 3.390 | 12 |
| – | – | 0.8 | – | – | – | – | – | – | – | – | – | 3.383 | 13 |
| – | – | – | 0.008 | – | – | – | – | – | – | – | – | 3.390 | 14 |
| – | – | – | 0.032 | – | – | – | – | – | – | – | – | 3.411 | 15 |
| – | – | – | 0.064 | – | – | – | – | – | – | – | – | 3.513 | 16 |
| – | – | – | 0.128 | – | – | – | – | – | – | – | – | 6.193 | 17 |
| – | – | – | – | – | 0 | – | – | – | – | – | – | 3.379 | 18 |
| – | – | – | – | – | 2.0 | – | – | – | – | – | – | 3.378 | 19 |
| – | – | – | – | – | – | – | 0.8 | – | – | – | – | 3.385 | 20 |
| – | – | – | – | – | – | – | 0.95 | – | – | – | – | 3.380 | 21 |
| – | – | – | – | – | – | – | 0.98 | – | – | – | – | 3.388 | 22 |
| – | – | – | – | – | – | – | – | 1e-20 | – | – | – | 3.379 | 23 |
| – | – | – | – | – | – | – | – | 1e-10 | – | – | – | 3.380 | 24 |
| – | – | – | – | – | – | – | – | 1e-05 | – | – | – | 3.380 | 25 |
| – | – | – | – | – | – | – | – | – | 256 | – | – | 3.391 | 26 |
| – | – | – | – | – | – | – | – | – | 512 | – | – | 3.436 | 27 |
| – | – | – | – | – | – | – | – | – | 1024 | – | – | 3.532 | 28 |
| – | – | – | – | – | – | – | – | – | – | – | 0 | 3.443 | 29 |
| – | – | – | – | – | – | – | – | – | – | – | 0.2 | 3.401 | 30 |

Table 91: Hyperparameter ablation for Scion on 130m on 4x Chinchilla Data

| $\eta_{adam}$ | $\beta_1$ | Decay(WSD) | $\eta$ | Schedule | $g_{\mathrm{norm}}$ | $\eta_{min}$ | $\beta_{\mathrm{muon}}$ | $\epsilon_{scion}$ | BSZ | warmup | $\lambda$ | Loss | Link |
|---|---|---|---|---|---|---|---|---|---|---|---|---|---|
| 0.0016 | 0.98 | 1 | 0.008 | linear | 2 | 0 | 0.9 | 1e-15 | 128 | 0 | 0.1 | 3.302 | 0 |
| 0.0008 | – | – | – | – | – | – | – | – | – | – | – | 3.302 | 1 |
| 0.0024 | – | – | – | – | – | – | – | – | – | – | – | 3.306 | 2 |
| 0.0032 | – | – | – | – | – | – | – | – | – | – | – | 3.307 | 3 |
| 0.004 | – | – | – | – | – | – | – | – | – | – | – | 3.306 | 4 |
| 0.0048 | – | – | – | – | – | – | – | – | – | – | – | 3.307 | 5 |
| – | 0.8 | – | – | – | – | – | – | – | – | – | – | 3.331 | 6 |
| – | 0.9 | – | – | – | – | – | – | – | – | – | – | 3.314 | 7 |
| – | 0.95 | – | – | – | – | – | – | – | – | – | – | 3.306 | 8 |
| – | – | 0.0 | – | – | – | – | – | – | – | – | – | 3.584 | 9 |
| – | – | 0.2 | – | – | – | – | – | – | – | – | – | 3.340 | 10 |
| – | – | 0.4 | – | – | – | – | – | – | – | – | – | 3.317 | 11 |
| – | – | 0.6 | – | – | – | – | – | – | – | – | – | 3.307 | 12 |
| – | – | 0.8 | – | – | – | – | – | – | – | – | – | 3.303 | 13 |
| – | – | – | 0.016 | – | – | – | – | – | – | – | – | 3.310 | 14 |
| – | – | – | 0.032 | – | – | – | – | – | – | – | – | 3.361 | 15 |
| – | – | – | 0.064 | – | – | – | – | – | – | – | – | 5.152 | 16 |
| – | – | – | 0.128 | – | – | – | – | – | – | – | – | 5.894 | 17 |
| – | – | – | – | – | 0 | – | – | – | – | – | – | 3.303 | 18 |
| – | – | – | – | – | 1.0 | – | – | – | – | – | – | 3.303 | 19 |
| – | – | – | – | – | – | – | 0.8 | – | – | – | – | 3.307 | 20 |
| – | – | – | – | – | – | – | 0.95 | – | – | – | – | 3.302 | 21 |
| – | – | – | – | – | – | – | 0.98 | – | – | – | – | 3.303 | 22 |
| – | – | – | – | – | – | – | – | 1e-20 | – | – | – | 3.302 | 23 |
| – | – | – | – | – | – | – | – | 1e-10 | – | – | – | 3.303 | 24 |
| – | – | – | – | – | – | – | – | 1e-05 | – | – | – | 3.302 | 25 |
| – | – | – | – | – | – | – | – | – | 256 | – | – | 3.323 | 26 |
| – | – | – | – | – | – | – | – | – | 512 | – | – | 3.365 | 27 |
| – | – | – | – | – | – | – | – | – | 1024 | – | – | 3.447 | 28 |
| – | – | – | – | – | – | – | – | – | – | – | 0 | 3.375 | 29 |
| – | – | – | – | – | – | – | – | – | – | – | 0.2 | 3.309 | 30 |

Table 92: Hyperparameter ablation for Scion on 130m on 8x Chinchilla Data

| $\eta_{adam}$ | $\beta_1$ | Decay(WSD) | $\eta$ | Schedule | $g_{\text{norm}}$ | $\eta_{min}$ | $\beta_{\text{muon}}$ | $\epsilon_{scion}$ | BSZ | warmup | $\lambda$ | Loss | Link |
|---|---|---|---|---|---|---|---|---|---|---|---|---|---|
| 0.0016 | 0.98 | 1 | 0.008 | linear | 2 | 0 | 0.9 | 1e-05 | 128 | 0 | 0.1 | 3.246 | 0 |
| 0.0008 | – | – | – | – | – | – | – | – | – | – | – | 3.245 | 1 |
| 0.0024 | – | – | – | – | – | – | – | – | – | – | – | 3.250 | 2 |
| 0.0032 | – | – | – | – | – | – | – | – | – | – | – | 3.249 | 3 |
| 0.004 | – | – | – | – | – | – | – | – | – | – | – | 3.251 | 4 |
| 0.0048 | – | – | – | – | – | – | – | – | – | – | – | 3.252 | 5 |
| – | 0.8 | – | – | – | – | – | – | – | – | – | – | 3.275 | 6 |
| – | 0.9 | – | – | – | – | – | – | – | – | – | – | 3.257 | 7 |
| – | 0.95 | – | – | – | – | – | – | – | – | – | – | 3.249 | 8 |
| – | – | 0.0 | – | – | – | – | – | – | – | – | – | 3.558 | 9 |
| – | – | 0.2 | – | – | – | – | – | – | – | – | – | 3.288 | 10 |
| – | – | 0.4 | – | – | – | – | – | – | – | – | – | 3.266 | 11 |
| – | – | 0.6 | – | – | – | – | – | – | – | – | – | 3.255 | 12 |
| – | – | 0.8 | – | – | – | – | – | – | – | – | – | 3.249 | 13 |
| – | – | – | 0.016 | – | – | – | – | – | – | – | – | 3.265 | 14 |
| – | – | – | 0.032 | – | – | – | – | – | – | – | – | 3.325 | 15 |
| – | – | – | 0.064 | – | – | – | – | – | – | – | – | 5.264 | 16 |
| – | – | – | 0.128 | – | – | – | – | – | – | – | – | 6.174 | 17 |
| – | – | – | – | – | 0 | – | – | – | – | – | – | 3.247 | 18 |
| – | – | – | – | – | 1.0 | – | – | – | – | – | – | 3.247 | 19 |
| – | – | – | – | – | – | – | 0.8 | – | – | – | – | 3.249 | 20 |
| – | – | – | – | – | – | – | 0.95 | – | – | – | – | 3.245 | 21 |
| – | – | – | – | – | – | – | 0.98 | – | – | – | – | 3.247 | 22 |
| – | – | – | – | – | – | – | – | 1e-20 | – | – | – | 3.246 | 23 |
| – | – | – | – | – | – | – | – | 1e-15 | – | – | – | 3.246 | 24 |
| – | – | – | – | – | – | – | – | 1e-10 | – | – | – | 3.247 | 25 |
| – | – | – | – | – | – | – | – | – | 256 | – | – | 3.250 | 26 |
| – | – | – | – | – | – | – | – | – | 512 | – | – | 3.272 | 27 |
| – | – | – | – | – | – | – | – | – | 1024 | – | – | 3.319 | 28 |
| – | – | – | – | – | – | – | – | – | – | – | 0 | 3.308 | 29 |
| – | – | – | – | – | – | – | – | – | – | – | 0.2 | 3.261 | 30 |

Table 93: Hyperparameter ablation for Scion on 300m on 1x Chinchilla Data

| $\eta_{adam}$ | $\beta_1$ | Decay(WSD) | $\eta$ | Schedule | $g_{\text{norm}}$ | $\eta_{min}$ | $\beta_{\text{muon}}$ | $\epsilon_{scion}$ | BSZ | warmup | $\lambda$ | Loss | Link |
|---|---|---|---|---|---|---|---|---|---|---|---|---|---|
| 0.0008 | 0.98 | 0.8 | 0.008 | linear | 2 | 0 | 0.9 | 1e-05 | 128 | 0 | 0.1 | 3.232 | 0 |
| 0.0016 | – | – | – | – | – | – | – | – | – | – | – | 3.237 | 1 |
| 0.0024 | – | – | – | – | – | – | – | – | – | – | – | 3.242 | 2 |
| 0.0032 | – | – | – | – | – | – | – | – | – | – | – | 3.242 | 3 |
| 0.004 | – | – | – | – | – | – | – | – | – | – | – | 3.241 | 4 |
| 0.0048 | – | – | – | – | – | – | – | – | – | – | – | 3.243 | 5 |
| – | 0.8 | – | – | – | – | – | – | – | – | – | – | 3.256 | 6 |
| – | 0.9 | – | – | – | – | – | – | – | – | – | – | 3.240 | 7 |
| – | 0.95 | – | – | – | – | – | – | – | – | – | – | 3.236 | 8 |
| – | – | 0.0 | – | – | – | – | – | – | – | – | – | 3.497 | 9 |
| – | – | 0.2 | – | – | – | – | – | – | – | – | – | 3.271 | 10 |
| – | – | 0.4 | – | – | – | – | – | – | – | – | – | 3.245 | 11 |
| – | – | 0.6 | – | – | – | – | – | – | – | – | – | 3.234 | 12 |
| – | – | 1.0 | – | – | – | – | – | – | – | – | – | 3.237 | 13 |
| – | – | – | 0.016 | – | – | – | – | – | – | – | – | 3.242 | 14 |
| – | – | – | 0.032 | – | – | – | – | – | – | – | – | 3.301 | 15 |
| – | – | – | 0.064 | – | – | – | – | – | – | – | – | 3.400 | 16 |
| – | – | – | 0.128 | – | – | – | – | – | – | – | – | 5.627 | 17 |
| – | – | – | – | – | 0 | – | – | – | – | – | – | 3.233 | 18 |
| – | – | – | – | – | 1.0 | – | – | – | – | – | – | 3.233 | 19 |
| – | – | – | – | – | – | – | 0.8 | – | – | – | – | 3.240 | 20 |
| – | – | – | – | – | – | – | 0.95 | – | – | – | – | 3.231 | 21 |
| – | – | – | – | – | – | – | 0.98 | – | – | – | – | 3.236 | 22 |
| – | – | – | – | – | – | – | – | 1e-20 | – | – | – | 3.232 | 23 |
| – | – | – | – | – | – | – | – | 1e-15 | – | – | – | 3.232 | 24 |
| – | – | – | – | – | – | – | – | 1e-10 | – | – | – | 3.233 | 25 |
| – | – | – | – | – | – | – | – | – | 256 | – | – | 3.259 | 26 |
| – | – | – | – | – | – | – | – | – | 512 | – | – | 3.319 | 27 |
| – | – | – | – | – | – | – | – | – | 1024 | – | – | 3.432 | 28 |
| – | – | – | – | – | – | – | – | – | – | – | 0 | 3.303 | 29 |
| – | – | – | – | – | – | – | – | – | – | – | 0.2 | 3.241 | 30 |

Table 94: Hyperparameter ablation for Scion on 520m on 1x Chinchilla Data

| $\eta_{adam}$ | $\beta_1$ | Decay(WSD) | $\eta$ | Schedule | $g_{\mathrm{norm}}$ | $\eta_{min}$ | $\beta_{\mathrm{muon}}$ | $\epsilon_{scion}$ | BSZ | warmup | $\lambda$ | Loss | Link |
|---|---|---|---|---|---|---|---|---|---|---|---|---|---|
| 0.0008 | 0.98 | 1 | 0.008 | linear | 2 | 0 | 0.9 | 1e-05 | 128 | 0 | 0.1 | 3.080 | 0 |
| 0.0016 | – | – | – | – | – | – | – | – | – | – | – | 3.089 | 1 |
| 0.0024 | – | – | – | – | – | – | – | – | – | – | – | 3.090 | 2 |
| 0.0032 | – | – | – | – | – | – | – | – | – | – | – | 3.090 | 3 |
| 0.004 | – | – | – | – | – | – | – | – | – | – | – | 3.090 | 4 |
| 0.0048 | – | – | – | – | – | – | – | – | – | – | – | 3.091 | 5 |
| – | 0.8 | – | – | – | – | – | – | – | – | – | – | 3.104 | 6 |
| – | 0.9 | – | – | – | – | – | – | – | – | – | – | 3.090 | 7 |
| – | 0.95 | – | – | – | – | – | – | – | – | – | – | 3.081 | 8 |
| – | – | 0.0 | – | – | – | – | – | – | – | – | – | 3.400 | 9 |
| – | – | 0.2 | – | – | – | – | – | – | – | – | – | 3.136 | 10 |
| – | – | 0.4 | – | – | – | – | – | – | – | – | – | 3.106 | 11 |
| – | – | 0.6 | – | – | – | – | – | – | – | – | – | 3.091 | 12 |
| – | – | 0.8 | – | – | – | – | – | – | – | – | – | 3.083 | 13 |
| – | – | – | 0.016 | – | – | – | – | – | – | – | – | 3.105 | 14 |
| – | – | – | 0.032 | – | – | – | – | – | – | – | – | 3.173 | 15 |
| – | – | – | 0.064 | – | – | – | – | – | – | – | – | 4.420 | 16 |
| – | – | – | 0.128 | – | – | – | – | – | – | – | – | 7.335 | 17 |
| – | – | – | – | – | – | 0 | – | – | – | – | – | 3.082 | 18 |
| – | – | – | – | – | – | 1.0 | – | – | – | – | – | 3.081 | 19 |
| – | – | – | – | – | – | – | 0.8 | – | – | – | – | 3.084 | 20 |
| – | – | – | – | – | – | – | 0.95 | – | – | – | – | 3.079 | 21 |
| – | – | – | – | – | – | – | 0.98 | – | – | – | – | 3.086 | 22 |
| – | – | – | – | – | – | – | – | 1e-20 | – | – | – | 3.081 | 23 |
| – | – | – | – | – | – | – | – | 1e-15 | – | – | – | 3.081 | 24 |
| – | – | – | – | – | – | – | – | 1e-10 | – | – | – | 3.080 | 25 |
| – | – | – | – | – | – | – | – | – | 256 | – | – | 3.093 | 26 |
| – | – | – | – | – | – | – | – | – | 512 | – | – | 3.134 | 27 |
| – | – | – | – | – | – | – | – | – | 1024 | – | – | 3.215 | 28 |
| – | – | – | – | – | – | – | – | – | – | – | 0 | 3.150 | 29 |
| – | – | – | – | – | – | – | – | – | – | – | 0.2 | 3.097 | 30 |

## E.11  Sweeping Results for Sophia

Table 95: Hyperparameter ablation for Sophia on 130m on 1x Chinchilla Data

| $\beta_1$ | $\beta_2$ | $\epsilon$ | $\gamma$ | $\eta$ | BSZ | warmup | $\lambda$ | Loss | Link |
|---|---|---|---|---|---|---|---|---|---|
| 0.95 | 0.9 | 1e-07 | 0.0125 | 0.004 | 128 | 4000 | 0 | 3.544 | 0 |
| 0.8 | – | – | – | – | – | – | – | 3.581 | 1 |
| 0.9 | – | – | – | – | – | – | – | 3.546 | 2 |
| 0.98 | – | – | – | – | – | – | – | 3.624 | 3 |
| – | 0.95 | – | – | – | – | – | – | 3.546 | 1 |
| – | 0.98 | – | – | – | – | – | – | 3.553 | 2 |
| – | 0.99 | – | – | – | – | – | – | 3.563 | 3 |
| – | 0.995 | – | – | – | – | – | – | 3.573 | 4 |
| – | – | 1e-17 | – | – | – | – | – | 3.559 | 1 |
| – | – | 1e-12 | – | – | – | – | – | 3.559 | 2 |
| – | – | – | 0.00625 | – | – | – | – | 3.546 | 1 |
| – | – | – | 0.025 | – | – | – | – | 3.546 | 2 |
| – | – | – | 0.05 | – | – | – | – | 3.554 | 3 |
| – | – | – | – | 0.002 | – | – | – | 3.551 | 1 |
| – | – | – | – | 0.008 | – | – | – | 3.576 | 2 |
| – | – | – | – | 0.016 | – | – | – | 7.769 | 3 |
| – | – | – | – | 0.032 | – | – | – | 7.745 | 4 |
| – | – | – | – | – | – | 500 | – | 7.754 | 1 |
| – | – | – | – | – | – | 1000 | – | 7.824 | 2 |
| – | – | – | – | – | – | 2000 | – | 3.576 | 3 |
| – | – | – | – | – | – | – | 0.1 | 3.553 | 1 |
| – | – | – | – | – | – | – | 0.2 | 3.566 | 2 |
| – | – | – | – | – | – | – | 0.3 | 3.579 | 3 |

Table 96: Hyperparameter ablation for Sophia on 130m on 2x Chinchilla Data

| $\beta_1$ | $\beta_2$ | $\epsilon$ | $\gamma$ | $\eta$ | BSZ | warmup | $\lambda$ | Loss | Link |
|---|---|---|---|---|---|---|---|---|---|
| 0.95 | 0.9 | 1e-07 | 0.0125 | 0.004 | 128 | 4000 | 0.1 | 3.414 | 0 |
| 0.8 | – | – | – | – | – | – | – | 3.439 | 1 |
| 0.9 | – | – | – | – | – | – | – | 3.416 | 2 |
| 0.98 | – | – | – | – | – | – | – | 3.469 | 3 |
| – | 0.95 | – | – | – | – | – | – | 3.414 | 1 |
| – | 0.98 | – | – | – | – | – | – | 3.418 | 2 |
| – | 0.99 | – | – | – | – | – | – | 3.421 | 3 |
| – | 0.995 | – | – | – | – | – | – | 3.428 | 4 |
| – | – | 1e-17 | – | – | – | – | – | 3.426 | 1 |
| – | – | 1e-12 | – | – | – | – | – | 3.424 | 2 |
| – | – | – | 0.00625 | – | – | – | – | 3.416 | 1 |
| – | – | – | 0.025 | – | – | – | – | 3.415 | 2 |
| – | – | – | 0.05 | – | – | – | – | 3.423 | 3 |
| – | – | – | – | 0.002 | – | – | – | 3.417 | 1 |
| – | – | – | – | 0.008 | – | – | – | 3.438 | 2 |
| – | – | – | – | 0.016 | – | – | – | 7.282 | 3 |
| – | – | – | – | 0.032 | – | – | – | 6.938 | 4 |
| – | – | – | – | – | 256 | – | – | 3.446 | 1 |
| – | – | – | – | – | – | 500 | – | 7.811 | 1 |
| – | – | – | – | – | – | 1000 | – | 7.549 | 2 |
| – | – | – | – | – | – | 2000 | – | 3.436 | 3 |
| – | – | – | – | – | – | – | 0 | 3.431 | 1 |
| – | – | – | – | – | – | – | 0.2 | 3.415 | 2 |
| – | – | – | – | – | – | – | 0.3 | 3.419 | 3 |

Table 97: Hyperparameter ablation for Sophia on 130m on 4x Chinchilla Data

| $\beta_1$ | $\beta_2$ | $\epsilon$ | $\gamma$ | $\eta$ | BSZ | warmup | $\lambda$ | Loss | Link |
|---|---|---|---|---|---|---|---|---|---|
| 0.95 | 0.99 | 1e-07 | 0.0125 | 0.004 | 128 | 4000 | 0.2 | 3.330 | 0 |
| 0.8 | – | – | – | – | – | – | – | 3.351 | 1 |
| 0.9 | – | – | – | – | – | – | – | 3.332 | 2 |
| 0.98 | – | – | – | – | – | – | – | 3.371 | 3 |
| – | 0.9 | – | – | – | – | – | – | 3.328 | 1 |
| – | 0.95 | – | – | – | – | – | – | 3.328 | 2 |
| – | 0.98 | – | – | – | – | – | – | 3.328 | 3 |
| – | – | 1e-17 | – | – | – | – | – | 3.337 | 1 |
| – | – | 1e-12 | – | – | – | – | – | 3.338 | 2 |
| – | – | – | 0.00625 | – | – | – | – | 3.330 | 1 |
| – | – | – | 0.025 | – | – | – | – | 3.330 | 2 |
| – | – | – | – | 0.002 | – | – | – | 3.329 | 1 |
| – | – | – | – | 0.016 | – | – | – | 7.059 | 3 |
| – | – | – | – | 0.032 | – | – | – | 6.664 | 4 |
| – | – | – | – | – | 256 | – | – | 3.340 | 1 |
| – | – | – | – | – | 512 | – | – | 3.390 | 2 |
| – | – | – | – | – | – | 500 | – | 7.345 | 1 |
| – | – | – | – | – | – | 1000 | – | 7.022 | 2 |
| – | – | – | – | – | – | 2000 | – | 3.349 | 3 |
| – | – | – | – | – | – | – | 0 | 3.367 | 1 |
| – | – | – | – | – | – | – | 0.1 | 3.330 | 2 |
| – | – | – | – | – | – | – | 0.3 | 3.332 | 3 |

Table 98: Hyperparameter ablation for Sophia on 130m on 8x Chinchilla Data

| $\beta_1$ | $\beta_2$ | $\epsilon$ | $\gamma$ | $\eta$ | BSZ | warmup | $\lambda$ | Loss | Link |
|------|------|------|--------|-------|-----|--------|-----|-------|------|
| 0.95 | 0.95 | 1e-07 | 0.0125 | 0.002 | 128 | 4000 | 0.2 | 3.259 | 0 |
| 0.8 | – | – | – | – | – | – | – | 3.291 | 1 |
| 0.9 | – | – | – | – | – | – | – | 3.265 | 2 |
| – | 0.9 | – | – | – | – | – | – | 3.259 | 1 |
| – | 0.98 | – | – | – | – | – | – | 3.260 | 2 |
| – | 0.99 | – | – | – | – | – | – | 3.260 | 3 |
| – | – | 1e-17 | – | – | – | – | – | 3.266 | 1 |
| – | – | 1e-12 | – | – | – | – | – | 3.265 | 2 |
| – | – | – | – | 0.004 | – | – | – | 3.265 | 1 |
| – | – | – | – | 0.008 | – | – | – | 3.308 | 2 |
| – | – | – | – | – | 256 | – | – | 3.265 | 1 |
| – | – | – | – | – | – | 1000 | – | 3.277 | 1 |
| – | – | – | – | – | – | 2000 | – | 3.260 | 2 |
| – | – | – | – | – | – | – | 0 | 3.300 | 1 |

Table 99: Hyperparameter ablation for Sophia on 300m on 1x Chinchilla Data

| $\beta_1$ | $\beta_2$ | $\epsilon$ | $\gamma$ | $\eta$ | BSZ | warmup | $\lambda$ | Loss | Link |
|------|-------|------|---------|-------|-----|--------|-----|-------|------|
| 0.9 | 0.9 | 1e-07 | 0.0125 | 0.004 | 128 | 4000 | 0.1 | 3.267 | 0 |
| 0.8 | – | – | – | – | – | – | – | 3.288 | 1 |
| 0.95 | – | – | – | – | – | – | – | 7.390 | 2 |
| 0.98 | – | – | – | – | – | – | – | 7.525 | 3 |
| – | 0.95 | – | – | – | – | – | – | 3.270 | 1 |
| – | 0.98 | – | – | – | – | – | – | 3.275 | 2 |
| – | 0.99 | – | – | – | – | – | – | 3.280 | 3 |
| – | 0.995 | – | – | – | – | – | – | 3.280 | 4 |
| – | – | 1e-17 | – | – | – | – | – | 3.289 | 1 |
| – | – | 1e-12 | – | – | – | – | – | 3.289 | 2 |
| – | – | – | 0.00625 | – | – | – | – | 3.273 | 1 |
| – | – | – | 0.025 | – | – | – | – | 3.271 | 2 |
| – | – | – | 0.05 | – | – | – | – | 3.274 | 3 |
| – | – | – | – | 0.002 | – | – | – | 3.276 | 1 |
| – | – | – | – | 0.008 | – | – | – | 6.966 | 2 |
| – | – | – | – | 0.016 | – | – | – | 7.142 | 3 |
| – | – | – | – | 0.032 | – | – | – | 6.811 | 4 |
| – | – | – | – | – | 256 | – | – | 3.298 | 1 |
| – | – | – | – | – | – | 500 | – | 7.149 | 1 |
| – | – | – | – | – | – | 1000 | – | 7.093 | 2 |
| – | – | – | – | – | – | 2000 | – | 7.382 | 3 |
| – | – | – | – | – | – | – | 0 | 3.288 | 1 |
| – | – | – | – | – | – | – | 0.2 | 3.274 | 2 |
| – | – | – | – | – | – | – | 0.3 | 3.281 | 3 |

Table 100: Hyperparameter ablation for Sophia on 520m on 1x Chinchilla Data

| $\beta_1$ | $\beta_2$ | $\epsilon$ | $\gamma$ | $\eta$ | BSZ | warmup | $\lambda$ | Loss | Link |
|------|------|-------|---------|-------|------|--------|------|-------|------|
| 0.95 | 0.9 | 1e-07 | 0.0125 | 0.002 | 128 | 4000 | 0.3 | 3.106 | 0 |
| 0.8 | – | – | – | – | – | – | – | 3.125 | 1 |
| 0.9 | – | – | – | – | – | – | – | 3.109 | 2 |
| 0.98 | – | – | – | – | – | – | – | 3.133 | 3 |
| – | 0.95 | – | – | – | – | – | – | 3.111 | 1 |
| – | 0.98 | – | – | – | – | – | – | 3.111 | 2 |
| – | 0.99 | – | – | – | – | – | – | 6.571 | 3 |
| – | – | 1e-17 | – | – | – | – | – | 3.116 | 1 |
| – | – | 1e-12 | – | – | – | – | – | 3.116 | 2 |
| – | – | – | 0.00625 | – | – | – | – | 3.107 | 1 |
| – | – | – | 0.025 | – | – | – | – | 3.107 | 2 |
| – | – | – | – | 0.004 | – | – | – | 6.940 | 1 |
| – | – | – | – | – | – | 1000 | – | 6.908 | 1 |
| – | – | – | – | – | – | 2000 | – | 6.823 | 2 |
| – | – | – | – | – | – | – | 0 | 3.148 | 1 |
| – | – | – | – | – | – | – | 0.1 | 3.113 | 2 |
| – | – | – | – | – | – | – | 0.2 | 3.105 | 3 |

# F  HYPERPARAMETER ABLATION IN PHASE II

We reported the results for the optimizers we swept in Phase II. The result is formulated in the same way as in Phase I.

## F.1  SWEEPING RESULTS FOR ADAMW

Table 101: Hyperparameter ablation for AdamW on 300m on 2x Chinchilla Data

| $\beta_1$ | $\beta_2$ | $\epsilon$ | $\eta$ | $g_{\mathrm{norm}}$ | $\eta_{min}$ | BSZ | warmup | $\lambda$ | Loss | Link |
|-----|------|-------|-------|-------|-------|-----|-------|------|-------|------|
| 0.9 | 0.98 | 1e-15 | 0.008 | 1 | 0 | 128 | 2000 | 0.1 | 3.166 | 0 |
| – | – | – | 0.004 | – | – | – | – | – | 3.167 | 1 |
| – | – | – | – | – | – | 256 | – | – | 3.170 | 2 |
| – | – | – | – | – | – | – | – | 0.2 | 3.183 | 3 |

Table 102: Hyperparameter ablation for AdamW on 300m on 4x Chinchilla Data

| $\beta_1$ | $\beta_2$ | $\epsilon$ | $\eta$ | $g_{\mathrm{norm}}$ | $\eta_{min}$ | BSZ | warmup | $\lambda$ | Loss | Link |
|-----|------|-------|-------|-------|-------|-----|-------|------|-------|------|
| 0.9 | 0.98 | 1e-10 | 0.008 | 2 | 0 | 256 | 2000 | 0.1 | 3.094 | 0 |
| – | – | – | 0.004 | – | – | – | – | – | 3.101 | 1 |
| – | – | – | – | – | – | 128 | – | – | 3.103 | 2 |
| – | – | – | – | – | – | – | – | 0.2 | 3.103 | 3 |

Table 103: Hyperparameter ablation for AdamW on 300m on 8x Chinchilla Data

| $\beta_1$ | $\beta_2$ | $\epsilon$ | $\eta$ | $g_{\mathrm{norm}}$ | $\eta_{min}$ | BSZ | warmup | $\lambda$ | Loss | Link |
|---|---|---|---|---|---|---|---|---|---|---|
| 0.9 | 0.98 | 1e-10 | 0.008 | 2 | 0 | 256 | 2000 | 0.1 | 3.043 | 0 |
| – | – | – | 0.004 | – | – | – | – | – | 3.042 | 1 |
| – | – | – | – | – | – | 128 | – | – | 3.057 | 2 |
| – | – | – | – | – | – | – | – | 0.2 | 3.059 | 3 |

Table 104: Hyperparameter ablation for AdamW on 520m on 2x Chinchilla Data

| $\beta_1$ | $\beta_2$ | $\epsilon$ | $\eta$ | $g_{\mathrm{norm}}$ | $\eta_{min}$ | BSZ | warmup | $\lambda$ | Loss | Link |
|---|---|---|---|---|---|---|---|---|---|---|
| 0.9 | 0.98 | 1e-10 | 0.004 | 1 | 0 | 256 | 1000 | 0.2 | 3.023 | 0 |
| – | – | – | – | – | – | 128 | – | – | 6.654 | 1 |
| – | – | – | – | – | – | – | – | 0.1 | 3.025 | 2 |

Table 105: Hyperparameter ablation for AdamW on 520m on 4x Chinchilla Data

| $\beta_1$ | $\beta_2$ | $\epsilon$ | $\eta$ | $g_{\mathrm{norm}}$ | $\eta_{min}$ | BSZ | warmup | $\lambda$ | Loss | Link |
|---|---|---|---|---|---|---|---|---|---|---|
| 0.9 | 0.98 | 1e-10 | 0.004 | 1 | 0 | 256 | 1000 | 0.1 | 2.958 | 0 |
| – | – | – | 0.008 | – | – | – | – | – | 7.075 | 1 |
| – | – | – | – | – | – | 128 | – | – | 7.139 | 2 |
| – | – | – | – | – | – | – | – | 0.2 | 2.962 | 3 |

Table 106: Hyperparameter ablation for AdamW on 520m on 8x Chinchilla Data

| $\beta_1$ | $\beta_2$ | $\epsilon$ | $\eta$ | $g_{\mathrm{norm}}$ | $\eta_{min}$ | BSZ | warmup | $\lambda$ | Loss | Link |
|---|---|---|---|---|---|---|---|---|---|---|
| 0.9 | 0.98 | 1e-10 | 0.004 | 1 | 0 | 256 | 1000 | 0.1 | 2.913 | 0 |
| – | – | – | 0.008 | – | – | – | – | – | 7.183 | 1 |
| – | – | – | – | – | – | 128 | – | – | 6.932 | 2 |
| – | – | – | – | – | – | – | – | 0.2 | 2.921 | 3 |

## F.2 SWEEPING RESULTS FOR CAUTIOUS

Table 107: Hyperparameter ablation for Cautious on 300m on 2x Chinchilla Data

| $\beta_1$ | $\beta_2$ | $\epsilon$ | $\eta$ | $g_{\mathrm{norm}}$ | $\eta_{min}$ | BSZ | warmup | $\lambda$ | Loss | Link |
|---|---|---|---|---|---|---|---|---|---|---|
| 0.98 | 0.98 | 1e-25 | 0.008 | 1 | 0 | 256 | 2000 | 0.1 | 3.165 | 0 |
| – | – | – | 0.004 | – | – | – | – | – | 3.175 | 1 |
| – | – | – | – | – | – | 128 | – | – | 3.171 | 2 |

Table 108: Hyperparameter ablation for Cautious on 300m on 4x Chinchilla Data

| $\beta_1$ | $\beta_2$ | $\epsilon$ | $\eta$ | $g_{\mathrm{norm}}$ | $\eta_{min}$ | BSZ | warmup | $\lambda$ | Loss | Link |
|---|---|---|---|---|---|---|---|---|---|---|
| 0.98 | 0.98 | 1e-25 | 0.008 | 1 | 0 | 256 | 2000 | 0.1 | 3.094 | 0 |
| – | – | – | 0.004 | – | – | – | – | – | 3.098 | 1 |
| – | – | – | – | – | – | 128 | – | – | 3.114 | 2 |

Table 109: Hyperparameter ablation for Cautious on 300m on 8x Chinchilla Data

| $\beta_1$ | $\beta_2$ | $\epsilon$ | $\eta$ | $g_{\mathrm{norm}}$ | $\eta_{min}$ | BSZ | warmup | $\lambda$ | Loss | Link |
|---|---|---|---|---|---|---|---|---|---|---|
| 0.98 | 0.98 | 1e-25 | 0.008 | 1 | 0 | 256 | 2000 | 0.1 | 3.043 | 0 |
| – | – | – | 0.004 | – | – | – | – | – | 3.041 | 1 |
| – | – | – | – | – | – | 128 | – | – | 3.071 | 2 |

Table 110: Hyperparameter ablation for Cautious on 520m on 2x Chinchilla Data

| $\beta_1$ | $\beta_2$ | $\epsilon$ | $\eta$ | $g_{\mathrm{norm}}$ | $\eta_{min}$ | BSZ | warmup | $\lambda$ | Loss | Link |
|---|---|---|---|---|---|---|---|---|---|---|
| 0.98 | 0.98 | 1e-25 | 0.008 | 1 | 0 | 256 | 2000 | 0.1 | 3.017 | 0 |
| – | – | – | 0.004 | – | – | – | – | – | 3.019 | 1 |
| – | – | – | – | – | – | 128 | – | – | >10 | 2 |

Table 111: Hyperparameter ablation for Cautious on 520m on 4x Chinchilla Data

| $\beta_1$ | $\beta_2$ | $\epsilon$ | $\eta$ | $g_{\mathrm{norm}}$ | $\eta_{min}$ | BSZ | warmup | $\lambda$ | Loss | Link |
|---|---|---|---|---|---|---|---|---|---|---|
| 0.98 | 0.98 | 1e-25 | 0.004 | 1 | 0 | 256 | 2000 | 0.1 | 2.956 | 0 |
| – | – | – | 0.008 | – | – | – | – | – | 2.959 | 1 |
| – | – | – | – | – | – | 128 | – | – | 2.971 | 2 |

Table 112: Hyperparameter ablation for Cautious on 520m on 8x Chinchilla Data

| $\beta_1$ | $\beta_2$ | $\epsilon$ | $\eta$ | $g_{\mathrm{norm}}$ | $\eta_{min}$ | BSZ | warmup | $\lambda$ | Loss | Link |
|---|---|---|---|---|---|---|---|---|---|---|
| 0.98 | 0.98 | 1e-25 | 0.004 | 1 | 0 | 256 | 2000 | 0.1 | 2.910 | 0 |
| – | – | – | 0.008 | – | – | – | – | – | 2.919 | 1 |
| – | – | – | – | – | – | 128 | – | – | 2.931 | 2 |

## F.3 SWEEPING RESULTS FOR LION

Table 113: Hyperparameter ablation for Lion on 300m on 2x Chinchilla Data

| $\beta_1$ | $\beta_2$ | $\eta$ | $g_{\mathrm{norm}}$ | $\eta_{min}$ | BSZ | warmup | $\lambda$ | Loss | Link |
|---|---|---|---|---|---|---|---|---|---|
| 0.9 | 0.98 | 0.001 | 1 | 0 | 128 | 2000 | 0.7 | 3.170 | 0 |
| – | 0.9 | – | – | – | – | – | – | 3.189 | 1 |
| – | 0.95 | – | – | – | – | – | – | 3.175 | 2 |
| – | – | 0.0005 | – | – | – | – | – | 3.172 | 3 |
| – | – | – | – | – | 256 | – | – | 3.183 | 4 |

Table 114: Hyperparameter ablation for Lion on 300m on 4x Chinchilla Data

| $\beta_1$ | $\beta_2$ | $\eta$ | $g_{\text{norm}}$ | $\eta_{min}$ | BSZ | warmup | $\lambda$ | Loss | Link |
|---|---|---|---|---|---|---|---|---|---|
| 0.9 | 0.98 | 0.001 | 1 | 0 | 256 | 2000 | 0.7 | 3.100 | 0 |
| – | 0.9 | – | – | – | – | – | – | 3.114 | 1 |
| – | 0.95 | – | – | – | – | – | – | 3.103 | 2 |
| – | – | 0.0005 | – | – | – | – | – | 3.105 | 3 |
| – | – | – | – | – | 128 | – | – | 3.104 | 4 |

Table 115: Hyperparameter ablation for Lion on 300m on 8x Chinchilla Data

| $\beta_1$ | $\beta_2$ | $\eta$ | $g_{\text{norm}}$ | $\eta_{min}$ | BSZ | warmup | $\lambda$ | Loss | Link |
|---|---|---|---|---|---|---|---|---|---|
| 0.9 | 0.98 | 0.001 | 1 | 0 | 256 | 2000 | 0.7 | 3.046 | 0 |
| – | 0.9 | – | – | – | – | – | – | 3.058 | 1 |
| – | 0.95 | – | – | – | – | – | – | 3.050 | 2 |
| – | – | 0.0005 | – | – | – | – | – | 3.043 | 3 |
| – | – | – | – | – | 128 | – | – | 3.061 | 4 |

Table 116: Hyperparameter ablation for Lion on 520m on 2x Chinchilla Data

| $\beta_1$ | $\beta_2$ | $\eta$ | $g_{\text{norm}}$ | $\eta_{min}$ | BSZ | warmup | $\lambda$ | Loss | Link |
|---|---|---|---|---|---|---|---|---|---|
| 0.9 | 0.95 | 0.001 | 1 | 0 | 128 | 2000 | 0.6 | 3.029 | 0 |
| – | 0.9 | – | – | – | – | – | – | 3.045 | 1 |
| – | 0.98 | – | – | – | – | – | – | 7.779 | 2 |
| – | – | 0.0005 | – | – | – | – | – | 3.028 | 3 |
| – | – | – | – | – | 256 | – | – | 3.030 | 4 |

Table 117: Hyperparameter ablation for Lion on 520m on 4x Chinchilla Data

| $\beta_1$ | $\beta_2$ | $\eta$ | $g_{\text{norm}}$ | $\eta_{min}$ | BSZ | warmup | $\lambda$ | Loss | Link |
|---|---|---|---|---|---|---|---|---|---|
| 0.9 | 0.95 | 0.001 | 1 | 0 | 256 | 2000 | 0.6 | 2.965 | 0 |
| – | 0.9 | – | – | – | – | – | – | 2.971 | 1 |
| – | 0.98 | – | – | – | – | – | – | 2.965 | 2 |
| – | – | 0.0005 | – | – | – | – | – | 2.966 | 3 |
| – | – | – | – | – | 128 | – | – | 2.975 | 4 |

Table 118: Hyperparameter ablation for Lion on 520m on 8x Chinchilla Data

| $\beta_1$ | $\beta_2$ | $\eta$ | $g_{\text{norm}}$ | $\eta_{min}$ | BSZ | warmup | $\lambda$ | Loss | Link |
|---|---|---|---|---|---|---|---|---|---|
| 0.9 | 0.95 | 0.0005 | 1 | 0 | 256 | 2000 | 0.6 | 2.915 | 0 |
| – | 0.9 | – | – | – | – | – | – | 2.922 | 1 |
| – | 0.98 | – | – | – | – | – | – | 2.915 | 2 |
| – | – | 0.001 | – | – | – | – | – | 2.920 | 3 |
| – | – | – | – | – | 128 | – | – | 2.922 | 4 |

### F.4 SWEEPING RESULTS FOR MARS

Table 119: Hyperparameter ablation for Mars on 300m on 2x Chinchilla Data

| $\beta_1$ | $\beta_2$ | $\epsilon$ | $\gamma$ | $\eta$ | $g_{\mathrm{norm}}$ | $\eta_{min}$ | BSZ | warmup | $\lambda$ | Loss | Link |
|---|---|---|---|---|---|---|---|---|---|---|---|
| 0.98 | 0.99 | 1e-25 | 0.05 | 0.008 | 1 | 0 | 128 | 1000 | 0.1 | 3.158 | 0 |
| 0.9 | – | – | – | – | – | – | – | – | – | 3.174 | 1 |
| 0.95 | – | – | – | – | – | – | – | – | – | 3.156 | 2 |
| – | – | – | – | 0.004 | – | – | – | – | – | 3.168 | 3 |
| – | – | – | – | – | – | – | 256 | – | – | 3.171 | 4 |
| – | – | – | – | – | – | – | – | 2000 | – | 3.159 | 5 |
| – | – | – | – | – | – | – | – | 4000 | – | 3.167 | 6 |

Table 120: Hyperparameter ablation for Mars on 300m on 4x Chinchilla Data

| $\beta_1$ | $\beta_2$ | $\epsilon$ | $\gamma$ | $\eta$ | $g_{\mathrm{norm}}$ | $\eta_{min}$ | BSZ | warmup | $\lambda$ | Loss | Link |
|---|---|---|---|---|---|---|---|---|---|---|---|
| 0.98 | 0.99 | 1e-25 | 0.05 | 0.008 | 1 | 0 | 128 | 1000 | 0.1 | 3.097 | 0 |
| 0.9 | – | – | – | – | – | – | – | – | – | 3.111 | 1 |
| 0.95 | – | – | – | – | – | – | – | – | – | 3.096 | 2 |
| – | – | – | – | 0.004 | – | – | – | – | – | 3.095 | 3 |
| – | – | – | – | – | – | – | 256 | – | – | 3.098 | 4 |
| – | – | – | – | – | – | – | – | 2000 | – | 3.097 | 5 |
| – | – | – | – | – | – | – | – | 4000 | – | 3.101 | 6 |

Table 121: Hyperparameter ablation for Mars on 300m on 8x Chinchilla Data

| $\beta_1$ | $\beta_2$ | $\epsilon$ | $\gamma$ | $\eta$ | $g_{\mathrm{norm}}$ | $\eta_{min}$ | BSZ | warmup | $\lambda$ | Loss | Link |
|---|---|---|---|---|---|---|---|---|---|---|---|
| 0.98 | 0.99 | 1e-25 | 0.05 | 0.008 | 1 | 0 | 256 | 1000 | 0.1 | 3.040 | 0 |
| 0.9 | – | – | – | – | – | – | – | – | – | 3.049 | 1 |
| 0.95 | – | – | – | – | – | – | – | – | – | 3.038 | 2 |
| – | – | – | – | 0.004 | – | – | – | – | – | 3.046 | 3 |
| – | – | – | – | – | – | – | 128 | – | – | 3.050 | 4 |
| – | – | – | – | – | – | – | – | 2000 | – | 3.049 | 5 |
| – | – | – | – | – | – | – | – | 4000 | – | 3.043 | 6 |

Table 122: Hyperparameter ablation for Mars on 520m on 2x Chinchilla Data

| $\beta_1$ | $\beta_2$ | $\epsilon$ | $\gamma$ | $\eta$ | $g_{\mathrm{norm}}$ | $\eta_{min}$ | BSZ | warmup | $\lambda$ | Loss | Link |
|---|---|---|---|---|---|---|---|---|---|---|---|
| 0.95 | 0.98 | 1e-25 | 0.025 | 0.008 | 1 | 0 | 256 | 2000 | 0.1 | 3.015 | 0 |
| 0.9 | – | – | – | – | – | – | – | – | – | 3.019 | 1 |
| 0.98 | – | – | – | – | – | – | – | – | – | 3.014 | 2 |
| – | – | – | – | 0.004 | – | – | – | – | – | 3.019 | 3 |
| – | – | – | – | – | – | – | 128 | – | – | 3.025 | 4 |
| – | – | – | – | – | – | – | – | 1000 | – | 3.023 | 5 |
| – | – | – | – | – | – | – | – | 4000 | – | 3.019 | 6 |

Table 123: Hyperparameter ablation for Mars on 520m on 4x Chinchilla Data

| $\beta_1$ | $\beta_2$ | $\epsilon$ | $\gamma$ | $\eta$ | $g_{\text{norm}}$ | $\eta_{min}$ | BSZ | warmup | $\lambda$ | Loss | Link |
|---|---|---|---|---|---|---|---|---|---|---|---|
| 0.95 | 0.98 | 1e-25 | 0.025 | 0.008 | 1 | 0 | 256 | 2000 | 0.1 | 2.955 | 0 |
| 0.9 | – | – | – | – | – | – | – | – | – | 2.960 | 1 |
| 0.98 | – | – | – | – | – | – | – | – | – | 2.953 | 2 |
| – | – | – | – | 0.004 | – | – | – | – | – | 2.953 | 3 |
| – | – | – | – | – | – | – | 128 | – | – | 2.974 | 4 |
| – | – | – | – | – | – | – | – | 1000 | – | 2.964 | 5 |
| – | – | – | – | – | – | – | – | 4000 | – | 2.956 | 6 |

Table 124: Hyperparameter ablation for Mars on 520m on 8x Chinchilla Data

| $\beta_1$ | $\beta_2$ | $\epsilon$ | $\gamma$ | $\eta$ | $g_{\text{norm}}$ | $\eta_{min}$ | BSZ | warmup | $\lambda$ | Loss | Link |
|---|---|---|---|---|---|---|---|---|---|---|---|
| 0.95 | 0.98 | 1e-25 | 0.025 | 0.004 | 1 | 0 | 256 | 2000 | 0.1 | 2.906 | 0 |
| 0.9 | – | – | – | – | – | – | – | – | – | 2.908 | 1 |
| 0.98 | – | – | – | – | – | – | – | – | – | 2.906 | 2 |
| – | – | – | – | 0.008 | – | – | – | – | – | 2.917 | 3 |
| – | – | – | – | – | – | – | 128 | – | – | 2.916 | 4 |
| – | – | – | – | – | – | – | – | 1000 | – | 2.906 | 5 |
| – | – | – | – | – | – | – | – | 4000 | – | 2.907 | 6 |

## F.5 SWEEPING RESULTS FOR NADAMW

Table 125: Hyperparameter ablation for NAdamW on 300m on 2x Chinchilla Data

| $\beta_1$ | $\beta_2$ | $\epsilon$ | $\eta$ | $g_{\text{norm}}$ | $\eta_{min}$ | BSZ | warmup | $\lambda$ | Loss | Link |
|---|---|---|---|---|---|---|---|---|---|---|
| 0.95 | 0.98 | 1e-10 | 0.008 | 1 | 0 | 128 | 2000 | 0.1 | 3.160 | 0 |
| – | – | – | 0.004 | – | – | – | – | – | 3.160 | 1 |
| – | – | – | – | – | – | 256 | – | – | 3.165 | 2 |

Table 126: Hyperparameter ablation for NAdamW on 300m on 4x Chinchilla Data

| $\beta_1$ | $\beta_2$ | $\epsilon$ | $\eta$ | $g_{\text{norm}}$ | $\eta_{min}$ | BSZ | warmup | $\lambda$ | Loss | Link |
|---|---|---|---|---|---|---|---|---|---|---|
| 0.95 | 0.98 | 1e-10 | 0.008 | 1 | 0 | 256 | 2000 | 0.1 | 3.090 | 0 |
| – | – | – | 0.004 | – | – | – | – | – | 3.097 | 1 |
| – | – | – | – | – | – | 128 | – | – | 3.098 | 2 |

Table 127: Hyperparameter ablation for NAdamW on 300m on 8x Chinchilla Data

| $\beta_1$ | $\beta_2$ | $\epsilon$ | $\eta$ | $g_{\text{norm}}$ | $\eta_{min}$ | BSZ | warmup | $\lambda$ | Loss | Link |
|---|---|---|---|---|---|---|---|---|---|---|
| 0.95 | 0.98 | 1e-10 | 0.008 | 1 | 0 | 256 | 2000 | 0.1 | 3.039 | 0 |
| – | – | – | 0.004 | – | – | – | – | – | 3.039 | 1 |
| – | – | – | – | – | – | 128 | – | – | 3.055 | 2 |

Table 128: Hyperparameter ablation for NAdamW on 520m on 2x Chinchilla Data

| $\beta_1$ | $\beta_2$ | $\epsilon$ | $\eta$ | $g_{\mathrm{norm}}$ | $\eta_{min}$ | BSZ | warmup | $\lambda$ | Loss | Link |
|---|---|---|---|---|---|---|---|---|---|---|
| 0.98 | 0.98 | 1e-10 | 0.004 | 1 | 0 | 128 | 4000 | 0.1 | 3.013 | 0 |
| – | – | – | 0.008 | – | – | – | – | – | 3.023 | 1 |
| – | – | – | – | – | – | 256 | – | – | 3.020 | 2 |

Table 129: Hyperparameter ablation for NAdamW on 520m on 4x Chinchilla Data

| $\beta_1$ | $\beta_2$ | $\epsilon$ | $\eta$ | $g_{\mathrm{norm}}$ | $\eta_{min}$ | BSZ | warmup | $\lambda$ | Loss | Link |
|---|---|---|---|---|---|---|---|---|---|---|
| 0.98 | 0.98 | 1e-10 | 0.004 | 1 | 0 | 128 | 4000 | 0.1 | 2.955 | 0 |
| – | – | – | 0.008 | – | – | – | – | – | 2.971 | 1 |
| – | – | – | – | – | – | 256 | – | – | 2.954 | 2 |

Table 130: Hyperparameter ablation for NAdamW on 520m on 8x Chinchilla Data

| $\beta_1$ | $\beta_2$ | $\epsilon$ | $\eta$ | $g_{\mathrm{norm}}$ | $\eta_{min}$ | BSZ | warmup | $\lambda$ | Loss | Link |
|---|---|---|---|---|---|---|---|---|---|---|
| 0.98 | 0.98 | 1e-10 | 0.004 | 1 | 0 | 256 | 4000 | 0.1 | 2.907 | 0 |
| – | – | – | 0.008 | – | – | – | – | – | 2.910 | 1 |
| – | – | – | – | – | – | 128 | – | – | 2.913 | 2 |

## F.6 SWEEPING RESULTS FOR ADAM-MINI

Table 131: Hyperparameter ablation for Adam-Mini on 300m on 2x Chinchilla Data

| $\beta_1$ | $\beta_2$ | $\epsilon$ | $\eta$ | $g_{\mathrm{norm}}$ | $\eta_{min}$ | BSZ | warmup | $\lambda$ | Loss | Link |
|---|---|---|---|---|---|---|---|---|---|---|
| 0.9 | 0.98 | 1e-25 | 0.004 | 2 | 0 | 128 | 2000 | 0.2 | 3.178 | 0 |
| – | – | – | 0.002 | – | – | – | – | – | 3.180 | 1 |
| – | – | – | – | – | – | 256 | – | – | 6.960 | 2 |
| – | – | – | – | – | – | – | 4000 | – | 3.183 | 3 |
| – | – | – | – | – | – | – | – | 0.1 | 3.179 | 4 |

Table 132: Hyperparameter ablation for Adam-Mini on 300m on 4x Chinchilla Data

| $\beta_1$ | $\beta_2$ | $\epsilon$ | $\eta$ | $g_{\mathrm{norm}}$ | $\eta_{min}$ | BSZ | warmup | $\lambda$ | Loss | Link |
|---|---|---|---|---|---|---|---|---|---|---|
| 0.9 | 0.98 | 1e-25 | 0.004 | 2 | 0 | 128 | 2000 | 0.1 | 3.103 | 0 |
| – | – | – | 0.002 | – | – | – | – | – | 3.111 | 1 |
| – | – | – | – | – | – | 256 | – | – | 3.109 | 2 |
| – | – | – | – | – | – | – | 4000 | – | 3.104 | 3 |
| – | – | – | – | – | – | – | – | 0.2 | 3.111 | 4 |

Table 133: Hyperparameter ablation for Adam-Mini on 300m on 8x Chinchilla Data

| $\beta_1$ | $\beta_2$ | $\epsilon$ | $\eta$ | $g_{\mathrm{norm}}$ | $\eta_{min}$ | BSZ | warmup | $\lambda$ | Loss | Link |
|---|---|---|---|---|---|---|---|---|---|---|
| 0.9 | 0.98 | 1e-25 | 0.002 | 2 | 0 | 128 | 2000 | 0.2 | 3.049 | 0 |
| – | – | – | 0.004 | – | – | – | – | – | 3.064 | 1 |
| – | – | – | – | – | – | 256 | – | – | 3.052 | 2 |
| – | – | – | – | – | – | – | 4000 | – | 3.050 | 3 |
| – | – | – | – | – | – | – | – | 0.1 | 3.051 | 4 |

Table 134: Hyperparameter ablation for Adam-Mini on 520m on 2x Chinchilla Data

| $\beta_1$ | $\beta_2$ | $\epsilon$ | $\eta$ | $g_{\mathrm{norm}}$ | $\eta_{min}$ | BSZ | warmup | $\lambda$ | Loss | Link |
|---|---|---|---|---|---|---|---|---|---|---|
| 0.9 | 0.98 | 1e-10 | 0.004 | 1 | 0 | 128 | 4000 | 0.1 | 3.027 | 0 |
| – | – | – | 0.002 | – | – | – | – | – | 3.031 | 1 |
| – | – | – | – | – | – | 256 | – | – | 3.032 | 2 |
| – | – | – | – | – | – | – | 2000 | – | 7.359 | 3 |
| – | – | – | – | – | – | – | – | 0.2 | 3.037 | 4 |

Table 135: Hyperparameter ablation for Adam-Mini on 520m on 4x Chinchilla Data

| $\beta_1$ | $\beta_2$ | $\epsilon$ | $\eta$ | $g_{\mathrm{norm}}$ | $\eta_{min}$ | BSZ | warmup | $\lambda$ | Loss | Link |
|---|---|---|---|---|---|---|---|---|---|---|
| 0.9 | 0.98 | 1e-10 | 0.004 | 1 | 0 | 128 | 4000 | 0.1 | 2.966 | 0 |
| – | – | – | 0.002 | – | – | – | – | – | 2.963 | 1 |
| – | – | – | – | – | – | 256 | – | – | 2.963 | 2 |
| – | – | – | – | – | – | – | 2000 | – | 7.529 | 3 |
| – | – | – | – | – | – | – | – | 0.2 | 2.981 | 4 |

Table 136: Hyperparameter ablation for Adam-Mini on 520m on 8x Chinchilla Data

| $\beta_1$ | $\beta_2$ | $\epsilon$ | $\eta$ | $g_{\mathrm{norm}}$ | $\eta_{min}$ | BSZ | warmup | $\lambda$ | Loss | Link |
|---|---|---|---|---|---|---|---|---|---|---|
| 0.9 | 0.98 | 1e-10 | 0.004 | 1 | 0 | 256 | 4000 | 0.1 | 2.912 | 0 |
| – | – | – | 0.002 | – | – | – | – | – | 2.918 | 1 |
| – | – | – | – | – | – | 128 | – | – | 2.921 | 2 |
| – | – | – | – | – | – | – | 2000 | – | 7.449 | 3 |
| – | – | – | – | – | – | – | – | 0.2 | 3.025 | 4 |

## F.7 SWEEPING RESULTS FOR MUON

Table 137: Hyperparameter ablation for Muon on 300m on 2x Chinchilla Data

| $\eta_{adam}$ | $\beta_1$ | $\beta_2$ | Decay(WSD) | $\epsilon$ | $\eta$ | Schedule | $g_{\mathrm{norm}}$ | $\eta_{min}$ | $\beta_{\mathrm{muon}}$ | $\epsilon_{muon}$ | BSZ | warmup | $\lambda$ | Loss | Link |
|---|---|---|---|---|---|---|---|---|---|---|---|---|---|---|---|
| 0.0024 | 0.8 | 0.98 | 0.8 | 1e-15 | 0.008 | linear | 1 | 0 | 0.98 | 1e-05 | 128 | 0 | 0.1 | 3.143 | 0 |
| – | – | – | – | – | 0.004 | – | – | – | – | – | – | – | – | 3.144 | 1 |
| – | – | – | – | – | – | – | – | – | – | – | 256 | – | – | 3.145 | 2 |

Table 138: Hyperparameter ablation for Muon on 300m on 4x Chinchilla Data

| $\eta_{adam}$ | $\beta_1$ | $\beta_2$ | Decay(WSD) | $\epsilon$ | $\eta$ | Schedule | $g_{norm}$ | $\eta_{min}$ | $\beta_{muon}$ | $\epsilon_{muon}$ | BSZ | warmup | $\lambda$ | Loss | Link |
|---|---|---|---|---|---|---|---|---|---|---|---|---|---|---|---|
| 0.0012 | 0.8 | 0.98 | 0.8 | 1e-15 | 0.004 | linear | 1 | 0 | 0.98 | 1e-05 | 128 | 0 | 0.1 | 3.079 | 0 |
| – | – | – | – | – | 0.008 | – | – | – | – | – | – | – | – | 3.088 | 1 |
| – | – | – | – | – | – | – | – | – | – | – | 256 | – | – | 3.083 | 2 |

Table 139: Hyperparameter ablation for Muon on 300m on 8x Chinchilla Data

| $\eta_{adam}$ | $\beta_1$ | $\beta_2$ | Decay(WSD) | $\epsilon$ | $\eta$ | Schedule | $g_{norm}$ | $\eta_{min}$ | $\beta_{muon}$ | $\epsilon_{muon}$ | BSZ | warmup | $\lambda$ | Loss | Link |
|---|---|---|---|---|---|---|---|---|---|---|---|---|---|---|---|
| 0.0024 | 0.8 | 0.98 | 0.8 | 1e-15 | 0.008 | linear | 1 | 0 | 0.98 | 1e-05 | 256 | 0 | 0.1 | 3.032 | 0 |
| – | – | – | – | – | 0.004 | – | – | – | – | – | – | – | – | 3.029 | 1 |
| – | – | – | – | – | – | – | – | – | – | – | 128 | – | – | 3.049 | 2 |

Table 140: Hyperparameter ablation for Muon on 520m on 2x Chinchilla Data

| $\eta_{adam}$ | $\beta_1$ | $\beta_2$ | Decay(WSD) | $\epsilon$ | $\eta$ | Schedule | $g_{norm}$ | $\eta_{min}$ | $\beta_{muon}$ | $\epsilon_{muon}$ | BSZ | warmup | $\lambda$ | Loss | Link |
|---|---|---|---|---|---|---|---|---|---|---|---|---|---|---|---|
| 0.0012 | 0.8 | 0.98 | 1 | 1e-25 | 0.004 | linear | 2 | 0 | 0.98 | 1e-05 | 128 | 0 | 0.1 | 3.002 | 0 |
| – | – | – | – | – | 0.008 | – | – | – | – | – | – | – | – | 3.008 | 1 |
| – | – | – | – | – | – | – | – | – | – | – | 256 | – | – | 3.009 | 2 |

Table 141: Hyperparameter ablation for Muon on 520m on 4x Chinchilla Data

| $\eta_{adam}$ | $\beta_1$ | $\beta_2$ | Decay(WSD) | $\epsilon$ | $\eta$ | Schedule | $g_{norm}$ | $\eta_{min}$ | $\beta_{muon}$ | $\epsilon_{muon}$ | BSZ | warmup | $\lambda$ | Loss | Link |
|---|---|---|---|---|---|---|---|---|---|---|---|---|---|---|---|
| 0.0024 | 0.8 | 0.98 | 1 | 1e-25 | 0.008 | linear | 2 | 0 | 0.98 | 1e-05 | 256 | 0 | 0.1 | 2.945 | 0 |
| – | – | – | – | – | 0.004 | – | – | – | – | – | – | – | – | 2.944 | 1 |
| – | – | – | – | – | – | – | – | – | – | – | 128 | – | – | 2.963 | 2 |

Table 142: Hyperparameter ablation for Muon on 520m on 8x Chinchilla Data

| $\eta_{adam}$ | $\beta_1$ | $\beta_2$ | Decay(WSD) | $\epsilon$ | $\eta$ | Schedule | $g_{norm}$ | $\eta_{min}$ | $\beta_{muon}$ | $\epsilon_{muon}$ | BSZ | warmup | $\lambda$ | Loss | Link |
|---|---|---|---|---|---|---|---|---|---|---|---|---|---|---|---|
| 0.0024 | 0.8 | 0.98 | 1 | 1e-25 | 0.008 | linear | 2 | 0 | 0.98 | 1e-05 | 256 | 0 | 0.1 | 2.906 | 0 |
| – | – | – | – | – | 0.004 | – | – | – | – | – | – | – | – | 2.900 | 1 |
| – | – | – | – | – | – | – | – | – | – | – | 128 | – | – | 2.930 | 2 |

## F.8 SWEEPING RESULTS FOR SCION

Table 143: Hyperparameter ablation for Scion on 300m on 2x Chinchilla Data

| $\eta_{adam}$ | $\beta_1$ | Decay(WSD) | $\eta$ | Schedule | $g_{norm}$ | $\eta_{min}$ | $\beta_{muon}$ | $\epsilon_{scion}$ | BSZ | warmup | $\lambda$ | Loss | Link |
|---|---|---|---|---|---|---|---|---|---|---|---|---|---|
| 0.0008 | 0.98 | 0.8 | 0.008 | linear | 2 | 0 | 0.95 | 1e-05 | 128 | 0 | 0.1 | 3.152 | 0 |
| – | – | – | 0.004 | – | – | – | – | – | – | – | – | 3.153 | 1 |
| – | – | – | – | – | – | – | – | – | 256 | – | – | 3.154 | 2 |

Table 144: Hyperparameter ablation for Scion on 300m on 4x Chinchilla Data

| $\eta_{adam}$ | $\beta_1$ | Decay(WSD) | $\eta$ | Schedule | $g_{norm}$ | $\eta_{min}$ | $\beta_{muon}$ | $\epsilon_{scion}$ | BSZ | warmup | $\lambda$ | Loss | Link |
|---|---|---|---|---|---|---|---|---|---|---|---|---|---|
| 0.0008 | 0.98 | 1 | 0.008 | linear | 2 | 0 | 0.95 | 1e-05 | 256 | 0 | 0.1 | 3.086 | 0 |
| – | – | – | 0.004 | – | – | – | – | – | – | – | – | 3.099 | 1 |
| – | – | – | – | – | – | – | – | – | 128 | – | – | 3.090 | 2 |

Table 145: Hyperparameter ablation for Scion on 300m on 8x Chinchilla Data

| $\eta_{adam}$ | $\beta_1$ | Decay(WSD) | $\eta$ | Schedule | $g_{norm}$ | $\eta_{min}$ | $\beta_{muon}$ | $\epsilon_{scion}$ | BSZ | warmup | $\lambda$ | Loss | Link |
|---|---|---|---|---|---|---|---|---|---|---|---|---|---|
| 0.0004 | 0.98 | 0.8 | 0.004 | linear | 2 | 0 | 0.95 | 1e-05 | 128 | 0 | 0.1 | 3.039 | 0 |
| – | – | – | 0.008 | – | – | – | – | – | – | – | – | 3.057 | 1 |
| – | – | – | – | – | – | – | – | – | 256 | – | – | 3.037 | 2 |

Table 146: Hyperparameter ablation for Scion on 520m on 2x Chinchilla Data

| $\eta_{adam}$ | $\beta_1$ | Decay(WSD) | $\eta$ | Schedule | $g_{norm}$ | $\eta_{min}$ | $\beta_{muon}$ | $\epsilon_{scion}$ | BSZ | warmup | $\lambda$ | Loss | Link |
|---|---|---|---|---|---|---|---|---|---|---|---|---|---|
| 0.0004 | 0.98 | 1 | 0.004 | linear | 2 | 0 | 0.95 | 1e-05 | 128 | 0 | 0.1 | 3.007 | 0 |
| – | – | – | 0.008 | – | – | – | – | – | – | – | – | 3.015 | 1 |
| – | – | – | – | – | – | – | – | – | 256 | – | – | 3.020 | 2 |

Table 147: Hyperparameter ablation for Scion on 520m on 4x Chinchilla Data

| $\eta_{adam}$ | $\beta_1$ | Decay(WSD) | $\eta$ | Schedule | $g_{norm}$ | $\eta_{min}$ | $\beta_{muon}$ | $\epsilon_{scion}$ | BSZ | warmup | $\lambda$ | Loss | Link |
|---|---|---|---|---|---|---|---|---|---|---|---|---|---|
| 0.0008 | 0.98 | 1 | 0.008 | linear | 2 | 0 | 0.95 | 1e-05 | 256 | 0 | 0.1 | 2.952 | 0 |
| – | – | – | 0.004 | – | – | – | – | – | – | – | – | 2.952 | 1 |
| – | – | – | – | – | – | – | – | – | 128 | – | – | 2.970 | 2 |

Table 148: Hyperparameter ablation for Scion on 520m on 8x Chinchilla Data

| $\eta_{adam}$ | $\beta_1$ | Decay(WSD) | $\eta$ | Schedule | $g_{norm}$ | $\eta_{min}$ | $\beta_{muon}$ | $\epsilon_{scion}$ | BSZ | warmup | $\lambda$ | Loss | Link |
|---|---|---|---|---|---|---|---|---|---|---|---|---|---|
| 0.0004 | 0.98 | 1 | 0.004 | linear | 2 | 0 | 0.95 | 1e-05 | 256 | 0 | 0.1 | 2.904 | 0 |
| – | – | – | 0.008 | – | – | – | – | – | – | – | – | 2.913 | 1 |
| – | – | – | – | – | – | – | – | – | 128 | – | – | 2.913 | 2 |

## F.9 SWEEPING RESULTS FOR KRON

Table 149: Hyperparameter ablation for Kron on 300m on 2x Chinchilla Data

| $\beta_1$ | blocksize | $\eta$ | $g_{norm}$ | $\eta_{min}$ | NormGrad | Blocking | $Init_{pc}$ | $\eta_{pc}$ | $p_{pc}$ | BSZ | $Step_{pc}$ | warmup | $\lambda$ | Loss | Link |
|---|---|---|---|---|---|---|---|---|---|---|---|---|---|---|---|
| 0.95 | 256 | 0.001 | 1 | 0 | True | True | 1 | 0.2 | 0.1 | 128 | 2000 | 1000 | 0.7 | 3.151 | 0 |
| – | – | 0.0005 | – | – | – | – | – | – | – | – | – | – | – | 3.157 | 1 |

Table 150: Hyperparameter ablation for Kron on 300m on 4x Chinchilla Data

| $\beta_1$ | blocksize | $\eta$ | $g_{\text{norm}}$ | $\eta_{min}$ | NormGrad | Blocking | $Init_{pc}$ | $\eta_{pc}$ | $p_{pc}$ | BSZ | $Step_{pc}$ | warmup | $\lambda$ | Loss | Link |
|---|---|---|---|---|---|---|---|---|---|---|---|---|---|---|---|
| 0.95 | 256 | 0.0005 | 1 | 0 | True | True | 1 | 0.2 | 0.1 | 128 | 2000 | 1000 | 0.7 | 3.083 | 0 |
| – | – | 0.001 | – | – | – | – | – | – | – | – | – | – | – | 3.090 | 1 |

Table 151: Hyperparameter ablation for Kron on 300m on 8x Chinchilla Data

| $\beta_1$ | blocksize | $\eta$ | $g_{\text{norm}}$ | $\eta_{min}$ | NormGrad | Blocking | $Init_{pc}$ | $\eta_{pc}$ | $p_{pc}$ | BSZ | $Step_{pc}$ | warmup | $\lambda$ | Loss | Link |
|---|---|---|---|---|---|---|---|---|---|---|---|---|---|---|---|
| 0.95 | 256 | 0.0005 | 1 | 0 | True | True | 1 | 0.2 | 0.1 | 128 | 2000 | 1000 | 0.7 | 3.031 | 0 |
| – | – | 0.001 | – | – | – | – | – | – | – | – | – | – | – | 3.074 | 1 |

Table 152: Hyperparameter ablation for Kron on 520m on 2x Chinchilla Data

| $\beta_1$ | blocksize | $\eta$ | $g_{\text{norm}}$ | $\eta_{min}$ | NormGrad | Blocking | $Init_{pc}$ | $\eta_{pc}$ | $p_{pc}$ | BSZ | $Step_{pc}$ | warmup | $\lambda$ | Loss | Link |
|---|---|---|---|---|---|---|---|---|---|---|---|---|---|---|---|
| 0.95 | 256 | 0.0005 | 1 | 0 | True | True | 1 | 0.2 | 0.1 | 128 | 2000 | 1000 | 0.5 | 3.009 | 0 |
| – | – | 0.001 | – | – | – | – | – | – | – | – | – | – | – | 3.009 | 1 |

Table 153: Hyperparameter ablation for Kron on 520m on 4x Chinchilla Data

| $\beta_1$ | blocksize | $\eta$ | $g_{\text{norm}}$ | $\eta_{min}$ | NormGrad | Blocking | $Init_{pc}$ | $\eta_{pc}$ | $p_{pc}$ | BSZ | $Step_{pc}$ | warmup | $\lambda$ | Loss | Link |
|---|---|---|---|---|---|---|---|---|---|---|---|---|---|---|---|
| 0.95 | 256 | 0.0005 | 1 | 0 | True | True | 1 | 0.2 | 0.1 | 128 | 2000 | 1000 | 0.5 | 2.946 | 0 |
| – | – | 0.001 | – | – | – | – | – | – | – | – | – | – | – | 2.950 | 1 |

Table 154: Hyperparameter ablation for Kron on 520m on 8x Chinchilla Data

| $\beta_1$ | blocksize | $\eta$ | $g_{\text{norm}}$ | $\eta_{min}$ | NormGrad | Blocking | $Init_{pc}$ | $\eta_{pc}$ | $p_{pc}$ | BSZ | $Step_{pc}$ | warmup | $\lambda$ | Loss | Link |
|---|---|---|---|---|---|---|---|---|---|---|---|---|---|---|---|
| 0.95 | 256 | 0.0005 | 1 | 0 | True | True | 1 | 0.2 | 0.1 | 128 | 2000 | 1000 | 0.5 | 2.900 | 0 |
| – | – | 0.001 | – | – | – | – | – | – | – | – | – | – | – | 2.909 | 1 |
| – | – | – | – | – | – | – | – | – | – | 256 | – | – | – | 2.902 | 2 |

## F.10 SWEEPING RESULTS FOR SOAP

Table 155: Hyperparameter ablation for Soap on 300m on 2x Chinchilla Data

| $\beta_1$ | $\beta_2$ | blocksize | $\epsilon$ | $\eta$ | $g_{\text{norm}}$ | $\eta_{min}$ | Blocking | $f_{pc}$ | $\beta_{shampoo}$ | BSZ | warmup | $\lambda$ | Loss | Link |
|---|---|---|---|---|---|---|---|---|---|---|---|---|---|---|
| 0.95 | 0.99 | 512 | 1e-10 | 0.008 | 1 | 0 | True | 10 | 0.9 | 128 | 1000 | 0.1 | 3.147 | 0 |
| – | – | 128 | – | – | – | – | – | – | – | – | – | – | 3.154 | 1 |
| – | – | 256 | – | – | – | – | – | – | – | – | – | – | 3.150 | 2 |
| – | – | – | – | 0.004 | – | – | – | – | – | – | – | – | 3.147 | 3 |
| – | – | – | – | – | – | – | – | – | – | 256 | – | – | 3.153 | 4 |

Table 156: Hyperparameter ablation for Soap on 300m on 4x Chinchilla Data

| $\beta_1$ | $\beta_2$ | blocksize | $\epsilon$ | $\eta$ | $g_{\mathrm{norm}}$ | $\eta_{min}$ | Blocking | $f_{pc}$ | $\beta_{shampoo}$ | BSZ | warmup | $\lambda$ | Loss | Link |
|---|---|---|---|---|---|---|---|---|---|---|---|---|---|---|
| 0.95 | 0.99 | 512 | 1e-10 | 0.008 | 1 | 0 | True | 10 | 0.9 | 256 | 1000 | 0.1 | 3.084 | 0 |
| – | – | 128 | – | – | – | – | – | – | – | – | – | – | 3.086 | 1 |
| – | – | 256 | – | – | – | – | – | – | – | – | – | – | 3.084 | 2 |
| – | – | – | – | 0.004 | – | – | – | – | – | – | – | – | 3.086 | 3 |
| – | – | – | – | – | – | – | – | – | – | 128 | – | – | 3.091 | 4 |

Table 157: Hyperparameter ablation for Soap on 300m on 8x Chinchilla Data

| $\beta_1$ | $\beta_2$ | blocksize | $\epsilon$ | $\eta$ | $g_{\mathrm{norm}}$ | $\eta_{min}$ | Blocking | $f_{pc}$ | $\beta_{shampoo}$ | BSZ | warmup | $\lambda$ | Loss | Link |
|---|---|---|---|---|---|---|---|---|---|---|---|---|---|---|
| 0.95 | 0.99 | 512 | 1e-10 | 0.008 | 1 | 0 | True | 10 | 0.9 | 256 | 1000 | 0.1 | 3.030 | 0 |
| – | – | 128 | – | – | – | – | – | – | – | – | – | – | 3.034 | 1 |
| – | – | 256 | – | – | – | – | – | – | – | – | – | – | 3.032 | 2 |
| – | – | – | – | 0.004 | – | – | – | – | – | – | – | – | 3.031 | 3 |
| – | – | – | – | – | – | – | – | – | – | 128 | – | – | 3.043 | 4 |

Table 158: Hyperparameter ablation for Soap on 520m on 2x Chinchilla Data

| $\beta_1$ | $\beta_2$ | blocksize | $\epsilon$ | $\eta$ | $g_{\mathrm{norm}}$ | $\eta_{min}$ | Blocking | $f_{pc}$ | $\beta_{shampoo}$ | BSZ | warmup | $\lambda$ | Loss | Link |
|---|---|---|---|---|---|---|---|---|---|---|---|---|---|---|
| 0.95 | 0.99 | 512 | 1e-10 | 0.008 | 1 | 0 | True | 10 | 0.95 | 256 | 1000 | 0.1 | 3.004 | 0 |
| – | – | 128 | – | – | – | – | – | – | – | – | – | – | 3.013 | 1 |
| – | – | 256 | – | – | – | – | – | – | – | – | – | – | 3.010 | 2 |
| – | – | – | – | 0.004 | – | – | – | – | – | – | – | – | 3.008 | 3 |
| – | – | – | – | – | – | – | – | – | – | 128 | – | – | 3.011 | 4 |

Table 159: Hyperparameter ablation for Soap on 520m on 4x Chinchilla Data

| $\beta_1$ | $\beta_2$ | blocksize | $\epsilon$ | $\eta$ | $g_{\mathrm{norm}}$ | $\eta_{min}$ | Blocking | $f_{pc}$ | $\beta_{shampoo}$ | BSZ | warmup | $\lambda$ | Loss | Link |
|---|---|---|---|---|---|---|---|---|---|---|---|---|---|---|
| 0.95 | 0.99 | 512 | 1e-10 | 0.004 | 1 | 0 | True | 10 | 0.95 | 256 | 1000 | 0.1 | 2.944 | 0 |
| – | – | 128 | – | – | – | – | – | – | – | – | – | – | 2.948 | 1 |
| – | – | 256 | – | – | – | – | – | – | – | – | – | – | 2.945 | 2 |
| – | – | – | – | 0.008 | – | – | – | – | – | – | – | – | 2.949 | 3 |
| – | – | – | – | – | – | – | – | – | – | 128 | – | – | 2.946 | 4 |

Table 160: Hyperparameter ablation for Soap on 520m on 8x Chinchilla Data

| $\beta_1$ | $\beta_2$ | blocksize | $\epsilon$ | $\eta$ | $g_{\mathrm{norm}}$ | $\eta_{min}$ | Blocking | $f_{pc}$ | $\beta_{shampoo}$ | BSZ | warmup | $\lambda$ | Loss | Link |
|---|---|---|---|---|---|---|---|---|---|---|---|---|---|---|
| 0.95 | 0.99 | 512 | 1e-10 | 0.004 | 1 | 0 | True | 10 | 0.95 | 256 | 1000 | 0.1 | 2.899 | 0 |
| – | – | – | – | 0.008 | – | – | – | – | – | – | – | – | 2.906 | 1 |

# G   Hyperparameter Ablation in Phase III 1.2B experiments

We reported the results for the optimizers we swept in Phase III. The result is formulated in the same way as in Phase I.

## G.1 SWEEPING RESULTS FOR ADAMW

Table 161: Hyperparameter ablation for AdamW on 1.2b on 1x Chinchilla Data

| $\beta_1$ | $\beta_2$ | $\epsilon$ | $\eta$ | $g_{\mathrm{norm}}$ | $\eta_{min}$ | BSZ | warmup | $\lambda$ | Loss | Link |
|---|---|---|---|---|---|---|---|---|---|---|
| 0.9 | 0.98 | 1e-10 | 0.002 | 2 | 0 | 256 | 2000 | 0.2 | 2.905 | 0 |
| 0.95 | – | – | – | – | – | – | – | – | 2.905 | 1 |
| 0.98 | – | – | – | – | – | – | – | – | 2.909 | 2 |
| – | 0.9 | – | – | – | – | – | – | – | 2.914 | 3 |
| – | 0.95 | – | – | – | – | – | – | – | 2.909 | 4 |
| – | – | 1e-25 | – | – | – | – | – | – | 2.907 | 5 |
| – | – | 1e-20 | – | – | – | – | – | – | 2.907 | 6 |
| – | – | 1e-15 | – | – | – | – | – | – | 2.907 | 7 |
| – | – | – | 0.004 | – | – | – | – | – | 2.916 | 8 |
| – | – | – | 0.008 | – | – | – | – | – | 7.347 | 9 |
| – | – | – | – | 0 | – | – | – | – | 2.909 | 10 |
| – | – | – | – | 1.0 | – | – | – | – | 2.908 | 11 |
| – | – | – | – | – | – | 128 | – | – | 2.904 | 12 |
| – | – | – | – | – | – | 512 | – | – | 2.928 | 13 |
| – | – | – | – | – | – | 1024 | – | – | 2.985 | 14 |
| – | – | – | – | – | – | – | 500 | – | 2.917 | 15 |
| – | – | – | – | – | – | – | 1000 | – | 2.910 | 16 |
| – | – | – | – | – | – | – | 4000 | – | 2.912 | 17 |
| – | – | – | – | – | – | – | – | 0 | 2.946 | 18 |
| – | – | – | – | – | – | – | – | 0.1 | 2.916 | 19 |

Table 162: Hyperparameter ablation for AdamW on 1.2b on 2x Chinchilla Data

| $\beta_1$ | $\beta_2$ | $\epsilon$ | $\eta$ | $g_{\mathrm{norm}}$ | $\eta_{min}$ | BSZ | warmup | $\lambda$ | Loss | Link |
|---|---|---|---|---|---|---|---|---|---|---|
| 0.9 | 0.98 | 1e-10 | 0.002 | 1 | 0 | 256 | 1000 | 0.2 | 2.836 | 0 |

Table 163: Hyperparameter ablation for AdamW on 1.2b on 4x Chinchilla Data

| $\beta_1$ | $\beta_2$ | $\epsilon$ | $\eta$ | $g_{\mathrm{norm}}$ | $\eta_{min}$ | BSZ | warmup | $\lambda$ | Loss | Link |
|---|---|---|---|---|---|---|---|---|---|---|
| 0.9 | 0.98 | 1e-10 | 0.002 | 1 | 0 | 256 | 1000 | 0.2 | 2.787 | 0 |

Table 164: Hyperparameter ablation for AdamW on 1.2b on 8x Chinchilla Data

| $\beta_1$ | $\beta_2$ | $\epsilon$ | $\eta$ | $g_{\mathrm{norm}}$ | $\eta_{min}$ | BSZ | warmup | $\lambda$ | Loss | Link |
|---|---|---|---|---|---|---|---|---|---|---|
| 0.9 | 0.98 | 1e-10 | 0.002 | 1 | 0 | 256 | 1000 | 0.2 | 2.752 | 0 |

## G.2 SWEEPING RESULTS FOR NADAMW

Table 165: Hyperparameter ablation for NAdamW on 1.2b on 1x Chinchilla Data

| $\beta_1$ | $\beta_2$ | $\epsilon$ | $\eta$ | $g_{\mathrm{norm}}$ | $\eta_{min}$ | BSZ | warmup | $\lambda$ | Loss | Link |
|---|---|---|---|---|---|---|---|---|---|---|
| 0.98 | 0.98 | 1e-10 | 0.004 | 1 | 0 | 256 | 4000 | 0.1 | 2.902 | 0 |

Table 166: Hyperparameter ablation for NAdamW on 1.2b on 2x Chinchilla Data

| $\beta_1$ | $\beta_2$ | $\epsilon$ | $\eta$ | $g_{\mathrm{norm}}$ | $\eta_{min}$ | BSZ | warmup | $\lambda$ | Loss | Link |
|------|------|-------|-------|------|---|-----|------|-----|-------|---|
| 0.98 | 0.98 | 1e-10 | 0.004 | 1 | 0 | 256 | 4000 | 0.1 | 2.833 | 0 |

Table 167: Hyperparameter ablation for NAdamW on 1.2b on 4x Chinchilla Data

| $\beta_1$ | $\beta_2$ | $\epsilon$ | $\eta$ | $g_{\mathrm{norm}}$ | $\eta_{min}$ | BSZ | warmup | $\lambda$ | Loss | Link |
|------|------|-------|-------|------|---|-----|------|-----|-------|---|
| 0.98 | 0.98 | 1e-10 | 0.004 | 1 | 0 | 256 | 4000 | 0.1 | 2.785 | 0 |

Table 168: Hyperparameter ablation for NAdamW on 1.2b on 8x Chinchilla Data

| $\beta_1$ | $\beta_2$ | $\epsilon$ | $\eta$ | $g_{\mathrm{norm}}$ | $\eta_{min}$ | BSZ | warmup | $\lambda$ | Loss | Link |
|------|------|-------|-------|------|---|-----|------|-----|-------|---|
| 0.98 | 0.98 | 1e-10 | 0.004 | 1 | 0 | 256 | 4000 | 0.1 | 2.749 | 0 |

## G.3 SWEEPING RESULTS FOR SOAP

Table 169: Hyperparameter ablation for Soap on 1.2b on 1x Chinchilla Data

| $\beta_1$ | $\beta_2$ | blocksize | $\epsilon$ | $\eta$ | $g_{\mathrm{norm}}$ | $\eta_{min}$ | Blocking | $f_{pc}$ | $\beta_{shampoo}$ | BSZ | warmup | $\lambda$ | Loss | Link |
|------|------|-----|-------|-------|---|---|------|----|-----|-----|------|-----|-------|---|
| 0.95 | 0.99 | 512 | 1e-10 | 0.004 | 1 | 0 | True | 10 | 0.9 | 256 | 1000 | 0.1 | 2.940 | 0 |

Table 170: Hyperparameter ablation for Soap on 1.2b on 2x Chinchilla Data

| $\beta_1$ | $\beta_2$ | blocksize | $\epsilon$ | $\eta$ | $g_{\mathrm{norm}}$ | $\eta_{min}$ | Blocking | $f_{pc}$ | $\beta_{shampoo}$ | BSZ | warmup | $\lambda$ | Loss | Link |
|------|------|-----|-------|-------|---|---|------|----|-----|-----|------|-----|-------|---|
| 0.95 | 0.99 | 512 | 1e-10 | 0.004 | 1 | 0 | True | 10 | 0.9 | 256 | 1000 | 0.1 | 2.829 | 0 |

Table 171: Hyperparameter ablation for Soap on 1.2b on 4x Chinchilla Data

| $\beta_1$ | $\beta_2$ | blocksize | $\epsilon$ | $\eta$ | $g_{\mathrm{norm}}$ | $\eta_{min}$ | Blocking | $f_{pc}$ | $\beta_{shampoo}$ | BSZ | warmup | $\lambda$ | Loss | Link |
|------|------|-----|-------|-------|---|---|------|----|-----|-----|------|-----|-------|---|
| 0.95 | 0.99 | 512 | 1e-10 | 0.004 | 1 | 0 | True | 10 | 0.9 | 256 | 1000 | 0.1 | 2.783 | 0 |

Table 172: Hyperparameter ablation for Soap on 1.2b on 8x Chinchilla Data

| $\beta_1$ | $\beta_2$ | blocksize | $\epsilon$ | $\eta$ | $g_{\mathrm{norm}}$ | $\eta_{min}$ | Blocking | $f_{pc}$ | $\beta_{shampoo}$ | BSZ | warmup | $\lambda$ | Loss | Link |
|------|------|-----|-------|-------|---|---|------|----|-----|-----|------|-----|-------|---|
| 0.95 | 0.99 | 512 | 1e-10 | 0.004 | 1 | 0 | True | 10 | 0.9 | 256 | 1000 | 0.1 | 2.749 | 0 |

## G.4 SWEEPING RESULTS FOR MUON

Table 173: Hyperparameter ablation for Muon on 1.2b on 1x Chinchilla Data

| $\eta_{adam}$ | $\beta_1$ | $\beta_2$ | Decay(WSD) | $\epsilon$ | $\eta$ | Schedule | $g_{\mathrm{norm}}$ | $\eta_{min}$ | $\beta_{\mathrm{muon}}$ | $\epsilon_{muon}$ | BSZ | warmup | $\lambda$ | Loss | Link |
|-------|-----|------|-----|-------|-------|--------|---|---|------|-------|-----|---|-----|-------|---|
| 0.0012 | 0.8 | 0.98 | 1 | 1e-15 | 0.004 | linear | 2 | 0 | 0.98 | 1e-05 | 256 | 0 | 0.1 | 2.891 | 0 |
| – | – | – | – | – | 0.008 | – | – | – | – | – | – | – | – | 2.886 | 1 |

Table 174: Hyperparameter ablation for Muon on 1.2b on 2x Chinchilla Data

| $\eta_{adam}$ | $\beta_1$ | $\beta_2$ | Decay(WSD) | $\epsilon$ | $\eta$ | Schedule | $g_{\mathrm{norm}}$ | $\eta_{min}$ | $\beta_{\mathrm{muon}}$ | $\epsilon_{muon}$ | BSZ | warmup | $\lambda$ | Loss | Link |
|---|---|---|---|---|---|---|---|---|---|---|---|---|---|---|---|
| 0.0012 | 0.8 | 0.98 | 1 | 1e-15 | 0.004 | linear | 2 | 0 | 0.98 | 1e-05 | 256 | 0 | 0.1 | 2.827 | 0 |
| – | – | – | – | – | 0.008 | – | – | – | – | – | – | – | – | 2.833 | 1 |

Table 175: Hyperparameter ablation for Muon on 1.2b on 4x Chinchilla Data

| $\eta_{adam}$ | $\beta_1$ | $\beta_2$ | Decay(WSD) | $\epsilon$ | $\eta$ | Schedule | $g_{\mathrm{norm}}$ | $\eta_{min}$ | $\beta_{\mathrm{muon}}$ | $\epsilon_{muon}$ | BSZ | warmup | $\lambda$ | Loss | Link |
|---|---|---|---|---|---|---|---|---|---|---|---|---|---|---|---|
| 0.0012 | 0.8 | 0.98 | 1 | 1e-15 | 0.004 | linear | 2 | 0 | 0.98 | 1e-05 | 256 | 0 | 0.1 | 2.780 | 0 |
| – | – | – | – | – | 0.008 | – | – | – | – | – | – | – | – | 2.793 | 1 |

Table 176: Hyperparameter ablation for Muon on 1.2b on 8x Chinchilla Data

| $\eta_{adam}$ | $\beta_1$ | $\beta_2$ | Decay(WSD) | $\epsilon$ | $\eta$ | Schedule | $g_{\mathrm{norm}}$ | $\eta_{min}$ | $\beta_{\mathrm{muon}}$ | $\epsilon_{muon}$ | BSZ | warmup | $\lambda$ | Loss | Link |
|---|---|---|---|---|---|---|---|---|---|---|---|---|---|---|---|
| 0.0012 | 0.8 | 0.98 | 1 | 1e-15 | 0.004 | linear | 2 | 0 | 0.98 | 1e-05 | 256 | 0 | 0.1 | 2.748 | 0 |

# H  HYPERPARAMETER ABLATION IN PHASE III 16X CHINCHILLA EXPERIMENTS

## H.1  SWEEPING RESULTS FOR ADAMW

Table 177: Hyperparameter ablation for AdamW on 130m on 16x Chinchilla Data

| $\beta_1$ | $\beta_2$ | $\epsilon$ | $\eta$ | $g_{\mathrm{norm}}$ | $\eta_{min}$ | BSZ | warmup | $\lambda$ | Loss | Link |
|---|---|---|---|---|---|---|---|---|---|---|
| 0.9 | 0.98 | 1e-10 | 0.008 | 1 | 0 | 256 | 1000 | 0.1 | 3.207 | 0 |

Table 178: Hyperparameter ablation for AdamW on 300m on 16x Chinchilla Data

| $\beta_1$ | $\beta_2$ | $\epsilon$ | $\eta$ | $g_{\mathrm{norm}}$ | $\eta_{min}$ | BSZ | warmup | $\lambda$ | Loss | Link |
|---|---|---|---|---|---|---|---|---|---|---|
| 0.9 | 0.98 | 1e-10 | 0.004 | 2 | 0 | 256 | 2000 | 0.1 | 3.001 | 0 |

## H.2  SWEEPING RESULTS FOR NADAMW

Table 179: Hyperparameter ablation for NAdamW on 130m on 16x Chinchilla Data

| $\beta_1$ | $\beta_2$ | $\epsilon$ | $\eta$ | $g_{\mathrm{norm}}$ | $\eta_{min}$ | BSZ | warmup | $\lambda$ | Loss | Link |
|---|---|---|---|---|---|---|---|---|---|---|
| 0.95 | 0.98 | 1e-10 | 0.008 | 1 | 0 | 256 | 2000 | 0.1 | 3.200 | 0 |

Table 180: Hyperparameter ablation for NAdamW on 300m on 16x Chinchilla Data

| $\beta_1$ | $\beta_2$ | $\epsilon$ | $\eta$ | $g_{\mathrm{norm}}$ | $\eta_{min}$ | BSZ | warmup | $\lambda$ | Loss | Link |
|---|---|---|---|---|---|---|---|---|---|---|
| 0.95 | 0.98 | 1e-10 | 0.004 | 1 | 0 | 256 | 2000 | 0.1 | 2.998 | 0 |

### H.3 SWEEPING RESULTS FOR SOAP

Table 181: Hyperparameter ablation for Soap on 130m on 16x Chinchilla Data

| $\beta_1$ | $\beta_2$ | blocksize | $\epsilon$ | $\eta$ | $g_{\mathrm{norm}}$ | $\eta_{min}$ | Blocking | $f_{pc}$ | $\beta_{shampoo}$ | BSZ | warmup | $\lambda$ | Loss | Link |
|---|---|---|---|---|---|---|---|---|---|---|---|---|---|---|
| 0.95 | 0.99 | 512 | 1e-10 | 0.008 | 1 | 0 | True | 10 | 0.98 | 256 | 1000 | 0.1 | 3.191 | 0 |

Table 182: Hyperparameter ablation for Soap on 300m on 16x Chinchilla Data

| $\beta_1$ | $\beta_2$ | blocksize | $\epsilon$ | $\eta$ | $g_{\mathrm{norm}}$ | $\eta_{min}$ | Blocking | $f_{pc}$ | $\beta_{shampoo}$ | BSZ | warmup | $\lambda$ | Loss | Link |
|---|---|---|---|---|---|---|---|---|---|---|---|---|---|---|
| 0.95 | 0.99 | 512 | 1e-10 | 0.004 | 1 | 0 | True | 10 | 0.9 | 256 | 1000 | 0.1 | 2.990 | 0 |

### H.4 SWEEPING RESULTS FOR MUON

Table 183: Hyperparameter ablation for Muon on 130m on 16x Chinchilla Data

| $\eta_{adam}$ | $\beta_1$ | $\beta_2$ | Decay(WSD) | $\epsilon$ | $\eta$ | Schedule | $g_{\mathrm{norm}}$ | $\eta_{min}$ | $\beta_{\mathrm{muon}}$ | $\epsilon_{muon}$ | BSZ | warmup | $\lambda$ | Loss | Link |
|---|---|---|---|---|---|---|---|---|---|---|---|---|---|---|---|
| 0.0012 | 0.8 | 0.98 | 1 | 1e-25 | 0.004 | linear | 1 | 0 | 0.98 | 1e-05 | 128 | 0 | 0.1 | 3.192 | 0 |
| – | – | – | – | – | 0.008 | – | – | – | – | – | – | – | – | 3.202 | 1 |

Table 184: Hyperparameter ablation for Muon on 300m on 16x Chinchilla Data

| $\eta_{adam}$ | $\beta_1$ | $\beta_2$ | Decay(WSD) | $\epsilon$ | $\eta$ | Schedule | $g_{\mathrm{norm}}$ | $\eta_{min}$ | $\beta_{\mathrm{muon}}$ | $\epsilon_{muon}$ | BSZ | warmup | $\lambda$ | Loss | Link |
|---|---|---|---|---|---|---|---|---|---|---|---|---|---|---|---|
| 0.0012 | 0.8 | 0.98 | 0.8 | 1e-15 | 0.004 | linear | 1 | 0 | 0.98 | 1e-05 | 256 | 0 | 0.1 | 2.991 | 0 |

## I COMPARISON WITH PRIOR WORK

This paper benchmarks the performance of 11 proposed optimizers and show vastly different speed-up ratio than prior works reported. In this section, we will compare the setup of our experiments with prior works with the hope of understanding the difference.

1. Sophia Liu et al. (2024a) (2×) This work utilizes a small peak learning rate of learning rate smaller than 6e-4 (similar to the one shown in Figure 1 Top Left). The reason for the small peak learning rate is likely 2-fold: (i) the authors are training on a pretraining dataset PILE with lower quality compared to the current pretraining dataset and (ii) in the implementation of Levanter that the authors used, the data shuffling is not completely random and instead is correlated on every compute node. Upon reproducing the results, we note that this difference can significantly impact the stability of the training process and a complete random shuffling is crucial for the usage of a large learning rate.

2. MARS Yuan et al. (2025) (2×) This papers considers a similar setup as Sophia and uses a similar AdamW baseline. We note that in the first version of the paper, the authors also reported that increasing the learning rate of AdamW to 3e-3 can significantly improve the performance of AdamW (see Figure 6 of Yuan et al. (2025) arxiv version 1).

3. Soap Vyas et al. (2025) (1.4×) The actual speedup of Soap on 300M and 520M models are 1.2 to 1.3×, which is only slightly lower than the claimed 1.4× speedup. We note that our implementation of Soap is slightly different from the one used in the paper that we performs blocking of weight in order to reduce the memory footprint and further uses bfloat16 for the momentum in the 1.2B experiments. Both modifications may lead to slightly lower step-wise performance.

4. Muon Jordan et al. (2024); Liu et al. (2025a) (2×) The speedup of Muon reported in different works are vastly different. In the original Nanogpt speedrun, Muon achieves $1.3\times$ speedup over AdamW. Later the reproduction of Kimi reported a much higher speedup of $2\times$. We note that the Kimi version utilizes a notably low learning rate for AdamW (8e-4 to 9e-4 for model between 400M to 1.5B) in the scaling experiments. We also note that the smaller learning rate is important for hyperparameter transfer from AdamW when roughly matching the update norm and AdamW can perform better higher learning rate in our experiments. Further, their comparison of Muon and AdamW on the MoE experiments compare two models with not fully decayed learning rate and this may significantly favors Muon, as shown in Figure 1. It is later shown independently in the work of Essential AI AI et al. (2025) that Muon's token efficiency compared to AdamW is only 1.1 to $1.2\times$.

5. Cautious Liang et al. (2025), Block-wise Learning Rate Adam Wang et al. (2025), FOCUS Liu et al. (2025c) report $2\times$ speedup over AdamW. These papers use a similar baseline as Sophia and MARS.

6. SWAN Ma et al. (2025) and DION Ahn et al. (2025) report 2-3× speedup over AdamW. The comparison between these two optimizers and AdamW is carried out on a smaller than $1\times$ Chinchilla regime, where the speed-up of matrix-based optimizer may be larger. We also note that we didn't consider the communication cost, which is the main focus of DION.

7. SPlus Frans et al. (2025) reports a $2\times$ speedup over AdamW. This work considers an atypical setup where the model is trained with a constant learning rate.

## J  VALIDITY OF COORDINATE DESCENT

### J.1  THEORETICAL GUARANTEES FOR COORDINATE DESCENT

Here, we provide a theoretical result showing that, assuming (i) each hyperparameter is discretized densely enough, and (ii) the hyperparameter-to-loss function contains no saddle points, classical results from block coordinate descent theory guarantee CD convergence to a locally optimal configuration.

**Definition 1** (Hyperparameter Search Space). *The hyperparameter search space $S$ is the set of all possible hyperparameter configurations $\mathbf{h} = (h_1, \ldots, h_d)$ where all $h_i \in [L_i, R_i]$ for some lower and upper bounds $L_i$ and $R_i$.*

**Definition 2** (Coordinate Descent). *The coordinate descent (CD) algorithm with grid size $\delta$ is an iterative algorithm that minimizes a loss function $\mathcal{L}(\mathbf{h})$ through the following steps:*

1. *Discretize the search space $S$ into a grid of size $\delta$ to get a set $\mathcal{S}_\delta = \{\mathbf{h} \in S : h_i = L_i + j\delta \text{ for some } j \in \{0, \ldots, \frac{R_i - L_i}{\delta}\}\}$.*

2. *Initialize $\mathbf{h}^{(0)} = (h_1^{(0)}, \ldots, h_d^{(0)})$ randomly on the grid $\mathcal{S}_\delta$.*

3. *Update $h_i^{(k+1)}$ to the value in $\mathcal{S}_\delta$ that minimizes $\mathcal{L}(\mathbf{h}^{(k+1)})$ and at most one coordinate of $\mathbf{h}^{(k+1)}$ can be different from $\mathbf{h}^{(k)}$.*

*Run this procedure until the loss value converges.*

**Assumption 3** (Regularity of the hyperparameter-to-loss function). *The hyperparameter-to-loss function $L : S \to \mathbb{R}$ satisfies the following conditions:*

1. *All stationary points are either strict local minimimizers or strict local maximizers.*

$$\nabla\mathcal{L}(h_\delta^*) = 0 \implies \nabla^2\mathcal{L}(h_\delta^*) \succeq \lambda \text{ or } \nabla^2\mathcal{L}(h_\delta^*) \preceq -\lambda.$$

2. *The hyperparameter on the boundary is not coordinate-wise optimal in the sense that, if $h \in \mathcal{S}_\delta$ and $h_i = L_i$, then $\nabla\mathcal{L}(h)_i < 0$. Similarly, if $h \in \mathcal{S}_\delta$ and $h_i = R_i$, then $\nabla\mathcal{L}(h)_i > 0$.*

3. *It is a $C^2$ smooth function, i.e., $\|\nabla\mathcal{L}(\mathbf{h}) - \nabla\mathcal{L}(\mathbf{h}')\|_2 \leq L\|\mathbf{h} - \mathbf{h}'\|_2$ and $\|\nabla^2\mathcal{L}(\mathbf{h}) - \nabla^2\mathcal{L}(\mathbf{h}')\|_* \leq M\|\mathbf{h} - \mathbf{h}'\|_2^2$ for some constants $L, M > 0$ and all $\mathbf{h}, \mathbf{h}' \in \mathcal{S}_\delta$.*

**Theorem 4.** *For any loss function $\mathcal{L}(\mathbf{h})$ that satisfies Theorem 3, there exists constant $C$, such that the converged hyperparameter $h_\delta^*$ found by the CD algorithm with grid size $\delta$ (Theorem 2), there exists a corresponding local minimum $h'$ of $\mathcal{L}(\mathbf{h})$ such that the regret on the loss value is bounded by $C\delta^2$.*

$$\mathcal{L}(h_\delta^*) - \mathcal{L}(h') \leq C\delta^2.$$

*Proof.* By Assumption 3, we know that the

**Lemma 5.** $h_\delta^*$ *satisfies that* $\|\nabla\mathcal{L}(h_\delta^*)\|_2 \leq dL\delta$.

*Proof.* The proof is by contradiction. Suppose that $\|\nabla\mathcal{L}(h_\delta^*)\|_2 > dL\delta$. Then, there exists a dimension $i$ such that $|\nabla\mathcal{L}(h_\delta^*)_i| > L\delta$. By symmetry, we can assume that $\nabla\mathcal{L}(h_\delta^*)_i > L\delta$.

We now discuss by cases, assume $e_i$ is the $i$-th standard basis vector.

- If $h_\delta^* - \delta e_i \in \mathcal{S}_\delta$, because $L$ is smooth, we have the following inequality:

$$\mathcal{L}(h_\delta^* - \delta e_i) \leq \mathcal{L}(h_\delta^*) + \nabla\mathcal{L}(h_\delta^*)^T(-\delta e_i) + \frac{L\delta^2}{2}\|e_i\|_2^2$$
$$< \mathcal{L}(h_\delta^*) - L\delta^2 + \frac{L\delta^2}{2}$$
$$< \mathcal{L}(h_\delta^*).$$

  This contradicts the assumption that $h_\delta^*$ is a converging point for the CD algorithm.

- If $h_\delta^* - \delta e_i \notin \mathcal{S}_\delta$, then let $h_{B,\delta}$ on the boundary $\mathcal{S}$ satisfy that $\|h_{B,\delta} - h_\delta^*\|_2 \leq \delta$ and $(h_{B,\delta})_i - (h_\delta^*)_i$ is parallel to $e_i$. By smoothness,

$$\|\nabla\mathcal{L}(h_{B,\delta}) - \nabla\mathcal{L}(h_\delta^*)\|_2 \leq L\|h_{B,\delta} - h_\delta^*\|_2 \leq L\delta.$$

  Meanwhile, due to Assumption 3, we know that $\nabla\mathcal{L}(h_{B,\delta})_i < 0$.

$$\|\nabla\mathcal{L}(h_{B,\delta}) - \nabla\mathcal{L}(h_\delta^*)\|_2 \geq \nabla\mathcal{L}(h_\delta^*)_i - (\nabla\mathcal{L}(h_{B,\delta}))_i > L\delta.$$

  This leads to a contradiction.

The proof is complete. $\qquad\square$

Further, we can show that the converged hyperparameter $h_\delta^*$ is not close to any of the local maximizers of $\mathcal{L}(\mathbf{h})$.

**Lemma 6.** *For every minimizer $h^*$ of $\mathcal{L}(\mathbf{h})$, there exists a radius $R = \sqrt{\frac{\lambda}{2M}} > 0$ around $h^*$, the loss $L$ is locally $\lambda/2$ strongly convex and satisfies the Polyak-Lojasiewicz condition at the ball $B(h^*, R)$.*

$$\mathcal{L}(h) - \mathcal{L}(h^*) \leq \frac{1}{2\lambda}\|\nabla\mathcal{L}(h)\|_2^2.$$

*Symmetrically, for every maximizer $h^*$, the negated loss $-L$ is locally $\lambda/2$ strongly convex and satisfies the Polyak-Lojasiewicz condition at the ball $B(h^*, R)$.*

*Proof.* Because $h^*$ is a minimizer, we have that $\nabla\mathcal{L}(h^*) = 0$ and by $Theorem\ 3$, we know that $\nabla^2\mathcal{L}(h^*) \succeq \lambda$.

Further as the loss $L$ is $C^2$ smooth, we have that

$$\forall h \in B(h^*, R), \|\nabla^2\mathcal{L}(h) - \nabla^2\mathcal{L}(h^*)\|_* \leq M\|h - h^*\|_2^2 \leq MR^2 = \frac{\lambda}{2}.$$

Therefore, $\forall h \in B(h^*, R), \nabla^2\mathcal{L}(h) \succeq \lambda/2$. Therefore, the Polyak-Lojasiewicz condition is satisfied as the loss $L$ is strongly convex when restricted to the ball $B(h^*, R)$. $\qquad\square$

We will first show that the converged hyperparameter $h_\delta^*$ is not close to any of the local maximizers of $\mathcal{L}(\mathbf{h})$.

**Lemma 7.** *There exists a radius $R = \sqrt{\frac{\lambda}{2M}} > 0$, If $h_\delta^*$ is the converged hyperparameter found by the CD algorithm with grid size $\delta$, for any local maximizer $h'$ of $\mathcal{L}(\mathbf{h})$, we have that $\|h_\delta^* - h'\|_2 \geq R/2$.*

*Proof.* We will prove by contradiction. Suppose that $\|h_\delta^* - h'\|_2 < R/2$. By Lemma 6, we know that the negated loss $-L$ is $\lambda/2$ strongly convex. And by Lemma 5, we know that $\|\nabla \mathcal{L}(h_\delta^*)\|_2 \leq dL\delta$. Therefore, we know that $h_\delta^*$ will be closed to $h'$ as strong convexity of $-L$ implies

$$\|h - h_\delta^*\|_2 \leq \frac{2}{\lambda}\|\nabla \mathcal{L}(h)\|_2 \leq \frac{2}{\lambda}dL\delta.$$

Smoothness then implies that $\mathcal{L}(h_\delta^*)$ is close to $\mathcal{L}(h')$.

$$\mathcal{L}(h_\delta^*) - \mathcal{L}(h') \geq -\frac{L}{2}\|h_\delta^* - h'\|_2^2 \geq -\frac{2d^2L^3}{\lambda^2}\delta^2.$$

Further, we can show that for any $h \in \left(B(h', R) - B(h', 4d(L/\lambda)^{3/2}\delta)\right) \cap \mathcal{S}$, the loss value will be smaller than $\mathcal{L}(h_\delta^*)$ due to the local strong concavity of $L$.

$$\mathcal{L}(h) - \mathcal{L}(h') \leq -\frac{\lambda}{4}\|h - h'\|_2^2.$$

As

$$\|h - h'\|_2^2 \geq \frac{8d^2L^3}{\lambda^3}\delta^2.$$

Then we have that,

$$L(h_\delta^*) \geq L(h') - \frac{2d^2L^3}{\lambda^2}\delta^2$$
$$> L(h') - \frac{\lambda}{4}\|h - h'\|_2^2 \geq L(h).$$

As $L(h_\delta^*)$ is a converged point for the CD algorithm, we know that it has the smallest loss value in the following set:

$$\{h_\delta^* - i\delta e_1 : i \in \mathbf{Z}\} \cap \mathcal{S}.$$

Let $N = \lceil 4d(L/\lambda)^{3/2} + 1 \rceil$, when $\delta$ is small enough, we know either $h_\delta^* - N\delta e_1$ or $h_\delta^* + N\delta e_1$ is in the set $\{h_\delta^* - i\delta e_1 : i \in \mathbf{Z}\} \cap \mathcal{S}$ and the distance between either of them to $h'$ is smaller than $R$ and larger $4d(L/\lambda)^{3/2}\delta$. This is a contradiction. The proof is then complete. $\square$

Now we can prove the main theorem, showing that the regret on the loss value is bounded by $C\delta^2$. Now consider the set of all stationary points of $\mathcal{L}(\mathbf{h})$ to be $S^* = \{h^* : \nabla \mathcal{L}(h^*) = 0\}$ and define the neighborhood of $h^*$ to be $N(S^*) = \{h \in S : \exists h^* \in S^*, \|h - h^*\|_2 \leq \sqrt{\frac{\lambda}{2M}}\}$. As the gradient of $L$ is continuous and $S$ is compact, we know that there exists a lower bound on the gradient norm in $S - N(S^*)$, i.e., there exists a constant $\gamma > 0$ such that

$$\forall h \in S - N(S^*), \|\nabla \mathcal{L}(h)\|_2 \geq \gamma.$$

For all $\delta < \frac{\gamma}{2L}$, by Lemma 5, we know that the converged hyperparameter $h_\delta^*$ found by the CD algorithm with grid size $\delta$ satisfies that $\|\nabla \mathcal{L}(h_\delta^*)\|_2 \leq dL\delta < \gamma$. Therefore, $h_\delta^* \in N(S^*)$. By Theorem 7, we know that $h_\delta^*$ is not close to any of the local maximizers of $\mathcal{L}(\mathbf{h})$. Therefore, there exists a corresponding local minimizer $h_\delta'$ of $\mathcal{L}(\mathbf{h})$ such that $\|h_\delta' - h_\delta^*\|_2 \leq \sqrt{\frac{\lambda}{2M}}$.

Further by Lemma 6, we know that

$$\mathcal{L}(h_\delta^*) - \mathcal{L}(h_\delta') \leq \frac{1}{2\lambda}\|\nabla \mathcal{L}(h_\delta^*)\|_2^2 \leq \frac{d^2L^2}{2\lambda}\delta^2.$$

Therefore, the regret on the loss value is bounded by $C\delta^2$ with $C = \frac{d^2L^2}{2\lambda}$. $\square$

## J.2 BENIGNITY OF THE HYPERPARAMETER OPTIMIZATION FROM PRIOR WORKS

Our theoretical result relies on the following benignity condition on the hyperparameter optimization problems: all stationary points are strict local minima. While it is not computationally feasible to check this condition for all hyperparameters, we investigate some prior works studying the interactions between two hyperparameters and find that this condition is often satisfied when constrained to the 2-dimensional hyperparameter search space.

1. You et al. (2017; 2019); Marek et al. (2025); Li et al. (2025a) study the interactions between batch size and learning rate. As shown in the Figure 1 of Li et al. (2025a), the loss surface is near quasi-convex with respect to learning rate and batch size and only contains one basin.

2. Bergsma et al. (2025) study the interactions between weight decay and batch size. As shown in the Figure 4 of Bergsma et al. (2025), the loss surface is near quasi-convex with respect to weight decay and batch size and only contains one basin.

3. Marek et al. (2025) study the interactions between learning rate and $\beta_2$. As shown in the Figure 1 of Marek et al. (2025), the loss surface is near quasi-convex with respect to learning rate and $\beta_2$ is near quasi-convex and only contains one basin.

## J.3 VALIDATION OF THE BENIGNITY OF THE HYPERPARAMETER OPTIMIZATION IN OUR EXPERIMENTS

We validate the benignity of the hyperparameter optimization in our experiments by performing 2 dimensional sweeping for the hyperparameters in Muon 130M 1x Chinchilla regime. We consider the following three pairs of hyperparameters: (learning rate, batch size), (learning rate, weight decay), and (learning rate, WSD decay ratio). The results are shown in the Figure 10.

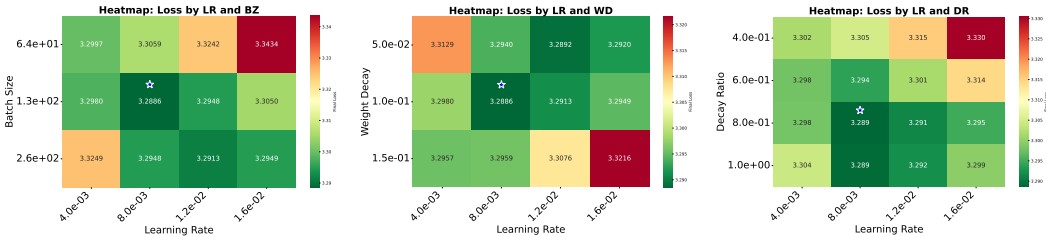

Figure 10: Validation of the benignity of the hyperparameter optimization in our experiments settings. We observe that the loss surface is near quasi-convex with respect to the learning rate and other hyperparameters. The blue star marks the optimal hyperparameter configuration in these sweeping and it is the one we found by our coordinate descent procedure.

# K ABLATIONS ON THE EFFECT OF BATCH SIZE

We conduct a sweep across different model sizes, training budgets, and batch sizes to understand how the speedup between AdamW and Muon changes with the batch size. Due to time constraints, we adopt the hyperparameter found in our experiments and only sweep the learning rate when we change the batch size.

Our experiments cover 130M and 300M parameter models trained with varying Chinchilla ratios using both AdamW and Muon optimizers. We also scale up to 1.2B parameter model in the 1x Chinchilla ratio to further confirm the results. We first show the aggregated results on the best performance of the two optimizers across different model sizes, training budgets, and batch sizes in Figure 11. We observe two phenomena: (i) the loss values increase as the batch size increases for both optimizers; (ii) Muon's loss increases slower than AdamW's, leading to a larger speedup between the two optimizers as the batch size increases.

## K.1 RESULTS FOR LEARNING RATE SWEEP

We make the following observations on how the learning rate scale with the batch size:

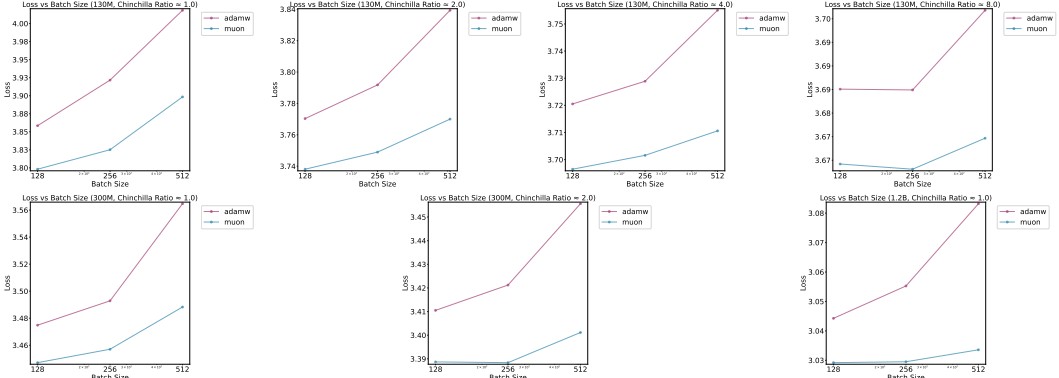

Figure 11: Minimum achievable loss as a function of batch size for different model sizes and Chinchilla ratios. We observe a general upward trend in the loss values as the batch size increases for both optimizers but Muon's loss decreases slower than AdamW's.

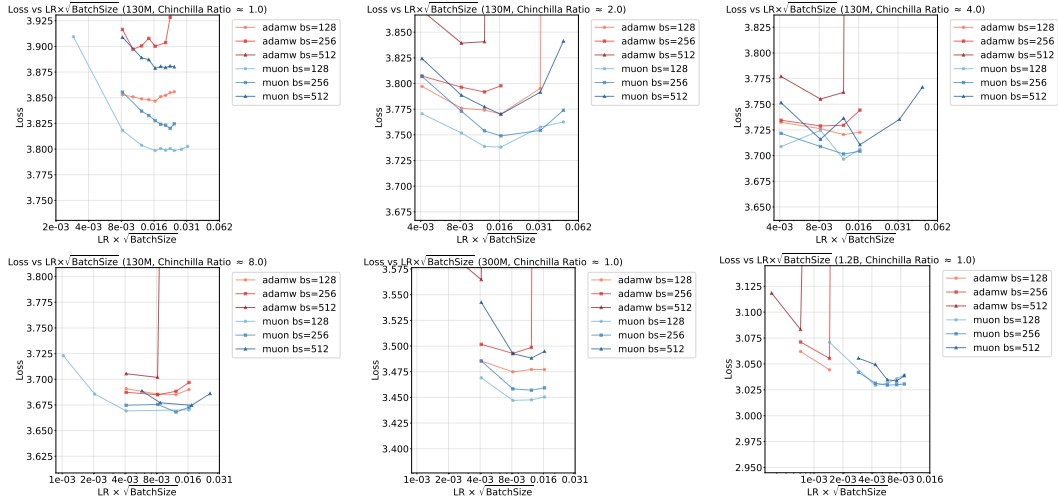

Figure 12: Learning rate sweep results. Each subplot shows evaluation loss on C4 as a function of learning rate (scaled by $\sqrt{\text{BatchSize}/128}$) for different training budgets. Different colors represent different batch sizes, with blue for Muon and red for AdamW.

- **Square root scaling until critical batch size.** We confirm prior works You et al. (2017; 2019); Marek et al. (2025) that the learning rate scales with the square root of the batch size until a certain batch size is reached, where the optimal learning rate tends to a constant. We observe this phenomenon in our experiments for both Muon and AdamW.

- **Training budget effects:** As the training budget increases (higher Chinchilla ratios), the optimal learning rate tends to decrease, and the loss curves become less sensitive to learning rate selection.

Based on our experiments, we provide the following recommendations for practitioners:

1. **Batch Size Scaling:**
   - Use $\text{LR}_{\text{new}} = \text{LR}_{\text{base}} \times \sqrt{\text{BatchSize}_{\text{new}}/\text{BatchSize}_{\text{base}}}$
   - Larger batch sizes (up to 512) can be used without performance loss when learning rate is properly scaled

2. **Optimizer Choice:**

- If the goal is to reach a decent loss as quickly as possible, use Muon with the largest batch size can be potentially beneficial as this can combine the benefits of Muon in larger batch size setting and Muon's memory efficiency. This also reduces the step-wise overhead of Muon over AdamW.
- If the goal is to reach the optimal loss given a fixed amount of data, use Muon with the smallest batch size will likely lead to the best performance.

## L  ADDITIONAL EXPERIMENTS ON THE LEARNING RATE SCHEDULE USED FOR MUON

In response to the reviewer's problem on the linear learning rate schedule used for Muon, we performed additional experiments to investigate the effect of the learning rate schedule. We consider swapping the linear learning rate schedule with the cosine learning rate schedule in the Muon 130M 1x Chinchilla regime. The results are shown in the Figure 13. We observe very minimal difference in the loss values between the two learning rate schedules (less than 2e-3 on the loss value).

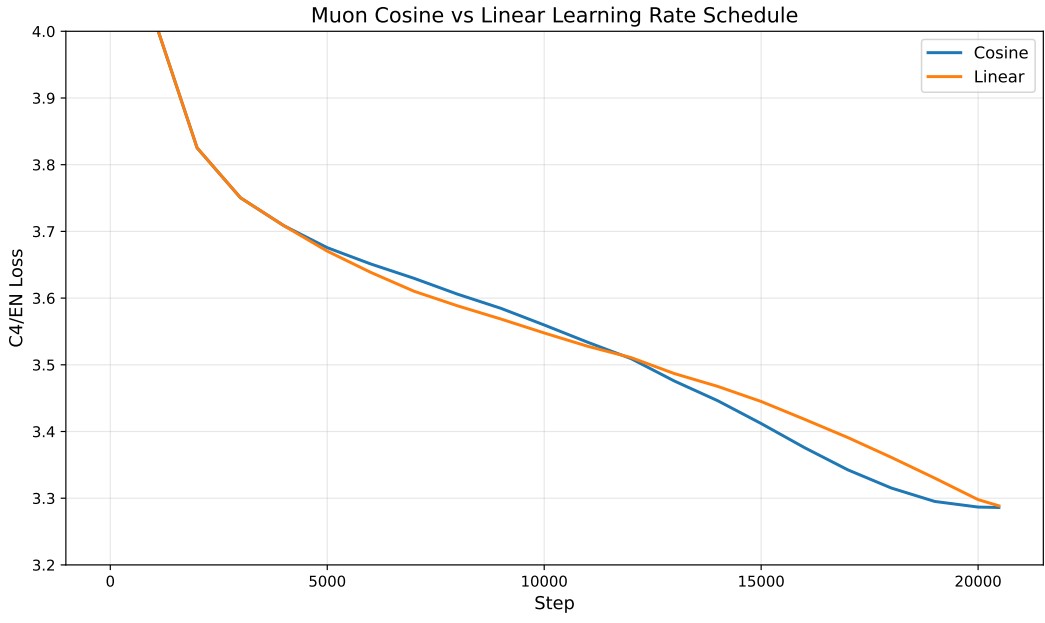

Figure 13: Validation of the effect of the cosine learning rate schedule in the Muon 130M 1x Chinchilla regime.

## M  COMPUTE BREAKDOWN

### M.1  SUMMARY

- Total unique tuples: 150
- Total runs: 4,281
- Total tracked hours: 7542.52
- Total GFLOPs: 186.42T

H100 GPU HOURS (USING 989 TFLOPS AS 1 H100 THROUGHPUT)

- H100 Equivalent Hours: 52,359.48
- H100 Equivalent Days: 2,181.64

- H100 Equivalent Years: 5.98

## M.2  PHASE I – 61 UNIQUE TUPLES

All 130m (except 42B and 48B), All 300m 6B, All 520m 10B

### PHASE SUMMARY

- Total GFLOPs: 79.31T
- Total Runs: 3,772
- Total Hours: 5,456.01
- H100 Equivalent Hours: 22,275.52
- H100 Equivalent Years: 2.54

| # | Model | Data | Optimizer | GFLOPs | Runs | Hours |
|---|-------|------|-----------|--------|------|-------|
| 1 | 520m | 10B | sophia | 6.0T | 129 | 704.55 |
| 2 | 520m | 10B | scion | 4.5T | 76 | 110.03 |
| 3 | 520m | 10B | mini | 4.5T | 90 | 150.58 |
| 4 | 130m | 21B | soap | 4.4T | 130 | 383.35 |
| 5 | 130m | 21B | sophia | 4.0T | 135 | 807.47 |
| 6 | 130m | 21B | mars | 2.9T | 73 | 111.81 |
| 7 | 130m | 21B | muon | 2.6T | 61 | 46.06 |
| 8 | 520m | 10B | mars | 2.4T | 83 | 52.30 |
| 9 | 520m | 10B | kron | 2.4T | 53 | 36.88 |
| 10 | 520m | 10B | muon | 2.3T | 40 | 64.85 |
| 11 | 520m | 10B | soap | 2.2T | 60 | 233.62 |
| 12 | 520m | 10B | nadam | 2.1T | 37 | 33.01 |
| 13 | 520m | 10B | adam | 2.0T | 46 | 33.66 |
| 14 | 130m | 21B | scion | 1.7T | 81 | 79.11 |
| 15 | 130m | 10B | scion | 1.7T | 95 | 73.27 |
| 16 | 130m | 10B | soap | 1.5T | 126 | 283.55 |
| 17 | 520m | 10B | cautious | 1.5T | 30 | 18.11 |
| 18 | 130m | 21B | cautious | 1.4T | 38 | 57.20 |
| 19 | 300m | 6B | sophia | 1.4T | 95 | 239.74 |
| 20 | 130m | 10B | muon | 1.3T | 102 | 113.12 |
| 21 | 300m | 6B | mars | 1.2T | 67 | 56.56 |
| 22 | 300m | 6B | mini | 1.2T | 61 | 46.15 |
| 23 | 300m | 6B | scion | 1.2T | 133 | 59.51 |
| 24 | 130m | 10B | adam | 1.1T | 53 | 45.03 |
| 25 | 130m | 21B | kron | 1.0T | 28 | 9.60 |
| 26 | 300m | 6B | soap | 1.0T | 62 | 114.65 |
| 27 | 130m | 10B | sophia | 1.0T | 72 | 133.75 |
| 28 | 130m | 10B | mars | 1.0T | 48 | 54.55 |
| 29 | 300m | 6B | cautious | 993.4B | 55 | 35.11 |
| 30 | 300m | 6B | adam | 991.9B | 47 | 29.86 |
| 31 | 130m | 21B | nadam | 951.5B | 22 | 10.37 |
| 32 | 130m | 10B | kron | 936.9B | 54 | 64.66 |
| 33 | 130m | 21B | adam | 908.3B | 21 | 8.98 |
| 34 | 130m | 21B | mini | 874.6B | 21 | 30.64 |
| 35 | 130m | 10B | mini | 802.5B | 38 | 19.16 |
| 36 | 300m | 6B | muon | 766.3B | 35 | 17.28 |
| 37 | 130m | 5B | soap | 745.4B | 76 | 186.13 |
| 38 | 130m | 2B | kron | 706.1B | 141 | 94.43 |
| 39 | 130m | 2B | muon | 671.0B | 134 | 81.45 |
| 40 | 130m | 2B | soap | 565.3B | 124 | 102.08 |

**Table 185 – continued from previous page**

| # | Model | Data | Optimizer | GFLOPs | Runs | Hours |
|---|-------|------|-----------|--------|------|-------|
| 41 | 130m | 2B | scion | 544.8B | 114 | 61.91 |
| 42 | 130m | 5B | scion | 543.0B | 52 | 25.65 |
| 43 | 130m | 5B | mars | 532.6B | 71 | 29.89 |
| 44 | 300m | 6B | kron | 503.6B | 36 | 9.28 |
| 45 | 130m | 2B | sophia | 476.6B | 103 | 133.70 |
| 46 | 130m | 10B | nadam | 475.8B | 22 | 31.92 |
| 47 | 130m | 10B | cautious | 475.8B | 22 | 37.10 |
| 48 | 300m | 6B | nadam | 459.8B | 21 | 13.76 |
| 49 | 130m | 5B | sophia | 428.3B | 50 | 87.75 |
| 50 | 130m | 5B | cautious | 400.1B | 37 | 15.60 |
| 51 | 130m | 5B | muon | 378.4B | 35 | 13.23 |
| 52 | 130m | 2B | cautious | 354.2B | 66 | 38.16 |
| 53 | 130m | 2B | adam | 298.2B | 61 | 31.99 |
| 54 | 130m | 198B | adam | 281.2B | 14 | 8.39 |
| 55 | 130m | 5B | kron | 248.7B | 25 | 8.03 |
| 56 | 130m | 5B | adam | 248.7B | 23 | 11.60 |
| 57 | 130m | 5B | nadam | 227.1B | 21 | 7.10 |
| 58 | 130m | 5B | mini | 227.1B | 21 | 8.03 |
| 59 | 130m | 2B | nadam | 195.7B | 43 | 21.50 |
| 60 | 130m | 2B | mini | 156.8B | 32 | 13.67 |
| 61 | 130m | 2B | mars | 145.1B | 31 | 15.52 |

## M.3    Phase II – 54 Unique Tuples

All other 300m and 520m (except 96B and 171B)

### Phase Summary

- Total GFLOPs: 70.98T
- Total Runs: 407
- Total Hours: 1,749.34
- H100 Equivalent Hours: 19,936.05
- H100 Equivalent Years: 2.28

| # | Model | Data | Optimizer | GFLOPs | Runs | Hours |
|---|-------|------|-----------|--------|------|-------|
| 1 | 520m | 85B | mini | 6.8T | 14 | 12.14 |
| 2 | 520m | 85B | mars | 6.2T | 13 | 42.15 |
| 3 | 520m | 85B | scion | 4.8T | 10 | 117.56 |
| 4 | 520m | 85B | soap | 4.2T | 14 | 115.94 |
| 5 | 520m | 42B | soap | 3.2T | 17 | 199.24 |
| 6 | 300m | 48B | mini | 2.8T | 16 | 159.03 |
| 7 | 520m | 85B | adam | 2.8T | 6 | 24.80 |
| 8 | 520m | 85B | nadam | 2.1T | 5 | 22.25 |
| 9 | 300m | 48B | mars | 2.0T | 12 | 83.16 |
| 10 | 520m | 85B | muon | 1.9T | 4 | 6.14 |
| 11 | 520m | 85B | cautious | 1.9T | 4 | 1.56 |
| 12 | 520m | 42B | mini | 1.9T | 15 | 7.24 |
| 13 | 520m | 42B | scion | 1.9T | 8 | 70.32 |
| 14 | 520m | 42B | mars | 1.7T | 7 | 2.53 |
| 15 | 300m | 48B | soap | 1.6T | 9 | 64.89 |
| 16 | 520m | 85B | kron | 1.5T | 3 | 103.75 |
| 17 | 520m | 42B | adam | 1.4T | 6 | 35.14 |
| 18 | 300m | 48B | scion | 1.4T | 8 | 21.83 |
| 19 | 300m | 24B | mini | 1.3T | 15 | 13.11 |
| 20 | 300m | 12B | adam | 1.2T | 29 | 19.60 |
| 21 | 520m | 21B | mini | 1.1T | 10 | 20.59 |
| 22 | 520m | 21B | soap | 1.0T | 10 | 73.19 |
| 23 | 300m | 48B | adam | 1.0T | 6 | 26.27 |
| 24 | 520m | 42B | muon | 1.0T | 5 | 22.76 |
| 25 | 520m | 42B | cautious | 930.2B | 4 | 27.42 |
| 26 | 520m | 21B | scion | 924.4B | 8 | 9.76 |
| 27 | 300m | 24B | soap | 913.9B | 12 | 63.52 |
| 28 | 520m | 21B | mars | 847.6B | 7 | 7.57 |
| 29 | 300m | 48B | muon | 804.4B | 5 | 29.87 |
| 30 | 520m | 42B | nadam | 726.5B | 3 | 23.79 |
| 31 | 300m | 24B | scion | 706.7B | 10 | 22.70 |
| 32 | 300m | 48B | nadam | 620.5B | 4 | 21.27 |
| 33 | 300m | 48B | cautious | 618.1B | 4 | 23.69 |
| 34 | 300m | 24B | mars | 613.0B | 7 | 8.48 |
| 35 | 300m | 48B | kron | 525.5B | 8 | 19.21 |
| 36 | 300m | 24B | adam | 493.5B | 6 | 22.71 |
| 37 | 520m | 21B | adam | 484.4B | 4 | 16.16 |
| 38 | 520m | 42B | kron | 484.4B | 2 | 16.26 |
| 39 | 520m | 21B | cautious | 408.4B | 4 | 2.95 |
| 40 | 300m | 12B | mini | 394.1B | 9 | 16.15 |
| 41 | 520m | 21B | nadam | 363.3B | 3 | 11.28 |
| 42 | 520m | 21B | muon | 363.3B | 3 | 1.22 |

Continued on next page

**Table 186 – continued from previous page**

| # | Model | Data | Optimizer | GFLOPs | Runs | Hours |
|---|-------|------|-----------|--------|------|-------|
| 43 | 300m | 12B | mars | 357.8B | 12 | 16.32 |
| 44 | 300m | 24B | cautious | 350.3B | 4 | 13.28 |
| 45 | 300m | 24B | nadam | 350.3B | 4 | 15.42 |
| 46 | 300m | 12B | soap | 262.8B | 12 | 41.53 |
| 47 | 300m | 24B | kron | 262.7B | 3 | 16.24 |
| 48 | 300m | 24B | muon | 262.7B | 3 | 7.06 |
| 49 | 520m | 21B | kron | 242.2B | 2 | 11.80 |
| 50 | 300m | 12B | scion | 218.9B | 5 | 1.96 |
| 51 | 300m | 12B | cautious | 175.2B | 4 | 3.43 |
| 52 | 300m | 12B | nadam | 131.4B | 3 | 1.31 |
| 53 | 300m | 12B | muon | 131.4B | 3 | 5.98 |
| 54 | 300m | 12B | kron | 131.4B | 3 | 5.81 |

## M.4    PHASE III – 32 UNIQUE TUPLES

All 1.2B, 130m 42B, 520m 171B, 300m 96B

PHASE SUMMARY

- Total GFLOPs: 36.13T
- Total Runs: 102
- Total Hours: 337.17
- H100 Equivalent Hours: 10,147.90
- H100 Equivalent Years: 1.16

| # | Model | Data | Optimizer | GFLOPs | Runs | Hours |
|---|-------|------|-----------|--------|------|-------|
| 1 | 1.2b | 24B | adam | 8.9T | 41 | 135.65 |
| 2 | 1.2b | 193B | muon | 3.2T | 3 | 11.88 |
| 3 | 1.2b | 193B | soap | 2.9T | 2 | 15.39 |
| 4 | 1.2b | 193B | adam | 2.1T | 2 | 1.13 |
| 5 | 1.2b | 96B | muon | 2.1T | 2 | 6.75 |
| 6 | 1.2b | 193B | nadam | 2.1T | 1 | 0.96 |
| 7 | 1.2b | 96B | soap | 1.3T | 2 | 19.93 |
| 8 | 1.2b | 96B | nadam | 1.1T | 2 | 1.13 |
| 9 | 1.2b | 96B | adam | 1.1T | 2 | 8.42 |
| 10 | 1.2b | 48B | muon | 1.1T | 2 | 1.03 |
| 11 | 1.2b | 48B | soap | 1.0T | 3 | 26.58 |
| 12 | 520m | 171B | soap | 968.7B | 1 | 12.07 |
| 13 | 520m | 171B | kron | 968.7B | 1 | 0.37 |
| 14 | 520m | 171B | nadam | 968.7B | 1 | 9.34 |
| 15 | 1.2b | 24B | soap | 693.8B | 10 | 21.04 |
| 16 | 520m | 171B | adam | 645.9B | 1 | 12.08 |
| 17 | 520m | 171B | muon | 600.7B | 1 | 7.22 |
| 18 | 1.2b | 48B | nadam | 528.5B | 2 | 0.91 |
| 19 | 1.2b | 24B | muon | 525.4B | 2 | 1.37 |
| 20 | 1.2b | 48B | adam | 525.4B | 1 | 0.78 |
| 21 | 300m | 96B | adam | 366.4B | 2 | 0.79 |
| 22 | 300m | 96B | nadam | 350.3B | 1 | 11.23 |
| 23 | 300m | 96B | soap | 350.3B | 1 | 1.38 |
| 24 | 300m | 96B | kron | 350.3B | 1 | 0.39 |
| 25 | 300m | 96B | muon | 350.3B | 1 | 0.77 |
| 26 | 130m | 42B | muon | 346.0B | 8 | 11.42 |
| 27 | 1.2b | 24B | kron | 262.7B | 1 | 7.45 |
| 28 | 1.2b | 24B | nadam | 262.7B | 1 | 0.70 |
| 29 | 130m | 42B | soap | 86.5B | 1 | 7.71 |
| 30 | 130m | 42B | nadam | 86.5B | 1 | 0.39 |
| 31 | 130m | 42B | kron | 86.5B | 1 | 0.49 |
| 32 | 130m | 42B | adam | 86.5B | 1 | 0.40 |

- Total Unique Tuples (normalized): 147
- Total GFLOPs: 186.42T
- Total Runs: 4,281
- Total Hours: 7,542.52
- H100 Equivalent Hours: 52,359.48
- H100 Equivalent Days: 2,181.64
- H100 Equivalent Years: 5.98

