# OpenReview forum: "Fantastic Pretraining Optimizers and Where to Find Them"
_ICLR.cc/2026/Conference — ICLR 2026 Poster_

### Official Review · Reviewer_DRRU · 2025-10-31

**Soundness:** 3
**Presentation:** 3
**Contribution:** 3
**Rating:** 6
**Confidence:** 3

**Summary:**

The paper benchmarks 11 pretraining optimizers (e.g., AdamW, Muon, Soap, Mars, Kron) across model scales up to 1.2B parameters and multiple Chinchilla ratios. It introduces a three-phase tuning protocol to ensure fair comparison and finds that most reported speedups over AdamW shrink to ~1.1–1.4× after proper tuning, especially at larger scales. Matrix-based optimizers perform slightly better on small models but lose advantage as size grows.

**Strengths:**

* The paper presents a comprehensive optimizer benchmark for pretraining, with consistent tuning and scaling analysis.
* The proposed three-stage hyperparameter search ensures that each optimizer is fairly tuned, reducing the common bias of under-optimized baselines in prior studies.
* The work clarifies that many claimed gains in prior work are overstated and provides actionable guidance for future optimizers' usage.

**Weaknesses:**

* The work is entirely empirical. Although this is acceptable for benchmarking, it lacks a theoretical analysis explaining why matrix-based optimizers lose advantage with scale or high data-to-model ratios. Even basic intuition about curvature estimation or variance scaling would enrich interpretation.
* Despite the emphasis on “LLM pretraining,” the largest model tested has 1.2B parameters. It remains unclear whether the diminishing returns of matrix-based optimizers persist at realistic scales (10B–70B+).
* The paper normalizes speedup by “tokens to reach a loss,” assuming step overhead is <10%. Please also report wall-clock time, throughput, optimizer step latency and memory footprint. This will show whether matrix preconditioning’s modest token gains translate to real training gains at scale.

**Questions:**

pls see weaknesses

---

> ### Author Response · Authors · 2025-11-22
> **Rebuttal**
>
> **W1:** The paper provides no theoretical explanation for why matrix-based optimizers lose advantage with scale or higher data/model ratios.
>
> **Response:** We acknowledge that our work does not include a rigorous theoretical analysis. Meanwhile we believe several observed phenomena are worthy of deeper theoretical investigation. Since matrix-based optimizers are motivated by better adapting to the loss landscape, we conjecture that the landscape-adaptation advantage plays a smaller role at the largest model scale we tested, where the noise factor may be more important for training dynamics, leading to similar performance between Muon and Nesterov AdamW. We include this conjecture in our paper with the hope that this can provide more insight.
>
> **W2:** The largest evaluated model is only 1.2B parameters, leaving unclear whether the observed diminishing returns persist at realistic large-model scales (10B–70B+).
>
> **Response:** Please refer to our **General Response W2.** Although our experiments use models up to 1.2B parameters, this scale is already on par with the largest academic pretraining efforts and comparable to some of the industry scaling-law studies.
>
> **W3:** Speedup is measured only by tokens-to-target-loss; wall-clock time, throughput, optimizer step latency, and memory footprint are not reported, limiting practical interpretability of speedups.
>
> **Response:** Please refer to our **General Response W3** where we benchmark the throughput for different optimizers using our current best implementations and show that Muon’s step-wise overhead is smaller than the provided speedup. Regarding the memory footprint, the following table contains a breakdown on the memory required to store optimizer state for a **m \* n** matrix.
>
> | Optimizer | Memory Required |
> | :---- | :---- |
> | AdamW  | 2mn |
> | NadamW | 2mn |
> | Mars | 3mn |
> | Cautious | 2mn |
> | Lion | mn |
> | Adam-mini  | mn \+ m (or mn \+ n) |
> | Muon  | mn |
> | Scion  | mn |
> | Kron (PSGD)  | 3mn (with blocking) or 2mn \+ $ (m^2 + n^2) $/2 (without blocking) |
> | Soap  | 4mn (with blocking) or 2mn \+ $m^2 \+ n^2$ (without blocking) |
> | Sophia | 2mn |

---

> ### Author Response · Authors · 2025-11-27
>
> As we approach the final week of the discussion phase, we would like to kindly remind the reviewer to engage with our rebuttal. We have tried to address the concerns raised and would greatly appreciate any further feedback.

---

### Official Review · Reviewer_pAs8 · 2025-10-31

**Soundness:** 4
**Presentation:** 4
**Contribution:** 3
**Rating:** 8
**Confidence:** 4

**Summary:**

The paper presents a comprehensive study of eleven deep learning optimizers to find the pareto frontier across four model sizes (0.1B–1.2B parameters) and data-to-model ratios (1–8× the Chinchilla optimum). It provides valuable practical insights and methodological guidance to find the optimal pretraining optimizer, though it is primarily an empirical study with limited theoretical contributions.

**Strengths:**

Strengths:
- The paper’s deep dive into evaluating multiple optimizers by individually tuning hyper-parameters is very interesting.
- The finding that speedups in newer optimizers over AdamW come down to 1.1X at 1.2B model scale at 8X Chinchilla is intriguing.
- The paper is methodical and analyses the optimizers in an under explored regime (of 1.2B model size and 8X Chinchilla data size) in terms of both model and data size. It has the potential to serve as a good reference for the research community while making decisions on pre-training large language models.
- The authors look at a range of benchmarks to evaluate the optimizers.

**Weaknesses:**

Scope for improvement:
- The largest models the authors experiment with is 1.2B parameters, while SOTA models are at-least 100X larger. This might not be directly applicable to SOTA open source models, because even the authors claim that ranking of optimizers changes with data / model scale.
- On a similar note, a recipe to evaluate optimizers on bigger models and higher data ratios would be very interesting to look at.
- The paper focusses a lot on empirical results, but doesn’t add any theoretical contributions.

**Questions:**

Like mentioned in the scope for improvement section, I am curious what the authors think about:
- a receipe to evaluate optimizers on SOTA models with even higher data ratios.
- theoretical understanding about some of the trends are missing. For instance, in Figure 1, the last line says "the three matrix-based optimizers converge to a similar loss in an overtrained setting." A more in depth theoretical analysis would be very helpful.

---

> ### Author Response · Authors · 2025-11-22
> **Rebuttal**
>
> We thank the reviewer for the encouraging review. Below we provide our responses.
>
> **Weaknesses**
>
> W1: The largest evaluated model is only 1.2B parameters, far smaller than SOTA models (\~100× larger), limiting direct applicability of findings to modern large-scale LLMs.
>
> **Response:** Please refer to our **General Response W2.** Although our experiments use models up to 1.2B parameters, this scale is already on par with the largest academic pretraining efforts and comparable to some of the industry scaling-law studies.
>
> **W2 & Q1:** The paper lacks a recipe or methodological guidance for extending optimizer evaluation to larger models and higher data ratios. What do the authors envision as a feasible recipe for evaluating optimizers on SOTA-scale models with higher data ratios?
>
> **Response:** Thank you for the insightful question. Our work does not focus on proposing a recipe for transferring hyperparameters to larger models. Empirically, we found that our power-law fit of the optimal hyperparameters extrapolates to 1.2B-parameter models. That said, predictability might be improved by incorporating techniques such as MuP. We leave a exploration of the interaction between hyperparameter transfer and optimization performance to future work.
>
> **W3 & Q2**: The work provides primarily empirical insights and does not contribute theoretical analysis to explain observed trends. Can the authors provide deeper theoretical understanding for observed trends, such as why matrix-based optimizers converge to similar losses in the overtrained regime (Figure 1)?
>
> **Response:** We acknowledge that our work does not include a rigorous theoretical analysis on the observed phenomenon. Meanwhile we believe several observed phenomenon—including the one highlighted by the reviewer—are worthy of deeper theoretical investigation. Regarding why matrix-based optimizers converge to similar losses, we conjecture that although their preconditioners follow different update rules, they may behave similarly in practice. For example, it has been shown that Muon can be viewed as Shampoo without momentum \[1\]. We expect that analogous, possibly more complex, approximations may exist for other optimizers as well.
>
> \[1\] Old Optimizer, New Norm: An Anthology

---

> > ### Author Response · Authors · 2025-11-27
> >
> > As we approach the final week of the discussion phase, we would like to kindly remind the reviewer to engage with our rebuttal. We have tried to address the concerns raised and would greatly appreciate any further feedback.

---

### Official Review · Reviewer_3kks · 2025-11-01

**Soundness:** 3
**Presentation:** 2
**Contribution:** 3
**Rating:** 6
**Confidence:** 4

**Summary:**

A thorough pretraining evaluation study is proposed, comparing and contrasting recent matrix-based optimizers such as Muon, SOAP, etc., with scalar-based optimizers such as AdamW. Different model sizes (130M–1.2B) and data budgets in terms of Chinchilla optimal (1x–8x) with the Llama 2 architecture are considered for benchmarking. Key findings include, among others, a decrease in the performance of matrix-based optimizers relative to AdamW as model size increases, and the relative performance differences between different matrix-based optimizers (Muon, SOAP, Kron) disappear in the over-trained regime.

**Strengths:**

- Optimizers are benchmarked under strong settings: diverse data mixtures (code, text, math), language, chinchilla-optimal data budgets (1-8x), various model scales (130m-1.2B) and modern matrix-based baselines (muon, scion, soap).
- Tuning framework to identify scale-sensitive hyperparameters for each optimizer in small-scale settings with sweeps and a scaling law to predict optimal scale-sensitive hyperparameter values for large-scale experiments based on model size and data.

**Weaknesses:**

- Optimizer optimal hyperparameters are chosen with a one-at-time sweep over hyperparameters which may miss correlation among hyperparameters as obtained from a dense grid search. This could have non-trivial downstream impact on the findings of the study, since optimal hyperparameters from a grid search could be quite different. The coordinate descent is done over too many hyperparameters including non-trivial ones with a few runs (~10-20 which is way less compared to prior work e.g. [1]), and I'm not sure if the final hyperparameters obtained from this one-at-a-time coordinate descent are actually optimal compared to a standard grid search over hyperparameters.
- Schedule for Muon in appendix tables (Table 83-88) is mentioned as "linear" -- does it mean just a linear decay? On the other hand, figure 6 shows cosine decay with warmup for different optimizers. Can authors confirm the choice of scheduler and if it's the standard cosine decay?
- It should be noted that benchmark is highly specific -- llama 2 architecture with text-only datasets, fixed sequence length (4096) and upto 1.2B scale. Hence, findings such as matrix-shaped optimizers decrease in speedup with model size, may change in multi-modal setup with vision tokens and different/modern architectures such as MoEs which are realistically used in the pretraining of modern LLMs.

[1] Small Batch Size Training for Language Models (NeurIPS 2025), https://arxiv.org/abs/2507.07101

**Questions:**

- Can authors please elaborate on the search space of hyperparameters used for each optimizer in their sweeps and how was it chosen?
- During coordinate descent - how are hyperparameters sampled and what scale is used (uniform or log-scale)?

---

> ### Author Response · Authors · 2025-11-22
> **Rebuttal**
>
> We thank the reviewer for the constructive feedback and thoughtful questions. Below we provide our responses.
>
> **W1:** The one-at-a-time coordinate descent tuning may miss correlations among hyperparameters, potentially leading to suboptimal configurations relative to dense grid search.
>
> **Response:** Please refer to our **General Response W1** for details. We show that
>
> 1. The stopping criterion we adopt for coordinate descent is widely used in prior work to show optimality of hyperparameter configuration\[1–5\].
> 2. In our new **Appendix I**, we validate our coordinate descent procedure against 2D grid searches (e.g., Learning Rate × Batch Size, Learning Rate × Weight Decay) with 130M-parameter models and Muon optimizer and find that our previously found hyperparameter remain optimal
> 3. Theoretically, coordinate descent can still converge to near-optimal hyperparameters even when coupling exists, under smoothness assumptions with respect to hyperparameters, which is empirically observed in prior studies.
>
> **W2:** The learning rate schedule for Muon is unclear: appendix tables say “linear,” while Figure 6 shows cosine decay with warmup.
>
> **Response:** We use a stable and decay learning rate schedule for Muon as suggested by the original authors \[1\]. In Figure 6, the plotted cosine decay schedule is used for other optimizers and the plotted learning rate curve for Muon is linear. We also include a new ablation on learning rate schedule (cosine vs linear decay) in new **Appendix K** and observe minimal difference.
>
> \[1\] Muon: An optimizer for hidden layers in neural networks
>
> **W3:** The benchmark is narrowly scoped—LLaMA-2 architecture, text-only data, fixed 4096 sequence length, and models up to 1.2B—limiting generalizability to multimodal settings and modern architectures such as MoEs.
>
> **Response:** Please refer to our **General Response W2** for concerns regarding model sizes. We acknowledge that we focus on text-only data and didn’t explore the effect of context lengths, other modalities, or architectures. This is indeed a limitation of the current work and we plan to continue to explore these settings in the future.
>
> **Q1:** What was the precise hyperparameter search space for each optimizer, and how were these ranges chosen?
>
> **Response:** Thank you for the question. We provide the exact hyperparameter search spaces for all settings studied in Appendices D–G. These ranges follow consistent patterns:
>
> 1. 2× power gridding for the learning rate,
> 2. 1e–5× power gridding for $\\epsilon$,
> 3. Uniform gridding for $(1/(1-\\beta))$, where $\beta$ is the first or second momentum term depending on the optimizer,
> 4. Uniform gridding for weight decay.
>
> **Q2:** During coordinate descent \- how are hyperparameters sampled and what scale is used (uniform or log-scale)?
>
> **Response:** Thank you for the question. During coordinate descent, we sweep each hyperparameter along its 1-dimensional grid while holding the others fixed. After evaluating all candidates, we select the run with the lowest loss and use it as the new starting point for the next coordinate descent.

---

> > ### Author Response · Authors · 2025-11-27
> >
> > As we approach the final week of the discussion phase, we would like to kindly remind the reviewer to engage with our rebuttal. We have tried to address the concerns raised and would greatly appreciate any further feedback.

---

### Official Review · Reviewer_xiBh · 2025-11-07

**Soundness:** 2
**Presentation:** 3
**Contribution:** 2
**Rating:** 2
**Confidence:** 4

**Summary:**

The paper benchmarks popular optimizers on LLM pretraining, including AdamW as a baseline. It tunes their hyperparameters using a three phased methodology. In phase 1 and phase 2 the hyperparameters are searched using coordinate descent i.e., at every iteration only a single hyperparameter is searched, while all others are held fixed at their current best values, and this is repeated until convergence. In phase 2, which involves larger scale experiments, only a subset of the hyperparameters are tuned, based on if they are determined scale sensitive using phase 1 results. Finally, phase 3 fits scaling laws from the hyperparameters found in phase 2. The benchmark trains Llama 2 architectures of four sizes, from 0.1B to 1.2B parameters and reports validation loss on C4-EN and on downstream tasks. The paper concludes that, while newer optimizers show non-negligible speedups over AdamW, the speedups are more moderate than those stated in existing literature and decrease as model size increases.

**Strengths:**

The paper addresses an important and timely topic. Pretraining is indeed the most resource-demanding stage in the LLM training process and thus increasing the optimizer's speed is of great research interest. I also agree with the authors that many works that propose new optimizers lack fair and rigorous evaluation, leading to unrealistic speedups (e.g., 2-3x speedups over AdamW).

A wide range of popular optimizers is evaluated, from different representative categories.

The experimental settings are clearly stated, and the accompanying code with the sweep configs ensures reproducibility of the results.

**Weaknesses:**

**Suitability of Coordinate Descent for hyperparameter tuning:**

My main concern is the use of Coordinate Descent (CD) for hyperparameter tuning. The paper's primary contribution, as stated in the introduction, is the claim that "the hyperparameters are near-optimal for each setting", and this is reiterated in the Phase 2 results section: "Combined with the results in Phase I, we obtain a set of near-optimal hyperparameters".

However, it is well established in the literature that many optimizer hyperparameters are coupled. For example, the learning rate and batch size are often considered to follow  a  linear or square root relationship [1,2]. Recent studies suggest that the weight decay also shares a similar coupling with the batch size [2]. [3] also demonstrates a strong relationship between the Adam beta2 parameter and batch size. Similar couplings are likely to exist for other hyperparameters, such as learning rate and warmup steps (fewer warmup steps could likely benefit from smaller learning rates) or learning rate and gradient norm clipping threshold.

In short, the objective (validation loss) is far from separable across coordinates (hyperparameters), making it uncertain whether CD can truly obtain "near-optimal" hyperparameters rather than getting stuck at "bad" local minima. In addition, the three phased methodology increases the likelihood of errors aggregating from one phase to another.

The paper includes no ablations or error analysis (e.g., as in [4]) to validate the tuning methodology and support the claim of "near-optimal hyperparameters". The ablations in the Appendix ("1-dimensional ablations centered around the found configuration") do not address such possible limitations of CD, since each step varies only a single parameter at a time. For example, the batch size is increased from 128 to 256 and 512, while the learning rate as well as all other hyperparameters remain fixed.

In short, since all of the paper's results rely on the robustness of the evaluation methodology, it is critical that this methodology is fully justified and potential limitations of CD in this setting be discussed. The paper makes a strong claim of "near-optimal hyperparameters", but absence of such analysis make this claim uncertain and this represents a major limitation of the work.

**Speedup Results Comparability:**

The paper defines the speedups based on the number of tokens needed to reach a given loss, without considering the time each optimizer needs to reach that loss. However, since the batch size is treated as a tunable hyperparameter, some optimizers use larger batch size than others in certain settings. For example, on 300M model for 4x Chinchilla Data, AdamW uses batch size 256 (Table 102) and Adam-Mini 128 (Table 132). As a result, it is unclear if the reported results are directly comparable, since optimizers with larger batch size may have increased throughput (assuming sufficient hardware), which could lead to a faster convergence in wall-clock time.

In short, it is debatable if the batch size should be treated as a regular hyperparameter, given that it influences the throughput and can affect the speedup in terms of wall-clock time. Furthermore, it is common for users to want to choose the best optimizer for a given batch size constrained by memory limitations, and the results provided do not help with this decision-making process.

**Single Architecture Type Used:**

The paper only tests the optimizers on four models of the same family (LLaMA 2). Including a broader range of model families would have been ideal to ensure the results are generalizable. For example, including more recent models (e.g., Gemma 2 or Qwen 3) or exploring different model types such as Mixture-of-Experts. Additionally, while scaling up to larger models (7B+) might be resource-prohibitive, testing on models at this range would more closely approximate production-level model sizes.

[1] LARGE BATCH OPTIMIZATION FOR DEEP LEARNING: TRAINING BERT IN 76 MINUTES, 2020.

[2] Power Lines: Scaling Laws for Weight Decay and Batch Size in LLM Pre-training, 2025.

[3] Small Batch Size Training for Language Models: When Vanilla SGD Works, and Why Gradient Accumulation Is Wasteful, 2025.

[4] Practical Efficiency of Muon for Pretraining, 2025.

**Questions:**

Is Coordinate Descent (CD) a common methodology for tunning optimizer hyperparameters? Could the authors cite relevant works (preferably in the setting of LLM pretraining) to support this? Could the authors provide supporting experimental evidence or analysis for the claim of "near-optimal hyperparameters" and also discuss potential limitations of CD (if any) for the given setting?

It would be useful if the authors could provide some statistics about the number of runs conducted per setting. For example, for each unique triplet (Optimizer, Model size, Chinchilla Ratio) how many pretraining experiments where performed in each phase? Also, could the authors provide how long each experiment took approximately (if possible translated in H100 GPU hours). This would help me get a better understanding of the scale of the benchmark.

It is mentioned in the paper that the speedup of matrix-based optimizers is inversely proportional to model scale, decreasing to only 1.1x for 1.2B parameter models. Could the authors discuss possible reasons for this trend? Additionally, for models larger than 1.2B parameters, do the authors expect the speedup to decrease further?

In Table 5 (Scaling-Sensitive Hyperparameters), some optimizers appear to have more scale-sensitive hyperparameters than others. For example, AdamW appears scale-sensitive to learning rate, warmup, weight decay and batch size, whereas Muon only to learning rate. Also, Adam-Mini is not scale-sensitive to batch size, while AdamW is. Could the authors provide any further insights on these differences

---

> ### Author Response · Authors · 2025-11-22
> **Rebuttal**
>
> We thank the reviewer for the constructive feedback and thoughtful questions. Below we provide our responses.
>
> **Coordinate descent related questions**
>
> **W1:** Coordinate descent (CD) may be unsuitable because optimizer hyperparameters are highly coupled.
>
> **Q1:** Is coordinate descent commonly used for tuning optimizer hyperparameters in LLM pretraining?
>
> **Response:** Please refer to our **General Response W1** for details. We show that
>
> 1. The stopping criterion we adopt for coordinate descent is widely used in prior work to show optimality of hyperparameter configuration\[1–5\].
> 2. In our new **Appendix I**, we validate our coordinate descent procedure against 2D grid searches (e.g., Learning Rate × Batch Size, Learning Rate × Weight Decay) with 130M-parameter models and Muon optimizer and find that our previously found hyperparameter remain optimal
> 3. Theoretically, coordinate descent can still converge to near-optimal hyperparameter even when coupling exists, under smoothness assumptions with respect to hyperparameters, which is empirically observed in prior studies.
>
> **W2 & Q2.1:** The claim of “near-optimal hyperparameters” lacks supporting validation.
>
> **Response:** In our new **Appendix I**, we strengthen our claim with direct comparisons to 2D grid searches using 130M-parameter models. In all examined settings, coordinate descent finds configurations matching the optima from 2-dimensional grid search.
>
> **Q2.2:** Can the authors discuss CD’s limitations?
>
> Response: As our theoretical analysis in the new **Appendix I** suggests, CD requires the hyperparameter loss landscape to be free of saddle points in order to guarantee convergence to a local minimum. We discuss in Appendix I and cite prior work \[10–12\], which shows that the pretraining loss with respect to many hyperparameters pairs satisfies this condition when constrained on the 2D surface.
>
> **Other questions**
>
> **W3:** Treating batch size as tunable complicates speedup comparability because throughput varies with batch size, and comparison fixing batch size can be helpful when making decision in memory constraint scenarios.
>
> **Response:** We agree this is an important practical setting. As described in **General Response W3** and further detailed in new **Appendix J**, we added experiments comparing AdamW and Muon fixing batch size. We find that Muon’s performance degrades more slowly as batch size increases, compared to AdamW. Consequently, the relative speedup of Muon over AdamW is smaller when the batch size is low, yet it is still larger than 1\.
>
> **W4:** The study only evaluates LLaMA-2–style architectures up to 1.2B, limiting generalizability.
>
> **Response:** Please refer to our **General Response W2**. While our experiments focus on LLaMA-2 models up to 1.2B parameters, this scale is already among the largest in academic pretraining efforts \[1,6–8\] and is comparable to scaling-law studies in industry \[5,9\]. LLaMA-2 is a widely used dense Transformer architecture, and we expect the qualitative behaviors we observe to hold for similar architectures that differ mainly in having additional normalization layers. We agree a more detailed investigation of how architectural choices interact with the optimizer is an important direction, which we leave for future work.
>
> **Q3:** How many runs were conducted per (optimizer, model size, Chinchilla ratio), and what were the approximate H100 GPU-hours per run?
>
> **Response:** Thank you for the question. We provide average statistics here for reference and include a detailed breakdown in new **Appendix L**. Here **Total GFLOPs** is computed as the theoretical flops only considering the model forward and backward passes and Equivalent H100 Hours is computed assuming full utilization at bfloat16 (989 TFLOPs/s)
>
> | Phase | Total GFLOPs | Num Runs | Unique Tuples | Equivalent H100 Hours | Avg Runs/Tuple |
> | :---- | :---- | :---- | :---- | :---- | :---- |
> | Phase I | 79309756738427.94 | 3772 | 61 | 22275.518688469816 | 61.84 |
> | Phase II | 70980330165785.42 | 407 | 54 | 19936.05498421116 | 7.54 |
> | Phase III | 36130599806483.79 | 102 | 32 | 10147.904675453261 | 3.19 |

---

> ### Author Response · Authors · 2025-11-22
> **Rebuttal (continued)**
>
> **Q4:** Why do speedups of matrix-based optimizers shrink with model scale, and what is expected beyond 1.2B?
>
> **Response:** Thank you for the question. We do not yet have a complete explanation for the shrinkage in speedup. However, since matrix-based optimizers are motivated by better adapting to the loss landscape, we conjecture that the landscape-adaptation advantage plays a smaller role at the largest model scale we tested, where the noise factor may be more important for training dynamics.  Extrapolating to even larger models, we expect the performance gap between small-batch Muon and NAdamW to remain small (and potentially shrink further), while both should continue to provide speedup over AdamW, as the benefit from Nesterov momentum appears roughly constant across settings. At the same time, because Muon tolerates a larger critical batch size, we expect it to remain empirically favorable for large-scale training with substantial compute.
>
> **Q5:** Why are some optimizers more sensitive to scale in their hyperparameters?
>
> **Response:** Thank you for the great question\! In our paper, the hyperparameters that are sensitive to scale are identified empirically based on the Phase I experiments, and we do not yet have a clear understanding of why some hyperparameters exhibit greater scale sensitivity than others. We agree that designing optimizers whose hyperparameters are robust to scale remains an important direction for future research.
>
> \[1\] (ICML 2025\) MARS: Unleashing the Power of Variance Reduction for Training Large Models
>
> \[2\] (ICLR 2025 Spotlight) u-μP: The Unit-Scaled Maximal Update Parametrization
>
> \[3\] (ICLR 2025\) How Does Critical Batch Size Scale in Pre-training?
>
> \[4\] (ICML 2024\) Scaling Exponents Across Parameterizations and Optimizers
>
> \[5\] (Kimi Technical Report) Muon is Scalable for LLM Training
>
> \[6\] (COLM 2024\) Mamba: Linear-Time Sequence Modeling with Selective State Spaces
>
> \[7\] (ICLR 2025\)  Gated Delta Networks: Improving Mamba2 with Delta Rule
>
> \[8\] (NeurIPS 2025\) The Road Less Scheduled
>
> \[9\] (Kimi Technical Report)  Kimi Linear: An Expressive, Efficient Attention Architecture
>
> \[10\] Large Batch Optimization for Deep Learning: Training BERT in 76 Minutes
>
> \[11\] Power Lines: Scaling Laws for Weight Decay and Batch Size in LLM Pre-training
>
> \[12\] Small Batch Size Training for Language Models: When Vanilla SGD Works, and Why gradient Accumulation Is Wasteful

---

> ### Author Response · Authors · 2025-11-27
>
> As we are in the final week of the discussion phase, we would like to kindly remind the reviewer to engage with our rebuttal. We have conducted experiments to address the concerns raised and would greatly appreciate any further feedback.

---

> > ### Comment · Reviewer_xiBh · 2025-11-27
> >
> > I thank the authors for the detailed rebuttal. I appreciate the detailed compute breakdown and recognize that the amount of compute used is among the highest in related studies. Also I appreciate the new experiments on batch size (Appendix J). I'm still a bit skeptical about the (near) optimality of the hyperparameters found from CD (I include my response to that part of the rebuttal bellow). Based on that, I will increase my score to 4 for now, but I'll be happy to further increase the score based on our discussion.
> >
> > **Comments on CD related responses:**
> >
> > > First, the stopping criterion we adopt for coordinate descent is widely used in prior work to show optimality of hyperparameter configuration[1–5].
> >
> > I read the references [1-5] and I'm not entirely convinced that they support the use of CD for optimizer benchmarking. Bellow are my comments on those references:
> >
> > [1]: In section "F Hyper-parameter Settings" it is mentioned that they perform grid searches only over learning rate and weight decay (except for AdamW where learning rates are taken from literature). In addition, in section "E.7 Sensitivity to Batch Size" different batch sizes are tested with fixed learning rate ("We fix other hyper-parameters the same as the main experiments"). So overall I find the hyperparameter search in this paper to be limited. This is even highlighted in this submission:
> >
> > *"This tuning process may result in a weak baseline (...) We validate this concern by showing that tuning just one hyperparameter in the widely adopted GPT-3 recipe (...)  used in (...) Yuan et al. (2025);"*
> >
> > Therefore I don't think [1] can be used to strengthen the case for CD. Similarly, [5] also seems to mostly focus on learning rate and batch size search to construct scaling laws.
> >
> > [2], [4]: To my understanding those two papers focus on learning rate tuning for muP, so they are not directly relevant to hyperparameter search methodology for optimizer benchmarking.
> >
> > [3]: Indeed in section "D Additional Details on Experiments" it is mentioned that due to compute constraints the hyperparameters are searched in sequence. However, [3] has extra results on Adam beta2. Specifically, it is mentioned in [3] that:
> >
> > *"When training a larger model with a longer duration (e.g. Chinchilla settings in Appendix Figure 8b), a high enough β2 is necessary. (...) Smaller β2 = 0.95 is helpful for large-batch training over short durations, while a large β2 = 0.99, 0.999 or 0.9995 is helpful for long-duration training and substantially improves small batch size training (<262k tokens)."*
> >
> > Also figure 8(b) shows that the 1.2B model in Chinchilla settings benefits from beta2=0.999.
> >
> > In contrast, based on appendices D-G of this submission, to my understanding all best AdamW beta2 are found to be 0.98, for all models sizes and Chinchilla settings. Could the authors comment on those differences between their paper and [3]? I recognize that the time until the time left for the rebuttal is limited, but I would appreciate any further ablations on AdamW (as the current 2D grid ablations only include Muon).
> >
> > > We examined prior works [7-11] in Appendix I on hyperparameter interactions and found in all of these papers, the empirically observed loss surface does not exhibit saddle points or spurious local minima—consistent with our assumptions. These works therefore do not contradict the applicability of CD; instead, they suggest that the relevant search problems are well behaved.
> >
> > I examined appendix "I.2 Benignity of the hyperparameter optimization from prior works", while I tend to agree with point 1 about Figure 1 of Li et al. (2025a), points 2. and 3. appear inconsistent with what the figures in the respective papers show. Please correct me if I'm mistaken in this case.
> >
> > *"2. Bergsma et al. (2025) study the interactions between weight decay and batch size. As shown in the Figure 4 of Bergsma et al. (2025), the loss surface is near quasi-convex with respect to weight decay and batch size and only contains one basin."*
> >
> > To my understanding Figure 4 of Bergsma et al. (2025) does not investigate the loss landscape of weight decay vs batch size, since weight decay is not one of the quantities shown in that plot.
> >
> > *"3. Marek et al. (2025) study the interactions between learning rate and β2. As shown in the Figure 1 of Marek et al. (2025), the loss surface is near quasi-convex with respect to learning rate and β2 is near quasi-convex and only contains one basin."*
> >
> > Figure 1 of Marek et al. (2025) studies the interactions between learning rate and first order momentum (beta1), not beta2. In addition, among beta1 and beta2, Marek et al. (2025) find beta2 to be much less robust (e.g., to batch size changes).

---

> > > ### Author Response · Authors · 2025-11-28
> > >
> > > We include a further experiment validating our coordinate descent hyperparameter tuning method on the AdamW 130M $1\times$ Chinchilla setting in Figure 10. We perform a two-dimensional sweep over fine-grained (learning rate, $\beta_2$) and (learning rate, weight decay) pairs. We observe that our previously found hyperparameters ($\eta = 0.008, \beta_2 = 0.98, \lambda = 0.1$) are within $0.0011$ w.r.t. the best (learning rate, weight decay) pair ($\eta = 0.012, \lambda = 0.05$) and within $0.003$ w.r.t. the best (learning rate, $\beta_2$) pair ($\eta = 0.008, \beta_2 = 0.99$). Both of these are within our tolerance bound $\Delta_1$ defined in Section 3.2.

---

> ### Author Response · Authors · 2025-11-28
>
> We would like to thank the reviewer for the careful reading after the rebuttal, the constructive comments and proposing to raise the score to 4. Here is our response to the remaining concerns.
>
> **1\. ​​The strength of cited works \[1-5\] to support validity of coordinate descent**
>
> We agree with the reviewer that the hyperparameter tuning in \[1\] is not as thorough as in this paper. Although \[2, 4, 5\] focus on hyperparameter transfer or scaling laws, their primary objective is still to find good hyperparameters, and they validate their choices using one-dimensional grid searches. \[3\], as reviewer mentioned, mostly adopted a simplified version of coordinate descent tuning strategy. We cite these works to highlight that our paper makes a greater effort to identify better hyperparameters than prior published work. Consequently, we believe these related works support our claim.
>
> **2\. Difference in choices of AdamW \\beta2 in \[3\]**
>
> The reviewer noted that our choice of $\\beta\_2$ in the AdamW experiments is different compared to \[3\]. We believe this discrepancy is largely due to differences in other hyperparameter settings. In particular, \[3\] does not use weight decay or learning-rate decay. As noted by Bergsma et al. \[6\], weight decay implicitly encourages weights to stay close to an exponential moving average of their updates. Consequently, differences in weight decay and learning-rate schedules can substantially alter the distribution of gradients, reducing the need to average over very long horizons to obtain a reliable second-moment estimate for AdamW. This may explain the discrepancy between the value of $\\beta\_2$ recommended in \[3\] and the value we found to work best in our experiments. We will conduct additional experiments to further validate our AdamW hyperparameter-tuning procedure, particularly with respect to $\\beta\_2$, and notify you when the experiment is finished.
>
> **3\. Questions related to Appendix J**
> We are sorry for the confusion\! You are correct that we made some mistakes in reference in Appendix J. We have corrected it now and here is the updated version with more details, with the bolded text being modified or added.
>
> *2\. Bergsma et al. (2025) study the interactions between weight decay and batch size. As shown in the Figure **2** of Bergsma et al. (2025), the loss surface is near quasi-convex with respect to weight decay and batch size and only contains one basin roughly aligning with $\\lambda \\propto B$. **This is consistent with the author’s observation that the weight decay makes the weight an exponential moving average update over $B/(\\eta\\lambda)$ data.***
>
> *3\. Marek et al. (2025) study the interactions between learning rate and $\beta_1$. As shown in the Figure 1 of Marek et al. (2025), the loss surface is near quasi-convex with respect to learning rate and **$\\beta\_1$** is near quasi-convex and only contains one basin. **While they observe higher sensitivity to $\\beta\_2 $ in Figure 4, the resulting landscape is still smooth and contains only one basin in the search space spanned by $\\log(1/(1-\\beta\_2))$ and batch size.***
>
> \[1\] (ICML 2025\) MARS: Unleashing the Power of Variance Reduction for Training Large Models
>
> \[2\] (ICLR 2025 Spotlight) u-μP: The Unit-Scaled Maximal Update Parametrization
>
> \[3\] (ICLR 2025\) How Does Critical Batch Size Scale in Pre-training?
>
> \[4\] (ICML 2024\) Scaling Exponents Across Parameterizations and Optimizers
>
> \[5\] (Kimi Technical Report) Muon is Scalable for LLM Training
>
> \[6\] Power Lines: Scaling Laws for Weight Decay and Batch Size in LLM Pre-training

---

### Author Response · Authors · 2025-11-22
**General Response**

We would like to thank the reviewers for their constructive reviews. We would like to answer some joint concerns below

**W1: Concerns about the hyperparameter tuning methodology (coordinate descent).**

Reviewer xiBh and 3kks raised concerns that CD might be suboptimal due to coupling between hyperparameters. We provide new theoretical and empirical results to clarify the following points.

**First**, the stopping criterion we adopt for coordinate descent is widely used in prior work to show optimality of hyperparameter configuration\[1–5\].

Our criterion ensures that, once CD terminates, changing *any* individual hyperparameter cannot significantly reduce the loss. Thus, our CD procedure generalizes the widely adopted one-dimensional grid-search methodology used to identify near-optimal hyperparameters.

**Second,** to further verify the robustness of our tuned hyperparameters, we conducted two-dimensional grid searches for (i) learning rate × batch size, (ii) learning rate × weight decay, and (iii) learning rate × decay ratio in WSD, for the Muon optimizer on 130M models in appendix I. In both cases, the hyperparameters identified by our CD procedure remained optimal under full 2D search, confirming that coordinate descent did not miss better coupled configurations.

**Third**, coordinate descent can still recover near-optimal hyperparameters even when hyperparameters interact with one another, under mild assumptions.

Reviewers xiBh and 3kks questioned whether CD can find a good configuration in the presence of couplings between hyperparameters. Intuitively, even when hyperparameters interact, a sufficiently smooth and well-behaved loss landscape (such as a convex one) lets coordinate descent reliably improve along each dimension without getting trapped, allowing it to converge to a near-optimal configuration. In the added Appendix I, we formalize this and show once (i) each hyperparameter is discretized densely enough, and (ii) the hyperparameter-to-loss function is smooth and contains no saddle points, CD convergence to a locally optimal configuration.

We examined prior works  \[7-11\]  in Appendix I on hyperparameter interactions and found in all of these papers, the empirically observed loss surface does *not* exhibit saddle points or spurious local minima—consistent with our assumptions. These works therefore do not contradict the applicability of CD; instead, they suggest that the relevant search problems are well behaved.

**W2: Scale of the experiments are limited to 1.2B**

Reviewers point out that our empirical studies have the limitation that the largest model sizes are only 1.2B. We acknowledge that the sizes of the models studied in this paper are far from the SOTA models. However, we would like to point out that

1. The model sizes studied here is among the largest in academic papers studying pretraining, including both architectures and optimizers study \[1,12-14\]
2. The model sizes are comparable with scaling law experiments in published technical reports of companies training large language model \[5, 16\]

**W3: Speedup definition lacks wall-clock time and system-level metrics.**

For our work, we deliberately choose to compare optimizers step-wise because the wall-clock time is dependent on many factors: chosen type and number of hardwares, implementation and batch size (as reviewer xiBh) points out. Meanwhile, we fully agree that wall-time comparison is crucial for informing the readers on which optimizer to choose. To incorporate this need, we add

1. A table summarizing the Avg tokens/second for the 1.2B model runs for different optimizers on the same hardware using fixed hardware (TPU v5litepod-64).

| Optimizer | Tokens/s @128 | Rel. to AdamW @128 | Tokens/s @256 | Rel. to AdamW @256 |
| :---- | ----: | ----: | ----: | ----: |
| **AdamW** | 669,039 | 1.000 | 673,535 | 1.000 |
| **Nadam** | 670,750 | 1.003 | 672,350 | 0.998 |
| **Muon** | 551,110 | 0.824 | 605,713 | 0.899 |
| **Lion** | 671,662 | 1.004 | 675,709 | 1.003 |
| **Mars** | 660,268 | 0.987 | 669,821 | 0.994 |
| **Cautious** | 667,314 | 0.997 | 673,750 | 1.000 |
| **SOAP** | 447,907 | 0.670 | 500,859 | 0.743 |

We found that all scalar-level optimizers typically exhibit negligible overhead compared to AdamW. In contrast, our implementation of Muon demonstrates roughly 20% overhead at a batch size of 128, though this drops to 10% when the batch size is increased to 256\. Soap incurs a larger overhead than Muon, likely due to a combination of its more complex update rule and the resulting lack of optimization in our current implementation.

---

> ### Author Response · Authors · 2025-11-22
> **General Response (continued)**
>
> 2. We further run extended experiments studying the effect of batch size on the step-wise speedup comparing Muon and AdamW.
>
> Our research focuses on token-wise comparison, and therefore considers descent along the batch-size dimension as well. However, as reviewer xiBh points out, batch size is not always a tunable factor. It is thus informative to examine the speedup under fixed batch sizes. In response, we conducted additional experiments analyzing how batch size affects the speedup between Muon and AdamW, representing matrix-level and scalar-level optimizers, respectively. The results are now provided in the new **Appendix J**. Across model sizes (130M–1.2B), we find that Muon’s performance degrades more slowly as batch size increases, compared to AdamW. Consequently, the relative speedup of Muon over AdamW is smaller when the batch size is low. We also verify that the batch sizes used in the main paper are generally below or equal to the critical batch size.
>
>
> [1] (ICML 2025) MARS: Unleashing the Power of Variance Reduction for Training Large Models
>
> [2] (ICLR 2025 Spotlight) u-μP: The Unit-Scaled Maximal Update Parametrization
>
> [3] (ICLR 2025) How Does Critical Batch Size Scale in Pre-training?
>
> [4] (ICML 2024) Scaling Exponents Across Parameterizations and Optimizers
>
> [5] (Kimi Technical Report) Muon is Scalable for LLM Training
>
> [6] Convergence of a Block Coordinate Descent Method for Nondifferentiable Minimization
>
> [7] Large Batch Optimization for Deep Learning: Training BERT in 76 Minutes
>
> [8] Power Lines: Scaling Laws for Weight Decay and Batch Size in LLM Pre-training
>
> [9] Small Batch Size Training for Language Models: When Vanilla SGD Works, and Why gradient Accumulation Is Wasteful
>
> [10] Practical Efficiency of Muon for Pretraining
>
> [11] Predictable Scale: Part I, Step Law -- Optimal Hyperparameter Scaling Law in Large Language Model Pretraining
>
> [12] (COLM 2024) Mamba: Linear-Time Sequence Modeling with Selective State Spaces
>
> [13] (ICLR 2025)  Gated Delta Networks: Improving Mamba2 with Delta Rule
>
> [14] (NeurIPS 2025) The Road Less Scheduled
>
> [15] (Kimi Technical Report)  Kimi Linear: An Expressive, Efficient Attention Architecture

---

### Author Response · Authors · 2025-12-01
**Summary of Rebuttal to AC**

Dear AC,

Due to the unfortunate incident this year, the reviewers is unable to further respond to our rebuttal. We would therefore like to summarize the current discussions we have had with the reviewers.

In our general response, we clarified the reviewers’ concerns regarding (1) the validity of our coordinate descent approach, (2) the scale of our experiments, and (3) wall-clock time comparisons, supported by new experimental and theoretical results.

We are encouraged that Reviewer xiBh *appreciates the detailed compute breakdown and recognizes that the amount of compute used is among the highest in related studies*, and has proposed raising the score from 2 to 4. Per their request, we also conducted additional experiments validating our coordinate descent methodology on AdamW.

While the other reviewers have not yet had the opportunity to respond to our rebuttal, we believe that our newly added experiments and clarifications address most of the concerns that were raised. We respectfully ask that you take these additional results and our responses into account in your overall assessment of the paper.

Thank you very much for your time and consideration.

Best regards,

The authors

---

### Meta-Review · Area_Chair_ae1V · 2026-01-06

**Summary:**

A thorough pretraining evaluation study is proposed, comparing and contrasting recent matrix-based optimizers such as Muon, SOAP, etc., with scalar-based optimizers such as AdamW. Different model sizes (130M–1.2B) and data budgets in terms of Chinchilla optimal (1x–8x) with the Llama 2 architecture are considered for benchmarking. Key findings include, among others, a decrease in the performance of matrix-based optimizers relative to AdamW as model size increases, and the relative performance differences between different matrix-based optimizers (Muon, SOAP, Kron) disappear in the over-trained regime.

**Reviewer Concerns:**

Reviewer xiBh and 3kks raised concerns that CD might be suboptimal due to coupling between hyperparameters.

**Reviewer Scores:**

No

---

### Decision · Program_Chairs · 2026-01-26

Accept (Poster)